# From Galactic Bars to the Hubble Tension: Weighing Up the Astrophysical Evidence for Milgromian Gravity

**Indranil Banik *** and **Hongsheng Zhao**

Scottish Universities Physics Alliance, University of Saint Andrews, North Haugh, Saint Andrews KY16 9SS, Fife, UK; hz4@st-andrews.ac.uk
* Correspondence: ib45@st-andrews.ac.uk

**Abstract:** Astronomical observations reveal a major deficiency in our understanding of physics—the detectable mass is insufficient to explain the observed motions in a huge variety of systems given our current understanding of gravity, Einstein's General theory of Relativity (GR). This missing gravity problem may indicate a breakdown of GR at low accelerations, as postulated by Milgromian dynamics (MOND). We review the MOND theory and its consequences, including in a cosmological context where we advocate a hybrid approach involving light sterile neutrinos to address MOND's cluster-scale issues. We then test the novel predictions of MOND using evidence from galaxies, galaxy groups, galaxy clusters, and the large-scale structure of the universe. We also consider whether the standard cosmological paradigm (ΛCDM) can explain the observations and review several previously published highly significant falsifications of it. Our overall assessment considers both the extent to which the data agree with each theory and how much flexibility each has when accommodating the data, with the gold standard being a clear a priori prediction not informed by the data in question. Our conclusion is that MOND is favoured by a wealth of data across a huge range of astrophysical scales, ranging from the kpc scales of galactic bars to the Gpc scale of the local supervoid and the Hubble tension, which is alleviated in MOND through enhanced cosmic variance. We also consider several future tests, mostly at scales much smaller than galaxies.

**Keywords:** gravitation; cosmology; galaxies: kinematics and dynamics; galaxies: evolution; galaxies: interactions; galaxies: groups; galaxies: clusters; large-scale structure of universe

## 1. Introduction

Our current understanding of physics is based on four fundamental forces: strong and weak nuclear forces, electromagnetism, and gravity. These must ultimately be different aspects of the same unified fundamental interaction. However, the long-sought unification with gravity has been hampered by theoretical difficulties and its relative weakness, which makes it extremely difficult to build accurate gravitational experiments. Gravity is so weak that two electrons 1000 km apart repel each other by $2.5 \times 10^{-10}$ m/s$^2$, slightly more than the galactocentric orbital acceleration of the solar system due to the combined gravitational pull of the entire galaxy [1]. The gravitational fields of individual particles are thus immeasurably small.

The relative weakness of gravity means that one must by necessity consider astronomical observations in order to understand it empirically, which is probably the first step to a deeper theoretical understanding. That gravity declines as inverse distance squared was first an empirical formula by Newton to explain Kepler's third law that each solar system planet's period$^2$ ∝ its orbital semi-major axis$^3$. A field theoretical basis was then put forth in terms of the Poisson equation, which clarified that the inverse square decline of both gravity and the electrostatic force can be understood in terms of field lines spreading in 3D. Newtonian gravity is an excellent description of non-relativistic motions in the solar

system, so much so that discrepancies it faced with the motion of Uranus were attributed to a previously unknown planet that was subsequently discovered.

While the successful a priori prediction of Neptune is widely known, a similar issue arose for the orbit of Mercury. The small but statistically significant $43''$ per century discrepancy between observations and the Newtonian expectation for its rate of perihelion precession was for a long time also attributed to an undiscovered planet (Vulcan), leading astronomers on an ultimately unsuccessful wild goose chase characterized by many claimed detections that were later proven incorrect [2]. The anomalous perihelion precession of Mercury was actually an important clue towards a fully relativistic theory of gravity, namely General Relativity (GR [3]). This was designed to reduce to Newtonian dynamics in the appropriate limit of low velocities and shallow potentials with escape velocities much smaller than $c$, the speed of light.

The predictions of GR have been verified one after another over the past century. Observations of the star S2 allow particularly stringent constraints because it approaches within 120 AU of the galactic centre black hole. Even in this strong field, GR correctly predicts the gravitational redshift of S2 [4] and its pericentre precession [5], which is much more extreme than for Mercury. Particularly striking have been the recent observations of a black hole shadow [6] and the measurement that gravitational waves (GWs) travel at $c$ to a precision of $\sim 10^{-15}$ [7], with $\sim$ used to mean equality only at the order of magnitude level (for higher precision but where there is still uncertainty with the stated significant figures, we will use $\approx$).

Despite these impressive successes, it is less clear whether GR can be extrapolated to the non-relativistic motions in the outskirts of galaxies and galaxy clusters. Wherever the relativistic corrections contribute insignificantly to the velocities in these systems, we will use the term 'Newtonian gravity' rather than GR because the two give very similar results for well-understood mathematical reasons [8,9]. The unexpectedly high velocity dispersion of galaxies relative to each other in the Coma Cluster could be the first evidence for a breakdown of Newtonian gravity [10,11], and thus GR, or it could indicate missing mass—though a combination of the two should not be lightly discarded.

Closer to home, the observation that the Andromeda galaxy (M31) is approaching the Milky Way (MW) despite their presumed recession shortly after the Big Bang also argues for significant missing gravity on Mpc scales, a line of evidence known as the Local Group (LG) timing argument [12]. Moreover, the rotation curves (RCs) of galaxies do not follow the expected Keplerian decline beyond the extent of their luminous matter distributions ([13], and references therein). This led to the widespread acceptance that astronomical systems require significantly more gravity than can be explained using the conventional gravity of their detectable mass (for a historical review, see [14]). Since one possible explanation is the existence of large amounts of invisible mass, this problem is usually called the dark matter (DM) problem. However, DM is by no means the only possible explanation. Therefore, we recommend the terminology 'missing gravity problem', with the missing gravity arising from DM, a modification to GR in a regime where it has not been tested, or both.

Beyond the problem of predicting a Keplerian decline in the outer RC, self-gravitating Newtonian discs without DM quickly develop an instability, as predicted analytically [15] and verified by numerical simulations [16]. However, the MW retains a thin disc despite being much older than its dynamical time of a few hundred Myr [17]. For an early review of work on this problem, we refer the reader to [18]. A possible solution is that disc galaxies have a dominant pressure-supported spheroidal halo, even though such a halo is not observed [19]. The currently conventional solution to the above-mentioned issues is still to design invisible pressure-supported DM halos that surround, dominate, and stabilize galaxy discs [20,21].

Despite these favourable reasons to hypothesize DM, its nature remains a mystery. It was hoped for a time that the DM can be accounted for by known objects such as faint stars or brown dwarfs [22]. A model in which the dark halo consists of massive compact halo objects (MACHOs) predicts occasional gravitational bending of light from background stars.

The EROS collaboration [23] conducted a microlensing survey in which they continuously monitored $7 \times 10^6$ stars over a period of 6.7 years in two fields of view towards our satellite galaxies the Large and Small Magellanic Clouds. One expects $\approx 39$ MACHOs out of the many millions in this general direction to move by chance into a milli-arcsecond alignment with a bright background star, leading to an achromatic $\gtrsim 30\%$ increase and then roughly symmetric decrease in its brightness over a few months for a solar mass MACHO (the two images are by definition unresolved in microlensing [24,25]). Only one such candidate event was found [26]. This confirmed an earlier result [27] that MACHOs almost certainly do not have enough mass to account for the missing gravity in our galaxy, or to stabilize its disc if it obeys Newtonian dynamics (for a historical review of microlensing, we refer the reader to [28]). Similar conclusions were later reached for primordial black holes (PBHs) over a wide range of mass [29], with GW results also ruling out the idea that the required DM consists primarily of PBHs if their mass is $\gtrsim 0.1 \, M_\odot$ [30]. These considerations led to the widespread assumption that the DM consists of undiscovered non-baryonic particles outside the well-tested standard model of particle physics (for a historical review of how this paradigm came about, see [31]). The DM should be dynamically cold to allow the formation of galaxy-scale halos, leading to the idea of cold dark matter (CDM). While the CDM could consist of low-mass axions [32] or other particles with a low mass [33], we will generally assume that the CDM should be weakly interacting massive particles (WIMPs [34]). However, most of our discussion will not depend on this distinction, especially when we consider scales much larger than individual galaxies.

Additional assumptions are required to reconcile GR with extragalactic observations. A well-known hypothesis is that a very small but positive cosmological constant $\Lambda$ should be included in GR. While in principle the field equations of GR allow a constant dark energy density anywhere within 60 orders of magnitude for a unification with quantum mechanics at high energy, the universe would gain symmetry if both its curvature and $\Lambda$ were exactly zero. However, it was realised in the 1990s that the observed ages of some globular clusters seem to exceed the age of a flat universe with $\Lambda = 0$ [35,36]. Invoking a positive but very small $c^2\sqrt{\Lambda} \sim cH_0 \sim 10^{-9}$ m/s$^2$ causes the dark energy density to dominate the late universe and increases its age for a given Hubble expansion rate $H_0$. The higher age arises from an accelerated cosmic expansion at late times, which was confirmed by observations of distant Type Ia supernovae [37,38]. Though these do not all have the same intrinsic luminosity, they are standardizable candles using the Phillips relation between peak luminosity and light curve shape [39,40]. Combined with other lines of evidence, this led to the currently standard $\Lambda$CDM paradigm [41,42].

The $\Lambda$CDM paradigm went on to fit the power spectrum of anisotropies in the cosmic microwave background (CMB) as observed by the satellites known as the Wilkinson Microwave Anisotropy Probe (WMAP [43]) and Planck [44]. These reveal a highly characteristic pattern of oscillations that are generally understood as acoustic oscillations in the baryon-photon plasma during the first 380 kyr of the universe. Extrapolating back to its first few minutes, the background temperature should have been high enough for nuclear fusion reactions to take place [45]. This process of Big Bang nucleosynthesis (BBN) lasted for only a few minutes because of falling temperatures due to cosmic expansion, with the conversion of hydrogen to helium further limited by the fact that free neutrons decay with an exponential decay time of 880 s [46]. As a result, BBN is very sensitive to all four fundamental forces: gravity sets the overall expansion history and thus background temperature, the strong nuclear force sets when neutrons and protons first condense out as (relatively) stable particles, the weak nuclear force is important to the decay of free neutrons into protons, while electromagnetism is important especially during the deuterium bottleneck when photons disrupted newly formed deuterium nuclei, delaying the formation of heavier elements and thus reducing the primordial helium abundance. It is therefore highly non-trivial that the primordial abundances of deuterium and helium are well accounted for under the standard assumption of a hot Big Bang where the expansion history follows the GR-based Friedmann equation [47]. Even the abundance of lithium seems to be consistent

with this model [48,49], though there are still some difficulties reconciling the predicted primordial lithium abundance with measurements in the atmospheres of low-mass stars (as reviewed in chapter 6 of [50]). One of their arguments for why these measurements reflect the primordial abundance (i.e., for little stellar processing of lithium) is the claimed detection of Li-6, but precise spectra published since then argue against the detection of this isotope [51]. The main adjustable parameter in BBN is the baryon:photon ratio, which can be independently constrained using the CMB. For a review of the ΛCDM paradigm, we refer the reader to [52].

In this review, we focus primarily on more recently gathered evidence and later cosmological epochs, leading to a far less rosy picture for ΛCDM (serious problems with it are reviewed in [53–55]). Our main goal is to compare ΛCDM with an alternative paradigm known as Milgromian dynamics (MOND [56]) using available lines of evidence which are theoretically expected to be discriminative [57,58], either because the paradigms predict different outcomes, or because at least one paradigm makes a strong prediction. Other modified gravity theories and possible observational tests of them have been reviewed elsewhere, e.g., [59]—we briefly consider a few of these in Section 3.6. Empirically confirming a breakdown of GR could provide much-needed guidance for theorists trying to better understand gravity and the nature of spacetime.

This review is organised as follows: We begin by describing the theoretical background to MOND, which is incomplete like all theories but can still be applied to a wide range of systems (Section 2). We then describe available evidence from the equilibrium dynamics of galaxies (Section 3) and their stability and long-term evolution (Section 4). This mostly concerns isolated galaxies, so we go on to discuss interacting galaxies and satellite planes (Section 5). We then consider galaxy groups (Section 6) and galaxy clusters (Section 7). Taking advantage of some important recent studies, we go on to consider the large-scale structure of the universe (Section 8) and the possible cosmological picture in the MOND framework (Section 9). We then evaluate the relative merits of ΛCDM and MOND using a 2D scoring system based on the theoretical flexibility each has when confronted with a certain phenomenon and how well the prediction agrees with observations (Section 10). While this review mainly discusses currently available evidence, we also mention some potentially exciting future investigations that could break the degeneracy between ΛCDM and MOND (Section 11). We then obtain a numerical score for each model based on the presently available tests (Section 12). There we conclude with our thoughts on what these tests seem to be telling us.

## 2. Theoretical Background to MOND

MOND was proposed to explain galaxy RCs without postulating CDM particles for which there still exists no direct (non-gravitational) evidence [56]. At that time, the Tully–Fisher relation (TFR [60]) indicated that rotational velocities $v_c$ scale approximately as $\sqrt[4]{L}$, where $L$ is the luminosity. This was not known to a high degree of certainty, with [57] stating in their Section 3 that the slope could lie in the range 0.2–0.4 based on the data available at that time (for a historical review, see, e.g., [14,61]). A scaling of the form $v_c \propto \sqrt[4]{L}$ combined with the assumption of a roughly similar stellar mass-to-light $(M_\star/L)$ ratio in different galaxies implies that any modified gravity alternative to CDM should depart from standard dynamics at low accelerations rather than beyond a fixed distance $r_0$. This is because the gravity law must be $\propto 1/r$ in the modified regime to get flat RCs, so continuity with the Newtonian regime implies that the gravity in the modified regime must be $GM_s/(rr_0)$, where $G$ is the Newtonian gravitational constant and $M_s$ is the mass of ordinary particles within the standard model of particle physics, i.e. without assuming CDM. $M_s$ is usually called the baryonic mass in the literature as leptons are similar in number yet much less massive. We follow this terminology for clarity, but prefer the more general term standard mass $M_s$, by which we mean a non-gravitational determination, often using stellar population synthesis. A mass estimated from the dynamics in a particular theory is called the dynamical mass. Using a fixed $r_0$ gives an incorrect scaling between

galaxies of $v_c \propto \sqrt{M_s}$. If instead the modification arises only at low accelerations (see also Section 3.1.1), it is necessary to postulate that in an isolated spherically symmetric system with Newtonian gravity $g_N$, the actual gravity is

$$g = \begin{cases} \sqrt{g_N a_0} & (g_N \ll a_0), \\ g_N & (g_N \gg a_0). \end{cases} \tag{1}$$

Another major motivation for MOND is that Newtonian gravity has not been tested to any acceptable precision in galaxies and galaxy clusters, where the dynamical discrepancies arise. Even our most distant spacecraft (Voyager 2) is at a distance of $r = 130$ AU, where the gravitational field of the Sun is $g_\odot = GM_\odot/r^2 = 3.5 \times 10^{-7}$ m/s$^2$, with $M_\odot$ being the solar mass. $g_\odot$ on Voyager 2 is much larger than the $(2.32 \pm 0.16) \times 10^{-10}$ m/s$^2$ acceleration of the solar system as a whole relative to distant quasars [1], with the dominant contribution no doubt arising from its galactocentric orbit given that the acceleration is almost exactly towards the galactic centre (see their Figure 10). Consequently, it is possible that Newtonian dynamics breaks down when the gravitational field $g$ drops three orders of magnitude below the solar gravity on Voyager 2. This might explain the observed flat RCs of disc galaxies and the similar problem in pressure-supported galaxies, which typically have a much higher internal velocity dispersion $\sigma_i$ than the Newtonian expectation without CDM.

The successes and limitations of both Newtonian gravity and MOND are similar—both lack a unique consistent description for photons and cosmic expansion, but do predict the right order of magnitude in both cases. The MOND theory has evolved somewhat over the years with various relativistic extensions (discussed later in Section 2.6), but its relation to GR and DM is not completely understood (for a historical review, see [61]). MOND necessarily entails deviations from a baryonic GR universe, though in a regime where GR has not been directly tested [62]. MOND is easier thought of as a generalization of Newtonian dynamics rather than GR. Its most user-friendly modern form (called quasi-linear MOND or QUMOND [63]) is often presented as a non-relativistic substitute for Newtonian gravity in galaxies, with the extra complexity offset somewhat by the absence of CDM in this picture. To minimize the adjustments to existing algorithms, the total gravitational potential at position $r$ can be expressed as

$$\Phi(r) = -\int \frac{G\rho_{\text{eff}}(x)d^3x}{|x - r|}, \tag{2}$$

where the effective density $\rho_{\text{eff}}$ at position $x$ can be decomposed into standard ($\rho_s$) and phantom ($\rho_p$) parts with total

$$\rho_{\text{eff}} \equiv \rho_s + \rho_p = -\nabla \cdot \left( \frac{\nu g_N}{4\pi G} \right). \tag{3}$$

The MOND interpolating function $\nu(y \equiv g_N/a_0)$ has the asymptotic limits

$$\nu = \begin{cases} 1 & (y \gg 1), \\ \frac{1}{\sqrt{y}} & (y \ll 1). \end{cases} \tag{4}$$

The mass density $\rho_s$ consists of only standard particles (baryons, photons, neutrinos, etc.) that source the Newtonian gravity $g_N(x)$. This must first be calculated in order to determine $\rho_{\text{eff}}$, which in general exceeds $\rho_s$. The remainder $\rho_p$ is the density of 'phantom dark matter' (PDM). For an isolated spherically symmetric system, the upshot is that the position $r$ of a test particle satisfies the equation of motion

$$|\ddot{r}| = g \equiv |\nabla\Phi| \approx \max\left[ \frac{GM}{r^2}, \sqrt{\frac{GMa_0}{r^2}} \right], \tag{5}$$

where the acceleration is radially inwards, $M$ is the mass interior to the particle, an overdot indicates a time derivative, and $r \equiv |r|$ for any vector $r$ in what follows. Note that this equation is only an approximation even in spherical symmetry—one should instead use a smooth interpolating function between the different asymptotic limits (see Section 2.3), yielding $g = \nu g_N$. The phantom density $\rho_p$ is generally positive, but it can be negative in more complicated geometries [64,65], potentially leading to a clear smoking gun signal for a breakdown of GR [66].

The value of $a_0$ can be estimated from a single RC, as first done by [67] for 16 galaxies. The author of [68] made some improvements to this, especially by including the gas mass and removing one galaxy with a significantly asymmetric RC between the approaching and receding sides. The median inferred $a_0$ from the 15 considered galaxies was $1.3 \times 10^{-10}$ m/s$^2$, which was shown to fit all 15 RCs quite well (see its Figure 1). A better estimate of $a_0$ can be obtained by jointly fitting the RCs of several galaxies with a single adjustable parameter $a_0$ that is consistent between galaxies. Only a handful of RCs are required to empirically determine that the common acceleration scale $a_0 = 1.2 \times 10^{-10}$ m/s$^2$ [69]. More recent studies confirm this value [70,71], which has therefore not changed for many decades.

An excellent review of MOND fits to galaxy RCs can be found in [72], which also considers other aspects of MOND. The older review of [73] contains useful discussions on cosmology and large-scale structure, mainly based on the work of [74]. A fairly thorough mostly theoretical review of MOND is provided by [75], with the author keeping this up to date. The philosophy of MOND is detailed in [50], which argues convincingly that a good scientific theory must make novel falsifiable predictions, or at least explain observations that it was not designed for. This excellent book considers published results until the end of 2017, while our review will also consider several crucial results obtained since then.

*2.1. Spacetime Scale Invariance*

Though empirically motivated, Equation (1) can be derived from a symmetry principle known as spacetime scale invariance if we also assume that $g$ is a function only of $g_N$ [76]. Suppose that a particle at very low acceleration follows some trajectory through spacetime defined by a list of $(r, t)$, where $r$ is the position and $t$ is the time. For any constant $\lambda$, another solution to the equations of motion in the low-acceleration (deep-MOND) regime is $(\lambda r, \lambda t)$. In other words, for any solution to the deep-MOND equations of motion, we may obtain another solution by performing the following scaling:

$$r \;\rightarrow\; \lambda r \,, \tag{6}$$

$$t \;\rightarrow\; \lambda t \,, \tag{7}$$

$$GMa_0 \;\rightarrow\; GMa_0 \,. \tag{8}$$

At the appropriately adjusted time, the scaled trajectory has the same velocity as the original. Consequently, if we take the case of uniform circular motion around a point mass, we see that the RC must be flat in order to satisfy spacetime scale invariance. One potentially important aspect of the above scalings is that since they do not alter the velocity, the introduction of a fundamental velocity scale $c$ preserves spacetime scale invariance.

In contrast, standard Newtonian mechanics does not obey spacetime scale invariance because the RC of a point mass follows a Keplerian $r^{-1/2}$ decline. Therefore, a revised trajectory can be obtained from the original only if $r$ and $t$ are scaled by different factors at high accelerations. The required scalings are instead:

$$r \;\rightarrow\; \lambda^{2/3} r \,, \tag{9}$$

$$t \;\rightarrow\; \lambda t \,, \tag{10}$$

$$GM \;\rightarrow\; GM \,. \tag{11}$$

Written in this way, the Newtonian scalings seem much less natural than those of the deep-MOND limit, partly because the Newtonian scalings must be broken at some point if we introduce a fundamental velocity scale.

The special symmetries of the deep-MOND limit may be related to cosmology and the presence of dark energy, which we discuss next. Note, however, that the deep-MOND limit and any symmetries of this limit are not an accurate representation of MOND because the deep-MOND limit is only a mathematical approximation—real-world accelerations are always a finite fraction of $a_0$. Moreover, theoretical elegance is a less important consideration than agreement with observations—the latter have always taken precedence in the whole Milgromian research program (see also [77]).

### 2.2. Possible Fundamental Basis

A more widely known theoretical justification for MOND is based on the empirical value of $a_0 = 1.2 \times 10^{-10}$ m/s$^2$. This very low value is similar to various acceleration scales of cosmological significance, including that defined by the inverse timescale that is the Hubble constant $H_0$. A particle accelerating at $a_0$ for a Hubble time would reach a speed

$$\frac{a_0}{H_0} \approx 0.18\,c\,. \tag{12}$$

The order of magnitude coincidence with the speed of light hints at a fundamental link between the empirical value of $a_0$ determined from galaxy RCs and the typical acceleration $cH_0$ of the cosmic horizon. This coincidence was known since MOND was first proposed (see the introduction to [56]).

Perhaps more physically relevant is the gravity energy density scale $g_\Lambda$ determined by the dark energy density $\rho_\Lambda c^2$ that conventionally explains the accelerated late-time expansion of the universe (Section 1). This scale is defined by where the classical energy density $g^2/(8\pi G)$ of a gravitational field (Equation 9 of [78]) becomes comparable to $\rho_\Lambda c^2 = \frac{c^4 \Lambda}{8\pi G} \approx \frac{2.25\,c^2 H_0{}^2}{8\pi G}$ if we assume that the dark energy density parameter is presently $\Omega_\Lambda \approx 0.75$ [79,80]. Assuming also that $H_0 = 69$ km/s/Mpc, equality occurs when $g = g_\Lambda$, where

$$g_\Lambda = c^2 \sqrt{\Lambda} = 1.5\,cH_0 = 8.4\,a_0\,. \tag{13}$$

The possible physical meaning of this coincidence was discussed by [81] and several subsequent works (e.g., [82–84]). If the gravitational field is extremely weak, then the dominant contribution to the energy density is $\rho_\Lambda c^2$, which is often identified with the quantum-mechanical zero point energy density of the vacuum. If this interpretation is correct, poorly understood quantum gravity effects could become rather important. Since quantum mechanics is totally neglected by GR, it may well be that currently not understood quantum corrections to GR cause its failure when $g \lesssim a_0$. MOND might then be an empirical way to include such quantum effects, much like the empirical formula for the blackbody spectrum was obtained long before quantum mechanics was developed.

In the $\Lambda$CDM context, the energy density of dark energy is presently comparable to that of baryons and of CDM. Equation (13) could then be interpreted as a coincidence with the acceleration scale defined by these densities. However, since they vary with the cosmic scale factor $a$ or equivalently with the redshift $z \equiv a^{-1} - 1$, the result would be that $a_0 \propto a^{-3/2}$, which is in tension with high-$z$ RCs [85] and several other lines of evidence (Section 9.1). It is also unclear why the behaviour of gravity might depart from classical expectations when the energy density of the gravitational field falls below the average density of baryons or of CDM, since these substances are thought to be significantly clustered. Therefore, we argue that the important threshold for the behaviour of gravity is that set by the dark energy density.

While the 'cosmic coincidence' of MOND may not be very compelling due to the agreement being only at the order of magnitude level even with arbitrary factors of $2\pi$

(Equation (13)), the above discussion raises the important question of whether the ultimately correct quantum theory of gravity will be numerically the same as GR in all regimes of astrophysical interest. Since the new theory that will ultimately replace GR is by definition different in order to include quantum effects where GR does not, exact numerical agreement with GR is obviously impossible. Astronomers generally make the assumption that any quantum corrections to GR will be very small in any system relevant to their work. However, there is currently no evidence that this is true. It is simply a working assumption, since if quantum corrections were important, it is not presently known how to include them. This assumption can be justified by appealing to the large size of astronomical systems such as galaxies compared to, e.g., the size of an atom, which is recognisably dominated by quantum effects. However, even rather large systems can be affected by quantum mechanics. The nuclear fusion reactions in the cores of stars occur over a rather large region by atomic standards and create radiation which affects the rest of the universe in often substantial ways. Similarly, it is easy to have a macroscopically large superconductor which would not function without quantum effects involving Cooper pairs. Therefore, the large size of a system does not guarantee that it can safely be treated classically, even if one is primarily interested in its gravitational dynamics [86,87]. The size of a superconductor is less relevant than its low temperature, which in a gravitational context corresponds to a weak field. Indeed, several studies argue for a fairly direct link between MOND and known low temperature quantum deviations from classical thermodynamics [82,88,89]. In this context, it is worth keeping an open mind to the possible breakdown of GR in spacetime regions where $g \lesssim g_\Lambda$ as defined in Equation (13).

### 2.3. Non-Relativistic Theories

There are a few similar but not equivalent ways to present MOND. The original idea behind it was that in spherically symmetric systems, the relation between $g$ and $g_N$ is such that $g = g_N$ at high accelerations but $g = \sqrt{g_N a_0}$ at low accelerations relative to some threshold $a_0$. To interpolate between these extreme cases, it is customary to write that $\mu g = g_N$ or $g = \nu g_N$, where $\mu$ is a smooth function whose so-called 'simple' form is

$$\mu(x) = \frac{x}{1+x}, \quad x \equiv \frac{g}{a_0}. \tag{14}$$

It is the spherical counterpart of the transition function

$$\nu(y) = \frac{1}{2} + \sqrt{\frac{1}{4} + \frac{1}{y}}, \quad y \equiv \frac{g_N}{a_0}. \tag{15}$$

This can be written in an alternative form that highlights the excess over the Newtonian case ($\nu = 1$).

$$\nu(y) \equiv 1 + \widetilde{\nu}(y) = 1 + \left[\frac{y}{2} + \sqrt{\frac{y^2}{4} + y}\right]^{-1}. \tag{16}$$

The definition using $\mu(x)$ is prevalent in older papers on MOND such as the one that introduced this function [90], but we generally use the more computer-friendly definition involving $\nu(y)$ in which $g = \nu g_N$. In what follows, the MOND interpolating function is used to mean $\nu(y)$. The functions are related by $\mu\nu = 1$ in spherical symmetry.

To handle systems which lack spherical symmetry, the authors of [91] came up with the aquadratic Lagrangian (AQUAL) formulation of MOND. They used a non-relativistic Lagrangian, thus guaranteeing the usual symmetries and conservation laws associated with linear and angular momentum and the energy. By retaining a standard kinetic term, standard inertia is retained in AQUAL, so it is a modified theory of gravity (modified inertia theories are briefly discussed in Section 2.5). The main result of AQUAL is that

the spherically symmetric relation $\mu g = g_N$ can be generalized by equating instead each side's divergence.

$$\nabla \cdot (\mu g) = \overbrace{-4\pi G\rho_s}^{\nabla \cdot g_N} .  \tag{17}$$

This retains the same results in spherical symmetry, but allows calculations in more complicated geometries. Far from an isolated matter distribution, we recover the result of Equation (1) that the gravitational field is radially inwards with magnitude

$$g = \frac{\sqrt{GMa_0}}{r}, \quad r \gg \overbrace{\sqrt{\frac{GM}{a_0}}}^{r_M},  \tag{18}$$

where $r$ is the distance from the barycentre of the system with total mass $M$ and MOND radius $r_M$, beyond which MOND effects become significant. Because $g \propto 1/r$, the potential diverges logarithmically with distance. For the simple interpolating function (Equation (15)), the isolated point mass potential was derived in equation 52 of [92].

$$\Phi = \sqrt{GMa_0}\left(\ln u - \frac{u}{\tilde{r}}\right), \, u \equiv 1 + \sqrt{1+\tilde{r}^2}, \, \tilde{r} \equiv \frac{2r}{r_M}.  \tag{19}$$

This yields the expected Newtonian behaviour $-GM/r$ for $r \ll r_M$. The logarithmic divergence beyond $r_M$ is not in general predicted in MOND because ultimately one must take into account other masses, breaking the assumption of isolation (Section 2.4).

There is also a lesser-known AQUAL-like theory which allows calculations of structure formation in a cosmological context [74]. This is reviewed further in the cosmology section of [73]. The coupling constant $\beta$ is usually set to 0 for simplicity, as discussed further in Section 5.2.3 of [93]. The cosmological context of MOND is discussed further in Section 9. Cosmological MOND simulations usually adopt the approach first outlined in [94], in which only the departures of the density from the cosmic mean enter into the gravitational field equation.

Compared to AQUAL with its non-linear Poisson equation, a more computer-friendly approach is provided by QUMOND, a modification to Newtonian gravity which starts from the approach that in spherical symmetry, $g = \nu g_N$. Similarly to AQUAL, this is generalized to less symmetric systems by instead equating the divergence of each side. This yields the QUMOND field equation

$$\nabla \cdot \overbrace{(-\nabla\Phi)}^{g} = \nabla \cdot (\nu g_N),  \tag{20}$$

which is an alternative form of Equations (2) and (3). It can also be obtained by applying the Euler—Lagrange equation with respect to variations of the auxiliary field $g_N$ if we use the non-relativistic action $S = \int \mathcal{L}\, d^3x\, dt$ with Lagrangian density

$$\mathcal{L} = \rho_s\Phi + \frac{g_N{}^2 - 2g_N \cdot \nabla\Phi + \int_{g_N{}^2}^{\infty} \tilde{\nu}(y)d\left(a_0{}^2y^2\right)}{8\pi G} .  \tag{21}$$

The use of an action principle ensures that QUMOND obeys the usual symmetries and conservation laws [63]. QUMOND and AQUAL can be derived from each other using a Legendre transform [95].

In the following, we try to give a physical meaning to the interpolating function $\nu$ based on the neutrino model of [96]. The massive neutrinos could be important to a viable

cosmological MOND framework (Section 9). We begin by defining the shifted Newtonian and true potentials and a rescaled acceleration parameter $a_U$.

$$N \equiv \Phi_N - \frac{C^2}{2}, \quad U \equiv \Phi - \frac{C^2}{2}, \quad \frac{a_U}{a_0} \equiv \sqrt{\frac{-2U}{C^2}}. \tag{22}$$

Using these definitions, we can rewrite the QUMOND action as

$$S \approx \int \left( \rho_N + \frac{C^2}{2N} \cdot \frac{\nabla^2 N}{4\pi G} \right) U d^3 \boldsymbol{r} \, d\eta, \tag{23}$$

where the variable $C$ is of order the light speed $c$ and is a spatially uniform function of $\eta$ (the conformal time), $\nabla^2 \rightarrow \left( \nabla^2 - \partial_\eta^2 \right) / a^2(\eta)$, $a$ is the cosmic scale factor, $S \rightarrow \int d^3 \boldsymbol{r} \, d\eta \, U \left[ \left( \rho - \frac{3P}{c^2} \right) + \frac{K}{16\pi G} \right]$ for $\frac{U}{a(\eta)^2} = \left( 1 - \frac{2\Phi}{c^2} \right) a(\eta)^2 = g_{33} = \frac{-2N}{c^2}$, $K = g^{ab} \partial_a g_{33} \partial_b g_{33} = \frac{1}{g_{33}} \left[ (\nabla g_{33})^2 - \left( \frac{g_{33}}{a^2} \partial_\eta g_{33} \right)^2 \right]$, while $U$ acts as a Lagrange multiplier such that varying the action $S$ with respect to $U$ gives the usual Newtonian Poisson equation sourced by the non-relativistic neutrino gas.

$$\rho_N = \frac{-C^2}{2N} \cdot \frac{\Box N}{4\pi G} \approx \frac{\nabla^2 N}{4\pi G}, \tag{24}$$

where $\Box$ denotes the d'Alembertian operator. The fermion gas Lagrangian density can be expressed as

$$-U\rho_N = \int_0^\infty d \left[ \frac{4\pi}{3} \left( \frac{\lambda_U}{\lambda} \right)^3 \right] u F(X), \tag{25}$$

$$F(X) = \frac{2}{1 + \exp \left[ \sqrt{ \left( \frac{\lambda}{\lambda_U} \right)^{-2} + \left( \cancel{\frac{mc^2}{kT_d}} \right)^2 } - \cancel{\frac{\mu}{kT_d}} \right]}, \tag{26}$$

$$(2\pi)^3 u = \left( \frac{a_U}{a_0} \right)^2 \cdot \frac{\overbrace{\left( \frac{\approx cH_0}{2\pi a_0} \right)^2}}{4\pi G}, \tag{27}$$

where $k$ is the Boltzmann constant, $\mu$ is the chemical potential, and $T_d$ is the decoupling temperature, which defines an energy $kT_d \sim 1$ MeV. This is orders of magnitude greater than the neutrino rest energy scale $mc^2$ and their momentum spread $2\pi\hbar/\lambda_U$. The crossed out term involving $m$ has been neglected because the neutrinos would be ultra-relativistic. This also implies that prior to decoupling, they were tightly coupled to photons with zero chemical potential, justifying our neglect of the cancelled out term involving $\mu$. The function $F(X)$ is reminiscent of the relativistic Fermi–Dirac (FD) distribution of neutrinos with a rescaled de Broglie wavelength $X^{-1/2} \equiv \lambda/\lambda_U$ (c.f., [96]). The above integration further reduces to

$$-U\rho_N = \int_{\frac{1}{\lambda} \geq \left| \frac{\nabla N}{a_U \lambda_U} \right|}^{\frac{1}{\lambda} < \frac{mc}{2\pi\hbar}} d \left[ \frac{(Xa_U)^2}{8\pi G} \right] \overbrace{\frac{1}{\sqrt{X}} \cdot \frac{2}{1 + e^{\sqrt{X}}}}^{\tilde{\nu}(X)}, \tag{28}$$

where we truncate the density of quantum mechanical energy $u$ within one typical wavelength $\lambda_U$ so that [96]

$$
\frac{E_U}{\left(\frac{\lambda_U}{2\pi}\right)^3} = 
\begin{cases}
0, \text{ if } \frac{\lambda_U}{\lambda} \equiv x < \sqrt{y} \equiv \sqrt{\left|\frac{\nabla N}{a_U}\right|}, \\[6pt]
0, \text{ if } \frac{2\pi\hbar}{\lambda} > mc, \\[6pt]
\left(\frac{a_U}{a_0}\right)^2 \cdot \dfrac{\overbrace{\left(\frac{\overbrace{2\pi a_0}^{\approx cH_0}}{}\right)^2}}{4\pi G}, \text{ otherwise.}
\end{cases}
\tag{29}
$$

The scale $\frac{c^2 H_0{}^2}{4\pi G} \sim \frac{11\,\text{eV}}{(1.4\,\text{mm})^3}$ is of order the critical density of the universe with Hubble constant $H_0 \approx 2\pi a_0/c$. Here, the energy density is tapered if the momentum $x\left(\frac{2\pi\hbar}{\lambda_U}\right)$ exceeds a threshold set by a small mass scale $m$ or if the neutrino is so slow that the work done by gravity $|\nabla N|$ over the length scale $c^2/a_U$ can heat it up. The typical energy $E_U \sim 0.06\,\text{eV}$, consistent with ordinary neutrino rest energy differences if using the present-day wavenumber $2\pi/\lambda_U \sim 2\pi/(1.4\,\text{mm})$ at the standard 1.9 K background temperature of massless neutrinos. Neutrinos with a mass of $0.06\,\text{eV}/c^2$ would today be non-relativistic in a homogeneous universe, while more massive neutrinos would be even more so.

Varying $4\pi GS$ with respect to $\nabla N$ gives

$$
\nabla \cdot [\widetilde{\nu}(y)\nabla N] = \nabla \cdot \left[\frac{\nabla N}{\sqrt{y}}\overbrace{\frac{2}{1+e^{\sqrt{y}}}}^{F(y)}\right] = \nabla^2 \left[\overbrace{\frac{U}{N} \cdot \frac{C^2}{2}}^{\Phi - \Phi_N}\right],
\tag{30}
$$

where $y = |\nabla N|/a_U$ as in the extended version of QUMOND described in [97] and $a_U{}^2 \equiv a_0{}^2\left(1 - \frac{2\Phi}{C^2}\right)$ is much bigger in regions with a deeper geodesic potential $U \ll \Phi < 0$, especially if $C \sim 0.01\,c \sim \sqrt{-2\Phi}$ as in galaxy clusters.

The most robust feature of FD neutrino-inspired models is that they naturally result in an interpolating function $\nu(y)$ which behaves as

$$
\nu(y) - 1 \equiv \widetilde{\nu}(y) \equiv \frac{F(y)}{\sqrt{y}} \rightarrow \left.\frac{1}{\sinh\sqrt{y}}\right|_{y\to 0}^{y\to\infty},
\tag{31}
$$

so the asymptotic behaviour is $1/\sinh\sqrt{y}$ in both the deep-MOND and Newtonian regimes. This leads to an *extremely fast* transition from $y^{-1/2}$ in the former to 1 in the latter (cf. Equation (30)), so the MOND corrections rapidly become very small in the Newtonian regime, helping to explain the null detection of MOND effects in solar system observations (Section 11.4). The spacetime scale invariance of the deep-MOND limit (Section 2.1) is reproduced by Equation (28) because the density of neutrinos is $\propto d(\lambda^{-3}) \propto d(x^3) \propto \frac{d(y^2)}{\sqrt{y}}$.

By taking the limit $y \to g_N \to 0$ such that $\Phi \to 0$ and $a_U/a_0 \equiv \sqrt{-2U/C^2} \to 1$, these FD-inspired models predict that a homogeneous universe has a positive dark energy density

$$
\mathcal{L} \to \int_0^\infty [\widetilde{\nu}(y)]\,d\left(\frac{a_U{}^2 y^2}{8\pi G}\right) = \left.\frac{g_\Lambda{}^2}{8\pi G}\right|_{g_\Lambda \approx 2.7 a_U}.
\tag{32}
$$

This is fairly close to the observed value $\sim \frac{(cH_0)^2}{8\pi G}$, which is the cosmic coincidence of MOND discussed in previous works [98,99] and in Section 2.2.

Numerical Solvers

The main advantage of the QUMOND approach is that standard techniques can be used to obtain $g_N$ and thus $\nu$ at all locations on a grid, which in turn allows calculation of the source term for the gravitational field, namely $\nabla \cdot (\nu g_N)$. We can think of this as an effective density $-4\pi G \rho_{\text{eff}}$ because the Newtonian gravity sourced by $\rho_{\text{eff}}$ is the same as the QUMOND gravity $g$ due to $\rho_s$ alone. Note that $\rho_{\text{eff}}$ is not a real density but a mathematical quantity which can be negative over rather large regions [65,66]. Thinking of QUMOND in this way can help, not least because it allows the use of standard codes where $\rho_{\text{eff}}$ is fed in as the density distribution in a second step after first obtaining $g_N$ from $\rho$ alone. Moreover, $\rho_{\text{eff}}$ is how much DM would be inferred in a Newtonian analysis of a system actually governed by QUMOND, which can help relate the results to studies in a conventional gravity context. Remembering to subtract the physical density $\rho_s$ as in Equation (3), this 'phantom' density is

$$\rho_p \equiv \rho_{\text{eff}} - \rho_s \,. \tag{33}$$

The older AQUAL and more computer-friendly QUMOND formulations of MOND are expected to give quite similar results [100] if one adopts the same interpolating function ≡ relation between $g$ and $g_N$ in spherical symmetry. The formulations differ little even in situations that are not spherically symmetric, as can be demonstrated analytically for a point mass embedded in a dominant external field [65] and the condition for local stability of a thin disc [101]. Due to the lower computational cost, QUMOND is often used to investigate MOND, especially using the publicly available algorithm called PHANTOM OF RAMSES (POR) developed by [102]. This is based on a modification to the gravity solver in standard RAMSES, a widely used grid-based *N*-body and hydrodynamical solver for astrophysics problems [103]. The POR algorithm is quite versatile, partly because it inherits advanced features of standard RAMSES such as parallel computing, adaptive mesh refinement (AMR), and a careful treatment of baryons. A similar algorithm called RAYMOND has been developed to handle both the AQUAL and QUMOND formulations of MOND [104], also by adapting RAMSES. For a user manual on how to conduct *N*-body and hydrodynamical simulations in QUMOND with POR, we refer the reader to [105], which briefly reviews all numerical MOND simulations and codes prior to its publication. It also explains how to initialize isolated and interacting disc galaxy models (the disc templates are discussed further in [106]).

Equation (1) implies that the gravitational field is a non-linear function of the mass distribution. This is required to fit the RCs of galaxies with different $M_s$ [56]. For these observations to be explained without CDM particles, it must be the case that the change to the gravitational field generated by a point mass depends on the already existing matter distribution. For instance, if an isolated mass $M$ at position $A$ causes some gravity $g$ at location $B$, then another mass at $A$ would increase the gravity at $B$ by only $0.41\,g$ in order to preserve that $g \propto \sqrt{M}$, which is the required deep-MOND behaviour. In situations with a low degree of symmetry, the non-linearity should be handled using Equations (17) or (20).

Unlike in Newtonian gravity, it is not possible to write down a general kernel such that integrating the mass distribution with respect to this kernel yields the MOND gravitational field. Algorithms to solve Newtonian gravity often exploit its linearity by superposing the gravitational fields from different masses—this is usually hard-wired into algorithms based on smoothed particle hydrodynamics (SPH). It is not possible to MONDify such codes, i.e., they cannot be straightforwardly generalized to MOND gravity. Only grid-based Newtonian codes can be generalized as they already involve relaxing the Poisson equation on a grid. In MOND, the Poisson equation can be modified (Equation (17)) or a two-step approach adopted (Equation (20)). Such techniques are necessary even for a problem with very few masses. It is likely that these computational difficulties slowed down the pace of MOND research in its early years.

### 2.4. The External Field Effect (EFE)

In addition to causing numerical difficulties, the non-linearity of MOND implies that the internal dynamics of a system is affected by the gravitational field experienced by the system as a whole, *even if tides can be neglected*. This effect is called the EFE. It preserves the weak equivalence principle because the gravitational dynamics of a test particle is still unaffected by its mass or internal composition. However, the EFE violates the strong equivalence principle because a local measurement internal to a system can be used to deduce its external acceleration $g_e$. In particular, a Cavendish-style active gravitational experiment would yield different results on an elevator freely falling on Earth compared to a spacecraft far from any mass. Since the latter version of the Cavendish experiment has never been conducted, this is entirely possible. If confirmed, this would be a direct violation of the strong equivalence principle at the heart of GR.

Ideally, it would not be necessary to include the EFE in a MOND simulation because its domain can in principle be extended to include the source of the EFE. In practice, this can be very computationally expensive, so the EFE is still a useful concept. This raises the issue of which frame should be used to quantify $g_N$, as this has implications for the value of $\nu$. The most logical interpretation is that the acceleration should be measured relative to the average matter content of the universe, which is similar to Mach's principle. For practical purposes, one can measure $g_N$ with respect to the CMB frame. The EFE on a system can be estimated from its peculiar velocity in this frame [92,107].

The EFE has always been part and parcel of MOND [56]. Their Section 3 mentions that in addition to the above theoretical reasoning, the velocity dispersions of the Pleiades [108] and Praesepe [109] open star clusters are lower than would be expected in MOND if one neglects the galactic EFE on these clusters. The historical importance of such constraints to the formulation of MOND is questionable because it appears very difficult to satisfy Equation (5) in any linear gravitational theory. This empirically motivated equation requires a theory that is non-linear with respect to at least the internal dynamics. However, since one can arbitrarily choose what is internal to a system and what is external by moving the adopted border, the non-linear gravitational behaviour of the larger system will appear on smaller scales as a dependence on the external field. We therefore argue that the EFE should be considered a fundamental prediction of MOND not motivated by any observations beyond the RC data used to motivate MOND in the first place. Observational tests of the EFE are discussed in Section 3.3 based on plausible assumptions for where one should place the border around a system.

Solutions to the MOND field equations including the EFE were discussed in [110]. The simplest case is a point mass in a dominant external field, which was considered by [65] for both AQUAL and QUMOND. The results are numerically quite similar in both formulations, even though the problem is not spherically symmetric. Their QUMOND result for the increment to the gravitational potential caused by a point mass $M$ in a dominant external field $g_e$ is

$$\Phi = -\frac{GM\nu_e}{r}\left(1 + \frac{K_e \sin^2\theta}{2}\right), \tag{34}$$

where $r$ is the position relative to the mass, $\nu_e \equiv \nu\left(g_{N,e}\right)$ is set by the background Newtonian gravity $g_{N,e}$ alone, and $\theta$ is the angle between $r$ and $g_{N,e}$. The rescaled quadrupole

$$K(g_N) \equiv \frac{\partial\nu}{\partial g_N} \div \frac{\nu}{g_N}, \tag{35}$$

with $K_e \equiv K\left(g_{N,e}\right)$ being the logarithmic derivative of $\nu$ with respect to its argument in the limit of EFE dominance. $K$ transitions between 0 in the Newtonian limit and $-1/2$ in the deep-MOND limit (Equation (4)), with typically $K \approx -1/3$ in the transition zone ($K \approx -0.26$ in the solar neighbourhood where $g_{N,e} = 1.2\,a_0$; see Section 3.6 of [111]).

Since QUMOND relies on knowing $\boldsymbol{g}_N$, it is logical that the EFE would depend on the background value of $\boldsymbol{g}_N$ in the absence of the mass under consideration, which we denote $\boldsymbol{g}_{N,e}$. As the EFE often comes from a distant point mass, it is possible to approximate that the actual external field

$$\boldsymbol{g}_e \; = \; \nu_e \boldsymbol{g}_{N,e} \, . \tag{36}$$

In other words, one can apply the spherically symmetric relation between $\boldsymbol{g}$ and $\boldsymbol{g}_N$ for the external field. While this is not entirely correct for more complicated geometries, we have found that it is usually accurate enough.

The azimuthally averaged strength of gravity in the radially inward direction is

$$\overline{g}_r \; = \; -\frac{GM\nu_e}{r^2}\left(1 + \frac{K_e}{3}\right). \tag{37}$$

Using numerical simulations detailed in [111], this was generalized to situations where $g_{N,e}$ is comparable to the internal Newtonian gravity $g_{N,i} \equiv GM/r^2$ assuming the simple interpolating function (Equation (15)). The fitting function obtained by [112] in their equation 23 is

$$\overline{g}_r \; = \; \nu g_{N,i}\left[1 + \frac{K}{3}\tanh^{3.7}\left(\frac{0.825\, g_{N,e}}{g_{N,i}}\right)\right], \tag{38}$$

where $\nu$ and $K$ must be calculated with argument $g_{N,t} \equiv \sqrt{g_{N,e}{}^2 + g_{N,i}{}^2}$, the estimated total $g_N$ in the problem. This semi-analytic fit is plotted in Figure 1, illustrating how the EFE weakens the point mass gravity below the isolated MOND case. This is because the $\nu$ function becomes 'saturated' at $\approx 1$ if the external field is sufficiently strong, making the problem effectively Newtonian. In general, a dominant EFE saturates the $\nu$ function at $\nu_e > 1$, leading to an inverse square gravity law.

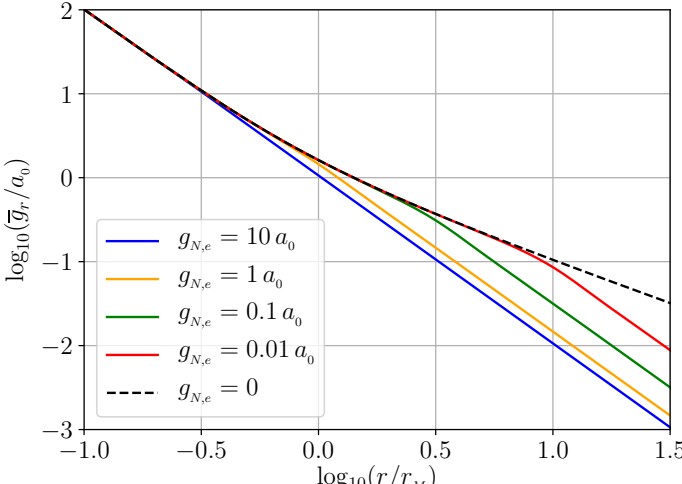

**Figure 1.** The azimuthally averaged inward radial gravity from a point mass as a function of distance from it, calculated with Equation (38) for different external field strengths (solid lines) and shown on logarithmic axes. The gravity is shown in units of $a_0$, while distances are in units of the MOND radius (Equation (18)). The dashed line shows the result without the EFE (Equation (15)). The result for the strongest considered EFE (solid blue line) is almost the same as the Newtonian prediction (not shown). Notice the Newtonian regime at low radii in all cases. At large radii, the predicted MOND boost depends on the EFE, but is limited to the boost in the isolated case (dashed line) of $r/r_M$. Credit: Elena Asencio.

Since the EFE breaks spherical symmetry, the point mass potential becomes angle-dependent (it is generally deeper on the axis defined by the external field; see Equation (34) and note $K < 0$). As a result, the gravity from a point mass is not in general directly towards it. The tangential component of the gravity can be handled in the deep-MOND limit using the fit to numerical results in appendix A of [92], though with the "$-$" sign changed to "$+$" in their equation A2 as discussed in Section 4.4 of [106]. Their Figure 27 clarifies that if $g_e$ is at right angles to the position of a test particle relative to a central mass, then the test particle also feels some gravity opposite to $g_e$ in addition to the usual radially inward component, as is clearly evident from the POR simulation results in their Figure 28. This means that the potential is asymmetric with respect to $\pm g_e$, which may lead to asymmetric tidal streams (Section 5.2) and cause galaxy discs to warp [106,113].

An important consequence of the EFE is that the gravitational field of a point mass asymptotically follows an inverse square law (Equation (34)), as also occurs conventionally. The normalization is different and there is also an angular dependence, but as there are clear similarities to the Newtonian regime, this limit is known as the quasi-Newtonian regime. Neglecting factors of order unity, the potential in the EFE-dominated case is

$$\Phi \approx -\frac{GM\nu_e}{r},\tag{39}$$

yielding similar behaviour to a Newtonian system where the mass is scaled up by $\nu_e$. The existence of this quasi-Newtonian regime leads to a finite potential depth even in MOND because every object experiences some gravity from other objects. While this is also true in Newtonian gravity, the longer-range nature of MOND gravity and its attendant EFE cause a system to be affected by much more distant objects than would be the case conventionally. For example, if we are considering a system and there is a point mass some distance $R$ away, its importance declines as $1/R^3$ conventionally because only the tidal effect is relevant to the internal dynamics. In MOND, the importance of the perturber would decline as $1/R$.

The Two-Body Force Law

Another manifestation of the EFE is in the gravity between two comparable point masses $A$ and $B$. The gravity from $B$ reduces the density of the PDM halo around $A$, and vice versa. The mutual gravity $g_{\mathrm{rel}}$ between two otherwise isolated point masses in the deep-MOND limit is [114]

$$g_{\mathrm{rel}} = \frac{Q\sqrt{GMa_0}}{r}, \quad Q = \frac{2\left(1 - q_A{}^{3/2} - q_B{}^{3/2}\right)}{3q_A q_B},\tag{40}$$

where $M$ is the total system mass, $q_i$ is the fraction of this in mass $i$ (so $q_A + q_B \equiv 1$), and $r$ is the distance between $A$ and $B$.

Equation (40) was originally derived in [115] for AQUAL and [63] for QUMOND. More recently, it was shown to hold for any modified gravity theory with the asymptotic deep-MOND behaviour [116]. Notice that $g_{\mathrm{rel}}$ is slightly below the result for a point mass acting on a test particle, so the mutual gravity between two masses with a fixed total mass depends on how this is distributed between the masses. This non-Newtonian dependence on the mass ratio is a consequence of the gravity from the secondary effectively imposing an EFE on the primary that reduces the density of its PDM halo. This is necessary because if we go out to distances $\gg r$ such that $A$ and $B$ can be considered as a single point mass, then simple superposition of their individual PDM halos must be violated in order to reproduce the empirical Equation (18).

*2.5. Modified Inertia*

To this point, we have focused on the more common modified gravity interpretation of MOND. It is also possible to modify the law of inertia at low accelerations [117]. In this case, standard Newtonian gravity applies to galaxies, but their flat RCs require a mismatch

between the acceleration $\ddot{\boldsymbol{r}}$ and the gravitational field $\boldsymbol{g} = \boldsymbol{g}_N$. Consistency with Equation (5) requires that for near-circular motion around a point mass, the acceleration of a test particle in the deep-MOND limit is

$$\ddot{\boldsymbol{r}}|\ddot{\boldsymbol{r}}| \;=\; \boldsymbol{g}_N a_0 \,. \tag{41}$$

Any version of MOND must address the basic issue of why a star with typical internal gravitational field $\gg a_0$ should have a centre of mass acceleration exceeding the $g_N$ sourced by the rest of the galaxy. To simplify our discussion, we will consider the case of the Sun, and assume that the galactic gravity is sourced by a distant point mass $A$ much more massive than the Sun. This issue was addressed in [91], but we briefly mention the solution here. In a modified gravity interpretation of MOND, gravity behaves similarly to Newtonian expectations near the Sun and $A$. However, the regions in between often have a total gravitational field $\lesssim a_0$. The gravitational field of $A$ thus follows a standard inverse square law close to $A$ and close to the Sun, but in principle an inverse distance law could apply elsewhere. As a result, it is easy to understand why the solar system as a whole accelerates towards the galactic centre faster than expected [1,118], even though standard gravity works well within the solar system [119,120] and near the galactic centre [4,5].

In a modified inertia theory, this logic does not apply because gravity is Newtonian. As the accelerations of planets in the solar system are $\gg a_0$, they should follow a standard law of inertia. Since the gravitational field from $A$ is not modified, the extra acceleration of the planet due to $A$ would be the Newtonian gravity $A$ generates at the location of the planet, even if this is $\ll a_0$. The same argument can also be applied to different parts of the Sun. It is thus very difficult to understand how the barycentric acceleration of the solar system is enhanced in a modified inertia interpretation of MOND, as would be required to yield flat galaxy RCs without CDM. Any such modified inertia theory must be strongly non-local [121], preventing test particle motion from being describable in terms of a potential [122]. No such theory currently exists that is capable of handling complicated systems such as interacting galaxies, though [121] describes a toy model obtained by working in frequency space (see their Section 3.1). Nonetheless, some consequences can still be deduced for near-circular motion, which is sufficient for traditional RC tests. Comparing such predictions with observations reveals a clear preference for the modified gravity interpretation of MOND [123]. Further, bearing in mind the above-mentioned theoretical issues, we assume in the rest of this review that if galaxies lack CDM halos, the most plausible alternative is a MOND-like modification to the gravitational field equation in the weak-field non-relativistic limit. We briefly consider other alternatives in Section 3.6.

### 2.6. Relativistic Theories

The widespread acceptance of GR dates back to the confirmation of its a priori prediction that a ray of starlight just grazing the solar limb would be deflected by $1.75''$ towards the Sun, with an inverse dependence between the deflection angle and the impact parameter [124]. This is twice the Newtonian prediction, even though GR and Newtonian dynamics give almost the same results for the non-relativistic planetary orbits. Clearly, gravitational lensing can place important constraints on the behaviour of gravity.

For GR effects to become important in a system not observed at such high precision, it should have a potential depth of order $c^2$. For MOND effects to become important, the system should also have $g \lesssim a_0$. Since we can take the ratio of the potential and its gradient to estimate the size $R$ of a system, these two considerations imply that

$$R \;\gtrsim\; \frac{c^2}{a_0} \;\gtrsim\; \frac{c}{H_0} \,, \tag{42}$$

with the latter inequality following from the numerical value of $a_0$ (Equation (12)). Therefore, a system strongly affected by both GR and MOND effects should be larger than the Hubble horizon, i.e., it should be cosmologically large. This makes it difficult to observationally probe certain aspects of relativistic MOND.

Nonetheless, as with the case of Mercury's orbit, higher-precision observations allow us to gain important insights even if GR or MOND effects are sub-dominant. In particular, it is possible to consider the motion of light through much smaller systems which have $g \lesssim a_0$, thereby probing MOND—but with the smaller size causing a potential depth $\ll c^2$. One of the most important such cases is gravitational lensing by galaxies and galaxy clusters.

The missing gravity problem is also apparent in gravitational lensing (Section 4.2 of [125]). Importantly, the enhancement to gravity implied by the lensing data is similar to that implied by non-relativistic tracers, posing an important constraint on extensions to GR [126]. That work proposed the addition of a vector field to explain why the missing gravity problem is also apparent in gravitational lensing. Making this time-like vector field dynamical and using a disformal metric led to the first fully relativistic theory of MOND, which came to be known as tensor-vector-scalar (TeVeS) gravity [127]. Lensing phenomena in TeVeS were discussed further in [128], while relativistic MOND theories were reviewed more generally in Section 7 of [72].

Subsequent work has confirmed that the non-relativistic gravitational field should cause light deflection in the same manner as GR to within a few percent (we discuss the observations in Section 3.4). Therefore, relativistic MOND theories are nowadays designed to have a GR-like relation between the angle of deflection $\Delta \widehat{n}$ and the non-relativistic $g$ felt by test particles [129,130], with $\widehat{r} \equiv r/r$ denoting the unit vector parallel to $r$ for any vector $r$. This result also holds in other relativistic MOND theories [131–134]. For a problem in which the metric is close to flat, the relation is that

$$\Delta \widehat{n} = \int \frac{2g}{c^2}\, dl\,, \tag{43}$$

where the integral must be taken along the photon trajectory with elements of length $dl$. The factor of 2 arises because GR predicts deviations from the flat Minkowski metric in both the space and time components. This aspect of GR should be carried over to any successful theory, though sub-percent deviations are still possible. Despite this similarity, GR and MOND typically predict different $\Delta \widehat{n}$ because they predict different $g$ for the same mass distribution.

As an example of how to build a relativistic model, consider a metric $g_{\alpha\beta}$ with determinant $g$ such that $\sqrt{|-g|} \approx -c^2 g^{00} = \left(1 - \frac{2\Phi}{c^2}\right) \to \frac{-2U}{c^2}$, where we have assumed $C = c$ for simplicity. We can make our non-relativistic action (Equation (23)) covariant by again using the scalar field $N$ with an initial perturbation around $N = -c^2/2$ everywhere. We also need a field $Y = \left|g^{\alpha\beta}\nabla_\alpha N \nabla_\beta N\right|$ such that $x^2 = y = \sqrt{Y}/a_0$ represents the Newtonian field strength of standard particles if we assume $a_U = a_0$ for simplicity. Replacing $\nabla^2 N \to \nabla^\alpha \nabla_\alpha N$ in the action, Equation (23) can be rewritten as a covariant action after adding the standard action of a particle with mass $m$:

$$\begin{aligned}
S &= \int m \sqrt{g_{ab}\dot{x}^a(t)\dot{x}^b(t)}\, dt \\
&\quad - \int \frac{R' + \int_{K'}^\infty \widetilde{v}\left(\frac{\sqrt{Y}}{a_0}\right) dY}{8\pi G} \sqrt{-g}\, d^3 r\, dt\,.
\end{aligned} \tag{44}$$

Here the usual torsion or Ricci scalar term in the Einstein–Hilbert action is recovered with $R' \leftarrow \frac{c^4}{2N}\nabla^\alpha \nabla_\alpha N$ and $K' \leftarrow \nabla^\alpha N \nabla_\alpha N$. One expects light bending to follow geodesics of $U$.

The original form of TeVeS is incompatible with the near-simultaneous detection of the event GW170817 and its electromagnetic counterpart GRB170817A, which demonstrates that GWs travel at $c$ to extremely high precision [7]. The very low false detection probability means that GW170817 rules out some relativistic versions of MOND [135] and a wide range of other modified gravity theories [136–138]. However, it does not rule out bimetric MOND [132]. Moreover, it is possible to explain the GW propagation speed with a slightly modified form of TeVeS [139]. Their model is probably the best currently available relativistic MOND theory in the scientific literature.

### 2.7. Theoretical Uncertainties in the Missing Gravity Problem

We are now in a position to delineate the scope of this review, which considers how different lines of evidence might be reconciled with ΛCDM and MOND in order to see which is more plausible. Only certain types of astronomical evidence are relevant to this discussion. Since MOND departs from Newtonian gravity at low accelerations, we focus primarily on systems with $g \lesssim a_0$. Another important distinction is that the MOND picture lacks CDM, which however is not expected to disappear in the ΛCDM picture as a direct consequence of $g \gg a_0$. While this may happen by coincidence in some cases, the huge range of currently available data should also identify other cases. In particular, ΛCDM requires significant amounts of CDM in the high-acceleration central regions of massive elliptical galaxies, seemingly contradicting observations [140]. We discuss this further in Section 3.2.1.

Table 1 shows the level of theoretical certainty in the predictions of ΛCDM and MOND for different astronomical systems and aspects of them. Our focus is on observables where differences are likely on theoretical grounds. We have included cases where it is currently unclear how a particular observable would behave in one theory if a strong prediction is made by the other. The reason is that observational agreement with this prediction would lend confidence to the theory, while disagreement would cast doubt on it or even falsify it. Actual observational results are not shown in Table 1, though we do consider the observational uncertainty and only list situations which might be observable at the requisite precision. To help future-proof this review, we also include some situations where no or very limited data are currently available, provided there are good prospects for obtaining decisive results in the future. Such tests are discussed further in Section 11.

**Table 1.** The extent to which ΛCDM (red) and MOND (blue) make clear predictions for various proposed tests in different astrophysical systems. The horizontal lines divide tests according to whether they probe smaller or larger scales than the indicated length. The open dots show systems from which data were crucial to theory construction or to fix free parameters, while other systems are shown with filled dots.

| Astrophysical Scenario | Clear Prior Expectation | Not Predicted, but Follows from Theory | Auxiliary Assumptions Needed, but These Have Little Effect | Auxiliary Assumptions Needed, and These Have a Discernible Effect | Auxiliary Assumptions Allow Theory to Fit Any Plausible Data |
|---|---|---|---|---|---|
| Big Bang nucleosynthesis | 🔴 | | | 🔵 | |
| Gravitational wave speed | 🔴 | | | | 🔵 |
| Wide binary velocities | 🔴 🔵 | | | | |
| Interstellar precursor mission trajectory | 🔴 🔵 | | | | |
| Cavendish experiment in saddle region | 🔴 🔵 | | | | |
| *pc* | | | | | |
| Tidal dwarf galaxy $\sigma_i$ | 🔴 🔵 | | | | |
| Splitting in tidal dwarf mass–size relation | 🔴 🔵 | | | | |
| Tidal limit to dwarf galaxy radii | 🔵 | | 🔴 | | |
| Prevalence of thin disc galaxies | | | | 🔴 | |
| Freeman limit to disc central density | 🔵 | | | | 🔴 |
| Number of spiral arms | 🔴 🔵 | | | | |
| Weakly barred galaxies | 🔵 | | 🔴 | | |
| Bar fraction in disc galaxies | | | 🔴 | | |
| Galaxy bar pattern speeds | 🔵 | 🔴 | | | |
| *kpc* | | | | | |
| Exponential profiles for disc galaxies | | | 🔵 | | 🔴 |
| Disc galaxy RCs | ⊙ (blue) | | | | 🔴 |
| Elliptical galaxy RCs | 🔵 | | | | 🔴 |
| Spheroidal galaxy $\sigma_i$ | 🔵 | | | | 🔴 |
| External field effect | 🔴 🔵 | | | | |
| Galactic escape velocity curve | | | | 🔵 | 🔴 |
| Number of satellite galaxies | | | | | 🔴 |
| Anisotropy of satellite distribution | 🔴 | | 🔵 | | |
| Weak lensing by galaxies | 🔵 | | | | 🔴 |
| Strong gravitational lensing | 🔴 | | | | 🔵 |

**Table 1.** *Cont.*

| Astrophysical Scenario | Clear Prior Expectation | Not Predicted, but Follows from Theory | Auxiliary Assumptions Needed, but These Have Little Effect | Auxiliary Assumptions Needed, and These Have a Discernible Effect | Auxiliary Assumptions Allow Theory to Fit Any Plausible Data |
|---|---|---|---|---|---|
| Polar ring and shell galaxies | | | | 🔵 | 🔴 |
| — Mpc — | | | | | |
| Local Group timing argument | | 🔵 | | | 🔴 |
| Hickson Compact Group abundance | | | | 🔴 | 🔴 |
| Binary galaxy relative velocity | 🔵 | | | | 🔴 |
| Galaxy group $\sigma_i$ | 🔵 | | | | 🔴 |
| Galaxy cluster internal dynamics | | | | 🔴🔵 | |
| Baryon-lensing offsets in galaxy clusters | 🔴 | | | 🔵 | |
| Galaxy cluster formation | 🔴 | 🔵 | | | |
| Galaxy two-point correlation function | 🔴 | | | | 🔴 |
| Weak lensing correlation function | 🔴 | | | | |
| CMB anisotropies | | | | ⊙ | 🔵 |
| Cosmic variance on 300 Mpc scale | 🔴🔵 | | | | |
| Local Hubble diagram slope and curvature | 🔴 | | 🔵 | | |
| — Gpc — | | | | | |
| Expansion history at $z \gtrsim 0.2$ | 🔴 | | | 🔵 | |

### 3. Equilibrium Galaxy Dynamics

At the centre of a thin disc with central surface density $\Sigma_0$, the vertical Newtonian gravity is $g_{N,z} = 2\pi G \Sigma_0$. The radial gravity is also of this order at a typical radius of one disc scale length. Therefore, significant MOND effects are expected if $\Sigma_0$ falls below the critical MOND surface density

$$\Sigma_M \equiv \frac{a_0}{2\pi G} = 137 \, M_\odot / \text{pc}^2 \,. \tag{45}$$

This also applies to a pressure-supported system, up to factors of order unity due to the different geometry. MOND and Newtonian gravity behave similarly if the surface density $\Sigma \gg \Sigma_M$.

Differences between $\Lambda$CDM and MOND can arise even if $\Sigma$ slightly exceeds $\Sigma_M$. This is partly because the simple interpolating function (Equation (15)) recommended in [111,141] implies a rather gradual transition between the Newtonian and MOND regimes. Moreover, the gravity law is not the only difference between the paradigms. Substantial amounts of DM might be expected in a high $\Sigma$ system in the $\Lambda$CDM paradigm where $\Sigma$ has no special significance, but the Newtonian gravity of the baryons alone should be almost sufficient in MOND. Nonetheless, we expect that differences between $\Lambda$CDM and MOND would generally be easier to detect in systems where $\Sigma \lesssim \Sigma_M$ [56].

#### 3.1. Disc Galaxies

Thin disc galaxies historically provided the main evidence for the missing gravity problem (as reviewed in [13]). Theoretically, they are quite tractable in MOND by applying Equations (17) or (20) to the detectable baryons. Observationally, the radially inward gravity $g_r$ can be inferred from the velocity field with some well-motivated assumptions, as we now explain.

It is usually assumed that $g_r = v_c^2 / r$, where $v_c$ is the circular rotational velocity or RC amplitude at galactocentric radius $r$. This relies on the assumption of a standard law of inertia (Section 2.5) and that the spectroscopic redshift gradient across a galaxy is indicative of coherent circular motion. If the latter assumption holds, then one can argue that long-term stability requires a centripetal acceleration of $v_c^2 / r$. Since gravity is expected to dominate on kpc scales, this acceleration can be equated with $g_r$.

Observations have confirmed critical links in this chain of logic. The rotation of a galaxy should in principle be directly detectable based on the proper motions in different parts of its disc revealing a rotating pattern. This has been observed in the Large Magellanic Cloud (LMC), with an implied RC amplitude of $\approx 90$ km/s [142–144]. This is consistent with the $72 \pm 7$ km/s estimate based on line of sight (LOS) velocities, which are called radial velocities (RVs) in the literature [145]. Spatially resolved proper motion measurements of M31 and M33 show that these galaxies are also rotating at roughly the speed previously inferred from RVs alone [146]. While non-circular motions might be more significant in other galaxies, these observations are very reassuring as they help to confirm the relation between spectroscopically determined redshift gradients across a galaxy and its 3D internal motions.

Another recent observational breakthrough is the direct detection of the solar system's barycentric acceleration relative to distant quasars, which mostly arises from motions within the MW [1]. Their detection was based on the aberration of light caused by the velocity of the solar system, whose time evolution causes the aberration angle to change by up to $5.05 \pm 0.35$ µas/yr towards the direction of the acceleration. The results show that the solar system does indeed accelerate towards a direction within $\approx 5°$ of the galactic centre at a rate close to $v_c^2 / r$ if the galactocentric distance of the Sun is $r = 8.18$ kpc [147] and $v_c = 233$ km/s, as suggested by kinematic observations (e.g., [118,148]).

While caution is required when generalizing results from the MW and LMC to other galaxies, the above-mentioned results make it very likely that orbital accelerations within galaxies can be reliably determined from observed redshift differences across them. With our assumption that a standard law of inertia applies even in the low-acceleration regime

(Section 2.5), it is then possible to test different ideas about the weak-field behaviour of gravity using RCs inferred from spatially resolved redshift maps. We therefore discuss what can be learned from such RC fits.

### 3.1.1. Rotation Curves

According to conventional physics, the RC of a galaxy should undergo a Keplerian decline beyond the extent of its luminous matter. In other words, we expect that $v_c \lesssim 1/\sqrt{r}$ analogously to the RC of the solar system (summarized by Kepler's Third Law). It therefore came as a great surprise that real galaxies do not behave in this way, as first noticed for M31 [149,150]. These optical studies did not go very far out because the gas density must exceed a certain threshold to form stars. Given the typically exponential surface density profiles of disc galaxies [151], this leads to an outer limit beyond which other tracers of the RC become necessary. Undoubtedly the most important is the 21 cm hyperfine transition of neutral hydrogen (reviewed in [152]). Observations of this spectral line showed that disc galaxies typically have flat outer RCs [153], including in the case of M31 [154]. This is also apparent in much larger galaxy samples, so a flat outer RC is a general feature of a disc galaxy [155,156].

Assuming a standard law of inertia (Section 2.5), these observations imply that our existing theories severely underpredict $g_r$. This missing gravity problem might be addressed by DM halos around galaxies, but another possibility is to use MOND. We illustrate this in Figure 2 (reproduced from Figure 6 of [157]) by considering CDM and MOND fits to the RC of NGC 2403. The photometry used in this fit is from a different galaxy (UGC 128). Though this has approximately the same $M_s$, its longer scale length means that its RC is expected to rise more gradually in MOND. Consequently, the MOND fit to the RC of NGC 2403 is very poor (left panel). The normalization has been adjusted by varying, e.g., the inclination, but even so, the longer scale length of the UGC 128 baryonic distribution makes it essentially impossible for its Milgromian RC to properly fit the observed RC of NGC 2403. In other words, MOND is unable to fit this fake combination of photometry and kinematics. However, the CDM fit (right panel) is quite good. This is due to the flexibility afforded by the dominant CDM halo, whose parameters are not otherwise constrained except from the same data that the model is attempting to fit. It is therefore clear that ΛCDM is not very predictive with regards to galaxy RCs—almost any conceivable measurement can be explained afterwards with an appropriately tuned distribution of particles that often have to dominate the mass budget but are not otherwise detectable. The predictive power of MOND is certainly a strong argument in its favour and has historically been a hallmark of major scientific breakthroughs [50].

A flat RC implies that $g_r \propto 1/r$. If this applies beyond the extent of the matter distribution, then the point mass gravity law must itself decline as $1/r$ instead of the conventional $1/r^2$. However, since the inverse square law works well in the solar system, one must decide on an appropriate parameter demarcating where the departure from conventional gravity sets in. If a fixed length scale $r_0$ is used, then the gravity at $r \gg r_0$ from a baryonic mass $M_s$ would be $GM_s/(rr_0)$, implying that $v_c \propto \sqrt{M_s}$. However, the TFR [60] indicated that rotational velocities scale approximately as $\sqrt[4]{L}$. Since $L$ should be roughly proportional to the stellar mass ($M_\star$) and galaxies such as the MW have much less mass in gas than in stars, it was clear that $v_c \lesssim \sqrt[4]{M_s}$ if a fundamental relation exists at all. Since $GM_s/r_0 = v_c^2$ in the outer flat region, the TFR can hold only if the transition radius $r_0 \propto M_s/v_c^2 \propto \sqrt{M_s}$, implying that $M_s/r_0^2$ should be the same in different galaxies. In other words, the RC of a galaxy should depart from conventional expectations only when $g_N$ falls below some particular threshold that is consistent between galaxies [158].

This line of reasoning led to the development of MOND, where the weak-field point mass gravity law is given by Equation (18). Equating this with $v_c^2/r$ implies that the outer flat part of the RC has an amplitude

$$v_f = \sqrt[4]{GM_s a_0}. \tag{46}$$

This is not an a priori prediction of MOND because the TFR was used to decide upon acceleration as the critical variable [14,61]. Nonetheless, the observations merely suggested a relation between $v_c$ and $M_\star$ with a somewhat uncertain power-law slope in the range 0.2–0.4. It was not clear that the correlation would persist when using instead the total baryonic mass $M_s$, since gas-rich galaxies were not yet well-studied—these tend to have a lower surface brightness. Moreover, RCs were generally not measured to large enough distances to reach the outer flat region. It was therefore not known a priori whether the baryonic Tully–Fisher relation (BTFR; Equation (46)) would hold once $v_f$ was plotted against the stellar + gas mass.

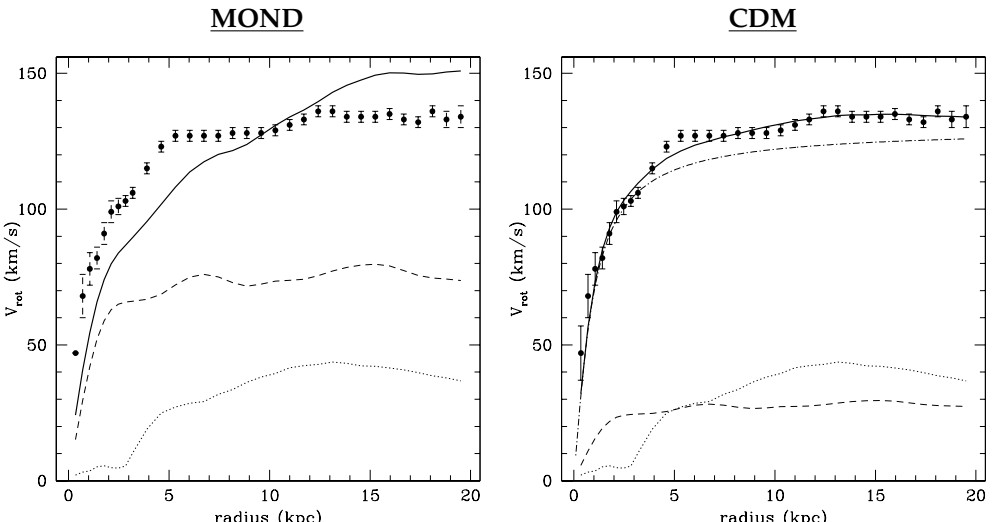

**Figure 2.** RC fit to NGC 2403 using MOND (**left**) and CDM (**right**), based on the photometry from a different galaxy (UGC 128). The observed RC (points with error bars) is far above the Newtonian prediction based on only the stars (dashed line) or gas (dotted line) in the best-fitting model. The high dot-dashed line in the right panel is the contribution of the halo. The solid line in both panels is the total RC. Notice that the fit is very poor in MOND due to the use of incorrect photometry. However, an acceptable fit can be obtained with CDM due to the flexibility in the halo parameters, suggesting that it can fit almost any RC with any run of surface brightness. Reproduced from Figure 6 of [157] by permission of Stacy McGaugh and the American Astronomical Society.

This question was addressed observationally by considering a large sample of galaxies with a wide range of gas fractions [159]. Those authors found that the relation between $v_f$ and $M_\star$ cannot be fit by a single power law, but this becomes possible once the gas mass is included and the total $M_s$ is plotted against $v_f$ (see their Figure 1). Since these are the quantities entering Equation (46), its prediction was verified over almost 4 orders of magnitude in $M_s$, or equivalently over the range $v_f \approx (30 - 300)$ km/s. More recent results continue to show that the dependence of $v_f$ on $M_s$ must be very close to the fourth root [160]. Their work highlights that the tightest correlation is obtained when using $v_f$ rather than other measures of a galaxy's rotation velocity, e.g., the maximum $v_c$. For a review of observations relating to the BTFR and possible interpretations in $\Lambda$CDM and MOND, we refer the reader to [161].

Combining Equations (18) and (46) allows us to define a characteristic specific angular momentum $j_M$ from only the basic postulates of MOND [162]. For any mass $M$ with MOND radius $\sqrt{GM/a_0}$, we get that

$$j_M = (GM)^{3/4} a_0^{1/4}. \tag{47}$$

This predicted scaling agrees quite well with the observational results of [163], especially for $M_s \gtrsim 10^{10} M_\odot$. Lower-mass galaxies tend to lie somewhat above this relation because they tend to have a lower surface brightness, which implies a larger size at fixed mass.

MOND can be used to predict the entire RC, not only its asymptotic level $v_f$. In this regard, MOND enjoys unparalleled predictive success (extensively reviewed in [72]). A particularly striking aspect is that some galaxies contain a feature in their $\Sigma(r)$ profile, e.g., a bump or a dip. The RC has a corresponding feature. In the case of NGC 6946, the feature occurs at a location where the proportion of missing gravity is small, so it is to be expected that the feature in the baryonic density profile causes a similar feature in the RC [164]. However, a similar phenomenon is apparent in NGC 1560, where the proportion of missing gravity is large [165]. This is also apparent in several other galaxies. It has been summarized as Renzo's Rule, which states that "for any feature in the luminosity profile there is a corresponding feature in the RC and vice versa" [166]. In a conventional gravity context, the validity of this rule in galaxies such as NGC 1560 implies that the dominant DM halo should have a feature in its density distribution that mimics the feature in the baryonic density profile. However, small-scale bumps and wiggles can be sustained only in a dynamically cold component such as baryons in a disc, not in a dynamically hot (pressure-supported) dissipationless DM halo. Moreover, the dark halo would be only slightly affected by features in the density distribution of the sub-dominant baryonic component. With a smooth halo that dominates the total gravity, even quite significant features in the baryonic surface density profile would have little effect on the RC.

A tight radial acceleration relation (RAR) between the radial components of $g$ and $g_N$ continues to hold in the Spitzer Photometry and Accurate Rotation Curves (SPARC) database of 175 galaxies [167]. Its major advantage is the use of 3.6 μm photometry, which reduces the uncertainty on $M_\star$ because $M_\star/L$ ratios have less scatter at near-infrared wavelengths [168,169]. Analysis of the SPARC database reveals a very tight RAR [71], which we show in our Figure 3 by reproducing their Figure 3. Their adopted functional form for the RAR is

$$\nu = \frac{1}{1 - \exp\left(-\sqrt{\frac{g_N}{a_0}}\right)} . \tag{48}$$

This MLS form of the interpolating function is numerically very similar to Equation (15), but with a faster exponential cutoff that strongly suppresses deviations in the solar system (Section 11.4) and leads to better consistency with observations at $g_N \approx 10\,a_0$ [140].

Equation (48) was used to fit the RCs of all SPARC galaxies, leading to quite good agreement given the uncertainties [170]. Allowing for known sources of error, those authors conservatively estimated that the intrinsic scatter in the RAR must be <0.057 dex despite the data covering approximately 3 dex in $g_N$ and 2 dex in $g$ centred approximately on $a_0$. Part of the uncertainty is due to variations in $M_\star/L$, which can be mitigated by focusing on gas-rich galaxies [171]. Doing so reveals that these galaxies also follow a tight BTFR, though the smaller sample size inflates the uncertainty on the inferred $a_0$ to $(1.3 \pm 0.3) \times 10^{-10}$ m/s$^2$ [172]. More generally, the RCs of gas-rich dwarf galaxies with low internal accelerations offer a powerful test of MOND because of reduced sensitivity to both the interpolating function and the $M_\star/L$ ratio. MOND fares well when confronted with the data for 12 such galaxies [173]. Another possible systematic is that galaxies might contain an additional undetected gas component. The authors of [174] approximately considered this by scaling up the conventionally estimated gas mass by some factor $f$, which was then inferred observationally from MOND fits to the SPARC sample of galaxy RCs. Those authors inferred that $f = 2.4 \pm 1.3$, indicating no strong preference for an additional component of "cold dark baryons". However, including such a component would reduce the best-fitting $a_0$ as there would be more baryonic mass for the same RC.

MOND assumes the presence of a universal acceleration scale $a_0$. It has been claimed that RCs can be fit better if $a_0$ is allowed to vary between galaxies, with a common acceleration scale ruled out at high significance [175]. Allowing $a_0$ to vary between galaxies but using a Gaussian rather than a flat prior on $a_0$ led to substantially weaker evidence against MOND [176], already casting severe doubt on the reliability of the strong claims

made by [175]. Another issue is their over-reliance on Bayesian statistics, which can lead to erroneous conclusions when it is inevitable that some sources of error are not accounted for, e.g., gradients in $M_\star/L$ and warps [177].

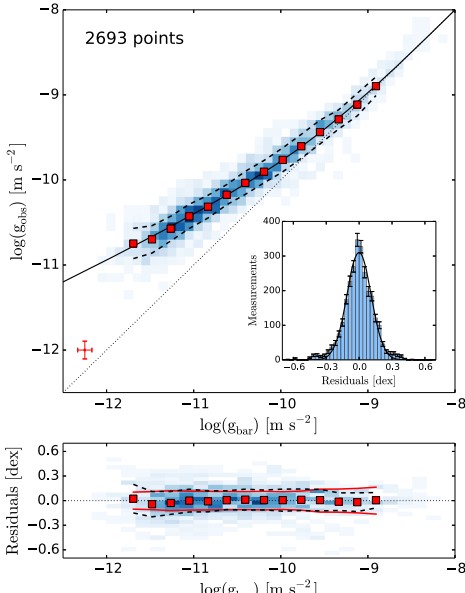

**Figure 3. Top**: The radial acceleration relation (RAR) between the observed centripetal acceleration $v_c{}^2/r$ inferred from a galaxy's RC and the radial component of $g_N$ generated by its observed baryons. Results are shown on logarithmic axes for 153 galaxies in the SPARC catalogue [167], yielding 2693 points. More frequently occurring 2D bins are shown with a darker blue, while the red squares show average results in different $g_N$ bins. The missing gravity problem is evident in that the data at low $g_N$ lie well above the line of equality (thin dotted line). The mean relation is shown as a solid black line (Equation (48)), while the dashed black lines around it show the width of the distribution at fixed $g_N$. The red cross at the lower left indicates the typical uncertainty of each point. The inset is a histogram of the deviations from the mean relation, along with Poisson uncertainties and a Gaussian of width 0.11 dex. **Bottom**: The RAR with the mean relation subtracted. The solid red lines show the expected dispersion due to variations in $M_\star/L$ and measurement errors, while the dashed black lines show the actual dispersion. Uncertainties in the binned results are too small to show here. Notice the lack of any trend, indicating Equation (48) is a good fit to the RAR. Reproduced from Figure 3 of [71] by permission of Stacy McGaugh and the American Physical Society.

Since MOND requires $a_0$ to be the same in different galaxies, it is irrelevant whether letting it vary improves the fit, which to some extent is inevitable. More important is whether MOND can plausibly fit observed RCs with a single value of $a_0$. The argument of [175] was essentially that MOND could not. However, no RCs were presented to support this claim. In addition, the analysis suffered from significant deficiencies related to the quality cuts applied to the data [178]. The most major problem was the use of Hubble flow distances for a large proportion of galaxies where no redshift-independent distance was available. Since the peculiar velocity of the LG relative to the CMB is 630 km/s [179] and this corresponds to a Hubble flow distance of 9 Mpc, it is clear that Hubble flow distances have an uncertainty of this order. Because many of the SPARC galaxies are closer than 50 Mpc, it is clear that the fractional distance uncertainty could easily exceed 20%, which was adopted by [175] as the maximum possible error in the estimated distance. Indeed, some of the galaxies they identified as problematic were previously well fit in MOND with distances slightly outside a ±20% range centred on the best estimate [170]. In addition, Figure 2 of [178] demonstrated that after applying well-motivated quality cuts, the claimed most problematic galaxy (NGC 3953) can actually be fit rather well in MOND, with a few points discrepant by slightly over 10 km/s in a galaxy where the typical rotation velocity is

$v_c \approx 200$ km/s. The very small published uncertainties could well be underestimates: RCs often have features that would be very difficult to explain in any theory. These features probably arise from small undetected systematic effects such as warps.

This raises the general issue that it would be premature to conclude against a theory because it disagrees with astronomical observations by 0.6% while the official uncertainty is 0.1%, since even a very small unknown or neglected systematic error could restore agreement. However, if the disagreement is 60% and the uncertainty is 10%, then it is much less likely that systematic errors will ever be uncovered that fully explain the disagreement, so there is good reason to suppose that the theory is falsified in its present form. Formally, the disagreement is $6\sigma$ in both cases, but that does not make the situations equally problematic for the theory given the complexities typical of astrophysics. This will be important to bear in mind when considering very high precision tests such as solar system ephemerides (Section 11.4).

A subsequent study claiming that MOND cannot fit observed RCs with a fixed value of $a_0$ [180] made some improvements over earlier work in terms of the quality cuts. However, it again followed the approach of [175] in not showing any RC fits to support the claim that galaxy RCs falsify MOND at $> 5\sigma$. This prevents readers from drawing their own conclusions about whether MOND is truly unable to fit galaxy RCs in light of their directly detected mass. Moreover, a similar statistical analysis can be applied to determine if Newton's gravitational constant $G$ varies between galaxies [181]. They found strong evidence for variation, but this disappeared upon conducting basic checks. For instance, the cumulative distribution of the reduced $\chi^2$ statistic hardly differed depending on whether $G$ was held fixed or allowed to vary (see their Figure 2). More importantly, the authors found that it was possible to use the same value of $G$ in all analysed galaxies.

With any scaling relation such as the BTFR, it is important to find the range over which it remains valid. In this regard, the galaxy samples of [182,183] could be valuable as they include a population of very luminous spiral galaxies (termed "super-spirals"), with $M_s$ reaching up to $6.3 \times 10^{11} M_\odot$ for OGC 0139 [184]. Those authors claimed to find evidence for a break in the BTFR at such high $M_s$. As described in Section 3 of [184], they used the maximum $v_c$ rather than the flatline level which enters into Equation (46). Moreover, the claimed break in the BTFR was primarily based on just six galaxies. This and other problems were discussed in detail by [185], which argued based on a larger galaxy sample (43 instead of 23) that there is no strong evidence of a break in the BTFR up to the highest masses probed. In their Section 7, they explain that two of the supposedly problematic galaxies have irregularities in their H$\alpha$ emission that make it difficult to unambiguously define a rotation velocity. For two more galaxies, a revised analysis of the data reduces the estimated rotation velocity by 50 km/s in one case and 60 km/s in the other. The remaining two galaxies remain somewhat problematic, but as shown in Figure 4 of [185], the discrepancies are not very severe and represent only a very small number of instances. This led those authors to conclude that the BTFR defined by lower mass disc galaxies extends to the highest $M_s$ probed, "without any strong evidence for a bend or break at the high-mass end" (see their conclusions). The small fraction of outliers is especially evident in their Figure 3, which shows that the much larger galaxy sample of [186] is quite consistent with a tight BTFR. Even the two outliers at the high-mass end might naturally be accounted for using the integrated galactic initial mass function (IGIMF [187–190]), a framework in which more massive galaxies would be expected to have a higher proportion of stellar remnants and thus a higher $M_\star/L$ (A. H. Zonoozi et al., in preparation). More massive spirals also tend to have a larger fraction of their baryons in a central bulge, which generally has a higher $M_\star/L$. Allowing the disc and bulge to have different $M_\star/L$ constrained using their observed colours leads to better agreement with MOND expectations, i.e., reduced curvature and scatter in the BTFR and a slope closer to the predicted $v_f \propto \sqrt[4]{M_s}$ [191].

There have also been a few claims that galaxies depart from the BTFR at the low-mass end (e.g., [192,193]). The main issue is the reliability of their reported RCs. The rotation periods are several Gyr due to the large sizes of these galaxies for their $M_s$, which may

mean that the galaxies are not yet in virial equilibrium. If the real or phantom halos of these galaxies have a cuspy inner profile, then this would create a sharp feature clearly identifiable in a velocity field that would make the analysis much more reliable. However, dwarf galaxies usually have an almost linearly rising RC in the central region (e.g., [194]), which in a conventional gravity context is interpreted as caused by a halo with a constant density central region, i.e., a core. The centre of the galaxy and its inclination are then quite difficult to determine. If the galaxies were slightly more face-on than reported, then the actual $v_f$ would be higher than estimated, which would bring the results closer to the BTFR defined by more massive galaxies (see Figure 9 of [193]). An overestimated inclination angle between disc and sky planes is actually quite likely as a face-on galaxy might not appear exactly circular. If instead the isophotes have an ellipticity of 10%, a face-on galaxy would appear to have an inclination of $i = \cos^{-1} 0.9 = 26°$. This phenomenon is apparent in the hydrodynamical MOND simulations with star formation and the EFE shown in Figure 4, which reproduces Figure 1 of [195]. The departures from axisymmetry presumably arise because discs are necessarily self-gravitating in MOND, allowing them to sustain non-axisymmetric features such as bars and spiral arms even in the dwarf galaxy regime. This appears to be the issue with a recent claim that the observed RC of AGC 114905 contradicts the expectations of both $\Lambda$CDM and MOND [196]. The tension with both theories can be alleviated by reducing the inclination below the observational estimate of $32°$ to a nearly face-on $11°$, which is quite possible in MOND. A similar downwards revision to $i$ was shown to be acceptable for UGC 11919 by changing the initial guess [197]. These situations are similar to the case of Holmberg II, whose RC was also claimed to be inconsistent with MOND [198]. However, it was later found that Holmberg II is closer to face-on, putting it in line with MOND expectations and the BTFR [199]. Over the decades, there have been several other "false alarms" for MOND where it seemed to fail, but it later turned out to agree with the data within uncertainties (e.g., [200], and references therein).

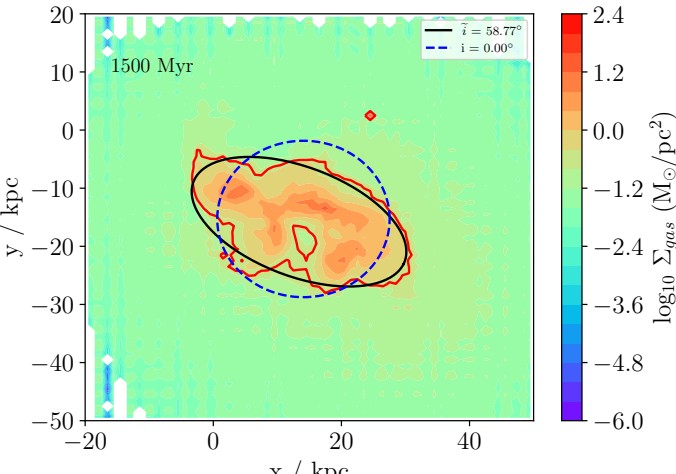

**Figure 4.** The 1.5 Gyr snapshot of a hydrodynamical MOND simulation of a disc galaxy with a density distribution and gas fraction similar to AGC 114905. The gas surface density in $M_\odot/\text{pc}^2$ is shown in the face-on view ($i = 0°$) on a $\log_{10}$ scale. The solid red line shows a contour at $-0.6$. The best-fitting circle (dashed blue) is a much poorer fit than the best-fitting ellipse (solid black). Its aspect ratio would lead to an inferred inclination of $\widetilde{i} = 59°$ if the galaxy is assumed to be circular when viewed face-on. The simulation shown here includes an external field of strength $0.05\,a_0$ at $30°$ to the disc plane, though results are similar in the isolated case. Reproduced from Figure 1 of [195].

While the reported RCs of AGC 114905 and Holmberg II appear to reach the outer flat portion, this is less clear in the larger sample of [193] that is necessary to claim a more general trend. It is possible that $v_f$ is higher than the reported value because $v_c$ rises with radius in galaxies with a low surface brightness (see, e.g., Figure 15 of [72]), as predicted in MOND (Figure 1 of [57]). Indeed, the authors of [193] do not provide detailed RCs that

clearly show an extended flat region from which $v_f$ can be measured, resorting instead to a comparison with $\Lambda$CDM simulations to justify their approach. Comparing observations with a forward model of $\Lambda$CDM galaxies should be a reasonable way to test $\Lambda$CDM, but obviously such an analysis cannot be used to reject MOND. Moreover, their study only considers six galaxies, so it bears strong similarities to the above-mentioned claims that high-mass galaxies deviate from MOND predictions. While the observational issues are surely different for low-mass galaxies, there is currently little evidence that these deviate systematically from MOND expectations. A handful of problematic galaxies are of course expected in the correct theory given imperfect data and the complexities of actual galaxies. A larger sample could help to test if the inclinations required by MOND statistically follow a nearly isotropic distribution, as face-on galaxies should not be too common.

RCs can certainly be used to falsify the highly predictive MOND theory, but this has not been demonstrated in a convincing manner. It appears unlikely that this will occur in any of the 175 galaxies (the full SPARC sample) for which [170] conducted RC fits using the empirical RAR (an approximation to MOND). The BTFR seems to hold up to the highest $M_s$ probed and also down to the lowest $M_s$ with reliable data [201]. Of course, the success of MOND in this regard does not prove it correct. Even so, it does seem difficult for Newtonian gravity to reproduce the observed common scale in the potential gradient in different galaxies across almost six orders of magnitude in $M_s$ [202].

In a $\Lambda$CDM context, the tightness of the RAR is in $3.6\sigma$ tension with expectations [203] based on a previously developed statistical framework [204]. A puzzling aspect is the diversity of RC shapes at fixed $v_f$, which essentially fixes halo properties such as the virial mass [205]. In dwarf galaxies with a large proportion of missing gravity, the baryons are sub-dominant tracers of the potential. Consequently, they should reveal the same RC in galaxies with the same $v_f$, even if the baryonic distribution differs. Since this expectation is not correct, one can postulate that baryonic feedback processes drove large fluctuations in the potential depth and that this led to redistribution of the dominant DM component [206]. However, the process would be stochastic, especially in dwarf galaxies where, e.g., only a few extra supernovae could make a large difference. As a result, it is very difficult to obtain a tight RAR.

In this regard, it is surprising that the RAR can apparently be reproduced in $\Lambda$CDM [207]. However, those authors did not consider the above-mentioned diversity problem. Section 4.2.1 of [208] demonstrates that it is extremely difficult to get a tight RAR simultaneously with a diversity of dwarf galaxy RCs at fixed $v_f$. This diversity can readily be understood in MOND as a consequence of differences in the baryonic surface density relative to the MOND threshold (Equation (45)), as discussed further in [161]. Essentially, the MOND RC of a deep-MOND galaxy has the same characteristic length scale as the baryon distribution, so differences in this at fixed $M_s$ (and thus $v_f$) nicely account for differences in the observed RC. For theories where a tight RAR is not obvious, claims to reproduce the RAR need to consider a wide range in surface brightness at fixed $M_s$.

Dwarf galaxies can be challenging to resolve in a large cosmological simulation. Results should be more reliable for larger galaxies, especially if the galaxies are resimulated with better resolution. One such attempt is the zoom-in $\Lambda$CDM resimulation project known as AURIGA, which considers 30 MW-like galaxies [209]. This has difficulty reproducing the BTFR, as is evident in their Figure 11. In addition to only covering the range $v_f \gtrsim 160$ km/s and thus not addressing lower-mass galaxies, the Figure reveals that $v_f$ rises more steeply with $M_s$ than the observed slope, which is very close to the MOND expectation of $1/4$ [210]. The sample size is perhaps too small to reliably address the question of intrinsic scatter, though that should be checked in subsequent work.

Despite a vast amount of effort by the $\Lambda$CDM community, it remains extremely challenging to match the small intrinsic scatter of the RAR over a very wide range of galaxy properties including mass, size, surface density, and gas fraction (Figure 4 of [211]). The latter is certainly a concern if the RAR is explained by loss of baryons from galaxies due to stellar feedback, because then one might expect galaxies that formed more stars to have lost

a larger fraction of their baryons, thus deviating from the BTFR defined by gas-rich galaxies. However, gas-rich and gas-poor galaxies lie at the same location on a BTFR diagram (see, e.g., Figure 3 of [72]). More generally, a tight RAR implies that the fraction of the total gravity that is missing in the outer region of a high-mass galaxy is the same as in the inner region of a low-mass galaxy provided the two locations have the same $g_r$. It remains very difficult to understand this if the proportion of missing gravity depends strongly on how many baryons were ejected from the galaxy by stochastic feedback processes such as individual supernovae and major mergers. Other studies such as [212] have also pointed out the tight relation between the luminous and dark components in galaxies if their dynamical discrepancies are attributed to CDM particles. The RAR is much easier to understand as a fundamental relation not contingent on the formation history of a galaxy, much like Newtonian gravity accounts for the motions of planets and moons in the solar system across a vast range of mass and size without knowing how these objects formed. It would be unusual if the generally favoured interpretation of solar system motions was that they were governed by an inverse cube gravity law supplemented by a distribution of invisible mass, the end result being identical to the action of an inverse square gravity law sourced only by the known masses.

### 3.1.2. Vertical Dynamics

In principle, the ΛCDM paradigm can match the RC of a galaxy satisfying the RAR if its DM halo is tuned to have a particular density profile. The RAR is also naturally explained in MOND. We can break this degeneracy by considering the vertical gravity $g_z$ felt by a star outside the disc mid-plane in the direction parallel to the disc spin axis. The basic principle is that very little of the mass in any CDM halo would lie close to the disc mid-plane because the DM is expected to be collisionless. Given the significant amount of DM required around the MW, postulating it to be collisional would create a DM disc, causing tension with observations [213]. Therefore, measurements of $g_z$ close to the disc mid-plane should reveal only a small amount of missing gravity commensurate with the amount of DM located even closer to the mid-plane. Since the scale height of the disc should be far smaller than the extent of the DM halo, it is possible for a star to lie outside the disc and yet have its $g_z$ little affected by the DM halo, even when this dominates $g_r$.

In MOND, $g_z$ would be enhanced over Newtonian expectations by the local value of $\nu$, which can greatly exceed 1 if the surface density is sufficiently small. Consequently, MOND typically predicts higher $g_z$ just outside the baryonic disc, as calculated for the MW by [214]. The higher predicted $g_z$ in MOND implies a higher divergence to the gravitational field, which can be attributed to a disc of PDM. The PDM distribution of thin exponential discs was visualized in, e.g., [102,215].

Observationally, $g_z$ can be inferred from the vertical profile of the vertical velocity dispersion $\sigma_z$. Combined with measurements of the tracer density $\rho_t$, the vertical momentum flux through a surface of constant height is $\rho_t \sigma_z^2$ per unit area. Considering a slab of finite thickness, the momentum flux though the top and bottom surfaces will differ slightly. If the galaxy is in equilibrium, this difference should be balanced by the slab's thickness multiplied by $\rho_t g_z$, which can therefore be determined from the observed kinematics. If we are considering the solar neighbourhood of the MW, a simplification is that $\sigma_z \approx \sigma_{\mathrm{LOS}}$ because observations towards the galactic poles will have the LOS aligned similarly to the desired component of the velocity dispersion. Even if this is not exactly correct, the corrections will be small provided the LOS is nearly along or opposite the disc spin axis, reducing sensitivity to proper motion uncertainties.

Inferring $g_z$ in this manner fundamentally relies on measuring gradients in $\rho_t$ and $\sigma_z$. Differentiating the data necessarily increases the uncertainties. Moreover, the assumption of dynamical equilibrium may not be accurate. Nonetheless, there have been several attempts to test the MOND-predicted enhancement to $\sigma_z$. There is currently no evidence for this enhancement, with the data actually supporting the ΛCDM model [216]. Their Figure 1 shows that MOND overpredicts $g_z$ by $2\sigma$, so this scenario cannot be ruled out either due

to the still significant uncertainty. Other works also reported difficulties when attempting to measure $g_z$ [217], partly because of an asymmetry between determinations towards the north and south [218] that indicates non-equilibrium processes such as the passage of Sagittarius through the galactic disc (see their Section 5.3.3).

Another consequence of the enhanced $g_z$ in MOND is that the velocity dispersion tensor would appear to tilt to a different extent as one moves away from the disc mid-plane. In general, the stronger the disc self-gravity, the closer the potential to cylindrical symmetry. If the disc self-gravity is weaker, then the geometry is closer to spherical symmetry. This has been proposed as another related test of MOND [214]. Observational measurements of the velocity ellipsoid's orientation remain uncertain [219], partly because of sensitivity to the Gaia parallax zero-point offset [220]. This should be clarified with future Gaia data releases, though departure from equilibrium would remain a concern. It may be necessary to address this using simulations of Sagittarius crossing the disc mid-plane in ΛCDM and MOND, with the disc response perhaps being a better discriminant [221–223].

Looking beyond the MW, the DiskMass survey [224] attempted to measure $g_z$ statistically by combining $\sigma_{\mathrm{LOS}}$ measurements from face-on galaxies with scale height measurements from edge-on galaxies. The idea was that the scale heights of the face-on galaxies would be known statistically based on the relation between scale length and scale height in edge-on galaxies. The analysis gave very low values for $g_z$ [225], though this could be due to underestimation of $\sigma_{\mathrm{LOS}}$ [226]. It was later shown that this is entirely possible because luminosity-weighted $\sigma_{\mathrm{LOS}}$ measurements give a higher weighting to more massive stars, but the mass-weighted $\sigma_{\mathrm{LOS}}$ entering a dynamical analysis is more sensitive to less massive stars [227,228]. In a galaxy such as NGC 6946 that is still forming stars, this would cause a mismatch between the typical ages of stars used to determine $\sigma_{\mathrm{LOS}}$ and the generally older stars that comprise the bulk of the mass [229]. As the stars in disc galaxies should be getting dynamically hotter with time due to interactions with molecular clouds [230] and other processes, the mass-weighted $\sigma_{\mathrm{LOS}}$ could indeed be higher than the luminosity-weighted $\sigma_{\mathrm{LOS}}$ if appropriate precautions are not taken. As a result, it has been argued that the DiskMass results are quite consistent with MOND [231].

It is also possible to measure $\sigma_z$ from the gas. This would be simpler in some ways as the velocity dispersion tensor should be isotropic, so the HI velocity dispersion $\sigma_{\mathrm{HI}} \approx \sigma_z$. Using this approach, the authors of [232] found plausible agreement between MOND and the observed flaring of the MW disc over galactocentric distances of $R = 16 - 40$ kpc, though the observed thickness exceeds the predicted value by $\approx 70\%$ at $R = 10 - 16$ kpc. Those authors attributed the mismatch to non-thermal sources of pressure support, which they argued would be quite plausible.

Gas kinematic and scale height measurements are also possible in external galaxies viewed close to edge on. This technique was attempted by [233] in four large galaxies and three dwarfs, with the latter probing the low-acceleration regime. The velocity dispersions in these cases are higher than expected in ΛCDM, leading the authors to conclude that a dark disc is present. This is qualitatively consistent with the phantom disc predicted by MOND, but more detailed analysis is required to check if MOND can explain the vertical dynamics of these galaxies. Even if ΛCDM underpredicts their $g_z$, it is possible that MOND will overpredict their $g_z$.

We conclude that the vertical dynamics of disc galaxies do not currently break the CDM-MOND degeneracy, but this remains a promising avenue for further investigation.

### 3.2. Elliptical Galaxies and Dwarf Spheroidals

Disc galaxies have historically been the focus of attempts to test MOND due to their relative simplicity from a theoretical perspective and the relative ease of interpreting the observations (Section 3.1). In principle, the gravitational fields of elliptical galaxies can be calculated in a similar manner to thin discs (Equation (17) or (20)). The task is actually simpler for spherical galaxies. Considering them makes it possible to study the missing gravity problem in dwarf galaxies, which tend to be pressure-supported and often have

quite low internal accelerations. In this section, we consider what can be learned from the internal dynamics of elliptical and dwarf spheroidal galaxies using a variety of tracers to probe their potential.

### 3.2.1. Velocity Dispersion

A test particle on a near-circular orbit has a well-known relation between rotational velocity $v_c$ and gravity $g$, namely $g = v_c^2/r$. The relation is more complicated for a pressure-supported system, where the typically eccentric orbit of each particle means that its instantaneous velocity is not a meaningful constraint on the potential. Instead, one must consider the problem statistically. This can be done using the MOND generalization of the virial theorem [115]. Its equation 14 states that for an isolated system in the deep-MOND limit with an isotropic velocity dispersion tensor, the mass-weighted LOS velocity dispersion

$$\sigma_{\text{LOS}} = \sqrt[4]{\frac{4GMa_0}{81}}. \tag{49}$$

This applies to any modified gravity theory of MOND [116], thereby providing a good starting point for estimating what MOND predicts about an isolated pressure-supported low-acceleration system. The 3D velocity dispersion is $\sigma_{\text{LOS}}\sqrt{3}$, so formulae in the literature may differ depending on which measure of velocity dispersion is intended. Observational studies generally focus on $\sigma_{\text{LOS}}$. As with Newtonian gravity, the application of the virial theorem is subject to the usual caveats, for instance that the system should be in equilibrium. Moreover, the velocity dispersion tensor could be anisotropic, which would require a more complicated treatment such as solving the Jeans equations (Equation (53) in spherical symmetry). In this case, the 3D velocity dispersion would still be $\sqrt[4]{4GMa_0/9}$.

An important aspect of Equation (49) is that the MOND dynamical mass $\propto \sigma_{\text{LOS}}^4$, which is much steeper than the Newtonian scaling of $\sigma_{\text{LOS}}^2$. As a result, the early claim of a severe discrepancy between MOND and the observed $\sigma_{\text{LOS}}$ for some dwarfs [234] was later resolved through better measurements and by properly taking into account their uncertainties [235]. This is just one of many examples where claims to have falsified MOND were later shown to be premature, with better data actually agreeing quite well with MOND.

The premature claims sometimes rely on an incomplete treatment of MOND, which reduces to Equation (49) only for isolated deep-MOND systems in virial equilibrium with an isotropic velocity dispersion tensor. The authors of [236] used $N$-body simulations conducted by [237] to overcome two important limitations of Equation (49), namely the assumption of having an isolated system in the deep-MOND limit. The authors of [236] analytically fit the earlier $N$-body results in their Section 2, coming up with a very useful series of formulae to predict $\sigma_{\text{LOS}}$ for a system where the external field is non-negligible and where the gravity is a possibly significant fraction of $a_0$. Their formulae yield the correct asymptotic limits. Though the underlying $N$-body simulations are based on an interpolating function with a sharper transition between the Newtonian and Milgromian regimes than Equation (15), this should have only a modest effect on the results, and even then only if the typical gravity is close to $a_0$ [237]. In this case, we expect $\sigma_{\text{LOS}}$ to be slightly higher with a more realistic interpolating function.

It is also possible to estimate how the $\sigma_{\text{LOS}}$ implied by Equation (49) should be altered to include departures from isolation and the deep-MOND limit, but without performing $N$-body simulations [238]. While their approach is less rigorous, it perhaps gives a more intuitive understanding of the corrections, which in their work are based on Equation 59 of [72]. This uses the AQUAL formulation (Equation (17)) with the simple interpolating function (Equation (14)). Their prescription is to solve for the internal gravity $g_i$ implicitly using

$$(g_i + g_e)\mu\left(\frac{g_i + g_e}{a_0}\right) = g_{N,i} + g_e\mu_e, \tag{50}$$

with $i$ and $e$ subscripts denoting internal and external contributions to the gravity as before.

The numerical values given by this approach are very similar to those obtained by [236] based on fitting $N$-body results, as demonstrated in their Figure 3 (reproduced here as our Figure 5). This shows $\sigma_{LOS}$ for a spherical Plummer distribution of stars with $M = 2 \times 10^8 \, M_\odot$ for different internal and external gravitational fields, considering both Equation (50) and the fit of [236] to the numerical AQUAL simulations of [237], which used the standard interpolating function $\mu = x/\sqrt{1 + x^2}$. The approaches broadly agree, but a systematic offset is apparent when the internal and external gravity are both very weak. The reason is that the angle-averaged radial gravity should be boosted by a factor of $\pi/(4\mu_e)$ if the EFE dominates, a consequence of the AQUAL EFE-dominated potential being [110]:

$$\Phi = -\frac{GM}{\mu_e r \sqrt{1 + L_e \sin^2\theta}}. \tag{51}$$

This is analogous to the QUMOND result in Equation (34), with $L_e$ being the logarithmic derivative of $\mu$ with respect to its argument. Since $\mu_e \nu_e \equiv 1$, there are only slight differences between the two approaches, e.g., the factor of $\pi/4$ in AQUAL becomes $5/6$ in QUMOND. Another useful relation is $(1 + L_e)(1 + K_e) = 1$ (Equation 38 of [111]). The EFE-dominated solutions in QUMOND and AQUAL were also compared in [63].

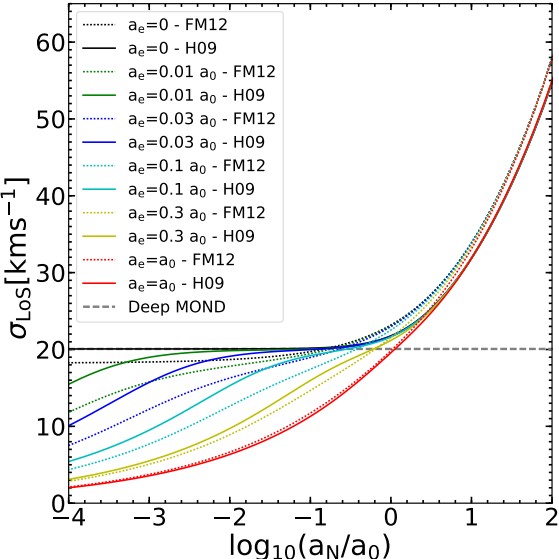

**Figure 5.** The $\sigma_{LOS}$ of a stellar Plummer sphere [239] with mass $M = 2 \times 10^8 M_\odot$ for a range of internal Newtonian gravity $a_N \equiv GM/(2r_h^2)$, which is varied by altering the half-mass radius $r_h$. Results for different external field strengths are shown with different coloured lines, as indicated in the legend. The solid lines show a fit to numerical AQUAL results [237], while the dotted lines are based on the ansatz proposed by [72] in their equation 59. The dashed grey horizontal line shows the isolated deep-MOND result (Equation (49)). For a different-mass galaxy, the values shown here should be scaled $\propto \sqrt[4]{M}$. Notice how the ansatz of [72] systematically underpredicts the numerical results when the internal and external gravity are both weak. The reason for this deficiency is explained analytically in the text. Reproduced from Figure 3 of [236].

However, the factor of $\pi/4$ is not correctly obtained from Equation (50) in the appropriate asymptotic limit $g_i \ll g_e$. By linearizing the $\mu$ function, it is straightforward to show that Equation (50) reduces to

$$\mu_e(1 + L_e)g_i \; = \; g_{N,i}. \tag{52}$$

Since $L_e \to 1$ in the deep-MOND limit, the approach of [72,238] implies $g_i = g_{N,i}/(2\mu_e)$, yielding a factor of 0.5 where there should be $\approx 0.8$ in AQUAL or QUMOND. An extra factor of $0.8/0.5 = 1.6$ in the gravity should translate to a factor of $\sqrt{1.6}$ in the velocity dispersion, so we expect $\sigma_{\mathrm{LOS}}$ to be $\approx 30\%$ higher in numerical simulations. This is indeed roughly the case in Figure 5. We therefore recommend using the fit of [236] to numerical determinations of $\sigma_{\mathrm{LOS}}$ as shown in this Figure. Its results can be used directly if a curve is available for the appropriate $g_e$, bearing in mind that the plotted $\sigma_{\mathrm{LOS}}$ should be scaled by $\sqrt[4]{M/(2 \times 10^8 M_\odot)}$. The MOND boost to the Newtonian $\sigma_{\mathrm{LOS}}$ can also be estimated by applying Equation (38) to find $\bar{\nu} \equiv \bar{g}_r/g_{N,r}$, the average factor by which the Newtonian radial gravity is enhanced in MOND. The enhancement to $\sigma_{\mathrm{LOS}}$ should then be $\approx \sqrt{\bar{\nu}}$, but this approach has never been demonstrated.

MOND fares well with $\sigma_{\mathrm{LOS}}$ measurements of M31 satellites, where the EFE sometimes plays an important role [240,241]. For the classical MW satellites, MOND agrees reasonably well in the sense that the luminosity profile and global velocity dispersion are mutually consistent [242]. For fainter satellites, care must be taken to ensure that the analysed object would be tidally stable in a MOND context (Section 5.1). Restricting attention to galaxies which should be tidally stable in MOND, it agrees reasonably well with the observed $\sigma_{\mathrm{LOS}}$ values of the classical MW satellites [243] and dwarf spheroidal satellites in the LG more generally [244]. $N$-body simulations show that the slightly problematic MW satellite Carina is in only mild tension with MOND because it requires $M_\star/L_V = 5.3 - 5.7$, somewhat higher than expected from stellar population synthesis modelling [245].

In addition to the satellite galaxies of the MW and M31, the LG also contains non-satellite dwarf galaxies that can be used to test MOND. The authors of [246] predicted $\sigma_{\mathrm{LOS}}$ for the isolated LG dwarfs Perseus I, Cetus, and Tucana to be 6.5, 8.2, and 5.5 km/s, respectively, with an uncertainty close to 1 km/s in all cases due to the uncertain $M_\star/L$ (see their Section 3). Observationally, $\sigma_{\mathrm{LOS}}$ of Perseus I is constrained to $4.2^{+3.6}_{-4.2}$ km/s, with a 90% confidence level upper limit of 10 km/s [247]. The reported $\sigma_{\mathrm{LOS}}$ for Cetus is $8.3 \pm 1.0$ km/s [248] or $11.0^{+1.6}_{-1.3}$ km/s [249]. The latest measurements for Tucana indicate that its $\sigma_{\mathrm{LOS}} = 6.2^{+1.6}_{-1.3}$ km/s [250], with the more careful analysis and larger sample size allowing the authors to rule out earlier claims that $\sigma_{\mathrm{LOS}} > 10$ km/s [251,252]. In all three cases, there is good agreement with the a priori MOND prediction, which is just Equation (49) as these dwarfs are quite isolated [246].

A more recent study considered a much larger sample of LG dwarfs, though again restricting to those which should be immune to tides [253]. Those authors defined a parameter $\beta_\star$ by which the $\sigma_{\mathrm{LOS}}$ of a dwarf galaxy would need to be scaled up for it to fall on the BTFR. In MOND, comparison of Equations (46) and (49) shows that we expect $v_f = \sigma_{\mathrm{LOS}} \sqrt[4]{81/4}$ for an isolated dwarf at low acceleration, which implies that scaling the velocity dispersions by $\beta_\star = 2.12$ should reconcile them with the BTFR. After first establishing that the BTFR holds for 9 rotationally supported LG galaxies not in the SPARC sample that underlies Figure 3, the authors then obtained $\beta_\star$ observationally. The values scatter in a narrow range around a median value of 2 if we assume that $M_\star/L_V = 2$. The median $\beta_\star$ reaches the MOND prediction of 2.12 if we assume a slightly higher $M_\star/L_V = 2.5$, which is quite reasonable for the typically old dwarf galaxies in the LG. Alternatively, the velocity dispersions could be higher than reported by 6% on average, which is also reasonable given the typical uncertainties and the sample size. Therefore, the main conclusion of [253] was that MOND is able to account for the velocity dispersions of LG dwarfs that should be immune to tides, which is necessary to allow an equilibrium

virial analysis. Since their sample generally avoids dwarfs near the major LG galaxies, the EFE should not have significantly influenced their results.

The application of Equation (49) to the galaxy known as NGC 1052-DF2 (hereafter DF2) has received considerable attention because the observational estimate is significantly lower [254]. However, their claim to have falsified MOND was swiftly rebutted in a brief commentary on the work because it did not consider the EFE from NGC 1052, which is at a projected separation of only 80 kpc [255] for a distance to both galaxies of 20 Mpc [256]. In addition to this deficiency, several choices were made by [254] which pushed the observational estimate of $\sigma_{\rm LOS}$ to very low values, worsening the discrepancy with Equation (49). One of the most serious problems unrelated to the misunderstanding of MOND is that the statistical methods used to infer $\sigma_{\rm LOS}$ were not well suited to the problem. Using instead basic Gaussian statistics returns a higher $\sigma_{\rm LOS}$ [236,257], though still below the result of Equation (49). Another issue with the work of [254] is their claim to have discovered DF2 in an earlier work, even though it was clearly marked in plate 1 of [258] and had the alternative designation [KKS2000]04. It is therefore clear that care must be taken before testing MOND with the photometry and observed $\sigma_{\rm LOS}$ of a dwarf galaxy.

DF2 is very gas-poor [259,260], thereby providing model-independent evidence that it feels a non-negligible external gravitational field from a massive host galaxy. This is almost certainly the nearby NGC 1052, whose projected separation of only 80 kpc implies an actual separation of $\approx 100$ kpc. This makes the orbital period rather short, causing the external field to be time-dependent on a scale not much longer than the internal dynamical time of DF2. Moreover, tides on DF2 should in principle also be considered as the virial theorem cannot be applied to an object undergoing tidal disruption, as seems to be the case for NGC 1052-DF4 [261,262]. These complications were handled using fully self-consistent *N*-body simulations with POR [102] in which DF2 was put on an orbit around NGC 1052 [236]. Those authors also conducted simulations with N-MODY [263] that include the EFE but not tides. The results of these simulations are shown in Figure 6, which reproduces Figure 5 of [236]. Their work clarified that non-equilibrium memory effects would be quite small in the case of DF2 for a wide range of plausible assumptions regarding its orbit around NGC 1052 (compare the POR and analytic results in the top panels). Such effects might be more significant elsewhere, but DF2 can be approximated as being in equilibrium with the present external field from NGC 1052. After pericentre passage, $\sigma_{\rm LOS}$ of the dwarf takes time to rise towards its equilibrium value. This memory effect [237,264] is partially counteracted by tides inflating $\sigma_{\rm LOS}$ (compare POR and N-MODY results in Figure 6). Before pericentre passage, both effects are expected to work in tandem to inflate $\sigma_{\rm LOS}$ above the equilibrium for the local EFE. The simulations of [236] therefore support earlier analytic and semi-analytic estimates that show the observed $\sigma_{\rm LOS}$ of DF2 is well accounted for in MOND given its observed luminosity, size, and position close to NGC 1052 [238,255]. The subsequently measured stellar body $\sigma_{\rm LOS}$ of $8.5^{+2.3}_{-3.1}$ km/s [265] is consistent with the globular cluster-based estimate of $8.0^{+4.3}_{-3.0}$ km/s shown in Figure 6.

Equation (49) and its generalization to higher acceleration systems feeling a non-negligible EFE (Figure 5) only pertain to the system as a whole. This is suitable for unresolved observations, but in general we expect that $\sigma_{\rm LOS}$ varies within a system. Assuming that it is spherically symmetric, collisionless, and in equilibrium, it would satisfy the Jeans equation

$$\frac{d\left(\rho\sigma_r{}^2\right)}{dr} + \frac{2\beta\rho\sigma_r{}^2}{r} = -\rho g\,, \quad \text{where } \beta \equiv 1 - \frac{\sigma_t{}^2}{2\sigma_r{}^2}\,, \tag{53}$$

$\rho$ is the tracer density, $r$ is the radius, $\sigma_r$ is the velocity dispersion in the radial direction, $\sigma_t$ is the total velocity dispersion in the two orthogonal (tangential) directions (hence the factor of 2), $\beta$ is the anisotropy parameter, and $g > 0$ is the inward gravity. The results are insensitive to an arbitrary rescaling of $\rho$.

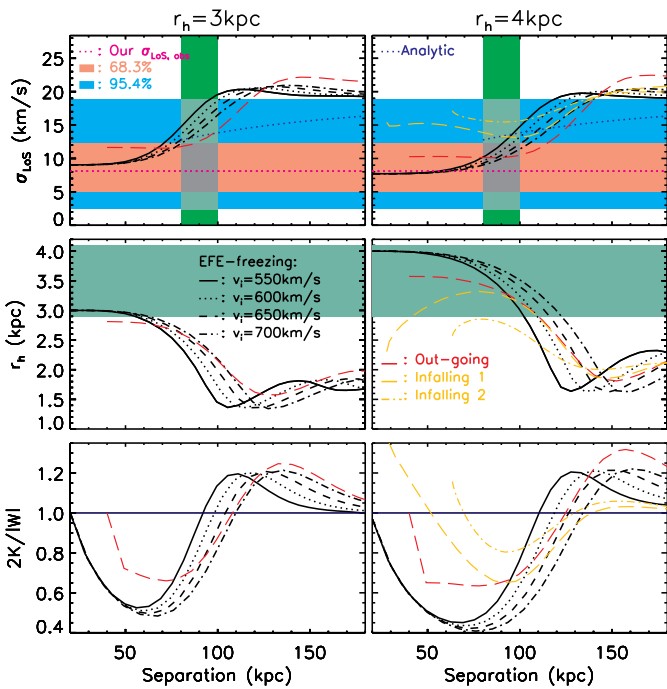

**Figure 6.** Results of numerical MOND simulations of DF2, showing $\sigma_{\text{LOS}}$ (**top**), half-mass radius $r_h$ (**middle**), and the virial ratio $2K/|W|$ between twice the kinetic energy $K$ and the magnitude of the virial $W \equiv \sum_i m_i(\mathbf{r}_i \cdot \mathbf{g}_i)$ in the dwarf reference frame (**bottom**), where $i$ labels different particles with mass $m_i$ at position $\mathbf{r}_i$ experiencing gravity $\mathbf{g}_i$ in excess of the external field from NGC 1052. The initial $r_h$ is 3 kpc (**left**) or 4 kpc (**right**). The vertical green shading in the top panels shows the range of likely distances between DF2 and NGC 1052 given their projected separation is 80 kpc at a heliocentric distance of $\approx$20 Mpc [256]. The horizontal shaded bands in the top panels show the observed $\sigma_{\text{LOS}}$ and its uncertainty based on ten globular clusters, while the shaded region in the middle panels covers the range between their $r_h$ and the slightly smaller stellar $r_h$. The dotted black line in each top panel shows the semi-analytic prediction based on the fitting formula illustrated in Figure 5. The other black lines show results of N-MODY simulations that consider the time-varying EFE but neglect tides on the dwarf, which is started at a pericentre of 20 kpc and launched with the velocity indicated in the middle left panel's legend. The dashed red lines show POR models launched outwards from a pericentre of 40 kpc. The golden models in the right panel show the most realistic POR models started at 200 kpc and launched inwards towards a pericentre of 64 kpc (dot-dashed) or 28 kpc (dashed), with the latter providing a good fit to the observables. In all cases, only the outgoing portion of the trajectory after pericentre passage is shown, so time flows from left to right. Reproduced from Figure 5 of [236].

Similarly to RCs, we expect the velocity dispersion profile to flatten at large radii in MOND rather than continue on a Keplerian decline, as would occur in Newtonian gravity. Thus, the flattened outer velocity dispersion profiles of some galactic globular clusters were used to argue in favour of MOND [266,267]. However, the outskirts of globular clusters would contain stars that are no longer gravitationally bound to the cluster but continue to follow its galactocentric orbit. These unbound 'potential escaper' stars can mimic an outer flattening of the velocity dispersion profile [268], so MOND might not be the only explanation. Moreover, we do not expect significant MOND effects in a nearby globular cluster due to the galactic EFE (Section 2.4).

A more distant globular cluster would be less affected by the EFE. If in addition its internal acceleration is small, then the cluster could serve as an important test of MOND [237,269]. One such example is NGC 2419, which is 87.5 kpc away [270]. Analysis of its internal kinematics led to the conclusion that it poses severe problems for MOND [271,272]. However, it was later shown that allowing a radially varying polytropic equation of state yields quite good

agreement with the observed luminosity and $\sigma_{\mathrm{LOS}}$ profiles [273,274]. More importantly, it is very misleading to consider the formal uncertainties alone because there is always the possibility of small systematic uncertainties such as rotation within the sky plane.

Another system with known $\sigma_{\mathrm{LOS}}$ profile is the ultra-diffuse dwarf galaxy DF44. Its global $\sigma_{\mathrm{LOS}}$ was claimed to exceed the MOND prediction [275] based on earlier observations [276]. However, subsequent observations reduced the estimated $\sigma_{\mathrm{LOS}}$ and revealed its radial profile [277]. This is reasonably consistent with MOND expectations [278]. Joint modelling of its internal dynamics and its star formation history in the IGIMF context showed DF44 to be in mild $2.40\sigma$ tension with MOND [279]. Its internal acceleration of $\approx 0.2\,a_0$ (see their Section 5) makes this a non-trivial success, though still somewhat subject to the mass-anisotropy degeneracy (Equation (53)).

The agreement of MOND with both the low $\sigma_{\mathrm{LOS}}$ of DF2 and the high $\sigma_{\mathrm{LOS}}$ of DF44 despite a similar baryonic content in both cases is due to the EFE playing an important role in DF2 but not in DF44. The stronger external field on DF2 can be deduced independently of MOND based on comparing their environments. We therefore see that dwarf spheroidal galaxies can provide strong tests of MOND due to their weak internal gravity, but this makes them more susceptible to the EFE and to tides. The lower surface density also makes accurate observations more challenging, but it should eventually be possible to test the predicted velocity dispersions of, e.g., the 22 ultra-diffuse dwarf galaxies considered in [280]. Note that since they used Equation (50), the predictions are likely slight underestimates if the EFE is important (Figure 5), which applies to the vast majority of their sample.

Observations are easier for a massive elliptical galaxy, but the central region tends to be in the Newtonian regime. Even so, MOND does still affect the overall size of the system. This is because if we assume that it is close to isothermal and in the Newtonian regime, then we run into the problem that a Newtonian isothermal sphere has a divergent mass distribution [281], which is also the case for the NFW profile expected in $\Lambda$CDM [282]. However, isothermal spheres in MOND have a finite total mass [283], as do all polytropes in the deep-MOND regime [284]. Thus, a nearly isothermal system that formed with a radius initially much smaller than its MOND radius (Equation (18)) should expand until it reaches its MOND radius. At that point, further expansion would be difficult due to the change in the gravity law. This argument does not apply to low-density systems initially larger than their MOND radius, but we might expect that a wide variety of systems were initially smaller. These would end up on a well-defined mass–radius relation. Elliptical galaxies do seem to follow such a relation, which perhaps explains why their internal accelerations only exceed $a_0$ by at most a factor of order unity [285]. Moreover, the DM halos of elliptical galaxies inferred in a Newtonian gravity context have rather similar scaling relations to the PDM halos predicted by MOND [286].

The stars in an elliptical galaxy can serve as tracers of its potential even if they cannot be resolved individually. This has led to observational projects such as ATLAS$^{3D}$ [287] and SDSS-IV MaNGA [288], an extension of the original Sloan Digital Sky Survey (SDSS [289]). The so-obtained stellar $\sigma_{\mathrm{LOS}}$ profiles of 19 galaxies reveal a characteristic acceleration scale consistent with $a_0$ as determined using spirals [290]. While the uncertainties are larger and their study consists of 387 data points rather than the 2693 in SPARC [167], it is still important to note that MOND works fairly well in elliptical galaxies.

This success may seem trivial due to the high accelerations. However, there is no reason for physical DM particles to avoid high-acceleration regions. As a result, it is entirely possible that the central regions of elliptical galaxies should contain more DM in the $\Lambda$CDM picture than PDM in the MOND picture. Indeed, the rather small amounts of missing gravity in the central regions of elliptical galaxies are actually problematic for $\Lambda$CDM, but this can be explained naturally in MOND if the interpolating function is chosen appropriately [140]. This is shown in their Figure 3, which we reproduce here as the left panel of our Figure 7. When $g_N \approx 10\,a_0$, the predicted extra gravity from the DM component is too large to sit comfortably with the data if a Burkert profile is assumed for the halo. The other panels in Figure 3 of [140] show that the discrepancy is worse for the NFW profile predicted in $\Lambda$CDM [282]. In MOND, it is possible to

accommodate the observations with a sufficiently sharp transition from the Milgromian to the Newtonian regime at these high accelerations, e.g., with an exponential cutoff to the MOND corrections (Equation (48)).

That ΛCDM yields too much DM in the central regions of massive galaxies is also evident from a different semi-analytic prescription to assign a baryonic component to purely DM halos [291]. As shown in their Figure 2 and stated in their equation 19, their simulated RAR is well fit by the interpolating function

$$
\nu = \left[ \frac{1}{2} + \sqrt{\frac{1}{4} + \frac{1}{y^{0.8}}} \right]^{1/0.8}, \quad y \equiv \frac{g_N}{a_0}.
\tag{54}
$$

**Figure 7.** **Left**: The RAR defined by disc galaxies and strong gravitational lenses ([292]; see also Section 3.4), shown here on logarithmic axes as the relation between $g_N$ of the baryons and the amount by which the observationally inferred gravity exceeds this (solid black points with uncertainties). The fit to the data is shown as a solid black line (subtracting 1 from Equation (48)). The light purple squares show results from 500 Monte Carlo trials of randomly selected points at radii of (0.01–2) effective radii in galaxies with a virial mass of $(10^{11}–10^{12})M_{\odot}$. A Burkert profile is assumed for the halo, with properties drawn from a ΛCDM distribution. The baryonic component is assigned using abundance matching [293]. The dotted curves show the radial run for three individual galaxies. The binned Monte Carlo results (purple squares with error bars) indicate that a significant contribution from the halo is expected at high $g_N$, at odds with observations. This discrepancy is more severe for the NFW profile (not shown here, but shown in original publication). Reproduced from Figure 3 of [140]. **Right**: Similar to the left panel, but now with the ΛCDM results (Equation (54); solid red line) obtained from a different semi-analytic prescription [291]. The dotted red lines show their estimated intrinsic dispersion of 0.2 dex, which corresponds to a smaller uncertainty of 0.075 dex in the total $g$ (see their Section 4.1). The mean simulated relation is known far more precisely due to the large number of mock galaxies they considered. The mean observed relation (black points with error bars) is from Table 1 of [140].

We use the right panel of Figure 7 to show this as a relation between $g_N$ and the halo contribution $g_h \equiv g - g_N$. This allows a comparison with the binned observational results from Table 1 of [140]. The dotted red lines show the estimated intrinsic dispersion of 0.2 dex in the simulated $g_h$ (Section 4.1 of [291]). The uncertainty in the mean simulated relation is much smaller due to a very large sample size. As a result, the simulated RAR is discrepant with observations at the high-acceleration end. In particular, the extra halo contribution must peak at $g_N \approx 3 a_0$ and start dropping thereafter, but the simulation results indicate that $g_h$ continues rising with $g_N$. This is essentially the same result as shown in the left panel of Figure 7, but with a different "baryonification" scheme [294]. Another issue is that the intrinsic dispersion in the total $g$ at fixed $g_N$ is expected to be 0.075 dex [291], but observationally "the intrinsic scatter in the RAR must be smaller than 0.057 dex" (Section 4.1

of [170]). It is therefore clear that the results obtained by [291] are actually very problematic for ΛCDM, contrary to what is stated in their abstract. The main reason seems to be that they only considered data on galaxy kinematics [295], but strong lenses also provide important constraints, especially at the high-acceleration end (Section 2 of [140]; see also Section 3.4). The discrepancy is related to the expected adiabatic compression of CDM halos due to the gravity from a centrally concentrated baryonic component [296,297]. If $M_\star/L$ is higher in elliptical galaxies than typically assumed due to a higher proportion of stellar remnants (as expected with the IGIMF theory), then $g_h$ would have to be smaller still, worsening the discrepancy with ΛCDM expectations.

Planetary nebulae can also serve as bright tracers that should have similar kinematics to the stars. This technique was used in three intermediate luminosity elliptical galaxies, revealing an unexpected lack of DM when analysed with Newtonian gravity [298]. These observations can be naturally understood in MOND due to the high acceleration [299]. The problem was revisited more recently with better data on those three galaxies and newly acquired data on four more galaxies [300]. Their analysis confirmed the earlier finding that MOND provides a good description of the observed kinematics.

Though fewer in number than stars, satellite galaxies must also trace the gravitational field modulo the mass-anisotropy degeneracy (Equation (53)). This technique was attempted by [243], who found that it is possible to fit the observations of [301] if the velocity dispersion tensor transitions from tangentially biased ($\beta < 0$) in the central regions to radially biased ($\beta > 0$) in the outskirts. This was expected a priori from MOND simulations of dissipationless collapse [302]. However, the mass-anisotropy degeneracy and possible LOS contamination mean that this is not a very sensitive test of the gravity law.

The satellite region can sometimes be probed using an X-ray halo, whose temperature and density profile allow for a determination of the gravitational field assuming hydrostatic equilibrium. The uncertainty is reduced somewhat as we expect $\beta = 0$ for collisional gas. This technique has been applied to NGC 720 and NGC 1521, yielding good agreement with MOND over the acceleration range $(0.1 - 10)a_0$ [303]. The fit is less good in the central few kpc of NGC 720, but it is quite likely that the assumption of hydrostatic equilibrium breaks down here [304], possibly due to feedback from stars and active galactic nuclei. Additional studies of X-ray halos around elliptical galaxies are reviewed in [305], with the results for 9 well-observed galaxies summarized in Figure 8 of [211]. We reproduce this as our Figure 8, showing how the results fall on the RAR traced by the RCs of spiral galaxies. The RAR is thus not unique to rotationally supported systems or to rotational motion, nor is it confined to thin disc galaxies.

### 3.2.2. Rotation of a Sub-Dominant Component

The mass-anisotropy degeneracy can be broken by considering a thin rotating gas disc. By definition, this can only be a sub-dominant component of an elliptical galaxy. Even so, such a disc would provide an excellent way to measure $g_r$. This is possible in 16 early-type galaxies from the ATLAS$^{3D}$ survey [306]. Using 21 cm HI observations extending beyond 5 effective radii, those authors showed that the analysed galaxies very closely follow the BTFR defined by spiral galaxies. The BTFR was again the most fundamental relation (smallest scatter), not, e.g., the relation between $M_\star$ and $v_f$. The results for these 16 galaxies are shown on an RAR diagram in Figure 8 of [211], revealing good agreement with the RAR defined by disc galaxies.

More recently, HI RCs have been obtained for three lenticular galaxies extending out to 10–20 effective radii [307]. Those authors used 3.6 μm photometry to reduce uncertainty on $M_\star/L$. Their Figure 8 (reproduced here as our Figure 9) shows that the 3 analysed galaxies fall on the spiral RAR, with the dataset probing $g_N$ out to $\approx 1.5$ dex either side of $a_0$ (the range in $g$ is slightly smaller).

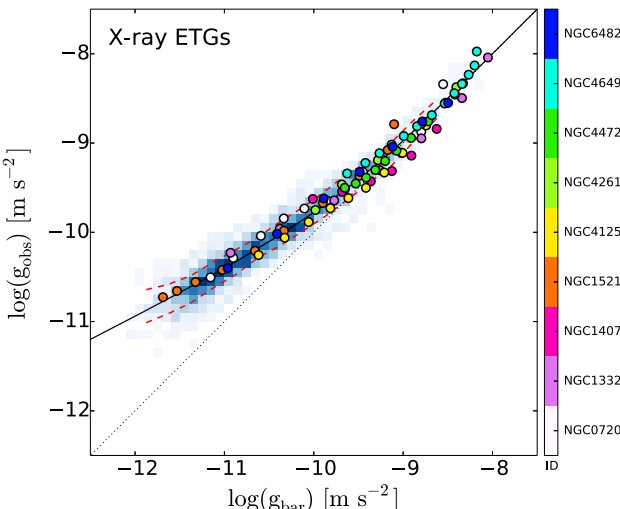

**Figure 8.** The elliptical galaxy RAR defined by the relation between $g_N$ and the gravity required for hydrostatic equilibrium of the X-ray halo (Equation (53) with $\beta = 0$). Results are shown as filled circles, with the colour indicating the galaxy (see the legend). The RAR of disc galaxies (Figure 3) is indicated by the shade of blue in the background, with the fit to it (Equation (48); solid black line) and the width of the distribution at fixed $g_N$ (dotted red lines) shown similarly to that figure. Notice that disc galaxies define the same RAR as X-ray halos around elliptical galaxies. Reproduced from Figure 8 of [211] by permission of Federico Lelli and the American Astronomical Society.

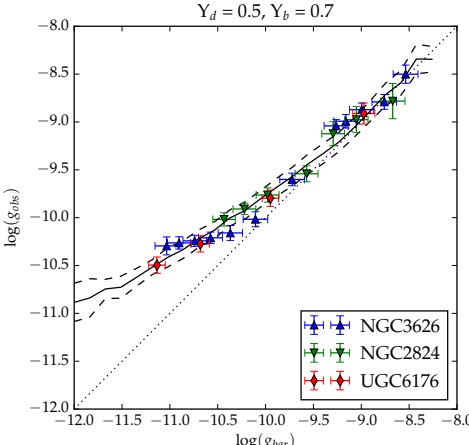

**Figure 9.** The RAR defined by the gas RC in three lenticular galaxies (points with error bars), as determined from CO in the inner regions and deep HI observations in the outer regions. The colour of each data point indicates which galaxy it comes from (see the legend). Notice the similarity to the mean RAR of disc galaxies (solid black line) and its standard deviation (dashed black lines). Reproduced from Figure 8 of [307] with permission from Astronomy & Astrophysics.

Perhaps because these results are still very recent, not much attention has been paid to whether ΛCDM can get the same tight RAR in elliptical and lenticular galaxies as that observed in spirals. Since the star formation histories are likely to have been different, the proportion of baryons lost by feedback should also be different. This makes it difficult to understand how the same tight RAR arises in galaxies of different Hubble types. However, this was the a priori prediction of MOND.

### 3.3. Observational Signatures of the EFE

In this section, we review the observational evidence for the EFE, which theoretically is an integral part of MOND (Section 2.4). A classic example of the EFE is when a satellite galaxy orbits around its host. The EFE was included in an early analysis of $\sigma_{LOS}$ in 7 galactic

satellites, some of which are dominated by the EFE [235]. Neglecting the EFE would enhance the self-gravity and thus reduce the MOND dynamical $M/L$, which could be problematic in the case of Fornax because the required $M/L$ is already on the low side.

The EFE should also be present in satellites of M31, which is near enough that some fairly detailed observations are now available. The authors of [241] considered the dynamics of three matched pairs of satellites which are photometrically similar but differ in their distance from M31 or in their size, leading to the EFE being less important in one satellite than for the other (see their Section 2.4). Since the considered dwarfs are deep in the MOND regime, neglecting the EFE would mean that $\sigma_{\mathrm{LOS}}$ depends only on the total luminosity, so it should be the same for both galaxies in each matched pair (Equation (49)). This is no longer the case once the EFE is considered. The difference between the predicted $\sigma_{\mathrm{LOS}}$ with or without the EFE was too small to be tested by observations in one matched pair, but for the other two pairs, the data seemed to indicate that the satellite more dominated by the EFE does indeed have a lower $\sigma_{\mathrm{LOS}}$.

Another strong hint for the EFE comes from the internal dynamics of the MW satellite Crater II at a galactocentric distance of $d \approx 120$ kpc [308]. Neglecting the galactic EFE and assuming that Crater II has $M_\star/L_V = 2$ with this being uncertain by a factor of 2, its predicted $\sigma_{\mathrm{LOS}}^{\mathrm{vir}} = 4.0_{-0.6}^{+0.8}$ km/s, but this decreases to $2.1_{-0.6}^{+0.9}$ km/s once the EFE is included [309]. The 'vir' superscript indicates that their calculation neglects the role of tides, which can be important for an ultra-diffuse galaxy such as Crater II because its half-mass radius $r_h = 1.07 \pm 0.08$ kpc (Table 4 of [308]) despite a very low $M_s = 3.2 \times 10^5\,M_\odot$ for $M_\star/L_V = 2$. Tides (discussed further in Section 5.1) can generally be expected to inflate $\sigma_{\mathrm{LOS}}$ further (Figure 4 of [310]). The amount by which this occurs can be estimated by considering the opposite limit in which self-gravity is negligible and the observed stars are considered to be travelling along independent galactocentric orbits. $\sigma_{\mathrm{LOS}}$ would then be set by the galactic RC and the range of orbital phases that we observe, which is roughly given by the angular size $r_h/d$. Using this to scale the galactic $v_c \approx 200$ km/s, we get that $\sigma_{\mathrm{LOS}}^{\mathrm{tid}} \approx v_c r_h/d = 1.8$ km/s, though this would vary somewhat around the orbit given the high eccentricity [311]. If we assume that the actual $\sigma_{\mathrm{LOS}}$ should be found by adding $\sigma_{\mathrm{LOS}}^{\mathrm{vir}}$ and $\sigma_{\mathrm{LOS}}^{\mathrm{tid}}$ in quadrature, we get that $\sigma_{\mathrm{LOS}}$ should be at least 3.9 km/s if the galactic EFE has no effect on the self-gravity of a satellite. In this case, our simple estimate shows that tides would be relatively unimportant due to the strong self-gravity of Crater II. However, including the EFE leads to its predicted $\sigma_{\mathrm{LOS}}$ dropping to 2.8 km/s. Observationally, Crater II has $\sigma_{\mathrm{LOS}}^{\mathrm{obs}} = 2.7 \pm 0.3$ km/s [312]. This shows that in MOND, Crater II must feel the galactic EFE. However, its internal kinematics are not a strong test of MOND because we can achieve plausible agreement with observations even if we neglect self-gravity altogether. The observations can thus be considered as putting a useful upper limit on the amount of self-gravity in Crater II.

Detailed predictions for the internal kinematics of dwarfs are difficult to obtain in $\Lambda$CDM, but it should still be possible to statistically predict roughly what $\sigma_{\mathrm{LOS}}$ Crater II should have. The authors of [309] predicted in their Section 2 that $\sigma_{\mathrm{LOS}} = 17.5$ km/s based on an empirical scaling relation [313]. The observed properties of Crater II remain difficult to reconcile with $\Lambda$CDM even if significant tidal disruption is considered, at least if it formed inside a CDM halo [314,315]. This is strongly suggested by the fact that its Newtonian dynamical $M_\star/L_V \approx 50$ [312,316]. If instead Crater II is purely baryonic (e.g., it is a globular cluster), then its large size and high observed $\sigma_{\mathrm{LOS}}$ imply that it must now be past perigalacticon. Proper motion data indicate that this would have been at $33 \pm 8$ kpc (Table 4 of [311]), which combined with the very large mass ratio between Crater II and a CDM-rich MW would have caused complete tidal destruction at that point (Equation (69)). This problem is somewhat alleviated in MOND because a purely baryonic Crater II is much more resilient to tides if gravity is Milgromian rather than Newtonian, making it more plausible that a very diffuse remnant would still be recognisable.

These issues with Crater II highlight that it is generally difficult to identify satellite galaxies that might be significantly affected by the EFE but not by tides, thus remaining amenable to an equilibrium virial analysis. A more promising approach might be to exploit

the differing levels of tidal stability in ΛCDM and MOND by comparing with some model-independent measure of whether a galaxy is affected by tides. This would test the EFE because the tidal radius is necessarily in the EFE-dominated regime. We explore such ideas further in Section 5.1. Another possibility is to construct detailed numerical simulations of tidal streams and compare with observations in order to search for specific signatures of the EFE, in particular an asymmetry between the leading and trailing arms (Section 5.2; see also [317]).

Turning now to disc galaxies, the EFE would cause the outer RC to decline, even though a flat RC is expected for an isolated system. Observational evidence for this was found by [318]. In a major breakthrough, the authors of [319] extended their work by cross-correlating the high-quality SPARC dataset with the large-scale gravitational field [320]. The analysis of [319] considered whether including the EFE leads to better RC fits, bearing in mind that allowing an extra model parameter inevitably improves the fit to some extent. Even so, they found very strong evidence for the EFE. This is clearly evident in their Figure 2, which reveals the expected flat outer RC for relatively isolated galaxies, but galaxies that should experience a stronger EFE clearly have a declining outer RC. These observations cannot easily be fit by instead altering conventional parameters such as the distance and inclination, leading [319] to conclude that they detected the EFE at $8\sigma - 11\sigma$ in some galaxies but not at all in others. The EFE strength inferred from the RC of a galaxy in a more isolated environment is typically much less than for a galaxy in a more crowded environment [321]. These pioneering studies test for the first time whether the strong equivalence principle at the heart of GR remains valid at the low accelerations typical of galactic outskirts. A significant failure of the principle could be analogous to the breakdown of energy equipartition at low temperatures in condensed matter physics (Section 2.2). Departures from GR are inevitable in some cases, so empirically finding precisely which ones could provide important clues to a better theory of gravity that incorporates quantum effects.

It is interesting to consider whether ΛCDM predicts a correlation between the RC of a galaxy and the external field on its barycentre from large-scale structure. While this cannot arise as a fundamental consequence of the gravity law, we might expect that in a high-density region, a CDM halo would typically have a higher density by virtue of forming earlier. At fixed $g_N$, this would make the observed $g$ higher, a result which was recently demonstrated using semi-analytic models [322]. The predicted sign of the EFE is thus opposite to observations [319,321].

In MOND, an important consequence of the EFE is that the point mass gravitational field transitions to an inverse square law once $g_e$ dominates (Equation (37)). Without the EFE, Equation (18) would apply, creating a logarithmically divergent potential (e.g., Equation (19)). Regardless of theoretical issues, an observational consequence of this would be an extremely deep galactic potential from which escape is virtually impossible. Including even a weak EFE would substantially alter this picture. These scenarios can be distinguished using the escape velocity from the solar circle of the MW [107,323]—and indeed the escape velocity curve more generally [324]. This agrees quite well with observations near the solar circle [325] and over galactocentric radii of 8–50 kpc [326], which is another indication that the EFE from more distant masses such as M31 does affect the internal dynamics of the MW in a MOND context.

The EFE can have important effects even in situations where it is sub-dominant to the internal gravity. For instance, the central regions of M33 would not be much affected by the EFE from M31, but even so a hydrodynamical MOND simulation of M33 agrees better with some observables once the EFE is approximately included [106]. A similar situation is apparent in AGC 114905 [195]. In addition to weakening the self-gravity of a system, the EFE also causes the point mass potential to become anisotropic (Equation (34)). If the internal and external gravity are comparable, the potential can even become asymmetric with respect to $\pm g_e$. This can leave a characteristic imprint on tidal streams, as discussed further in Section 5.2. The asymmetry could also cause discs to become warped in an external field [106]. Indeed, the warp of the galactic disc might be due to the EFE from the LMC [113].

Besides the open cluster data discussed in [56], the properties of Ly-$\alpha$ absorbers also require the EFE in a MOND context (Section 8; see also [327]). On large scales, the MOND scenario for the KBC void and Hubble tension requires the EFE to make the void dynamics consistent with observations (Section 9.2.2; see also the inference on the EFE in Figure 4 of [93]).

Similarly to galactic star clusters in the solar neighbourhood of the MW, local wide binary stars would also experience too much self-gravity in MOND if the galactic EFE is neglected. Wide binaries provide a promising test of MOND (Section 11.3). By definition, these systems have low internal accelerations, which would cause a large departure from Newtonian expectations if the binaries were isolated [328]. However, the difference is much smaller once the galactic EFE is considered [111]. The sky-projected separation $r_{sky}$ of each binary is very small compared to its galactocentric radius, so considering the binary in isolation would indeed involve neglecting the galactic EFE for a very wide range of plausible choices on where to put the system's boundary. The authors of [329] showed that Gaia data are strongly inconsistent with the large 'MOND' signal that would be expected if we neglect the galactic EFE, especially for the more widely separated binaries where the internal accelerations are lower (see their Figure 11, reproduced here as our Figure 10). Including the EFE makes the observations consistent with MOND, though they also agree with Newtonian expectations. There have also been many terrestrial experiments at low internal accelerations that show no sign of departure from Newtonian dynamics [330]. We therefore conclude that there are compelling theoretical and empirical reasons for supposing that the EFE is an important aspect of MOND—and probably of the real universe as well.

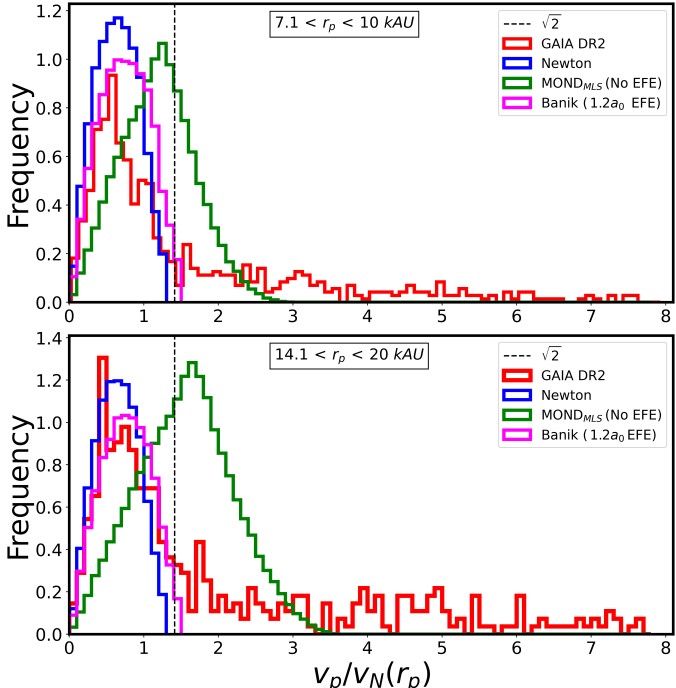

**Figure 10.** Histogram of the sky-projected relative velocity of wide binary stars in the solar neighbourhood divided by the predicted Keplerian velocity at their sky-projected separation $r_p$, shown for different $r_p$ bins as indicated at the top of each panel. Observational results from Gaia data release 2 (red) are compared with the prediction of Newtonian gravity (blue) and MOND (magenta) for $g_{N,e} = 1.2\,a_0$ as appropriate for the solar neighbourhood. The green distribution is for a toy version of MOND which neglects the galactic EFE. The peak of this distribution shifts to the right at larger $r_p$ (bottom panel) because the binary circular velocity no longer declines with separation, analogously to the flat RCs of isolated galaxies. The predicted peak shift is strongly excluded by the data. Note that MOND is not excluded here as the EFE is an integral part of it. Including the EFE considerably narrows the gap between the Newtonian and Milgromian predictions (Section 2.4), causing both to be in reasonable agreement with Gaia data. Adapted from Figure 11 of [329] by its lead author.

### 3.4. Strong Gravitational Lensing

A gravitational lens is said to be strong if it leads to multiple resolved images of the same background source. This definition is somewhat technology-dependent because gravitational microlensing events [24] should yield multiple images which could in principle be resolved with future technology. Nonetheless, the definition is useful because important constraints can be obtained from the angular separation between the multiple images in a strong gravitational lens.

Due to the cosmic coincidence of MOND (Equation (12)), photons involved in strong lensing are necessarily in the Newtonian regime at the point of closest approach if the lens and source are at cosmological distances [331]. This is not true for the entire trajectory, so some MOND enhancement is expected. Moreover, we expect some departure from Newtonian expectations even if $g > a_0$ because the RAR (Figure 3) implies a fairly gradual transition between the Newtonian and Milgromian regimes. The extra light deflection due to the enhanced MOND gravity is expected to be $\approx$15% for the simple interpolating function (Equation (15)).

The best-known examples of gravitational lenses are Einstein rings, where the lens and unlensed source are at almost the same location on our sky. This makes the problem nearly axisymmetric, leading to a wide range of possible photon paths that reach the Earth. If the lens is a massive galaxy, we can constrain its potential using the angular size of the Einstein ring and the redshifts of the source and lens, bearing in mind that the deflection angle and non-relativistic $g$ are related in the same way as in GR (Section 2.6). Due to the typically cosmological distances, it is necessary to assume a background cosmology, with the choice generally being a standard flat background cosmology because this should be appropriate to both $\Lambda$CDM and MOND (Section 9). If the redshift is not too large, the angular diameter distance can be determined with little dependence on cosmology provided the Hubble constant is known.

MOND was shown to be consistent with 10 lenses in the Centre for Astrophysics-Arizona Space Telescope Lens (CASTLES [332]) catalogue of strong lenses [333]. A much larger sample was provided by the Sloan Lens Advanced Camera for Surveys (SLACS) catalogue [334,335]. MOND provides a good fit to all 36 [336], 65 [331], or 57 [292] lenses analysed, with the studies adopting different quality cuts. MOND has also been applied to the time delays in three gravitational lenses [130], yielding broad agreement for a standard background cosmology. As with the deflection angle, the MONDian Shapiro delay [337] is expected to have a standard dependence on the non-relativistic $g$.

### 3.5. Weak Gravitational Lensing

The surface density rarely reaches the threshold required for strong gravitational lensing. Nonetheless, any source is always distorted to some extent by a foreground lens. This regime is known as weak lensing. Since the true shape of the source is rarely known, it is necessary to use statistical techniques to infer how much distortion has been induced by the lens, usually by stacking many sources around many lenses. The idea is that the source (typically a thin disc galaxy) would have its sky-projected major axis oriented in a random direction. In contrast, a lens induces a subtle spacetime distortion that causes the major axes of background galaxies to preferentially align in a certain direction, which for a point mass lens would be the tangential direction. One can imagine that the lens tries to distort background galaxies into an Einstein ring, but is insufficiently powerful to do so.

Weak lensing effects arise because the direction and magnitude of the deflection angle vary over different parts of the source. In the deep-MOND limit around an isolated point mass $M$, the deflection angle in the weak deflection limit follows from Equations (1) and (43).

$$\Delta \widehat{\boldsymbol{n}} \;=\; -\,\frac{2\pi\sqrt{GMa_0}}{c^2}\,\widehat{\boldsymbol{r}}_p \;=\; -\,2\pi\left(\frac{v_f}{c}\right)^2\widehat{\boldsymbol{r}}_p\,, \tag{55}$$

where $\widehat{\boldsymbol{r}}_p$ is the direction from the lensing mass to the photon at closest approach. We used Equation (46) to write the result in terms of $v_f$, which might be more readily observable. Unlike in the Newtonian case, the deep-MOND deflection angle is independent of the impact parameter. As a result, shear arises only due to changes in the direction of the deflection caused by changes in $\widehat{\boldsymbol{r}}_p$, or equivalently because the radial direction is in general different.

Galaxy–galaxy weak lensing was detected at high significance in a stacked analysis of 12 million sources [338]. Their results agree quite well with the MOND prediction (Equation (55)) if the blue lens subsample (presumably spiral galaxies) has $M_\star/L_B$ in the range 1–3, which covers the range expected from stellar population synthesis modelling [339]. The red lens subsample (presumably elliptical galaxies) requires a higher $M_\star/L_B$ of 3–6, in line with expectations for an older population. The data cover out to projected separations of $\approx$280 kpc, thus reaching accelerations down to only a few percent of $a_0$. This is close to the limit beyond which the assumption of isolation is expected to break down due to the EFE from large-scale structure. The consistency of MOND with galaxy-galaxy weak lensing was again demonstrated using 33,613 isolated central galaxies [340]. Recently, an even larger sample was used to obtain similar results [341]. We reproduce their Figure 4 in our Figure 11, showing that different surveys and sample selections give fairly similar results. The very significant amount of missing gravity evident at $g_N \ll a_0$ is correctly reproduced in MOND down to the limit of the data. At the very low acceleration end, there is a hint of a downturn from the RC-based RAR that might be due to the EFE.

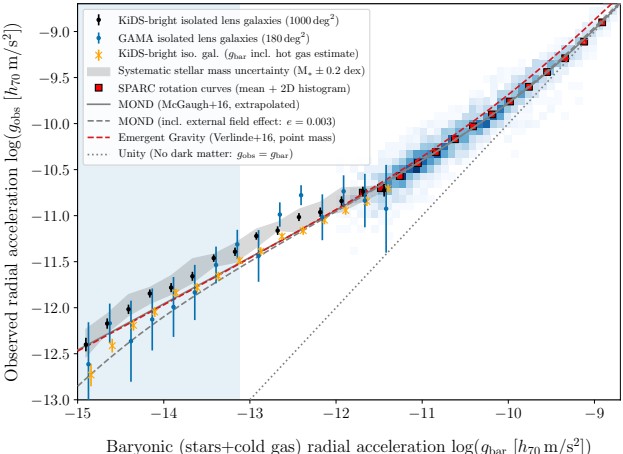

**Figure 11.** The RAR determined from stacked galaxy-galaxy weak gravitational lensing using Equation (43) (points with error bars). Different surveys and sample selections are used, as described in the legend. The relation for disc galaxy RCs is shown towards the upper right using blue shading and red squares for the binned results (similarly to Figure 3), with the fit (Equation (48)) shown as a solid grey line. The dashed grey curve shows the estimated deviation due to the EFE assuming $g_{N,e} = 0.003\,a_0$, reasonable for the fairly isolated galaxies considered here. In MOND, the EFE is expected to affect the results in the blue shaded region on the left (cf. Figure 1). Reproduced from Figure 4 of [341] with permission from Astronomy & Astrophysics.

It has been claimed that the weak lensing signal is anisotropic relative to the lens, which in a Newtonian analysis implies a DM halo flattened similarly to the baryonic disc [342]. The anisotropic signal has not been conclusively detected, but there is strong evidence for it over the radial range 45–200 kpc. Since the disc is typically much smaller, this may pose problems for the MOND scenario. However, there could be an extended halo of hot gas, which we might expect would be flattened along the disc spin axis. Moreover, the EFE from large-scale structure can also induce anisotropy in the potential, even for a point mass lens (Equation (34); see also [66]). Since the orientation of the central disc is likely to correlate with large-scale structure due to the EFE and other effects, it is possible that the PDM halo of a galaxy is typically flattened along the same axis as its disc. Weak

lensing is also sensitive to other structures along the LOS [343], so the signal might not be directly associated with the lens galaxy. Further work would help to clarify if the results are genuinely problematic for MOND, though this is likely to require a cosmological hydrodynamical simulation to assess the large-scale structure and the possibility of a hot gas halo.

### 3.6. Implications for Alternatives to ΛCDM and MOND

The results discussed thus far broadly favour MOND over ΛCDM, but it is helpful to consider other solutions to the missing gravity problem. These are generally much less theoretically mature, partly due to being proposed more recently. Meaningful predictions can still be obtained in some systems, so we briefly consider such 'third ways' in this section.

In the emergent gravity (EG) theory, gravity is an emergent phenomenon arising from some currently not understood microphysical processes [83]. EG is currently not as well developed as MOND, but its application to spherically symmetric non-relativistic systems should be clear. It predicts the same gravitational field around a point mass, but departs from MOND inside the mass distribution of a galaxy due to an extra dependence on the density. Observationally, EG works fairly well with the dwarf spheroidal satellites of the MW [344], but it fails when confronted with kinematic measurements from a larger sample of dwarfs [345,346]. For plausible assumptions about how EG should be applied to disc galaxies, it predicts a distinct hook-shaped feature in the RAR, leading to significant tension with observations [347]. In particular, the required $M_\star/L$ becomes much lower than expectations from stellar population synthesis modelling. Once EG is developed further so that a non-relativistic field equation is available (e.g., [348]), this should be applied to the SPARC sample to check whether the above-mentioned criticisms are valid. EG also faces significant problems explaining the behaviour of ultracold neutrons in the terrestrial gravitational field [349]. While these results have been questioned [350], significant experimental problems for EG were later reaffirmed [351]. The basic assumptions underpinning the thermodynamic approach to gravity advocated in EG also appear not to hold outside of spherical symmetry [352].

Unlike EG, a relativistic modified gravity theory is Moffat gravity (MOG [353]). This is ruled out by the fact that GWs travel at a speed close to $c$ [135]. Focusing on non-relativistic tests, the main problem with MOG is the lack of a fundamental acceleration scale, making it extremely difficult to match the observed RAR. As a result, the MW RC is in strong tension with MOG—for any plausible baryonic distribution, the predicted circular velocity at the solar circle is $v_{c,\odot} < 200$ km/s [354]. However, the observed value is >200 km/s at overwhelming significance (e.g., [118,148]). Even the directly measured acceleration of the solar system barycentre relative to distant quasars proves that $v_{c,\odot} > 200$ km/s at almost $5\sigma$ confidence [1,355]. Moreover, MOG is incompatible with DF44 at $5.49\sigma$ based on a joint fit to its star formation history and observed $\sigma_{\text{LOS}}$ profile [279]. This is evident in their Figure 3, which we reproduce as our Figure 12. It is possible to justify a higher $M_\star/L_I$ with a very short star formation timescale because this would raise $M_\star/L_I$ in the IGIMF context. However, the kinematics are such that an exceptionally short timescale would be required that is very uncommon for dwarfs [356], and even then only a poor fit can be obtained to the observed $\sigma_{\text{LOS}}$ profile. Several other very serious problems for MOG were reviewed in Section 5.2 of [357]. Therefore, MOG is currently ruled out as a viable theory on galactic scales.

More generally, Figure 10 of [72] shows that the proportion of gravity that is unaccounted for on galaxy scales does not correlate with the galactocentric radius of the data point. We have reproduced the relevant panel in our Figure 13, highlighting the difficulty of explaining the tight observed correlation with $g_N$ in a theory that supposes a fixed length $r_0$ beyond which the departure from Newtonian dynamics sets in. Since a flat RC implies that the gravity at larger radii must be $GM/(rr_0)$, such a theory would predict that $M_s \propto v_f^2$. However, observations indicate that $M_s \propto v_f^{3.85\pm0.09}$, with systematic uncertainties perhaps allowing exponents in the range 3.5–4 [160]. This is consistent with the MOND

prediction of 4, but by now it is quite clear that $M_s \propto v_f{}^2$ is strongly incompatible with observations, a fact which has been known for many decades [159]. This also rules out volume corrections to the Hawking–Bekenstein entropy relation as a cause for the missing gravity problem [358].

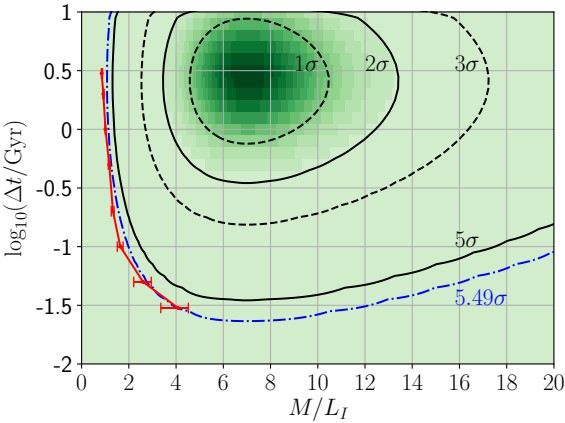

**Figure 12.** Joint inference on the *I*-band $M_\star/L$ of DF44 and the duration $\Delta t$ over which it formed its stars, based on its kinematics [277] assuming MOG and the typical $\Delta t$ values for other galaxies [356], respectively. The probability density is shown using different shades of green, with different confidence regions also plotted as indicated. The red points show the results of stellar population synthesis models in the framework of the integrated galactic initial mass function [187], with the error bars arising from the assumed 20% uncertainty on the luminosity of DF44. This is already factored into the uncertainty on the kinematically inferred $M_\star/L_I$. The lowest tension is obtained for the shortest considered $\Delta t$, in which case there is still $5.49\sigma$ tension between the photometrically expected $M_\star/L_I$ and that required to reproduce the kinematics of DF44 in MOG, also bearing in mind that such a short $\Delta t$ is atypical. Reproduced from Figure 3 of [279] using ApJL author republication rights.

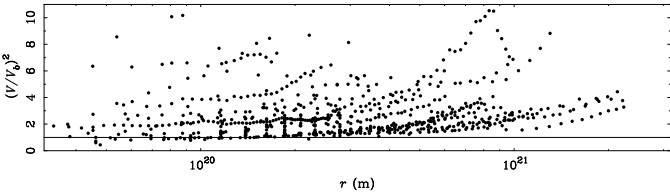

**Figure 13.** The factor by which the radial gravity inferred from a galaxy's RC exceeds the $g_N$ of its detected baryons, shown as a function of the galactocentric radius of the data point. There is no clear correlation, unlike when the dynamical discrepancy is plotted against $g_N$ (Figure 3). Reproduced from Figure 10 of [72].

Another possibility is called superfluid dark matter (SFDM [359,360]). The basic idea is that galaxies reside in DM halos which dominate the mass, but these condense into a superfluid state in the central $\approx$100 kpc, with the baryons dominating the mass in the region probed by RCs (out to $\approx$30 kpc). The gravity between baryons is supplemented by extra forces from phonons propagating in the superfluid, so essentially the missing gravity problem is solved using additional non-gravitational forces. The SFDM parameters have been estimated using galaxy clusters [361] where the superfluid nature of the DM is not very relevant, making the paradigm similar to $\Lambda$CDM. As discussed in Section 5.6 of [362], the LG satellite planes extend beyond plausible estimates for the superfluid core radii of the MW and M31 [360,363]. Since the planar geometry and coherently rotating velocity trend make a primordial origin unlikely, the satellite galaxies in these structures are most likely tidal in origin regardless of the gravitational framework (Section 5.6). In SFDM, the more distant members of these structures would experience just the Newtonian gravity of their baryons, contradicting their high observed $\sigma_{LOS}$ [240,241,244]. Another issue is

that the phonon force acts only on the baryons, so it is difficult to understand why the RAR is also evident in weak gravitational lensing of background galaxies (Section 3.5). It is possible that the gravity of the SFDM (excluding phonon forces) conspires to yield the RC-based RAR at the much larger distances probed by weak lensing [338–341], but this would undermine the aim of SFDM to reproduce the RAR naturally. It has also been argued that since stars on circular orbits in the outer disc would generally move slower than the sound speed of the superfluid, they would emit Cherenkov radiation, with the resulting energy loss causing the orbit to shrink in a small fraction of a Hubble time [364]. Moreover, the fact that MOND effects would persist only in the superfluid region implies that a galaxy does not feel the EFE from another galaxy hundreds of kpc away. This makes it difficult to explain the detection of the EFE by comparing RCs of galaxies in isolated and more crowded environments [319,321]. Even if these issues could be overcome, since the phonon force must be added to the Newtonian gravity, the deep-MOND behaviour can only be reproduced if SFDM reduces to a MOND-like description in which the interpolating function is $\nu = 1 + 1/\sqrt{y}$, where $y \equiv g_N/a_0$. This gives $\nu(1) = 2$, but the simple $\nu$ function (Equation (15)) and the MLS form based on fitting the RAR (Equation (48)) both give $\nu(1) \approx 1.6$. Given the small observational uncertainties (Figure 3), the transition between the Newtonian and low-acceleration regimes appears to be too gradual in the SFDM picture. The inferred value of the acceleration constant could be reduced to address this, but we would need a reduction by a factor of $0.6^2$. There is limited scope for this given that the data go down to very low $g_N$ and thus probe the low-acceleration regime, with the BTFR normalization essentially fixing $a_0$ (e.g., [160]). This argument also applies to EG, and more generally to any theory that attempts to reproduce the RAR by simply adding an extra force to the Newtonian gravity—the extra force would need to have the deep-MOND behaviour (Equation (1)), leading to a MOND-like description with interpolating function $\nu = 1 + 1/\sqrt{y}$. This does not agree with the observed RAR, perhaps explaining why SFDM has difficulty matching galaxy RCs with plausible $M_\star/L$ ratios—especially if lensing constraints are also included [365]. In addition, serious theoretical difficulties have been identified with SFDM when trying to make it covariant [366].

The observationally required boost factor to $g_N$ depends only on $g_N$ across a wide range of galaxies and is large if $g_N \ll a_0$. This regime can be probed either by considering large radii in a high-mass galaxy or the inner regions of a dwarf galaxy with a typically low surface brightness. The two situations are rather different in many ways, but $g_N$ falls short of the actual $g$ by the same factor. Whatever causes this boost to $g_N$ seems to care only about the strength of the gravitational field. We have seen that alternative explanations for this "one law to rule them all" [211] beyond ΛCDM and MOND face serious difficulties given the diversity of the so-called MOND phenomenology, e.g., the fact that it also applies to weak lensing (Section 3.5). In addition, gravitational dipole dark matter [367] and scale-invariant dynamics [368] have been strongly excluded on the basis of solar system ephemerides [369,370]. We therefore focus on MOND as the main alternative to ΛCDM when it comes to addressing the missing gravity problem.

## 4. Disc Galaxy Stability and Secular Evolution

When we observe the RC of a galaxy, there is a direct degeneracy between its matter distribution and the the gravity law. This degeneracy could be broken by the vertical forces, which are expected to be stronger in MOND (Section 3.1.2). This increases the relative importance of disc self-gravity, which would affect the disc's dynamical stability. Consequently, we may gain important insights from the number of spiral arms and properties of the central bar, especially the bar strength and pattern speed. These differences are related to the gravity law, but the other side of this coin is that the matter distribution of a galaxy also differs substantially between the ΛCDM and MOND scenarios. We will see that the combination of these factors leads to distinct predicted outcomes.

### 4.1. Survival of Thin Disc Galaxies

The continued survival of thin disc galaxies provided an important motivation for the hypothesis of CDM halos around galaxies [16,19]. These massive extended halos would cause galaxies to have a large collision cross-section, making mergers quite common due to dynamical friction between overlapping halos [55,371]. Minor mergers with the predicted dark satellites would also be common. These mergers can be very disruptive for any embedded disc galaxies, so their observed frequency and morphology might provide clues to how often mergers actually occur. In particular, the ΛCDM scenario seems to imply that relatively fragile thin disc galaxies are rather rare. Even if gas is accreted after a major merger and subsequently reforms a thin disc, there should still be a central bulge.

Therefore, the high prevalence of bulgeless thin disc galaxies appears difficult to explain [372,373]. Locally, only 4/19 large galaxies ($v_c > 150$ km/s) are ellipticals or contain a significant classical bulge [374]. Moreover, galaxy interactions would eject stars into the halo, but the very low stellar halo mass of M94 [375] and M101 [376] argues against this scenario. Similar results have been obtained around NGC 1042 and NGC 3351 [377]. Even these low-mass halos might not have been accreted. If they formed in situ by ejection of stars from near the galaxy centre, then one possible way to confirm this is a steep radial metallicity gradient [378].

A related problem is that if thin discs form in ΛCDM, they would typically have a significant fraction of material on orbits that deviate substantially from circular motion, e.g., in a pressure-supported bulge [379]. In its own words, "there is reasonably persuasive evidence that simulations of galaxy formation based on the ΛCDM theory with Gaussian initial conditions produce unacceptably large fractions of stars in hot distributions of orbits." Importantly, the fact that this conclusion is based on many different simulations "suggests this is characteristic of ΛCDM in a considerable range of ways to treat the complexity" of subgrid baryonic physics processes.

Lower-mass halos would be more common in ΛCDM, especially due to the implied presence of many purely DM halos with a negligible amount of baryons (e.g., [380]). Consequently, we expect there to have been many more minor mergers than major mergers. Even minor mergers could be quite disruptive, so it seems very difficult to avoid substantial dynamical heating of thin galactic discs. One possibility is that if the perturber is gas-rich, it is more likely for the remnant to contain a dominant thin disc [381].

To check if the proportion of thin disc galaxies is correct in ΛCDM, it is necessary to compare the distribution of the sky-projected aspect ratio $q_{sky}$ in a large cosmological hydrodynamical simulation with that in actual observations. This reveals a very significant $12.52\sigma$ discrepancy between the GAMA survey and TNG50-1, the highest-resolution ΛCDM simulation suitable for such a comparison [382]. The "−1" suffix distinguishes the simulation from TNG50-2, TNG50-3, and TNG50-4, which have successively lower resolution. This allows for a test of numerical convergence. The aspect ratio distribution differs somewhat depending on the resolution because thin galactic discs are still somewhat challenging to resolve in large simulations [383]. By using the different TNG50 runs, it is possible to extrapolate to the result of a higher-resolution simulation than TNG50-1, as done by [382] in their Figure 8 (reproduced in our Figure 14). If the CDM mass resolution is improved by a factor of $8^5$ with corresponding improvements elsewhere (mimicking the improvements from TNG50-2 to TNG50-1 repeated five times), the tension with observations is reduced to $5.58\sigma$ using the more accurate quadratic extrapolation discussed in their work. It is therefore clear that improvements to the resolution of the simulations are likely to prove insufficient: ΛCDM faces a genuine problem reproducing the high observed fraction of thin disc galaxies. The authors of [382] also highlighted severe deficiencies with earlier studies that claimed good agreement in this regard [384–386]. The problem might be alleviated in MOND due to a reduced frequency of major mergers and because of the stronger vertical gravity from the disc making it thinner for the same $\sigma_z$ [387].

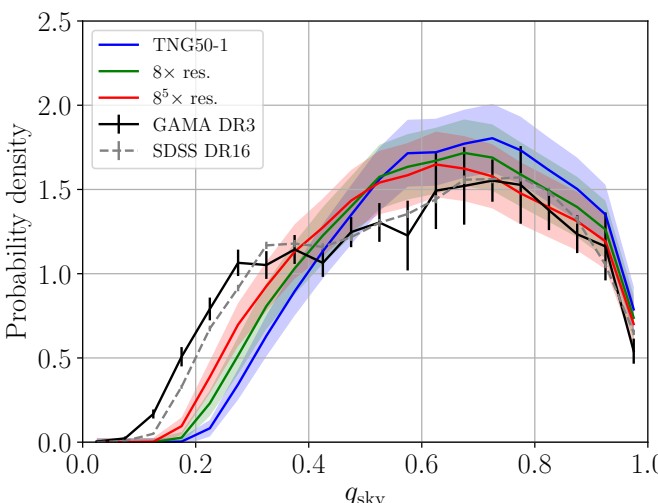

**Figure 14.** The sky-projected aspect ratio distribution of galaxies with $10^{10.0} < M_\star/M_\odot < 10^{11.65}$ according to the Galaxy and Mass Assembly (GAMA [388,389]) and SDSS surveys (black and grey lines, respectively), with GAMA having a higher resolution. The Poisson error bars are shown for both surveys, but these are very small for SDSS. The galaxies have been statistically weighted to match the $M_\star$ distribution of TNG50-1, the highest-resolution hydrodynamical cosmological $\Lambda$CDM simulation suitable for a comparison (solid blue line). The green line shows the extrapolation to a simulation with $8\times$ better mass resolution in the CDM, with corresponding improvements elsewhere. The red line shows the estimated result of improving the TNG50-1 CDM mass resolution by a factor of $8^5$, beyond which results should change little as such a simulation should be numerically converged. The shaded bands show the Poisson uncertainties on simulated distributions. Reproduced from Figure 8 of [382] using ApJ author republication rights.

Since MOND reduces to Newtonian dynamics at high surface density and acceleration, MOND cannot explain the continued survival of thin disc galaxies with a surface density much above the MOND threshold $\Sigma_M = 137\, M_\odot/\mathrm{pc}^2$ (Equation (45)). Such a galaxy would behave like a self-gravitating Newtonian disc, which is known to be unstable [16]. Fortunately for MOND, disc galaxies do not have central surface densities very far above $\Sigma_M$, which was observationally first noticed as a very narrow concentration in the distribution of surface brightness [151,390]. This 'Freeman limit' is almost certainly caused by selection effects [391] since it later became clear that many galaxies exist with a much lower surface brightness [392]. Even so, the upper limit appears to be real [393–395] and is not easily assigned to a selection effect. We illustrate this in Figure 15, which reproduces Figure 5 of [393]. The sharp break evident at $\approx 22\,B$ magnitudes per square arcsecond corresponds to $100\,L_\odot/\mathrm{pc}^2$ [396]. Assuming $M_\star/L_B \approx 1$, this implies a transition in the behaviour of galaxies at a critical surface density of $\approx 100\,M_\odot/\mathrm{pc}^2$, which is very similar to the predicted stability threshold in MOND. Note that disc galaxies somewhat above this threshold $\Sigma$ can still be stable in MOND, as demonstrated in [362] who considered a case where the central $\Sigma = 10\,\Sigma_M$. This is partly because the dynamical stability of a galaxy would depend on the average surface density within roughly the central scale length. For a typical exponential profile, this average surface density is smaller than the central value by a factor of $2(1 - 2/e) \approx 0.5$. In a MOND context, we therefore do not expect a complete lack of high surface brightness galaxies (HSBs), but we do expect that galaxies with a central $\Sigma \gg \Sigma_M$ are much more likely to have become unstable at some point, thereby contributing little towards the overall disc galaxy population.

In principle, MOND should also explain the formation of disc galaxies out of collapsing gas clouds. This was investigated for the first time by [397], leading to an exponential surface density profile after 10 Gyr of evolution. However, the work was not done in a cosmological context. Hydrodynamical cosmological MOND simulations are currently underway (N. Wittenburg et al., in preparation) based on the cosmological paradigm discussed in Section 9.2. Gas accretion in a cosmological context is expected to yield

different results compared to starting with all the gas in a rotating spherical cloud. Prior MOND simulations indicate that gas-rich clumpy galaxies in the early universe do not secularly form bulges from the coalescence of clumps [398], helping to explain the high fraction of discs with little to no bulge [374]. Hydrodynamical MOND simulations of M33 were also able to avoid the formation of a significant central bulge [106]. MOND appears more consistent with the high prevalence of thin disc galaxies due to a reduced rate of mergers and the lack of bar-halo resonances that can cause a strong bar at late times (Section 4.3).

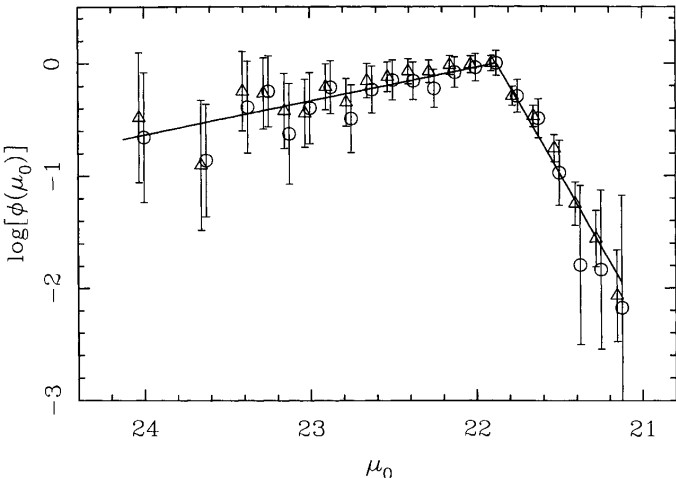

**Figure 15.** The logarithmic number density of disc galaxies as a function of central surface brightness in *B* magnitudes per square arcsecond, with error bars showing the Poisson noise. The solid lines show a broken power-law fit, with a sharp break clearly evident at a luminosity close to $100 \, L_\odot/\text{pc}^2$, the significance of which is discussed in the text. Reproduced from Figure 5 of [393].

*4.2. Number of Spiral Arms*

The greater importance of disc self-gravity in MOND manifests itself in the different Toomre disc stability condition [15] for AQUAL [399] and QUMOND [101]. In what follows, we will focus on QUMOND, but the results are very similar in AQUAL. The local stability of a stellar disc is determined by three main factors:

1. The radial velocity dispersion $\sigma_r$,
2. Disc self-gravity, and
3. Shear caused by differential rotation of the disc.

Disc galaxy RCs are well understood in MOND (Section 3.1.1). Since the CDM halo is chosen to match the RC, this is the same in both paradigms by construction. Therefore, differences in disc self-gravity feed through to differences in the minimum $\sigma_r$ required for local stability.

The Toomre condition relies on the notion that there is a particular wavelength $\lambda_c$ which is least stable to perturbations. This is because very short wavelength modes are essentially just sound waves little affected by gravity due to the short crossing time, while long wavelength modes rapidly wind up in a differentially rotating disc. Intermediate length modes are significantly affected by self-gravity, yet are not so long as to be easily wound up. This is explained further in [281], whose Section 6.2.3 shows how these considerations lead to a parabolic dispersion relation between the inverse length scale of perturbations and the square of their oscillation frequency (negative values indicate the existence of an unstable growing mode). Though their result is valid for an isothermal fluid, the result for a collisionless stellar disc is very similar. According to equation 22 of [15], a Newtonian disc with the lowest $\sigma_r$ necessary for local stability has

$$\lambda_c = 0.55 \times \frac{4\pi^2 G \Sigma}{\Omega_r^2} \, , \tag{56}$$

where $\Sigma$ is the disc surface density and $\Omega_r$ is the radial epicyclic frequency determined by the RC. The factor of 0.55 was derived numerically in Section 5c of [15]. According to its equation 65, this mode is stable in the local approximation if

$$\sigma_r \geq \frac{3.36\,G\Sigma}{\Omega_r}\,.$$ (57)

The ratio of $\sigma_r$ to this critical threshold is known as the Toomre parameter $Q$, which in Newtonian gravity is thus

$$Q_N \equiv \frac{\sigma_r \Omega_r}{3.36\,G\Sigma}\,.$$ (58)

Discs are locally stable if $Q > 1$ everywhere, including near the centre, where $Q$ is typically lower than in the outskirts. Note that Equation (56) assumes $Q = 1$, which we expect due to self-regulatory mechanisms (discs with $Q \ll 1$ should become dynamically hotter). In general, $\lambda_c \propto Q^2$.

As derived in [101], the above-mentioned classical Toomre condition for a thin disc can be generalized to QUMOND by redefining $G \rightarrow G\nu(1 + K/2)$, where the MOND interpolating function $\nu$ (e.g., Equation (15)) and its logarithmic derivative $K$ (Equation (35)) must be evaluated just outside the disc plane so as to include the vertical component of the Newtonian gravity. With this adjustment, the QUMOND version of Equation (58) is

$$Q_M \equiv \frac{\sigma_r \Omega_r}{3.36\,G\Sigma\nu\left(1 + \frac{K}{2}\right)}\,.$$ (59)

The extra factor of $\nu$ can be very large in low surface brightness (LSB) discs, where $K$ is close to the deep-MOND value of $-1/2$. In the Newtonian regime, $\nu \rightarrow 1$ and $K \rightarrow 0$, recovering the standard behaviour.

By considering QUMOND models of M33 with different values of the Toomre parameter, the authors of [106] demonstrated that using Equation (59) as the QUMOND Toomre condition provides a good guide to where the transition lies between stable and unstable discs (see their Figure 24). This is reproduced here as our Figure 16, which shows the effect of lowering the minimum allowed $Q$ in the initial conditions (this parameter is called $Q_{\text{lim}}$ in the code). When $Q_{\text{lim}}$ is reduced from 1.25 (black curve) to 1.1 (blue curve), the disc is initially dynamically colder and also ends up with a lower $\sigma_z$. However, if $Q_{\text{lim}}$ is reduced further from 1.1 to 1 (red curve), then $\sigma_z$ starts rising rapidly after $\approx 5$ Gyr, ending up higher than in the other models despite being dynamically colder initially. This is suggestive of a dynamical instability which is avoided by the higher $Q_{\text{lim}}$ models. To ensure stability and leave some safety margin, we recommend using $Q_{\text{lim}} = 1.25$ in MOND simulations where the goal is to have a stable disc.

The higher effective $G$ in MOND makes discs less stable for a given velocity dispersion, so they must be dynamically hotter to remain locally stable [387]. In practice, this means they would heat up to a greater extent through dynamical instabilities until they become Toomre stable. Therefore, the observation of an extremely low velocity dispersion could challenge the MOND scenario. However, if observations confirm the higher velocity dispersion it requires, this could be interpreted in the standard context as a disc which is dynamically hotter than required for local stability. Since the Toomre condition only provides a lower limit on $\sigma_r$, this is plausible, though other considerations might break the degeneracy.

Interestingly, observed LSB galaxies (LSBs) have rather higher velocity dispersions than required in the CDM picture—they appear to be dynamically overheated [400]. If so, it would be difficult for LSBs to have ongoing star formation and sustain spiral density waves, the leading explanation for observed spiral features in HSBs such as the MW and M31 [401]. However, LSBs also have spiral features [402–405]. This and other features of LSBs suggest that their gravitating mass mostly resides in their disc, contradicting $\Lambda$CDM expectations and instead confirming the MOND prediction (Section 3.3 of [387]).

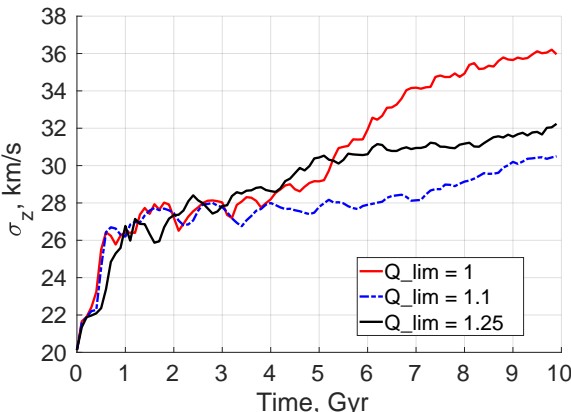

**Figure 16.** Time evolution of the vertical velocity dispersion $\sigma_z$ of stars at radii of 0.5–3 kpc in a hydrodynamical MOND simulation of M33 that includes a weak EFE representative of the gravity from M31. Different lines show different values for $Q_{\text{lim}}$, the minimum Toomre parameter (Equation (59)) imposed on the initial conditions. Notice how reducing $Q_{\text{lim}}$ from 1.25 (solid black) to 1.1 (dotted blue) reduces the final $\sigma_z$, as expected for a dynamically colder initial disc. However, $\sigma_z$ ends up higher when $Q_{\text{lim}} = 1$ (solid red), suggesting that the MOND generalization of the Toomre condition does indeed provide a good guide to which models are stable and which are not. Reproduced from Figure 24 of [106] using ApJ author republication rights.

The extra MOND factors in Equation (59) also affect $\lambda_c$, which in QUMOND becomes

$$\lambda_c = \frac{2.2\pi^2 G \Sigma \nu \left(1 + \frac{K}{2}\right)}{\Omega_r^2} . \tag{60}$$

The least stable mode is thus longer in MOND (cf. Equation (56)). Since the amplitude of density fluctuations is presumably larger for a less stable mode, we expect that the most prominent structures in a galaxy would differ depending on the gravity law. In a disc galaxy, the most striking feature is usually its pattern of spiral arms. Assuming the Lin–Shu density wave theory for spiral structures [401], the number of spiral arms allows for an estimate of $\lambda_c$ [406,407]. Using this principle, the author of [408] analytically predicted the number of spiral arms in galaxies observed as part of the DiskMass survey. She found good agreement with observations for a reasonable $M_\star/L$. The DiskMass survey "selects against LSB discs", making the work of [408] an important check on the validity of the mode-counting technique in a regime where $\Lambda$CDM and MOND give similar predictions.

The density wave theory should also apply in LSBs. Applying a similar technique to LSBs under the assumption of Newtonian gravity, the authors of [409,410] found that LSB discs would be much too stable to permit the formation of their observed spiral arms for plausible $M_\star/L$ ratios consistent with stellar population synthesis models (e.g., [168]). The spiral structures seem to imply that much of the mass needed to explain the elevated RCs with standard gravity resides not in a stabilizing CDM halo but rather within the disc itself. This would make the discs very massive, sometimes requiring $M_\star/L_R > 10\times$ the solar value. Physically, this arises because density perturbations would wind up quickly in a system where a dominant dark halo elevates the RC but not the disc self-gravity. This scenario would make the disc very stable and thus unable to easily form a grand-design spiral, which however is very common observationally [411]. While some cases such as the well-known M51 are tidally induced [412], it seems unlikely that all grand-design spirals are of tidal origin. Their high prevalence is more easily understood as a sign that the least stable wavelength (Equation (56) or (60)) is rather long, favouring an enhancement to the effective value of $G$.

### 4.3. Bar Strength

In the previous section, we assumed that CDM halos (if present) would act to damp perturbations in the disc. This is reasonable because a physical halo is distinct from the disc, unlike a phantom halo which would respond instantaneously to changes in the disc (the response would occur within a few light crossing times). However, a physical halo would also respond, albeit over a longer timescale. An important characteristic of a real halo is that it can absorb angular momentum from the disc, which is not possible for a phantom halo. Moreover, a physical halo opens up the possibility of resonances with the disc, i.e., that perturbations to the disc affect the halo which then further affects the disc. We therefore discuss if the secular evolution of disc galaxies is more easily understood with a real surrounding halo, or if postulating such a halo would cause effects that are not observed. We begin by describing ΛCDM expectations, then discuss the observations, and finally consider the MOND scenario, focusing in particular on how the observed morphology of M33 might be understood in both paradigms.

Early *N*-body simulations of self-gravitating Newtonian discs quickly became dominated by bars [16,413,414]. These works suggested an almost unavoidable global instability of a self-gravitating rotating stellar disc. It can be made more stable if it is embedded within a dominant CDM halo [19]. However, their equation 6 highlights an important deficiency in the analysis—the halo was assumed to provide a time-independent contribution to the potential. It was not treated using live particles. This is sometimes known as a rigid halo model, which can be thought of as including extra halo particles that provide gravity but do not move.

Subsequent simulations with live halos have shown that over long periods, a disc inside a massive halo is bar unstable due to a resonance between the orbits of stars in the bar and DM particles in the inner halo [415]. The reduced role of disc self-gravity means that a massive halo can delay the bar instability, but it is even stronger when it does occur, which would still be well within the lifetime of the universe. This runs contrary to the role of a halo in reducing the relative importance of disc self-gravity, which might naively be expected to stabilize the disc (Section 4.2). While this is true on short timescales, discs within a more centrally concentrated halo end up with a stronger bar that is longer and thinner [416].

For a bar to develop, it is necessary for stellar orbits to become significantly non-circular. While the orbits need not be as elliptical as the bar itself, some ellipticity is needed because otherwise any non-axisymmetric feature would quickly wind up and disappear due to differential rotation of the disc [401]. Since elliptical orbits have less angular momentum than a circular orbit for the same apocentre, it is clear that redistribution of angular momentum is important to the bar instability. A critical issue is how much angular momentum can be transferred by the bar, which in turn depends on where it is being transferred to. If the sink is the outer disc where there is relatively little material, the bar is unlikely to remain very strong over an extended period—the outer disc would expand further, making it less tightly coupled to the bar region. However, if the sink is a dominant halo, then a very large amount of angular momentum can be transferred out of the inner disc, leading to a rather strong bar [417]. Therefore, a halo serves to increase the maximum possible amplitude of the bar to very high values, even if it slows down the bar instability.

The halo would typically have a small amount of net rotation in the same sense as the disc, because both would have been spun up by tidal torques from large-scale structure [418,419]. A rotating halo further worsens the bar instability because it allows more DM to be on a resonant orbit with the rotating bar [420]. Those authors found that a counter-rotating halo could suppress the bar instability, but this would be a rare scenario as tidal torques from large-scale structure would typically spin up the baryonic and DM components in the same sense. This is ultimately just a consequence of the weak equivalence principle or the universality of free fall (which remains valid even in MOND).

The ΛCDM expectation of typically rather strong bars was confirmed by more recent *N*-body simulations [421,422]. Scattering by giant molecular clouds delays the onset of

the bar instability, but is unable to prevent it [230]. This suggests that including the gas component would have little impact on the results.

The larger-scale cosmological environment could be important, though more computationally demanding to include. This was attempted by [423] using a technique to insert a disc into a DM-based cosmological simulation [424]. Section 5.2 of [423] describes the setup of their two cosmological disc simulations. Their Figure 12 shows a rapid increase in the bar strength once the disc is switched from rigid to live. Bars in cosmologically motivated disc environments are therefore expected to develop in only a few Gyr [423], somewhat contrary to what is stated in their abstract.

Since the bar is a self-gravitating instability, a dynamically hotter and thicker disc could suppress the development of a bar. However, any bar in a thicker disc would itself be thicker, making the bar less prone to buckle. This leads to an even longer bar [425]. Indeed, their Figure 12 shows that the bar in their thick disc model ends up covering the entire disc. It therefore seems clear that if disc galaxies have a substantial amount of CDM in a spheroidal halo, then gravitational interactions with the halo would lead to a quite strong bar on a timescale much shorter than the age of the universe. Differences in the timescale of this instability are less important than what limits the magnitude of the bar, simply because the universe is many dynamical timescales old.

Turning now to the real universe, the authors of [426] found that $\approx$30% of disc galaxies have bars based on optical SDSS images [289]. The bar fraction doubles in the higher angular resolution Spitzer Survey of Stellar Structure in Galaxies (S$^4$G [427]). Their estimated bar fraction of $\approx$60% is in agreement with earlier estimates using near-infrared data [428,429]. Moreover, the authors of [428] found that "strong bars are nearly twice as prevalent in the near-infrared as in the optical." Similar conclusions were reached by [427], who argued that the superior resolution of the Spitzer Space Telescope allowed the detection of shorter bars that might be missed in SDSS images. This may well explain the higher bar fraction in higher-resolution images, though the reduced dust extinction at longer wavelengths might help as well. It is therefore clear that the majority of disc galaxies have a bar, but $\approx$40% are unbarred or have only a very weak bar.

One such weakly barred galaxy is the rather isolated M33 [430,431]. Its RC requires a substantial amount of DM in conventional gravity [72,432,433], as expected in MOND for its surface brightness. However, unavoidable gravitational interactions between the disc and the required dark halo make it very difficult to explain the observed properties of M33 [434]. Those authors conducted a detailed investigation into the global stability of M33 in the $\Lambda$CDM context, finding that a strong bar develops in only a few Gyr. Their Section 3.2 discusses how they obtained similar results when they included gas with thermal or mechanical feedback from the stars. The authors were able to suppress the bar instability by using a rigid halo which provides a fixed potential, corresponding to a halo whose particles do not move. Though unphysical, this demonstrated that the instability is indeed due to the halo (see their Section 4.1). A more plausible solution is to reduce the disc mass, but this entails an unrealistically low $M_\star/L$ of only 0.6 (0.23) in the $V$ ($K$) band [435,436]. Moreover, a lower mass disc would be less maximal and require a larger halo contribution, reducing the least stable wavelength (Equation (56)). This caused [434] to conclude in their Section 4.2.1 that "all the spiral patterns in our simulation of this low-mass disc were multi-armed", making it difficult to understand the observed bisymmetric spiral in M33 [430,437–440]. It could be made bar stable with a higher velocity dispersion and thicker disc in which $Q = 2$, showing that combining such a hot disc with a dominant DM halo might perhaps give M33 the sought-after stability in a $\Lambda$CDM context. However, it is difficult to understand how such a dynamically overheated disc is currently forming stars [441]. In addition, the simulation does not yield a grand-design spiral because the least stable wavelength is too short. The difficulties reproducing the observed properties of M33 led [434] to conclude that "none of the ideas proposed over the subsequent years can account for the absence of a bar in M33" in the $\Lambda$CDM context. Generalizing the

implications to other weakly barred galaxies, the authors went on to conclude that "it is shocking that we still do not understand how the bar instability is avoided in real galaxies."

In a MOND context, the enhanced role of disc self-gravity would make the most unstable wavelength longer (Equation (60)), helping to address the dominant two-armed spiral. The lack of bar-halo angular momentum exchange would remove the main mechanism causing difficulty for the simulations of [434]. However, the enhanced disc self-gravity means that a MOND model of M33 would be much less stable at early times. MOND is therefore unable to suppress the bar instability, potentially leading to a conflict with observations. Consistency would require the bar strength to saturate at a rather low value consistent with observations, or for the bar to become very strong and then get destroyed. This is theoretically possible because if angular momentum can only be redistributed within the disc, then the outer disc might not have enough mass to sustain a very strong bar over a Hubble time. Clearly, numerical simulations are required to address such subtle issues.

The first *N*-body MOND simulation of an isolated disc was the seminal work of [442], who used a custom potential solver to advance particles in 2D. This helped to clarify that MOND discs would be more stable than a DM-free Newtonian disc, but less stable than a Newtonian disc in a DM halo adjusted to reproduce the MOND RC. Perhaps the main outcome of the study was that the stability problem of Newtonian self-gravitating discs (e.g., [16]) might be solved in MOND, consistent with earlier analytic expectations [399]. Clearly, a dominant DM halo around galaxies is not the only way to stabilize an embedded disc.

The 3D dynamics of disc galaxies were explored in a series of simulations by [443]. These revealed the expected rapid strengthening of the bar due to enhanced disc self-gravity, but the bar then weakened as $\sigma_z$ increased (see their Section 5.1). The same basic picture was obtained when those authors included gas as sticky particles in their simulations, though including dissipation in this way weakened the bar somewhat [444].

The stability of M33 was addressed in a recent hydrodynamical MOND simulation [106]. MOND is expected to have a significant effect on M33 because its surface density is below the critical MOND threshold (Equation (45)) outside the very central region (see their Figure 1). The gravity law governing M33 is thus rather different in MOND, with corresponding differences in its total mass (i.e., the lack of a dominant dark halo). Despite very little fine-tuning, Figure 12 of [106] demonstrates that the bar strength has the sought-after behaviour of a sharp rise at early times followed by a decline to a very low strength well within a Hubble time. We show this in our Figure 17, which reproduces their Figure 19 in which the EFE from M31 is also approximately included. The bar strength evolved similarly in earlier simulations at lower resolution, which also show a sharp initial rise in the bar strength followed by a more gradual decline [443,444].

In addition to a realistic bar strength at late times, Figure 9 of [106] demonstrates that their model develops a bisymmetric spiral, which is probably related to the most unstable wavelength being longer in MOND (Equation (56)). Interestingly, the authors of [106] also considered the EFE from M31, finding that it improves the agreement with some observables such as the RC in the central 2 kpc (see their Figure 13). The more realistic models did not develop a central bulge, with the EFE further suppressing the tendency for material to concentrate towards the centre (see their Figure 14). Therefore, the global properties of M33 are reasonably well reproduced in MOND despite the smaller theoretical flexibility it offers (videos of their isolated simulations are available here: https://seafile.unistra.fr/d/843b0b8ba5a648c2bd05/, accessed on 23 May 2022). While this by no means proves the MOND model correct, various aspects of M33 such as its weak bar argue against the presence of a dominant DM halo [434].

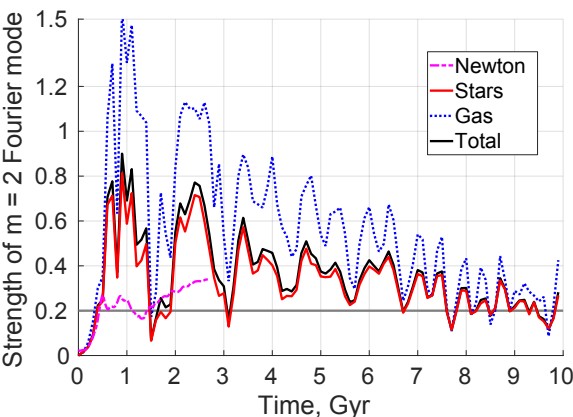

**Figure 17.** Time evolution of the m $=$ 2 azimuthal Fourier mode in a hydrodynamical MOND simulation of M33 [106], found from a Fourier decomposition of material at radii of 0.5–3 kpc. Results are shown for the stars (solid red), gas (dotted blue), and the total (solid black). For comparison, the Figure also shows a purely stellar $N$-body simulation (dotted red) and the most realistic CDM simulation of [434] in Newtonian gravity (dot-dashed magenta), which they called their "full mass" model. Notice that the CDM model develops a stronger bar than the observationally estimated strength of 0.2 [430], shown here as a grey line. The bar is expected to strengthen further with time. In MOND, after a very strong bar instability at early times, the bar ends up being rather weak. Reproduced from Figure 19 of [106] using ApJ author republication rights.

### 4.4. Bar Fraction

The properties of bars can be considered more generally, at least in the $\Lambda$CDM paradigm where the galaxy population can be investigated in the proper cosmological context. A major error with the evolution of bars should be evident in the fraction of galaxies which have a bar, defined in [362] as those galaxies where the strength of the m $=$ 2 azimuthal Fourier mode ($A_2$) in units of the azimuthal average ($A_0$) is $A_2/A_0 > 0.2$ at the radius where this ratio peaks. The study made use of the cosmological $\Lambda$CDM simulation known as Illustris The Next Generation (hereafter TNG [445]), an improvement on the earlier Illustris simulation [384]. For the comparison, the authors of [362] considered TNG100 and TNG50, the highest-resolution simulation in the TNG suite. They also considered an earlier simulation called Evolution and Assembly of Galaxies and their Environments (EAGLE [446,447]), as well as observational results based on SDSS [448] and a much more recent study called S$^4$G that uses data from the Spitzer Space Telescope [427]. As argued in Section 4.3, S$^4$G should provide a much more accurate estimate, especially at the low-mass end, where shorter bars can be difficult to resolve in SDSS. The bar fractions in different numerical and observational studies are shown in Figure 18, which reproduces Figure 1 of [449]. It is clear that there is a very significant disagreement between the latest observations and the bar fraction expected in $\Lambda$CDM, which is similar between TNG100 and the much higher-resolution TNG50 (albeit slightly higher than in EAGLE100). The numbers refer to the approximate side length of the cubic simulation volume, expressed in co-moving Mpc. A similar analysis for TNG100 had previously been conducted by [450] and gave similar results (the grey curve). The zoom-in NewHorizon simulation [451] also produces too few barred galaxies at the low-mass end [452] despite having a multi-phase interstellar medium.

The bar fraction decreases at high $M_\star$, but in $\Lambda$CDM it is expected to rise. This could be due to lower-mass galaxies typically having a larger proportion of CDM to baryons, which overstabilizes the disc and prevents bar formation. However, the results of [434] indicate that the DM halo can promote bar growth. Since the disc needs to be quite cold dynamically to form a bar, the low bar fraction at low $M_\star$ might instead be related to such discs generally being thicker than in reality, i.e., having a greater degree of pressure support. This may be related to the underpredicted fraction of galaxies which are thin discs, possibly due to baryonic feedback effects and heating by DM subhalos (Section 4.1).

Lacking subhalo heating, the isolated model of [434] was able to retain a rather thin disc in their M33 simulation—and thus a strong bar.

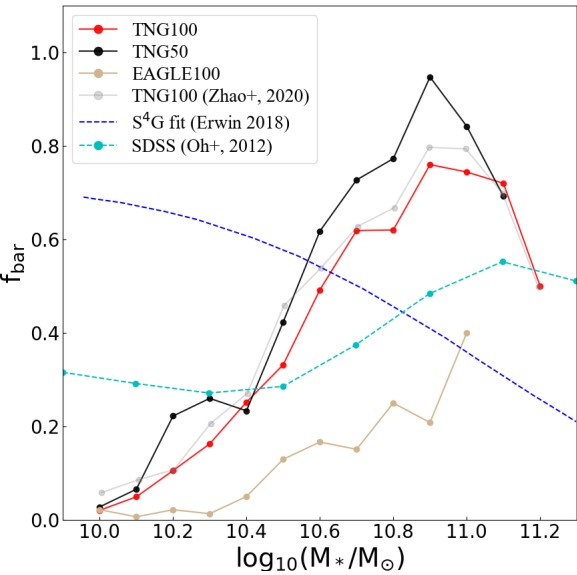

**Figure 18.** The fraction of disc galaxies with a bar ($A_2/A_0 > 0.2$) in the surveys known as SDSS (cyan [448]) and $S^4G$ (blue [427]), with the latter having a much higher angular resolution due to the use of data from beyond the atmosphere. These observational results are shown with dashed lines, while solid lines are used for simulation results. The bar fractions are similar in the TNG50 simulation (black) and TNG100 (red), with a similar determination for TNG100 by a different team shown in grey [450]. The TNG results appear to have converged, though the bar fraction is somewhat higher than in the older EAGLE100 simulation (brown). Reproduced from Figure 1 of [449].

At the high-mass end, the high bar fraction in $\Lambda$CDM simulations (Figure 18) might be due to the disc being dynamically colder as subhalo heating and feedback would be less disruptive for a more massive galaxy. A major merger would thicken the disc, but in this case the galaxy might not be considered a disc any more—and thus removed from the sample used to estimate the fraction of disc galaxies with a bar. Massive simulated disc galaxies seem to have much more DM than they need to maintain consistency with the observed RAR [453], which is the disc galaxy analogue to the similar problem for ellipticals (Figure 7). This makes it possible for the disc to be Toomre stable with a rather low velocity dispersion, and thus able to sustain a strong bar. Resonant bar-halo interactions can then strengthen the bar further (Section 4.3).

The interpretation of Figure 18 is also far from clear in MOND, which did not predict the observed trend. In MOND, we expect that bar properties would differ between galaxies depending mainly on whether their central surface density $\Sigma_0$ is below or above the critical threshold $\Sigma_M$ (Equation (45)). Since higher-mass galaxies generally have a higher $\Sigma_0$, we interpret Figure 18 as showing a lower bar fraction at high $\Sigma_0$ rather than at high $M_\star$. It would be useful to plot the observed bar fraction in terms of central surface brightness rather than $M_\star$, but we will assume in what follows that the trend would remain similar. In the study of [106] on M33, $\Sigma_0 \approx \Sigma_M$ (see their Figure 1). The exponential disc galaxy model considered in [362] had $\Sigma_0 = 10\Sigma_M$, which as discussed in their Section 3.3 is intermediate between the values for the MW and M31, thereby representing a much higher mass galaxy than M33. In the lower $\Sigma_0$ model representative of M33, the bar strength rises rapidly to $\approx 1$ (Figure 17), so there should be a high likelihood of observing a strong bar if we consider the time evolution of a single model as giving similar statistics to a galaxy population observed at the same epoch. However, in the higher $\Sigma_0$ simulation representative of the major LG galaxies, the peak bar strength is only $\approx 0.3$ (see Figure 13 of [362]). The bar weakens substantially by the end of both simulations to a similar final

strength, but there is a significant difference in the peak strength. For a comparison with the purely stellar simulations in [362], it is best to consider the stellar-only model of M33 shown in Figure 10 of [106], which gives an even stronger bar that is still well above the typically used 0.2 threshold by the end of the simulation. As a result, we might expect that we are much more likely to observe a bar in a lower $\Sigma_0$ galaxy such as M33. This is in line with the observed declining trend in the bar fraction as a function of $M_\star$ (Figure 18).

As with the $\Lambda$CDM case, we assign the different behaviour to differences in the degree of random motion. For a fair comparison, we first need to consider the typical circular velocity of each galaxy, for which we use Equation (46). The M33 model of [106] has $M = 6.5 \times 10^9 \, M_\odot$ ($v_f = 101$ km/s) while the MOND model of [362] has $M = 8.57 \times 10^9 \, M_\odot$ ($v_f = 108$ km/s), so both models have a similar RC amplitude. The random motions in the initial conditions used by [362] are given in their Figure 5. We will focus on $\sigma_z$, which is shown in the bottom right panel. The central $\sigma_z \approx 30$ km/s, much higher than the $\approx 20$ km/s evident in Figure 16 for the M33 model. This Figure also shows that $\sigma_z$ rises by a factor of $\approx 1.6$ over the full duration, with the factor being $\approx 2$ in the stellar-only model shown in Figure 6 of [106]. The disc heating in the model of [362] is shown in their Figure 26 in terms of the disc thickness $h$. We expect that $\sigma_z{}^2$ is proportional to the product of $h$ and the vertical gravity. Assuming a thin disc, the vertical gravity remains similar as we move up from the disc plane, so $\sigma_z \propto \sqrt{h}$. In reality, we expect a slightly steeper scaling because at large radii, the vertical gravity is not related to the local surface density but instead arises from the change in angle towards the central regions of the galaxy, i.e., we can estimate the result by considering the galaxy to be a point mass. In this regime, the vertical gravity $\propto h$, so we expect that $\sigma_z \propto h$. However, since the disc thickness shown in Figure 26 of [362] is not measured that far out, it should be more accurate to use $\sigma_z \propto \sqrt{h}$, which will give a conservative estimate of the rise in $\sigma_z$ during the simulation. Since this Figure shows $h$ rising $\approx 4\times$, we can safely say that $\sigma_z$ doubles. This is a similar rate of disc heating to the M33 model of [106], if not even higher. As a result, the initially dynamically hotter disc in [362] remains dynamically hotter by the end, which is also apparent from comparing the edge-on projections. The higher $\Sigma_0$ in their model requires it to be dynamically hotter to remain Toomre stable, as can be understood analytically [101]. We would thus expect that in MOND, higher surface density galaxies would be less likely to have a detectable bar due to being dynamically hotter. This is borne out by comparison of the bar strength evolution between Figure 13 of [362] and the lower $\Sigma_0$ model shown in our Figure 17.

At this stage, it is far from certain that MOND can solve the very significant failure of $\Lambda$CDM with respect to the fraction of disc galaxies that have a bar (Figure 18). Existing results suggest that MOND should give the right declining trend with $M_\star$, even though the simulation of [106] was motivated by a desire to obtain a weak bar resembling M33, while the work of [362] considered only one model and was targeted at the pattern speed of the bar rather than its strength. Though these unlooked-for results are encouraging, further MOND simulations are needed covering a wider range of galaxy properties. Eventually, cosmological simulations will be necessary to handle issues such as the EFE and interactions between galaxies, and more generally to put galaxies into their proper cosmological context (Section 9).

*4.5. Bar Pattern Speed*

Another crucial property of a galactic bar is its pattern speed $\Omega_p$, the angular frequency at which the bar rotates almost as a solid body. Since galaxies cover a wide range of mass, size, and rotation speed, they have a huge range of characteristic angular frequency. It is therefore necessary to normalize $\Omega_p$ in some way to arrive at a dimensionless quantity. For this, we follow [454] in using the parameter

$$\mathcal{R} \equiv \frac{R_c}{R_b}, \tag{61}$$

where $R_b$ is the semi-major axis of the bar and $R_c$ is its corotation radius, the radius where $\Omega_p$ is also the angular velocity $v_c/r$ of a star on a circular orbit in the galactic potential.

Bars are said to be fast if $\mathcal{R} = 1.0 - 1.4$ [281], while bars with $\mathcal{R} > 1.4$ are said to be slow as $\Omega_p < v_c/r$ at the end of the bar. Theoretically, an ultrafast bar ($\mathcal{R} < 1$) should be unstable, so bars should not extend beyond their corotation radius [455–457].

A bar rotating through a CDM halo is expected to experience dynamical friction for much the same reason as any other massive moving object [458]. Their Figure 11 shows that the disc rapidly starts losing angular momentum to the halo as soon as the bar forms, though substantial bar slowdown was prevented by this pioneering simulation having a limited duration of 1.2 Gyr. Rapid slowdown of the bar was demonstrated in the work of [459] and confirmed in later studies [425,460–462]. However, not all such simulations give slow bars (e.g., [417,463]). For a review of isolated CDM-based models of bar evolution in disc galaxies, we refer the reader to Section 5.4.1 of [362], which also considers possible reasons for the different results. One of the most crucial factors they identified was whether the halo parameters are "truly what one expects in the ΛCDM paradigm", or if they are unrealistic due to inaccuracies such as a small truncation radius for the CDM halo.

These uncertainties can be reduced by considering a large galaxy population drawn from a high-resolution cosmological hydrodynamical simulation of the ΛCDM paradigm. Before turning to this, we discuss the results of isolated galaxy simulations conducted by [362] which give a deeper insight into the problem. If ΛCDM is correct, the halo parameters must reproduce the RAR (Figure 3), leaving little wiggle room. Importantly, those authors also considered several other modified gravity theories (summarized in their Section 2). The evolution of $\Omega_p$ in these models is shown in our Figure 19, which reproduces their Figure 17. Two models were considered in Newtonian gravity: a self-consistent live Plummer halo (LPH) model and a model with a rigid Hernquist halo (RHH) that provides no dynamical friction, achieved by treating the halo as providing a fixed extra contribution to the potential. In both cases, the halo parameters were chosen to reproduce the RAR in an exponential disc galaxy with $\Sigma_0 = 10\,\Sigma_M$, intermediate between the MW and M31 values (see their Section 3.3). The unphysical RHH model illustrates how suppressing dynamical friction from the halo allows the bar to maintain a nearly constant $\Omega_p$, whereas the bar in the LPH model decelerates significantly, as is particularly apparent in the extra 2 Gyr covered by the inset to Figure 19. All considered modified gravity theories maintain a nearly constant $\Omega_p$.

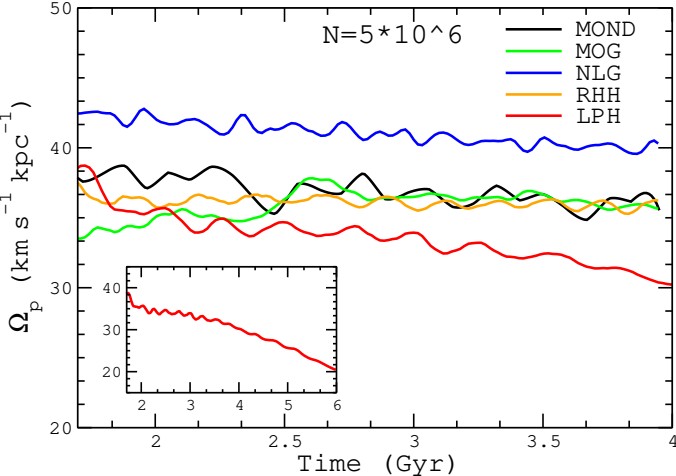

**Figure 19.** Time evolution of the bar pattern speed $\Omega_p$ in the modified gravity theories MOND (black), MOG (green), and non-local gravity (NLG; blue). Two Newtonian models are shown in which the CDM halo is either included self-consistently (red) or as a fixed extra contribution to the potential (orange), which can be thought of as the halo consisting of rigid particles that provide but do not respond to gravity. This unphysical model yields a nearly constant $\Omega_p$, as do the modified gravity theories. The bar decelerates significantly in the realistic CDM model, as shown also in the inset which evolves it for longer. Reproduced from Figure 17 of [362].

These differences in the evolution of $\Omega_p$ translate directly into differences in the distribution of $\mathcal{R}$. This is because the RCs shown in Figure 4 of [362] rise almost linearly with radius in the central region, so a small decrease in $\Omega_p$ translates into a significant rise in $R_c$ and thus also in $\mathcal{R}$ (Equation (61)). To build up its statistics, those authors treated the same galaxy viewed at different times as representative of a population of galaxies viewed at the same time in the theory under consideration. The distribution of $\mathcal{R}$ was then analysed in log-space to yield a mean value and intrinsic dispersion. For comparison, the study also considered different observational studies and $\mathcal{R}$ parameter determinations for barred galaxies in the EAGLE100 simulation, as summarized in their Table 2. We show the results in our Figure 20, which reproduces Figure 20 of [362]. The modified gravity theories yield $\mathcal{R} \approx 1$ with little intrinsic dispersion, quite similar to the observations. In contrast, the decelerating bar in the LPH model has a rather high $\mathcal{R}$, which rises further as the simulation is evolved. This is because of angular momentum exchange between the bar and halo, as demonstrated directly in Figure 12 of [362] by considering how the angular momentum of each evolves over time. An important aspect of these results is the distribution of $\mathcal{R}$ in EAGLE100 galaxies. This also shows a rising trend with time, and at $z = 0$ is strongly inconsistent with observations at $7.96\sigma$ confidence.

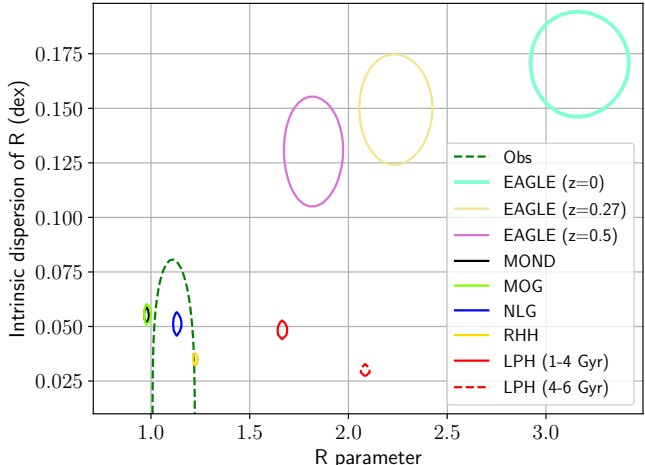

**Figure 20.** Posterior inference on the mean value of the $\mathcal{R}$ parameter (Equation (61)) and its intrinsic dispersion in log-space based on considering different snapshots of the simulations shown in Figure 19, with the same colour used for the same model. The $1\sigma$ confidence region is shown in all cases. The dotted red contour shows the extended evolution of the CDM model in a live halo, whose shift to higher $\mathcal{R}$ is directly linked with the bar deceleration evident in the inset to Figure 19. The dotted green contour shows the observational estimate for the barred disc galaxy population. While this cannot be directly compared to the above-mentioned models, it can be compared to the galaxy population in the EAGLE100 cosmological hydrodynamical simulation of $\Lambda$CDM, results of which are shown at redshift 0.5 (purple), 0.27 (yellow), and 0 (thick cyan) based on $\mathcal{R}$ determinations by an independent team [464]. Here also the mean $\mathcal{R}$ rises with time, and at $z = 0$ is in $7.96\sigma$ tension with local observations. Reproduced from Figure 20 of [362].

Given the high significance of this falsification, we briefly describe how $\mathcal{R}$ is determined observationally. Available observational determinations of the $\mathcal{R}$ parameter were compiled in the study of [465], which also contains a good review of how observers determine the bar length and corotation radius. We therefore summarize only the main points below.

Since the RC is generally well known in comparison to the other parameters, obtaining $R_c$ is mainly a question of reliably determining $\Omega_p$. This is easy from a movie of a galaxy, but remarkably is also possible using a single snapshot with the model-independent Tremaine–Weinberg (TW) method [466]. The basic idea is to determine the so-called kinematic and

photometric integrals, defined as the luminosity-weighted average LOS velocity $V_{\mathrm{LOS}}$ and position $X$ parallel to the major axis of the particles/stars, respectively.

$$\langle V \rangle \equiv \frac{\int V_{\mathrm{LOS}}\Sigma\,dX}{\int \Sigma\,dX}, \tag{62}$$

$$\langle X \rangle \equiv \frac{\int X\Sigma\,dX}{\int \Sigma\,dX}, \tag{63}$$

with $\Sigma$ being the surface density of the tracer in arbitrary units. The integrals have to be measured along apertures (or pseudo-slits) parallel to the line of nodes between the disc and sky planes, between which the inclination angle is $i$. The units of $\Sigma$ are not relevant, reducing sensitivity to the assumed $M_\star/L$. The pattern speed is calculated from the kinematic and photometric integrals as

$$\Omega_p \sin i = \frac{\langle V \rangle}{\langle X \rangle}. \tag{64}$$

The bar length $R_b$ can be obtained in a few different ways from a photograph. The main methods are isophotal ellipse fitting and Fourier analysis. In the former, isophotes are drawn joining points of constant surface brightness, with the bar length defined as the semi-major axis of the isophote having the highest ellipticity [467]. In the latter, the azimuthal surface brightness profile is decomposed into Fourier modes, which are then used to estimate the ratio $I$ of surface brightness in the bar direction and at right angles. $I$ usually rises rapidly to a maximum before declining to a low value in the outskirts. Typically the idea is to find the mean between the minimum and maximum $I$ and then find the corresponding radius within the disc at which $I$ has this value, with the highest determination used in case of multiple solutions [468].

To check if the EAGLE100-based falsification of $\Lambda$CDM persists with more recent higher-resolution simulations, the authors of [449] analysed all $z = 0$ disc galaxies in EAGLE50, EAGLE100, TNG50, and TNG100 using the TW method, thereby allowing a direct comparison with measurements of $\mathcal{R}$ in real galaxies. Their Figure 2 illustrates the steps involved in determining $R_b$ and $\Omega_p$ for a single simulated galaxy with identifier number 229935 in TNG50. The reliability of the TW method is confirmed in their Figure 3, which considers a simulated galaxy with known $\Omega_p$ from time-resolved data. With $\Omega_p$ in hand, it is not difficult to find $R_c$, as illustrated in their Figure 4. On the observational side, 42 galaxies were used which passed all the quality cuts to be included in the final statistical analysis. The highest-resolution simulation considered in the study was TNG50, from which 209 galaxies were usable. EAGLE100 had fewer usable galaxies, while TNG100 had more (see their Table 2). EAGLE50 had only 5 usable galaxies, so we do not discuss it further given also that it has the same resolution as EAGLE100.

The final result of [449] is their Figure 8, reproduced here as our Figure 21. The different $\Lambda$CDM simulations all give consistent results, but the region they converge on is strongly excluded observationally. For TNG50, the tension is $12.62\sigma$ based on 209 galaxies. Due to the larger sample size of 745, TNG100 yields a slightly higher tension of $13.56\sigma$, while the 70 EAGLE100 galaxies used in the final analysis yield a slightly smaller tension of $9.69\sigma$. The authors also used a different method to obtain the RC that should capture the maximum possible uncertainty in it (see their Section 3.10), but this only reduces the significance levels by $\approx 1.5\sigma$ (see their Table 5). Given also the similarity in bar fraction between TNG100 and the higher-resolution TNG50 (Figure 18), there is compelling evidence that the latest highest-resolution cosmological $\Lambda$CDM simulations have converged with respect to the $\mathcal{R}$ parameter distribution in barred disc galaxies, thereby falsifying $\Lambda$CDM at overwhelming significance when confronted with the observational compilation in [465]. For $\Lambda$CDM to be consistent with the observed approximate equality between bar length and corotation radius, the bar length would have to rise over time to keep up with the rising corotation radius. However, observed bar lengths evolve little over time [469,470].

This argument is statistical in nature, leaving plenty of room for *some* galaxies to have fast bars in ΛCDM. Indeed, many such cases are evident in Figures 5 and 6 of [449]. It is therefore not completely surprising that some studies found fast bars in the ΛCDM paradigm (e.g., [471]). Those authors used 16 barred galaxies from the 30 Auriga zoom-in galaxy simulations [209], finding a mean $\mathcal{R}$ close to 1. Possible reasons for this result were discussed in Section 6 of [449], whose main points we review here. Since a much larger sample of galaxies is available from TNG50, the main advantage of a zoom-in simulation would be if it reaches a higher resolution. In this regard, the authors of [471] checked if resolution effects were responsible for their surprising result, but concluded in their Section 3 that "resolution is likely not the main culprit for the high $\mathcal{R}$ values found in previous simulations, such as EAGLE and Illustris." TNG50 is not mentioned because their work pre-dates the publication of [449], but the resolution in TNG50 is almost as good as in the lower-resolution simulations of [471], which reveal no major difference with the higher-resolution runs. Instead of resolution, the typically fast bars reported by [471] are almost certainly related to the sample selection. Indeed, those authors mention that the galaxies they study are not on the abundance matching relation expected in ΛCDM, even though the full simulation volume usually is because the adjustable parameters are calibrated partly to ensure this is so.

More generally, it is unclear how the authors of [471] selected their sample—and whether this introduced biases. For example, requiring a strong bar may mean that the galaxy necessarily had an interaction in the not too distant past. If so, there would be less time for the bar to slow down via dynamical friction against the halo, thus closing the gap with observations in a fake way (Figure 20). Moreover, it is possible that strongly barred galaxies make some aspects of the TW analysis less robust, with the analysis perhaps giving an incorrectly long bar that leads to an underestimated $\mathcal{R}$ [472]. Those authors showed how observed galaxies with apparently ultrafast bars can largely be attributed to complications with the TW analysis for galaxies with fast bars if they also have features such as an inner ring and/or strong spiral arms, artificially raising the estimated bar length. It is easy to imagine that this also occurs in some simulated galaxies, where it is known that bar–spiral arm alignment can lead to an erroneously low measurement of $\mathcal{R}$ [473]. Selecting strongly barred galaxies might have inadvertently selected galaxies where just such issues arise, perhaps because a bar aligned with the spiral arm would also lead to an overestimated bar strength, raising the likelihood that the galaxy passes the criteria in appendix A of [471]. It is also possible that the original Auriga sample was itself not representative of the full EAGLE100 disc galaxy population, whose $\mathcal{R}$ parameter distribution is in $9.69\sigma$ tension with observations (Figure 21). Other zoom-in simulations of ΛCDM galaxies also yield a significant fraction of slow bars (e.g., [474,475]), which seems inevitable given the predicted dynamical friction against the halo. A resimulation of a typical TNG50 disc galaxy with a slow bar might help to clarify this, though the resolution is already sufficient according to [471]. Removing DM from the central region through enhanced feedback might reduce dynamical friction on the bar, but this would make it difficult to satisfy the RAR, especially for lower-mass galaxies. Indeed, their study does not go down to as low a mass as the observational compilation of [465].

The failure of ΛCDM with respect to bar pattern speeds could be rectified by assuming that the DM consists of low-mass particles such as axions with a long de Broglie wavelength, suppressing dynamical friction on kpc scales (see [476] for a review). However, a significant reduction in the density fluctuations on this scale would make it difficult to form dwarf galaxies in sufficient numbers (see Section 6 of [449]). It is also necessary to solve other serious problems with ΛCDM discussed elsewhere in this review, many of which are on much larger scales where the above models would lead to much the same behaviour as ΛCDM (see in particular Sections 7 and 8). We therefore argue that it is untenable to solve the many problems faced by ΛCDM merely by making a small adjustment to the properties of DM on kpc scales. Indeed, this is unlikely to be a viable solution even if we only consider evidence on this scale [477].

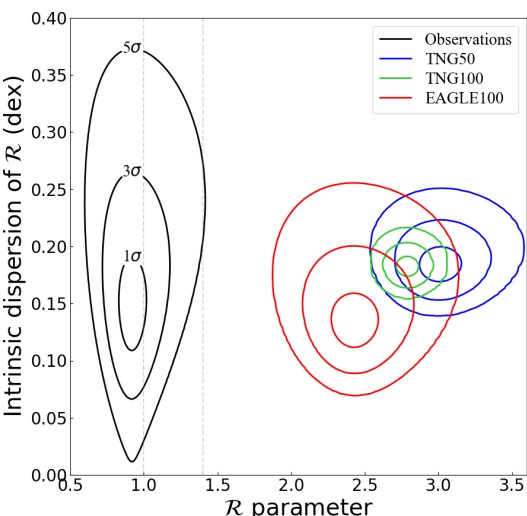

**Figure 21.** Posterior inference on the mean $\mathcal{R}$ and its intrinsic dispersion for the low-redshift galaxy population according to observations (black [465]) and the $\Lambda$CDM cosmological simulations EAGLE100 (red), TNG100 (green), and TNG50 (blue). In all cases, the $1\sigma$, $3\sigma$, and $5\sigma$ confidence regions are shown. Notice the consistency between the different $\Lambda$CDM simulations, whose differently sized error ellipses are correlated with differences in the sample size. Reproduced from Figure 8 of [449].

In summary, $\Lambda$CDM appears to get incorrect the fraction of galaxies that are thin discs (Figure 14), the fraction of these discs that have bars (Figure 18), and the pattern speeds of the bars which do form (Figure 21). The number of spiral arms is also less than predicted, indicative of an incorrect dispersion relation for perturbations to the disc (Section 4.2). Since the RC and velocity dispersion are measurable, the issue is almost certainly with the self-gravity term. Therefore, $\Lambda$CDM does not correctly address gravitational dynamics in galaxies. Meanwhile, MOND provides a good physical explanation for the general trends apparent in observations, with our intuitive understanding backed up by several pioneering $N$-body and hydrodynamical simulations. Further work is required to address the behaviour of Milgromian disc galaxies in a cosmological context.

## 5. Interacting Galaxies and Satellite Planes

We can learn a great deal about the properties of particles by colliding them at high energy. Similarly deep insights might be gained by studying interactions between galaxies. In this section, we consider what such studies reveal about the likely composition of galaxies and the gravity law that governs them. As observations improve, an important test using interacting galaxies may become possible (Section 11.2).

### 5.1. Tidal Stability

One of the most common types of galaxy interaction is when a primary galaxy with mass $M_p$ tidally disturbs an orbiting dwarf companion with mass $M_c \ll M_p$. Tidal stability considerations imply a maximum companion size or tidal radius $r_t$ at fixed $M_c$ and distance $R$ from the host. The basic principle is that the dwarf self-gravity should be comparable to the difference in host gravity across the spatial extent of the dwarf. If the host exerts gravity $g_e$ on the dwarf, then the approximate Newtonian condition for the maximum allowed dwarf size $r_t$ is

$$\overbrace{\frac{GM_c}{r_t^2}}^{g_{N,i}} \approx g_e' r_t \,, \tag{65}$$

where $'$ is used to denote a radial derivative. Notice that the host properties and the host-centric distance are relevant only insofar as they affect the tidal stress $g_e'$ on the dwarf.

The basic idea is the same in MOND. Since the difference in host gravity across the dwarf is comparable to its self-gravity $g_i$ at the tidal radius, it is clear that the host gravity itself must be much stronger than $g_i$. This means that at the tidal radius of a dwarf, its internal dynamics are necessarily EFE-dominated ($g_e \gg g_i$). Consequently, we can use Equation (34) to estimate $g_i$. Neglecting factors of order unity such as the angular dependence, we have that

$$g_i \approx \frac{GM_c a_0 \nu_e}{r_t^2},$$
(66)

where $\nu_e$ is the value of $\nu$ at the location of the dwarf due to $\mathbf{g}_e$ alone. For a point-like host and assuming the deep-MOND limit,

$$\nu_e = \sqrt{\frac{a_0}{g_{N,e}}} = R\sqrt{\frac{a_0}{GM_p}}.$$
(67)

In this configuration, the tidal stress is $g_e' = \sqrt{GM_p a_0}/R^2$, so the tidal radius of the dwarf is:

$$r_t \approx R\left(\frac{M_c}{M_p}\right)^{1/3}.$$
(68)

Including the factors of order unity that we have neglected so far, the author of [478] derived the following more accurate estimate (see its equation 12):

$$\frac{r_t}{R} = \overbrace{0.374}^{2^{1/6}/3}\left(\frac{M_c}{M_p}\right)^{1/3}.$$
(69)

The authors of [479] derived a generalization of this in their equation 27 under the approximation that $g_e$ follows a power law in $R$. This leads to additional corrections of order unity to account for the fact that in general, the host gravity does not follow an inverse distance law because it is an extended source not fully in the deep-MOND regime. It is possible to write their result in terms of $g_e$ and $g_e'$, the only aspects of the host potential which affect the internal dynamics of the dwarf. Notice that both $g_e$ and $g_e'$ are relevant in MOND due to the EFE, while only $g_e'$ is relevant in Newtonian gravity because it satisfies the strong equivalence principle (see Equation (65)).

In MOND, the tidal radius $r_t$ of a dwarf can be calculated without reference to its internal dynamics or its actual radius $r_h$. For a virial analysis (e.g., Equation (49)) to be valid in any theory, it is necessary that $r_h \ll r_t$. The authors of [244] showed in their Figure 6 that many dwarf spheroidal satellites of the MW would not satisfy this condition in MOND, which may well explain why they deviate from the BTFR (Equation (46)), or more generally from MOND expectations for objects in dynamical equilibrium. Indeed, there is a fairly tight correlation between the extent to which the dwarfs deviate from MOND equilibrium expectations and their sky-projected ellipticity (see their Figure 3). Such a correlation would have to be a coincidence in the ΛCDM framework because the analysed dwarfs should all have $r_h \lesssim 0.1\,r_t$, so tidal effects would be negligible and the elliptical images should have a different cause. It would be helpful to distinguish these scenarios by identifying or ruling out clearer signatures of ongoing tidal disruption.

An important consistency check on the MOND scenario is that none of the dwarfs analysed by [244] have $r_h \gg r_t$, as shown in their Figure 6. Interestingly, this shows that the distribution of $r_h/r_t$ marginally reaches 1 in MOND. This could be an indication that dwarf galaxies formed around the MW with a range of sizes and galactocentric radii, but dwarfs which were tidally unstable are no longer detectable. The fact that the distribution of $r_h/r_t$ has an upper limit of 1 in the MOND context would need to be a complete coincidence

in the ΛCDM context because the upper limit lies at 0.1 instead, which would have to be understood in some other way.

Tides are expected to be important in cases other than the MW, for instance in the Fornax galaxy cluster 20 Mpc away [480]. The dwarfs in this cluster have been surveyed quite deeply as part of the Fornax Deep Survey (FDS [481]). When the dwarfs are plotted in the space of surface brightness and projected separation from the cluster centre, a clear tidal edge is apparent (Figure 22). The location of this tidal edge could be a good way to understand how much self-gravity the dwarfs have, which is expected to differ substantially depending on the gravity law and whether dwarf galaxies have their own CDM halos. The FDS could thus provide a promising way to test ΛCDM and MOND based on their ability to reproduce this feature and other aspects of the Fornax dwarf population, especially given that a recent study found strong evidence tor tidal interactions playing a role (see Section 7.4 of [482]). Preliminary results suggest a significant problem for ΛCDM because the dwarfs should be immune to tides, but this issue seems to be resolved in MOND where they would be much more easily disrupted (see https://darkmattercrisis.wordpress.com/2021/08/02/, accessed on 23 May 2022). A similar analysis might also be possible in the Coma cluster [483,484]. In both cases, the results would be sensitive to the additional physics required to reconcile MOND with galaxy cluster observations, be this sterile neutrinos or something else (Section 7.1). While the discrepancy between the observed gravity and the MOND gravity of the baryons alone is larger in Coma, it is also apparent in Fornax [485].

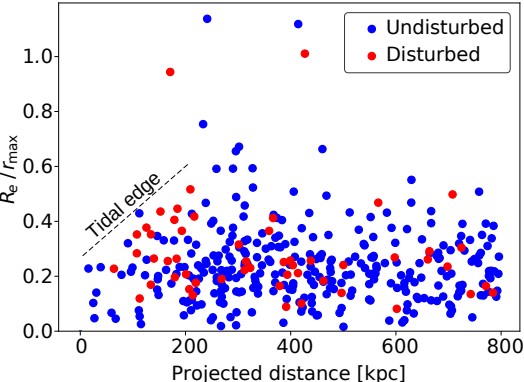

**Figure 22.** Data on dwarf galaxies in the Fornax cluster [481]. The *x*-axis shows the sky-projected separation from the cluster centre. The *y*-axis shows the ratio between the optical effective radius $R_e$ and the maximum radius $r_{max}$ that the dwarf could have given its stellar mass to still remain detectable above the survey sensitivity. The colour of each point indicates whether the dwarf appears disturbed (red) or undisturbed (blue) in a manual visual classification [482]. There is a clear lack of dwarfs above the diagonal line marked as a tidal edge. Notice that dwarfs just below this edge are much more likely to appear disturbed. LSB dwarfs are missing close to the cluster centre, but these are clearly detectable in the survey as many such dwarfs are evident further out. The predicted location of the tidal edge is expected to differ between theories, so comparison with the observed location should be a promising way to distinguish them. Credit: Elena Asencio.

The higher tidal susceptibility of dwarfs in MOND is due to the EFE and the lack of protective CDM halos. As a result, dwarfs which should be suitable for an equilibrium analysis in ΛCDM might not be amenable to this in MOND. This fact was not fully appreciated in the MOND virial analysis of [486], leading to the erroneous conclusion that MOND faces a "possibly insurmountable challenge" explaining the high observed $\sigma_i$ in some cases. While they considered the EFE, they did not consider whether tides invalidated the assumption of equilibrium, even though tides were extensively discussed in the ΛCDM part of their work. In principle, it is much simpler to estimate the tidal susceptibility of a galaxy in MOND because the purely baryonic nature of galaxies makes it much easier to estimate their mass. Only dwarfs with a low tidal susceptibility in MOND are suitable for

analysis with Equation (49) or its generalization in Figure 5. More generally, this can be inaccurate if the internal crossing time within a dwarf becomes comparable to its orbital period around its host [244,310]. This is especially true for an eccentric orbit because if the EFE changes sufficiently quickly, it is difficult for the dwarf to adjust, leading to a memory effect lasting a few internal crossing times [264]. However, our results indicate that this is only a major issue if the dwarf $r_h$ exceeds its MOND tidal radius [478,479]. This is supported by the very small memory effect experienced by DF2 as it orbits NGC 1052 in the simulation results shown in Figure 6.

### 5.2. Tidal Streams and EFE-Induced Asymmetry

If a dwarf satellite galaxy has $r_h > r_t$, we expect it to undergo tidal disruption. By this, we mean that stars would escape through the inner or outer Lagrange point and enter an independent host-centric orbit little affected by the disrupting dwarf. This would lead to a tidal stream, which might be detectable for a long time—possibly even outliving the satellite itself. The shape and kinematics of the tidal stream can be used to set constraints on the host potential, especially if a gravitationally bound remnant is still recognisable and its kinematics can be measured. Due to the faintness of tidal features and the importance of observing them in detail to obtain possibly fundamental insights about gravity, we will focus on tidal streams around the MW. However, it is sometimes possible to conduct meaningful analyses of tidal streams around other galaxies (e.g., around M31 [487]).

One of the best-studied and most prominent galactic tidal streams is that of the Sagittarius dwarf satellite, which goes out far beyond the galactic disc [488]. Using $N$-body models in a $\Lambda$CDM context to constrain the shape of the galactic potential, those authors found that the MW halo needs to be quite round, with an aspect ratio $> 0.7$. This was a priori not expected, as mentioned in their abstract. Another argument for a near-spherical halo is the observed bifurcation of the Sagittarius tidal stream [489], which suggests that its orbital pole has precessed very little. As orbital pole precession is generically expected in a non-spherical halo, they argued for a nearly spherical halo.

Since there is no direct evidence for the DM halo of the MW, it is possible to choose a near-spherical halo consistent with the Sagittarius tidal stream. This would make for a similar halo to the phantom halo in MOND, which would be almost spherical at distances exceeding a few disc scale lengths [102], though the EFE from other galaxies becomes relevant at distances of several hundred kpc and causes the potential to become flattened in a well-defined manner (as discussed in Section 2.4; see also [66]). Consequently, it is difficult to distinguish between $\Lambda$CDM and MOND using the Sagittarius tidal stream [490]. Even so, tidal streams provide an important test of the much less flexible MOND scenario. Moreover, other constraints can be used to check whether the DM halo properties required to fit the Sagittarius tidal stream are reasonable in a $\Lambda$CDM context.

The benchmark $\Lambda$CDM model of the Sagittarius tidal stream involves a triaxial DM halo whose intermediate axis aligns with the disc [491]. The failure to reproduce the observed stream bifurcation might indicate that the Sagittarius dwarf was originally rotating [492]. Rotation has indeed been detected in its present-day remnant [493]. Even so, the intermediate axis configuration proposed by [491] appears to be unstable [494] and is not consistent with the tidal stream of the globular cluster Palomar 5 [495]. They identified 'stream fanning' as a characteristic signature of a triaxial halo, but the observed lack of stream fanning in Palomar 5 argues against this. One reason is that the lower mass of Palomar 5 makes for a colder tidal stream, allowing its morphology to be studied in more detail. This bodes well for future surveys that should uncover additional faint tidal streams. A reasonably good match can be obtained to the Sagittarius tidal stream in CDM-based models by including a massive LMC [496] and allowing more flexibility in the halo density distribution. Their high required LMC mass of $(1.3 \pm 0.3) \times 10^{11} M_\odot$ is consistent with the mass inferred from a $\Lambda$CDM timing argument analysis of the LG [497–499]. These results are consistent with a more thorough 3D timing argument analysis that gave the LMC a mass of $2.03 \times 10^{11} M_\odot$ [500], as explained in footnote 1 to [501].

The first study to consider the Sagittarius tidal stream in MOND was able to fit most of its observed properties quite well despite very little flexibility [502]. For instance, the total luminosity of the stream tightly constrains the initial progenitor mass, with the stellar mass–size relation for field galaxies then constraining the initial size [503]. The only major deficiency identified by [502] was with the RVs in part of the leading arm, an issue also faced by Newtonian models (see their conclusions). In addition to the possibility that stars in this sky region assigned to the stream are not truly part of it, the leading arm properties are better reproduced if the MW is assumed to have a hot gas corona flattened similarly to the MW satellite plane (Section 5.6). Though the models used QUMOND, results should be similar in AQUAL [65].

The Sagittarius tidal stream is sufficiently distant that dynamical friction on the progenitor would be negligible in the MOND scenario. This is not necessarily true in ΛCDM, where a sufficiently massive progenitor moving through a sufficiently dense halo would experience strong dynamical friction, altering the path of the tidal stream and its kinematics. As a result, observations place an upper limit on how much dynamical friction the Sagittarius progenitor might have experienced, which in turn constrains a combination of its mass and that of the MW halo [504]. Their work highlights a path to a potentially clear falsification of MOND: unambiguous evidence for a satellite experiencing dynamical friction despite being outside the baryonic distribution of the host galaxy. Null detection would set an upper limit to the dynamical friction, which may be more stringent than how much dynamical friction is expected in ΛCDM. This would favour the MOND scenario.

MOND has also been applied to the tidal stream of Palomar 5 [317], though without attempting to obtain a detailed fit. Those authors explored the EFE in greater detail, finding that it might explain the observed asymmetry between the leading and trailing arms. While the EFE-dominated point mass potential (Equation (34)) is symmetric with respect to $\pm g_e$, the potential becomes asymmetric if the internal and external gravity are comparable, as discussed in more detail in Sections 4.4 and 4.5 of [106]. The asymmetry of the Palomar 5 tidal stream might have other explanations such as a perturbation by the galactic bar [505] or by a subhalo consisting almost entirely of CDM [506]. However, the latter would be a random process because the subhalo could pass through the leading or trailing arm, so the asymmetry could well be in the opposite sense to that expected in MOND. More such examples would be required to identify if the agreement with MOND in the case of Palomar 5 is purely coincidental or part of a more general trend.

*5.3. Polar Ring Galaxies*

Polar ring galaxies [507] consist of a disc galaxy with a substantially misaligned ring of usually gaseous material (for a review, see [508]). These offer an interesting probe of gravity because RCs can be determined for two separate planes in the same system. Dynamical models of polar rings in Newtonian and Milgromian gravity reveal that MOND can naturally explain the higher rotation velocity in the polar ring compared to the host [215]. While this can be explained to some extent in the Newtonian model, the MOND interpretation was favoured by [509] due to its smaller degree of flexibility. As explained in their conclusions, the higher rotation velocities in the polar ring arise from the orbits here being close to circular, while orbits in the host galaxy are more eccentric due to perturbations caused by the polar ring. The difference arises mainly because the polar ring tends to be more extended than the host galaxy, presumably to avoid destructive disc crossings. As a result, the host galaxy can be considered a point mass at the centre of the polar ring, thus yielding a nearly axisymmetric potential around the ring. However, if the polar ring has a non-negligible mass compared to the host galaxy and is not too distant, then the potential in the host galaxy cannot be axisymmetric with respect to its rotation axis [215]. The higher induced eccentricity of orbits in the host reduces its observationally inferred rotation velocity, which theoretically can be calculated by considering a test particle on a closed orbit moving through the numerically determined potential (see their Section 4).

## 5.4. Shell Galaxies

The photographs of some galaxies reveal shells [510], which likely arose when a dwarf galaxy encountered a much more massive host [511]. The basic idea is that a dwarf galaxy on a nearly radial orbit is disrupted at pericentre. When the liberated stars reach apocentre, they spend a relatively large amount of time in a narrow range of radii, leading to a much higher surface brightness in a photograph [512].

Shell galaxies test several aspects of galaxy physics. The dwarf is expected to undergo dynamical friction, affecting the pericentre times. Some dynamical friction is expected in MOND as the baryonic parts of the galaxies would typically overlap at pericentre, but in the outer parts of the orbit, dynamical friction would be significant only in the CDM scenario. In addition, the galactic potential also affects the shell positions.

The authors of [512] argued that the observed shells in NGC 3923 rule out MOND, but their work suffered from several very serious problems that completely invalidate their conclusion [513]. In addition to observational difficulties and the possibility of missing shells, the model may not be realistic at very small radii because the dwarf would in general not be on a purely radial orbit and would generally start losing stars before reaching pericentre. While these issues can be mitigated with deeper imaging and excluding very small radii from the fits, a more complicated issue is that the dwarf may not be completely destroyed at pericentre. As a result, each shell can be assigned to one of many pericentric passages of the dwarf. The particular passage chosen is known as the generation of the shell. A plausible dynamical model should fit all the shell radii quite well using only a small number of generations, assume only a small number of missing shells, and perform poorly only at very small radii.

The shells of NGC 3923 have been analysed more recently in the MOND context [514]. Using a semi-analytic method, those authors obtained a very good fit to the positions of 25 shells out of the observed 27 using a model with three generations. The maximum deviation in shell radius was only 5.4%, with the mismatch correlating well with the shell's positional uncertainty. The two shells that could not be matched well are at very low radii, but these might be part of the fourth generation—a possibility not considered by the authors in order to limit the flexibility of their model.

The use of shells to test MOND was discussed further in [515,516], with the latter authors also considering MOG (subsequently falsified in Figure 12). In general, the strong dynamical friction expected in the ΛCDM scenario suggests that large faint shells should be accompanied by small bright shells because the launch velocity would drop substantially between pericentres [516]. A more uniform distribution of shell radii would be expected in the absence of CDM, which is more in line with observations of NGC 3923 [517]. More information can be gained by taking spectra and determining the LOS velocity distribution of the shells, which are expected to have a quadruple-peaked profile [518].

Some of these techniques were recently applied to the shell galaxy NGC 474 [519]. Their *N*-body MOND simulation of it agrees well with the shell positions and even with the velocity dispersion of the shell where a comparison was made. In general, the authors "were able to find a scenario that fits all available constraints well." The flexibility of the model is fairly small due to the lack of CDM and the use of only two shell generations arising from two pericentre passages, a consequence of the first interaction having been only 1.3 Gyr ago. The more flexible Newtonian model also yielded an acceptable match, though it proved difficult to produce a long tidal stream similar to the northern stream, possibly due to the stronger dynamical friction at large separation in this model. Further exploration of the parameter space may yield an improved match. Even if that were the case, we consider the less flexible MOND approach to be favoured by its fairly good agreement with the tests conducted so far in shell galaxies (Equation (72)).

## 5.5. Tidal Dwarf Galaxies (TDGs)

When gas-rich galaxies interact, tidal forces can pull out a thin stream of stars and gas known as a tidal tail [520]. If this tail is dense enough, part of it can undergo self-

gravitating Jeans collapse into a bound structure, which for some range of properties would be considered a TDG. The formation of TDGs has long been known from an observational perspective, e.g., in the interacting galaxies known as the Antennae [521] and the Seashell [522]. For a review of TDGs, we refer the reader to [523].

If only a few TDGs are produced in each interaction and survive to the present, then most dwarf galaxies might actually be TDGs [524]. Later studies suggested that TDGs probably comprise only a few percent of dwarf galaxies at low redshift [525]. However, this is only a lower limit because the tidal features that allow dwarfs to be confirmed as TDGs are expected to fade away as the tidally expelled material becomes more diffuse. Whether the TDG also gets tidally disrupted is less clear, but the existence of confirmed TDGs up to 4 Gyr old suggests that they can be long-lived [526]. It can therefore be very difficult to distinguish ancient TDGs from primordial dwarfs, especially if their properties are similar because both are purely baryonic. As a result, the actual frequency of TDGs is not reliably known. The strong correlation between the interaction histories of nearby galaxies and how many satellites they have could be an indication that TDGs are actually commonplace, with the interaction history quantified from the bulge fraction [53,527,528] or from more detailed stellar population studies [529].

One possible distinguishing feature of TDGs regardless of the gravity law is that they are expected to be relatively metal-rich for their mass [526,530,531]. Unfortunately, even this signature would be undetectable for a really ancient TDG because the progenitor disc galaxies would not have been very enriched at sufficiently early times [530]. This raises the possibility that many nearby dwarf galaxies are actually ancient TDGs [524], which in turn means that we might be able to measure the $\sigma_i$ of a nearby TDG. In the $\Lambda$CDM context, $\sigma_i$ would be rather low [532–535]. This would cause a significant discrepancy with MOND expectations as dwarfs generally have a low surface brightness that puts them in the MOND regime (Equation (45)). It is therefore possible to falsify MOND by observing a virialized TDG if the $\Lambda$CDM scenario is correct, even if the tidal origin of the dwarf is unclear. However, if MOND is correct, the $\sigma_i$ of a TDG would be typical of a primordial dwarf, so a TDG governed by MOND would generally be misinterpreted as a primordial dwarf in the $\Lambda$CDM context. Table 2 clarifies the $\sigma_i$ expectations of both paradigms for dwarfs that formed in different ways.

**Table 2.** Expectations for the internal velocity dispersion of a dwarf galaxy depending on how it formed and the gravity law. If Newtonian gravity applies, the cosmological context is assumed to be the $\Lambda$CDM paradigm in which primordial galaxies form inside CDM halos. TDGs would be more common in MOND due to their stronger self-gravity making them more difficult to disrupt.

| | Expected $\sigma_i$ if Gravity . . . | |
| Origin of Dwarf | Newtonian | Milgromian |
| --- | --- | --- |
| Primordial | High | High |
| Tidal | Low | High |

Unlike in MOND, the self-gravity of a $\Lambda$CDM dwarf galaxy differs substantially depending on how it formed. We discuss later the possibility of directly testing this through $\sigma_i$ or $v_c$ measurements of a virialized TDG, which represents a very challenging measurement (Section 11.2). The difference between the matter content of similarly luminous primordial and tidal dwarfs in the $\Lambda$CDM context causes them to follow distinct tracks in the mass–radius plane [531]. However, the splitting evident in their Figure 9 is not observed when we compare field dwarfs with confirmed TDGs [503]. This problem is evident in their Figure 2, reproduced here as our Figure 23. These results strongly suggest that there is no fundamental distinction between dwarf galaxies that formed primordially or out of tidal debris. Since TDGs should be purely baryonic, the most logical conclusion is that all dwarfs are purely baryonic, thus favouring the modified gravity solution to the missing gravity problem on galaxy scales.

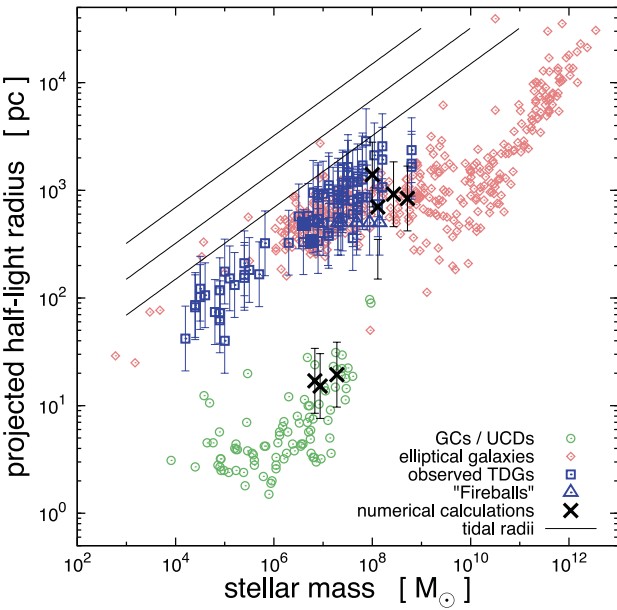

**Figure 23.** The observed mass–radius relation of globular clusters and ultra-compact dwarfs (green circles), ordinary and dwarf elliptical galaxies (red diamonds), and TDG candidates (blue squares). Ram pressure-stripped TDGs (blue triangles) were discussed further in [531]. The black crosses show TDGs in numerical CDM simulations [534]. The solid black lines show tidal radii 100 kpc from a galaxy whose mass in $M_\odot$ is (from top to bottom) $10^{10}$, $10^{11}$, and $10^{12}$. Notice that observed TDGs do not define a distinct sequence in the mass–radius plane, even though this is expected in $\Lambda$CDM [531] due to their lack of CDM, which makes their self-gravity much weaker for the same baryonic content (see Table 2). Reproduced from Figure 2 of [503].

### 5.6. The Local Group Satellite Planes

Due to the lack of any fundamental difference between primordial and tidal dwarfs in MOND, the critical issue becomes how to demonstrate that a dwarf galaxy is a TDG. One silver lining is that as with a primordial dwarf, the self-gravity of a TDG would be enhanced by the modification to gravity, making TDGs much more robust to perturbations such as tides and gas removal by stellar feedback. TDGs are thus expected to be more common in MOND (as shown with numerical simulations [536]). This raises the possibility that a rather nearby dwarf galaxy is actually a TDG, allowing it to be observed in sufficient detail to both measure $\sigma_i$ and confirm its tidal origin.

The nearest dwarfs are the satellites of the MW and M31, which have elevated $\sigma_i$ values that reveal large amounts of missing gravity. With the MW satellites, it has been demonstrated that this cannot be explained by tides in a Newtonian context [244]. It would moreover be very unusual if we were observing so many dwarfs right at the moment of dissolution, even if this might be a viable explanation in some cases [537,538]. The dwarf spheroidal satellites of M31 also have rather high $\sigma_i$ values consistent with MOND expectations [240,241]. This is not obviously problematic for $\Lambda$CDM if the LG satellites are mostly primordial dwarfs (Table 2). If on the other hand they are mostly TDGs, this would rule out $\Lambda$CDM. We therefore review whether the LG satellites are better understood as primordial or as tidal dwarfs.

To try and distinguish these possibilities, we exploit the fact that TDGs would typically form a phase space-correlated structure such as a satellite plane [531,539]. Anisotropy of the satellite system might therefore be one of the longest-lasting lines of evidence for the TDG nature of a dwarf. Such evidence would necessarily be statistical in nature as many individual dwarfs are needed to detect the anisotropy. Nonetheless, the large number of LG dwarfs (e.g., [540]) suggests that this could be a promising approach.

The classical satellite galaxies of the MW do indeed form a highly flattened system orthogonal to the galactic disc, as was already evident prior to the formulation of

MOND [541,542]. This was later confirmed from the positions of subsequently discovered satellites [543]. Their larger sample size of 16 finally allowed the authors to conclude that the anisotropic distribution of galactic satellites strongly contradicts ΛCDM expectations. Today, it is known that the ultra-faint satellites, globular clusters, and tidal streams also align with the plane of classical satellites [544]. This realization has led to the notion that the orbital motion of material in this plane came about due to a past encounter between the MW and another galaxy [539]. Importantly, proper motion data confirm that most classical MW satellites share a common orbital plane [545,546] aligned with the plane normal defined by the satellite positions alone [547]. Their velocities show a significant bias towards the tangential direction, as occurs for a rotating disc [548–550]. This is highly suggestive of a dissipational origin, and was certainly not predicted a priori in the ΛCDM context [551,552]. The velocity data are quite crucial to the argument because if the galactic satellites are distributed isotropically but observations cover only one great plane on the sky, the distribution of velocities would still be random relative to this plane.

Such incomplete sky coverage effects can be minimized by considering instead M31, whose satellite region subtends a much smaller region of the sky. M31 has been homogeneously observed by the Pan-Andromeda Archaeological Survey [553,554]. This allowed [555] to confirm earlier hints of a satellite plane around M31 [556]. Approximately half of the M31 satellites are part of its dominant satellite plane, a much smaller fraction than for the MW. Since we observe the M31 satellite plane almost edge-on, a coherently rotating satellite plane should have a gradient in RV across it, which is observed [555]. Proper motions have recently become available for the plane members NGC 147 and NGC 185, demonstrating that their 3D velocities align with the plane within uncertainties [557]. This significantly worsens the problem for ΛCDM [558]. For a review of the LG satellite planes and the recently discovered one around Centaurus A (Section 5.7), we refer the reader to [559], who also considered whether various proposed scenarios might or might not provide viable explanations for these structures. The authors of [560] explored correlations between satellites more broadly, in particular by also considering satellite pairs and asymmetry of the satellite distribution. Some of the review is spent on discussing possible explanations to the observations which do not sit well with ΛCDM, while also highlighting the areas in good agreement. The satellite planes sit alongside other problems for ΛCDM on galaxy scales, many of which seem to be resolved in MOND [357]. Their review also considers other modified gravity theories such as MOG and standard gravity solutions where the dark halos of galaxies have constituents with different properties to CDM.

After carefully considering several proposed scenarios for how primordial CDM-rich satellites might come to lie in a thin plane, the authors of [561] concluded that none of them agree with observations for either the MW or M31. Structures as anisotropic as their satellite planes are extremely rare in cosmological simulations [562,563], including hydrodynamical simulations [546,564,565] and simulations which approximately include the effects of a central disc galaxy [566]. Previous arguments against the group infall and filamentary accretion scenarios [561,567] were later independently confirmed [568] using the EAGLE simulation [446,447], thereby including hydrodynamics in a ΛCDM cosmological context. If group infall is somehow the solution, then the most plausible scenario is that the LMC brought in a significant number of the classical MW satellites, in which case they cannot be considered independently. Group infall is already included in cosmological ΛCDM simulations, but it was nonetheless suggested that the LMC could have brought in enough MW satellites to explain the MW satellite plane in ΛCDM [569]. However, the LMC could not plausibly have brought in enough satellites to explain the whole galactic satellite plane [570]. Indeed, a more recent study showed that the LMC should have brought in "about 2 satellites with $M_\star > 10^5 \, M_\odot$" [571]. Given also that the direct gravitational effect of the LMC is insufficient to induce a sufficiently strong clustering of orbital poles [572–574], some other explanation is required for the 8/11 classical MW satellites that orbit within a common plane [546].

In summary, each LG satellite plane is in $3.55\sigma$ tension with ΛCDM (Table 3 of [501]). The problems faced by ΛCDM with these structures motivate us to reconsider their origin,

especially given past misunderstandings that if corrected lead to a much bleaker assessment for standard cosmology [575]. The anisotropy of the LG satellite planes strongly suggests a tidal origin in both cases, but the TDG scenario is not viable in ΛCDM because TDGs are already included in hydrodynamical cosmological simulations such as Illustris [531]. Their very low frequency makes this an unlikely explanation. Even if TDGs were more common, the high $\sigma_i$ of the MW and M31 satellites rules out this possibility if gravity is Newtonian at low accelerations (Table 2).

Both objections to the TDG scenario disappear in MOND. It therefore offers the possibility that the highly flattened LG satellite planes formed as TDGs, which we know from observations can form a flattened satellite system [521,522]. We have seen that it is extremely difficult to come up with an alternative explanation. Moreover, MOND provides a very natural answer to the question of which galactic interaction(s) formed the LG satellite planes. This is because the stronger long-range gravity between the MW and M31 acting on their nearly radial orbit [146,576,577] implies that they must have experienced a close encounter $9 \pm 2$ Gyr ago [578]. This is based on the two-body force law in the deep-MOND limit (Equation (40)). For the MW–M31 system, the MW mass fraction is close to 0.3, in which case the factor $Q = 0.7937$ [92].

The MW and M31 are so widely separated that the EFE on the whole LG must be considered. If the EFE dominates, the point mass potentials can be superposed because the gravitational field is linear in the matter distribution (Equation (34)). However, neither the isolated nor the EFE-dominated approximation is valid because the LG is in the intermediate regime. This requires numerical calculations of $g_{\rm rel}$, the results of which can be fit fairly well analytically [92]. In addition, tides from M81, IC 342, and Centaurus A should also be considered, though fortunately the EFE must be provided by more distant sources and can therefore be treated as acting on all these objects in a similar manner. By taking these factors into account, the authors of [92] showed that MOND is consistent with the classical LG timing argument (Section 6.1) for a timing argument mass compatible with baryons alone. Their work reaffirmed that a past MW–M31 flyby is required by MOND, or else the timing argument mass would become extremely small and fall below the directly observed baryonic mass within the LG (see also [579,580]).

*N*-body simulations of the MW–M31 flyby in MOND produce a tidal tail between the galaxies, suggesting that the LG satellite planes may have formed due to the flyby [581]. This problem was also investigated using restricted *N*-body models where the MW and M31 were treated as point masses surrounded by test particle discs [92]. The tidal debris around each galaxy generally had a preferred orbital pole. In some models, this aligned with the observed satellite plane orbital pole for both the MW and M31 (see their Figure 5). This work has recently been followed up with hydrodynamical MOND simulations of a flyby between two disc galaxies carefully chosen to resemble the MW and M31 as they might have looked 9 Gyr ago [582]. Their major result is the orbital pole distribution of the material in the satellite region of each galaxy, i.e., within 250 kpc but outside the disc (see their Figures 7 and 8, the important aspects of which are reproduced in our Figure 24). Taken in combination with several other results from their paper which are not shown here, the model provides a good description of the MW and M31 discs and satellite planes, including their present-day orientations, disc scale lengths, and the MW–M31 separation. Despite not being considered when selecting the best model, it is consistent with the observed M31 proper motion [577]. Note that since the models have a rather high temperature floor and do not allow star formation, individual TDGs do not form, so the main aim was to compare the preferred orbital pole of the tidal debris with the corresponding observations [545,583]. It seems very likely that satellite planes of some sort would be produced from a past MW–M31 encounter, especially as this could also have triggered the rapid formation of the galactic bulge [584] and thick disc [585], which could be related to the formation and subsequent buckling of the galactic bar due to tidal torques [586]. The flyby might also explain the secondary peak in the age distribution of young halo globular clusters in the galactic halo (Figure 9 of [587]). In MOND, TDGs

would naturally have high $\sigma_i$, just like any other galaxy in the weak-field regime. Besides explaining their high observed $\sigma_{LOS}$, their enhanced self-gravity would make them more robust to tides and feedback than TDGs in ΛCDM [536]. On a larger scale, the flyby is a promising explanation for why the RVs of galaxies in the NGC 3109 association exceed the ΛCDM prediction by $\approx 100$ km/s (Section 6.1).

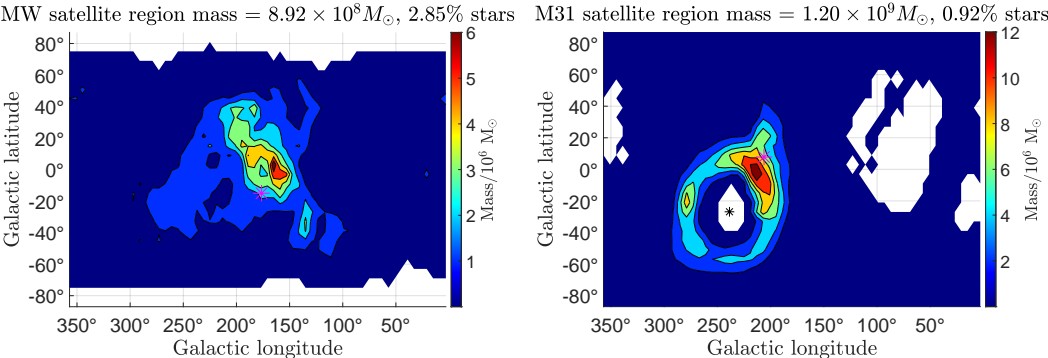

**Figure 24.** Results from a hydrodynamical MOND simulation of the MW and M31 discs interacting with each other 7.65 Gyr ago, showing the present-day orbital pole distribution of material within 250 kpc but outside the disc of the MW (**left**) and M31 (**right**). We show the mass in $6° \times 6°$ squares in galactic coordinates, as indicated in the colour bar. The total mass in the considered region is indicated at the top, along with the fraction of this in stars. The MW disc spin vector is at the south galactic pole by convention (not shown). The preferred angular momentum of its satellite galaxies is indicated with a magenta star in the left panel. The M31 disc spin vector is shown with a black star in the right panel, while the preferred angular momentum of the galaxies in its satellite plane is shown with a magenta star. The lack of material whose angular momentum closely aligns or anti-aligns with that of each disc is caused by the disc-shaped excluded region having its symmetry axis aligned with that of the corresponding LG galaxy. For both the MW and M31, notice the good agreement between the preferred tidal debris orbital pole in the simulation and the preferred orbital pole of its observed satellite galaxies, which were obtained from Section 3 of [545] and Section 4 of [583], respectively. Adapted from Figures 7 and 8 of [582] by its lead author.

Regardless of the specific MOND scenario of a past MW–M31 flyby, the LG satellite planes are strongly suggestive of a tidal origin. The high $\sigma_i$ of their members then requires a departure from GR. Given the large body of prior work on the satellite plane issue (reviewed in [559,560]), this reasoning is likely to withstand the test of time. Moreover, any scenario for the LG satellite planes should not be too contrived because satellite galaxy planes are not unique to the LG.

### 5.7. Satellite Planes beyond the Local Group

One of the closest major galaxies beyond the LG is Centaurus A. Its satellites were initially thought to define two planes [588], but later observations clarified that both are part of one thicker plane which includes Centaurus A [589]. Subsequent results confirm the existence of a thin kinematically coherent satellite plane around Centaurus A [590]. Such a structure is expected in ΛCDM in only 0.2% of cases [591]. This work also showed the found analogues are just chance alignments, presumably because TDGs are very rare in ΛCDM (see Figure 16 of [531]). Therefore, the Centaurus A satellite plane poses a significant challenge to ΛCDM despite the present lack of $\sigma_i$ measurements.

If this structure is interpreted as caused by a past interaction, then the perturber may have already merged with Centaurus A. Alternatively, its satellite plane might have formed due to a past interaction with M83, around which there is some evidence for a satellite plane [592].

Some hints for anisotropy in the satellite distribution are also evident around NGC 2750 based on 7 satellites, which may form part of a lower mass analogue to the satellite plane

around Centaurus A [593]. There is also evidence for a satellite plane around M81 [594] and NGC 253 [595,596].

It is difficult to constrain the 3D shape of the satellite distribution around a more distant host galaxy. However, if we are fortunate enough to view a satellite plane close to edge-on, then satellites on opposite sides of the host galaxy would have RVs with opposite signs once the host RV is subtracted. Such an anti-correlation has been reported [597]. It was later claimed to not be a significant detection [598], but the concerns raised in this work were later addressed [599]. Those authors used a large galaxy sample from SDSS to confirm the signal and its unexpected nature in ΛCDM. While this remains a matter of conjecture, there is certainly at least one satellite plane beyond the two in the LG, so these structures should be reasonably common in a viable cosmological model.

## 6. Galaxy Groups

MOND was originally motivated by the flat outer RCs of disc galaxies [56]. By definition, these regions must lie well into the MOND regime. Since gravity declines with distance even in MOND, the gravity between galaxies should lie deep in the MOND regime, as is the case for the MW and M31 [92]. If the mutual acceleration between galaxies can be measured, it might help to distinguish Newtonian from Milgromian gravity. For this, we should primarily consider the phase space coordinates of galaxy barycentres, thus treating them as point masses. This is especially helpful when studying groups of galaxies, which can be analysed in detail if they are sufficiently nearby to know the internal structure of the group. We describe some previous work in this regard for the LG (Section 6.1) and other galaxy groups (Section 6.2). Further afield, binary galaxies offer a rather direct way to test Equation (40) (see Section 6.3). It is also possible to analyse a distant galaxy group statistically by applying the virial theorem, provided it has enough members (Section 6.4).

### 6.1. The Local Group and the NGC 3109 Association

One of the oldest yet still valid lines of evidence for the missing gravity problem is the LG timing argument [12]. The basic idea is that the gravity between the MW and M31 must explain how they turned around from their presumed post-Big Bang recession as they are currently approaching each other (e.g., [576]). We first discuss the timing argument in ΛCDM to highlight certain problems, before explaining how these might be resolved in MOND.

If we consider the MW and M31 as the only two point masses in an otherwise homogeneous ΛCDM universe and assume a radial orbit, it is straightforward to show that their total mass is $M_{\rm LG} \approx 5 \times 10^{12} M_\odot$ (e.g., [600]). This estimate relies on the usual assumption of no past close MW–M31 flyby because a past flyby would cause a much higher $M_{\rm LG}$ [601] and lead to significant dynamical friction during the encounter, inevitably causing a merger rather than subsequent separation to large distance (e.g., [55,371]). Since $M_{\rm LG}$ and the unknown initial separation can both be adjusted, we have two model parameters. These are sufficient to match the two observational constraints, namely the galactocentric distance and RV of M31. Clearly, this is not a strong test of ΛCDM—agreement with observations is guaranteed in a model where galaxies can have arbitrarily large amounts of invisible mass. Moreover, the mass ratio between the MW and M31 is impossible to constrain based purely on the mutual gravity between them.

Fortunately, we can consider other LG galaxies, including especially dwarf galaxies that can be used as tracers of the velocity field. Such analyses have a long history, with an early attempt by [602] finding it difficult to simultaneously explain all the data available at that time. Improvements in survey capabilities led to an updated catalogue of LG dwarfs [540], which typically forms the basis of more recent timing argument analyses. One such example is the study of [497], which found that matching observations requires an additional source of uncertainty with magnitude $35 \pm 5$ km/s. Both these analyses treat the LG potential as spherically symmetric, thus assuming the tracer dwarf galaxies are

much further from the MW and M31 than their mutual separation of $783 \pm 25$ kpc [540]. This is only approximately correct for dwarf galaxies within 3 Mpc.

The first axisymmetric dynamical timing argument analysis of the LG was conducted by [603], who showed how the equations governing the Newtonian timing argument can be derived from GR. By then, it was clear that the MW–M31 orbit is indeed almost radial [576], which was later confirmed [146,577]. Centaurus A can also be included in an axisymmetric model of the LG because it lies close to the MW–M31 line [604]. This allowed [603] to consider LG dwarfs as test particles in the gravitational field of these three massive moving objects. The computational costs were kept low, allowing them to investigate a wide range of model parameters using a full grid search. Despite this, a good fit was never obtained, even when considering that the model is not a perfect representation of ΛCDM—the expected model uncertainty is only 30 km/s based on the scatter about the Hubble flow in detailed *N*-body simulations [605]. However, some LG dwarfs have a receding RV $> 100$ km/s in excess of the best-fitting model prediction [603]. While several LG dwarfs have unexpectedly high RVs, there are very few cases where the RV is unexpectedly low (see their Figure 9). The typical mismatch between predicted and observed RVs was estimated to be $45^{+7}_{-6}$ km/s.

To check whether this apparently serious problem for ΛCDM might be alleviated in a more advanced 3D model of the LG, the authors of [499] applied the algorithm described in [606]. In the best-fitting model of [499], the typical RV mismatch was slightly higher than in the 2D case, with a clear asymmetry evident in their Figures 7 and 9 such that an unexpectedly high RV occurs much more often than an unexpectedly low RV. This is despite including all the major galaxies in and around the LG out to almost 8 Mpc, as listed in their Table 3. Though phrased more positively for ΛCDM, similar results were obtained by [607] using a similar algorithm. The authors of [500] borrowed this algorithm from Peebles and improved it in certain important respects to give it the best possible chance of finding dwarf galaxy trajectories compatible with the ΛCDM timing argument (see their Section 4.1). However, this had only a small impact on the results.

The high RVs of some LG dwarfs outside the MW and M31 virial volumes can partly be understood simply by reducing the LG timing argument mass. This is of course not a solution to the problem as the masses of LG galaxies are already treated as free parameters in timing argument analyses, but it is helpful to bear this in mind when considering other works. In particular, attempts to constrain $M_{\mathrm{LG}}$ by considering all LG dwarfs can be expected to yield a lower $M_{\mathrm{LG}}$ than estimates considering only the MW and M31. This could explain the unusually low $M_{\mathrm{LG}} = (1.6 \pm 0.2) \times 10^{12} M_{\odot}$ inferred in the former manner by [608]. However, the authors of [609] obtained a much higher timing argument mass of $M_{\mathrm{LG}} = 4.4^{+2.4}_{-1.5} \times 10^{12} M_{\odot}$ by searching cosmological ΛCDM simulations for LG analogues based on properties of the MW and M31 alone, especially with regards to their relative separation and velocity. This estimate is in line with earlier results and simple analytic estimates that consider only the MW and M31 [600]. The mass of M31 alone has been estimated at $1.9^{+0.5}_{-0.4} \times 10^{12} M_{\odot}$ based on its giant southern tidal stream [487]. $M_{\mathrm{LG}}$ must be higher as we also need to include the MW and material outside the major LG galaxies. Thus, several timing argument analyses of the whole LG have found that the RVs of some LG dwarfs are too high to easily explain, though the tension is sometimes phrased as an anomalously low $M_{\mathrm{LG}}$ rather than as a problem for the currently popular ΛCDM paradigm.

In addition to the timing argument, ΛCDM also faces a problem with the abundance matching relation between the stellar and halo masses of nearby galaxies. The prescription is that the observed number density of dwarf galaxies per unit logarithmic interval in $M_{\star}$ tells us their halo mass based on which DM halos in a cosmological simulation have the same number density per unit logarithmic interval in the halo mass [293,610]. This simple idea encounters difficulties in the LG because the Newtonian dynamical mass of the MW [611] is lower than its abundance matching mass given its $M_{\star}$ [612]. Those authors showed that the same problem arises in a more significant way around M31, whose

dynamical mass can be estimated from modelling its giant southern tidal stream [487]. Lower values arise from considering the M31 RC, worsening the problem [613,614]. Another way to look at the discrepancy is that given the combined dynamical mass of the MW and M31, the stellar mass in the LG should be just that of the MW, not the MW and M31. Reducing the LG mass to better fit the kinematics of non-satellite LG dwarfs would worsen this problem.

The high-velocity galaxies (HVGs) identified by [500] are all part of the NGC 3109 association, at least if we exclude HIZSS 3 due to its very low galactic latitude of $0.09°$ and KKH 98 due to the RV mismatch being only $65.5 \pm 9.1$ km/s (see their Table 3). This almost linear association [615] consists of several dwarf galaxies moving away from the LG with rather high RVs. The NGC 3109 association was studied in more detail by [246] in light of other LG structures such as the satellite planes (Section 5.6). Their work reached a similar conclusion regarding the high RVs. Viewing the problem backwards in time, the association should have been close to the MW in the past. This suggests that an earlier gravitationally bound NGC 3109 galaxy group came close to the MW and gained orbital energy at that time, which is most easily understood as caused by a three-body interaction. Indeed, timing argument calculations such as that of [500] do not perfectly represent ΛCDM because they lack dynamical friction and thus galaxy mergers, which are however expected in ΛCDM. During such mergers, galaxies can temporarily have high relative velocities. A dwarf near the spacetime location of the merger could then gain orbital energy from the time-dependent LG potential. This leads to 'backsplash galaxies' or backsplashers, defined as objects on rather extreme orbits that were once within the virial radius of their host but were subsequently carried outside of it. This backsplash process was studied in detail by [616], who found it very difficult to get backsplashers at the $1.30 \pm 0.02$ Mpc distance of NGC 3109 [617]. This is also evident in Figure 3 of [618], which additionally shows that backsplashers are very rarely more massive than $10^{10} M_{\odot}$ regardless of their present position. In a ΛCDM context, NGC 3109 has a mass of at least $4.0 \times 10^{10} M_{\odot}$ [619], so both its mass and its galactocentric distance are unusually high for a backsplasher. The distance from M31 is even higher at almost 2 Mpc, so backsplash from M31 appears even less plausible.

Early studies of the backsplash process considered only a small number of analogues to the LG in ΛCDM simulations, e.g., the authors of [618] considered just one. Recently, it has become possible to revisit this issue with the advent of high-resolution hydrodynamical cosmological ΛCDM simulations such as IllustrisTNG [445]. TNG300 is well suited for this because its volume is large enough to capture many thousand galaxies similar to the MW. The authors of [501] used TNG300 to quantify the distribution of backsplashers in terms of present mass and distance from the host within whose virial radius the backsplasher once was. The results are shown in our Figure 25, which reproduces their Figure 8. To account for the high RV of NGC 3109, it is necessary to gain energy during a past interaction with the host, so the backsplasher considered as a point mass is required to leave the host with at least as much energy as it came in with (see their Section 5.3). The end result is that there are no backsplashers with similar properties to NGC 3109 around any of the 13225 identified analogues to the MW or M31, of which 640 are in 320 LG-like paired configurations. The results remain similar when considering the DM-only version of TNG300, demonstrating their robustness to how baryonic physics is handled. The issue with the backsplash scenario is that dynamical friction between the CDM halo of NGC 3109 and the MW or M31 makes it likely that orbital energy would be lost rather than gained during any close interaction. However, a more distant interaction would be rather weak, so its effect should be correctly handled in a few-body timing argument analysis of the LG. It is therefore very difficult to understand the anomalously high RVs of galaxies in the NGC 3109 association compared to ΛCDM expectations [499,500,607]. The problem is evident even without doing a detailed timing argument calculation and thereby dropping the requirement for NGC 3109 to be a backsplasher: its observed RV is higher than the typical RVs of galaxies with a similar mass and at a similar distance to a host resembling the MW or M31 ($P = 1.09\%$; see Section 4.1

of [501]). If the NGC 3109 analogue is also required to be rather isolated (as is the case for the actual NGC 3109 group), then the tension worsens ($P = 0.72\%$).

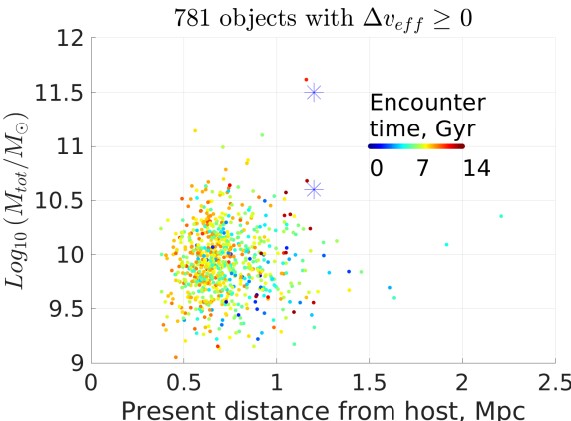

**Figure 25.** The distribution of total mass and distance from the host for backsplashers in the TNG300 ΛCDM cosmological simulation. The colour is used to indicate the cosmic time when the backsplasher interacted with its host, from which it may well be unbound today. Results are based on 13225 host galaxies similar to the MW or M31, not all of which have an associated backsplasher. The blue stars show different estimates for the mass of NGC 3109 in this framework. Its present galactocentric distance is well known [617,620], with the $5\sigma$ lower limit used here. The backsplashers are required to have not lost energy during the interaction with their host, as this would make it even more difficult to explain the observed high RV of NGC 3109 compared to ΛCDM expectations (see the text). There are no backsplashers with a higher mass and distance from the host than NGC 3109 is from the MW. The distance from M31 is ≈2.0 Mpc (not shown). Reproduced from Figure 8 of [501].

In the MOND context, the lack of CDM halos around galaxies removes the objection regarding dynamical friction. Moreover, MOND requires a past close MW–M31 flyby (Section 5.6), during which their high relative velocity would have created substantial time variation of the LG potential. This would have substantial effects on any dwarf near the spacetime location of the flyby. Using a restricted *N*-body model of the LG in MOND, it was shown that some LG dwarfs might well have been gravitationally slingshot outwards at high speed [500]. These dwarfs could certainly reach the distance of NGC 3109 (and even Leo P) relative to the LG barycentre, with the gravitational slingshot causing a much higher present RV than galaxies at the same position that never closely approached the MW or M31. Moreover, their Figure 6 shows that it would not be necessary for a dwarf galaxy that experienced such an encounter to have passed very close to the MW or M31 in order to reach a large present distance with a high RV, thus limiting damage to, e.g., the disc of NGC 3109. Even so, it seems inevitable that a galaxy group would be tidally disrupted, perhaps explaining the filamentary nature of the NGC 3109 association [615] as an imprint of a past physical association that is gravitationally unbound today [621]. It could be a structure analogous to a tidal stream, but traced by galaxies rather than stars. MOND therefore appears to provide a more holistic explanation for the LG satellite planes and the larger-scale velocity field out to ≈2 Mpc, all of which are very difficult to understand in ΛCDM despite much effort by the community over many decades.

### 6.2. M81 and Hickson Compact Groups

One of the nearest galaxy groups is the M81 group, which lies at a distance of only 3.6 Mpc [622]. Its properties are well explained by an interaction between its three main members (M81, M82, and NGC 3077) in a Newtonian context [623], with good agreement between the times of close interactions and properties of the star formation histories [624]. However, the models of [623] neglected dynamical friction, which is certainly not valid in ΛCDM as including this process leads to the rapid merging of all three galaxies [625].

This problem has been revisited more recently, but the conclusions remain similar [626]. The only possible scenario is if all three galaxies recently encountered each other from large distances, which seems quite unlikely—yet needs to have occurred in one of the nearest galaxy groups beyond the LG. Moreover, galaxies should have been receding after the Big Bang, but the models require them to have been approaching each other for several Gyr at nearly constant velocity (see their Figure 14). This implies a significant velocity perturbation at very early times, but cosmic structures and peculiar velocities then were much less pronounced than today, posing a similar problem to NGC 3109 and the LG (Section 6.1). In other words, the timing argument is likely to be violated in a scenario where three galaxies are currently rapidly approaching each other for the first time, as would be required to allow for dynamical friction.

The existence of compact galaxy groups may pose problems for the ΛCDM scenario in which galaxies are surrounded by extended CDM halos, since dynamical friction between these halos would cause a rapid merger [55,371]. We would therefore need to observe the group immediately before it merges, possibly posing a fine-tuning problem similar to the M81 group. However, many so-called Hickson Compact Groups have been identified [627–629]. It has been argued that their observed frequency can be explained in ΛCDM because some are only chance alignments [630]. This is surely not always the case, and genuine compact groups should merge within a few Gyr. However, those authors showed that it is possible for new groups to form out of accretion along filaments.

This highlights the degeneracy between scenarios such as ΛCDM with frequent mergers that are only detectable as such for a brief period, and scenarios where galaxy interactions are rare but would be detectable over most of a Hubble time due to very weak dynamical friction [536]. The observed frequency of interacting galaxies may be similar in both cases. Further work is required to break this degeneracy. One possibility is that frequent mergers between galaxies would lead to a low proportion of fragile thin discs, so their high observed fraction argues against this scenario (Section 4.1).

*6.3. Binary Galaxies*

An isolated binary galaxy offers a geometrically simple configuration in which the accelerations are typically deep into the MOND regime. Since we also know that MOND was not originally formulated with binary galaxies in mind, these can offer a strong test of the paradigm. The important MOND prediction is that the mutual gravity $g_{rel}$ between two otherwise isolated point masses in the deep-MOND regime is given by Equation (40). This can be translated directly into a prediction for the relative velocity $v_c = \sqrt{r g_{rel}}$ if the galaxies are on a circular orbit with separation $r$. The predicted $v_c$ is independent of $r$. We expect the actual distribution of the relative velocity $v_{rel}$ to be somewhat broader than a $\delta$-function at $v_c$ due to orbital eccentricities and inaccuracies in estimating $M_s$. However, these are rather modest effects compared to the much larger variations possible in Newtonian gravity due to scatter in the relation between baryonic and halo masses and the fact that $v_c$ follows a Keplerian decline, so a more widely separated pair should have a lower $v_{rel}$.

One complication is that observations from afar are sensitive to only the sky-projected relative velocity. However, this can be accounted for statistically because the relative LOS velocity should be uniformly distributed over the range 0–$v_{rel}$, allowing the distribution of $v_{rel}$ to be recovered using a large sample [631]. Those authors applied this technique to the HyperLEDA database [632] to obtain the Isolated Galaxy Pairs Catalog (IGPC [633]). The deprojection algorithm is discussed in more detail in [634]. Their main result is that the $v_{rel}$ distribution shows a clear peak at ≈150 km/s. The reason for this was unclear—their Section 5 goes off on a tangent about how exoplanets have a similar peak in their distribution of Keplerian velocities, but this is completely irrelevant to understanding galaxy dynamics. The Section also states that the results support the equivalence principle, even though this leads to GR and its consequent prediction that $v_{rel}$ should undergo a Keplerian decline with increasing separation, contrary to the observations.

To take advantage of subsequent additions to the HyperLEDA database, the authors of [635] reanalysed it to identify galaxy pairs using a similar technique to [634]. This confirmed the existence of a peak in the $v_{rel}$ distribution at $\approx 150$ km/s (see Figure 1 of [635]). Those authors pointed out that such a clear peak is rather unlikely in $\Lambda$CDM according to the prior CDM-only Millennium simulation [636], most likely due to the reasons discussed above. However, the peak is in line with MOND expectations due to the rather narrow range in $M_s$ and the fact that the predicted $v_{rel} \propto \sqrt[4]{M_s}$ (see Figure 7 of [635]). An updated version of this analysis has recently been published [637]. Its Figure 1 (reproduced here as our Figure 26) shows the distribution of $v_{rel}$, which has a clear peak at $\approx 130$ km/s for all three choices of width for the velocity bins used in the reconstruction.

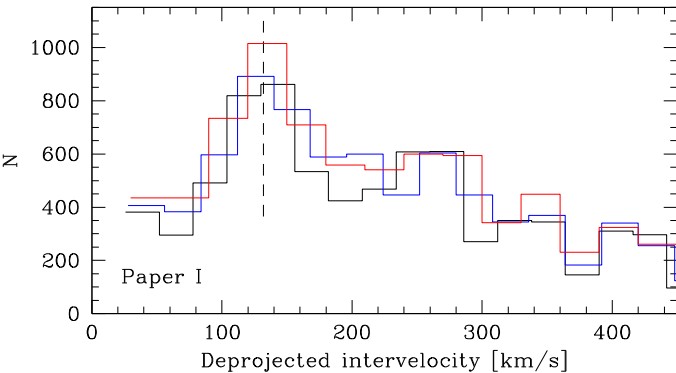

**Figure 26.** Deprojected distribution of $v_{rel}$ between 8571 isolated galaxy pairs. Different colours show different velocity bin widths used in the reconstruction. The dashed vertical line at 132 km/s marks the location of the very clear peak in the distribution, which is at a similar location in all three cases. Reproduced from Figure 1 of [637].

To assess whether these results are consistent with MOND, it is necessary to compare them with the circular velocities predicted by Equation (40). Even for the case of purely circular orbits and a perfect reconstruction of the $v_{rel}$ distribution, we still expect a spread in values because of differences in the total mass and mass ratio between galaxy pairs. The predicted distribution of $v_c$ is shown in Figure 27, which reproduces Figure 5 of [637]. The black line shows a Gaussian fit to the peak region in the observationally reconstructed $v_{rel}$ distribution. The red histogram shows the predicted distribution of $v_c$. This provides a rather good match to the observations. There is an excess of systems with a lower velocity than the predicted $v_c$, which could well be due to the fact that masses on an eccentric orbit spend more time near apocentre, where they are moving slower. In general, the clear peak in the $v_{rel}$ distribution is well understood in MOND as a consequence of a narrow range of total $M_s$ for each pair combined with only a weak dependence on this and little sensitivity to the mass ratio (Equation (40)).

These results only consider a narrow range of galaxy luminosity, so the galaxy pairs all have a similar total $M_s$. This makes it difficult to test the predicted scaling behaviour $v_c \propto \sqrt[4]{M_s}$. We can test this using galaxy pairs in the GAMA survey [389] thanks to the work of [638]. Their abstract indicates that the typical difference in the LOS peculiar velocity between two galaxies increases by a factor of 3 for a $100\times$ increase in luminosity, which translates to approximately the same increase in $M_s$. This would imply that the logarithmic slope of the relation is close to $\ln 3/\ln 100 = 0.24$, which is almost identical to the MOND prediction of $1/4$. In a $\Lambda$CDM context, there appears to be a problem at the low-mass end due to the abundance matching halo masses being too large, a problem which is also evident in the LG (Section 6.1).

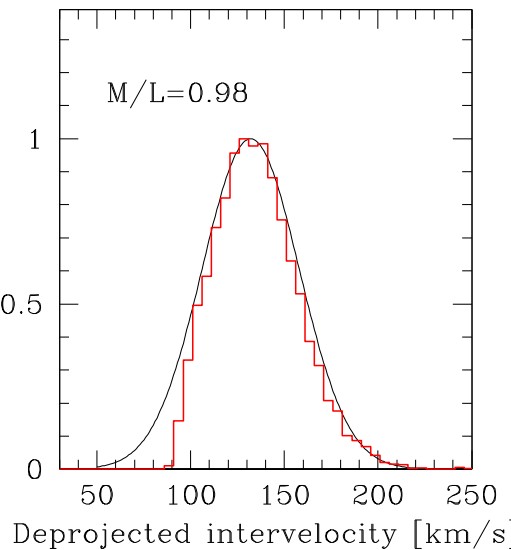

**Figure 27.** The red histogram shows the distribution of the MOND-predicted circular velocity (Equation (40)) between the binary galaxies considered in Figure 26. The black curve shows the Gaussian fit to the deprojected $v_{\mathrm{rel}}$ distribution shown in that Figure after binning more finely in the peak region. The best-fitting $M_\star/L_B$ is shown at the top left. Reproduced from Figure 5 of [637].

### 6.4. Virial Analysis of Galaxy Groups

In more distant galaxy groups, the data quality becomes poor such that only the most basic analyses remain possible. Assuming a galaxy group is in equilibrium, the simplest dynamical analysis is an application of the virial theorem to deduce the group dynamical mass. This can be compared with the mass estimated in other ways, including especially the photometric $M_s$.

Galaxy groups are generally in the deep-MOND limit because the gravity at, e.g., the solar circle of the MW is comparable to $a_0$ [1], so the gravity on neighbouring galaxies is much smaller still. Therefore, MOND analyses of galaxy groups generally make use of Equation (49), which is valid for an isolated spherically symmetric system in the deep-MOND limit. The Equation assumes virial equilibrium and an isotropic velocity dispersion tensor. These assumptions may not hold exactly, partly because the longer dynamical timescales of galaxy groups compared to individual galaxies can make the assumption of equilibrium less accurate. Even so, Equation (49) provides a good starting point to check whether MOND gives reasonable results in systems where the departure from Newtonian gravity is expected to be large.

This Equation was first applied to galaxy groups by [639], who found a dynamical $M/L < 10$ that might be explained using baryons alone. The author of [640] analysed 9 galaxy groups within 5 Mpc which exhibit often very large amounts of missing gravity in a conventional context. The study found that the MOND dynamical $M/L$ ratios were order unity. This remained true for groups with and without "luminous" galaxies, i.e., dwarf-only groups and groups containing more massive galaxies. However, these two types of group have significantly different proportions of missing gravity [641], which in a MOND context can be understood due to the dwarf-only groups having lower internal accelerations.

This topic was revisited in a MOND analysis of 53 galaxy groups [642], which was later extended to 56 medium-richness groups despite strict quality cuts [643]. The MOND dynamical $M/L$ ratios were found to be order unity in the near-infrared $K$ band within uncertainties, suggesting reasonable agreement. These results should be considered in light of dwarf galaxies also following Equation (49) (e.g., [253]). We provide a holistic picture in our Figure 28, which reproduces Figure 6 of [643]. It is clear that Equation (49) provides a good description for the dynamics of pressure-supported systems across 7 dex in $M_\star$, with the low-mass end probed by stars in dwarf galaxies and the high-mass end by galaxies in galaxy groups.

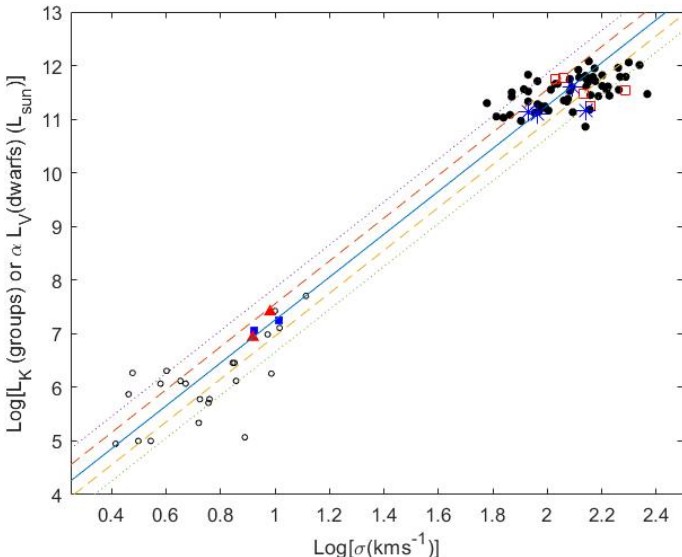

**Figure 28.** The *K*-band luminosity of galaxy groups in solar units as a function of their $\sigma_{\rm LOS}$ in km/s, with both shown on logarithmic axes (solid black dots towards the upper right). The open circles towards the lower left show dwarf galaxies, whose observed *V*-band luminosities have been scaled by $\alpha = 2/0.7$ to account for the higher expected $M_\star/L$ in the *V*-band, the goal being to correlate $\sigma_{\rm LOS}$ with mass. The solid blue line is the prediction of Equation (49) for $M_\star/L_K = 0.7$ [644], with the dashed (dotted) lines showing a factor of 2 (4) variation on either side. Notice the huge dynamic range in luminosity shown here. Reproduced from Figure 6 of [643] by permission of Moti Milgrom and the American Physical Society.

The author of [643] also attempted to detect the EFE in terms of groups with a very small internal acceleration having a lower $\sigma_{\rm LOS}$ than implied by Equation (49), which is valid only for isolated systems. However, no strong evidence was found for or against the EFE assuming a typical external field strength of $0.03\,a_0$ (e.g., [92,93,107]). One reason could be that groups would experience a different EFE depending on their environment, so assuming the same EFE on all groups is inaccurate. At the galaxy scale, the EFE preferred by MOND RC fits depends on the environment of the galaxy [319,321], so it would be natural if the same applies to galaxy groups. Checking this would require a large-scale map of the gravitational field (e.g., [320]).

While the 3D structure of a galaxy group is generally unknown, recent observations have clarified the 3D structure of the Centaurus A galaxy group and allowed a kinematic analysis [645]. Allowing for both rotation and random motion, those authors detected both at high significance based on the RVs of 28 member galaxies and distances for 27 of them. Both contribute similarly to supporting the system against gravity, whose strength can thus be inferred from the kinematics. The estimated group $v_c$ of $258 \pm 57$ km/s is consistent with the BTFR (Equation (46)). Since the internal gravity should dominate over the EFE out to 500 kpc [66] and most of the used tracers are much closer in, the kinematics of the Centaurus A group are consistent with MOND expectations [645]. This success is on a much larger scale than Centaurus A itself, which is also consistent with MOND [646]. In a $\Lambda$CDM context, the large size of this group and the low accelerations at its outskirts mean that a significant proportion of DM is required, much more than the cosmic fraction inferred from, e.g., fits to the CMB power spectrum. As a result, there should be $\approx 8 \times 10^{11} M_\odot$ of hot gas in this very nearby galaxy group [645]. X-ray observations indicate that the actual amount is $\approx 4\%$ of this out to the virial radius of 300 kpc [647].

In addition to roughly spherical galaxy groups, it is also possible to test MOND with filamentary structures such as the Perseus–Pisces superfilament [648]. The observations were found to be broadly consistent with MOND, though the uncertainties were large because it was necessary to make various simplifying assumptions [649]. Nonetheless, the study is interesting because it probed very low accelerations (only a few percent of $a_0$),

which in a Newtonian analysis leads to a very large proportion of missing gravity [650]. Moreover, the linear mass distribution is rather unique—typical RC analyses only probe a disc geometry, and even then the critical outer parts of the RC are mainly sensitive to the total baryonic mass. While the inner parts of the RC are somewhat more sensitive to the disc geometry, such analyses mainly test the force law for an isolated point mass (Equation (5)), especially if the focus is on the outer regions (e.g., [160]).

In the ΛCDM paradigm, the above-mentioned observational results merely indicate how much DM each system should have. It is difficult to judge whether this is reasonable. As with galaxies, in principle a wide range of DM fractions would be possible, so the agreement with MOND is not a priori expected. We conclude that this agreement could pose a fine-tuning problem similar to that faced by ΛCDM in galaxies (Section 3). The problem is less severe in galaxy groups due to the possibility of interlopers and the smaller number of tracers in each group, among other difficulties.

## 7. Galaxy Clusters

The largest gravitationally bound structures in the universe are galaxy clusters. Their importance to the missing gravity problem goes back almost a century [10,11]. The very large amounts of missing gravity claimed in the earlier works were mostly due to an erroneous assumption that most of the baryonic mass lies in the galaxies. Galaxy clusters actually contain a very significant component of intracluster gas, but this only became detectable much later with the advent of space-based X-ray telescopes [651]. However, this very substantial addition to the baryonic mass budget of galaxy clusters is still not sufficient to resolve the missing gravity problem on this scale (for a review, see [652]). In a standard context, the missing gravity is assumed to come from CDM. As we discuss below, this problem is reduced but not completely alleviated in MOND. We will argue that this does not rule out the MOND scenario, but it does place important constraints on how MOND can be made to work in a broader cosmological context.

### 7.1. Internal Dynamics

Galaxy clusters have internal accelerations that are typically of order $a_0$ or slightly above, so MOND can provide only a small amount of extra gravity. It has long been known that this is insufficient to completely solve the missing gravity problem in galaxy clusters [327,653,654]. It is sometimes argued that the MOND dynamical mass is only twice the detectable $M_s$, making for a much smaller discrepancy than with Newtonian gravity (e.g., [655]). While this may be correct, the discrepancy in MOND must still be understood. We argue it is unlikely that the very substantial mismatch between the MOND dynamical mass and the directly detected baryonic mass in galaxy clusters arises from some form of baryonic matter that has thus far evaded detection (though see [655]).

An important clue is that if we assume the MOND dynamical mass is correct and also that our inventory of baryons in galaxy clusters is nearly complete, then the implied baryon fraction matches that required by standard fits to the CMB power spectrum and the acoustic oscillations therein [656]. Part of the cluster baryon budget is the intracluster light due to stars outside galaxies [657]. Including this component, the authors of [658] obtained baryon fractions consistent with the Planck determination [80]. Other studies also assign galaxy clusters a baryon fraction quite close to that implied by the CMB, especially at the high-mass end [659,660]. Since the accelerations there are generally large enough for MOND to have only a small impact, it seems logical that the universe as a whole contains a dominant non-baryonic component.

Colliding galaxy clusters provide important clues regarding where the extra gravity comes from, mainly because they allow for more detailed studies beyond just the radial profiles of the gravitational field and the enclosed baryonic mass. A crucial observation in this regard is the Bullet Cluster (discovered by [661]). Its interacting nature was noted by [662], who identified the Bullet sub-component and thereby revealed the ongoing interaction. This is also apparent in Chandra X-ray observations of a clear bow shock [663].

The importance of the Bullet Cluster stems from the offset between its X-ray and weak lensing peaks, which has been detected at high significance [664,665].

The conventional interpretation is that the clusters consisted of hot gas embedded in DM halos. When the clusters collided at high velocity, the gas experienced substantial hydrodynamical drag, causing it to slow down. However, the collisionless DM halos were not decelerated in this manner. As a result, the DM halos reached a larger post-interaction separation than the gas, explaining why the weak lensing peaks are more widely separated than the X-ray peaks. This morphology can be reproduced in idealized non-cosmological Newtonian simulations of the merger [666].

It is of course not wise to place undue emphasis on any individual system, especially when the Train Wreck cluster seems to have the opposite properties to the Bullet Cluster (as briefly reviewed in [667]). One reason is that filaments near a system in projection can have non-trivial effects on the weak lensing signature in MOND due to its non-linearity [343]. In any case, structures along the LOS can also create a weak lensing peak, even if this may seem a lucky coincidence that is not the most favoured interpretation. However, a chance alignment of this sort is often advocated for the Train Wreck cluster because a central concentration of DM would not be expected. In general, disagreement of a theory with observations can often be attributed to errors with the observations or their interpretation, but this logic is rarely applied when a popular theory agrees with observations. Just as a claimed falsification of a theory might be a mistake, so too can a seemingly correct prediction be a mistake caused by a coincidence between incorrect observations and incorrect predictions. Nonetheless, it could be argued that only one chance alignment is needed in the Train Wreck cluster but two are needed in the Bullet Cluster to explain its two observed weak lensing peaks. It is therefore probable that there is indeed an offset between its X-ray gas and $\nabla \cdot \boldsymbol{g}$, the source of the gravitational field. Such offsets have also been detected in 37 other clusters [668].

Mainly due to this offset, the Bullet Cluster is quite difficult to reproduce in MOND using only its directly detected baryonic matter [669]. However, the Bullet Cluster can be reconciled with MOND using extra collisionless matter in the form of neutrinos with a mass of $2 \, \mathrm{eV}/c^2$ [670]. A large offset between the weak lensing and X-ray peaks is clearly evident in their Figure 1, which we reproduce as our Figure 29. The authors of [670] suggested that ordinary neutrinos could be sufficient, but $2 \, \mathrm{eV}/c^2$ is now known to be above the Katrin cosmology-independent upper limit on the ordinary neutrino mass, which is only $0.8 \, \mathrm{eV}/c^2$ at 90% confidence [671,672]. Moreover, even neutrinos with a mass of $7 \, \mathrm{eV}/c^2$ would be insufficient to explain the lensing caused by some galaxy clusters in a MOND context [673].

These difficulties led to the hypothesis of an undiscovered sterile neutrino with a mass of $11 \, \mathrm{eV}/c^2$ [674]. While significantly more massive than the ordinary neutrinos, this is still so low that galaxy RC fits would not be affected [675]. This is not due to the high velocities of the neutrinos—the CMB temperature of $T = 2.725 \, \mathrm{K}$ corresponds to an energy of $kT = 2.35 \times 10^{-4} \, \mathrm{eV}$, implying that the universe has expanded $47{,}000\times$ since any such neutrinos became non-relativistic. In the absence of structure, their typical velocities today would therefore be approximately $c/47{,}000 = 6.4 \, \mathrm{km/s}$. This is only $\approx 1\%$ of the local escape velocity from the MW [325,326]. Sterile neutrinos with a mass of $11 \, \mathrm{eV}/c^2$ are nonetheless unable to form substantial galactic halos around MW-like discs due to the Tremaine–Gunn limit [676]. The important constraint is the phase space density of the neutrinos, which are subject to the Pauli Exclusion Principle. As a result, halos of gravitationally bound sterile neutrinos are actually expected around galaxies, but they would be dynamically irrelevant even if all available phase space states were filled. Galaxy clusters are not only larger, they also have a deeper potential and thus a higher velocity dispersion, increasing the available phase space volume. Therefore, it is possible for sterile neutrinos to be dynamically relevant in galaxy clusters but not in galaxies.

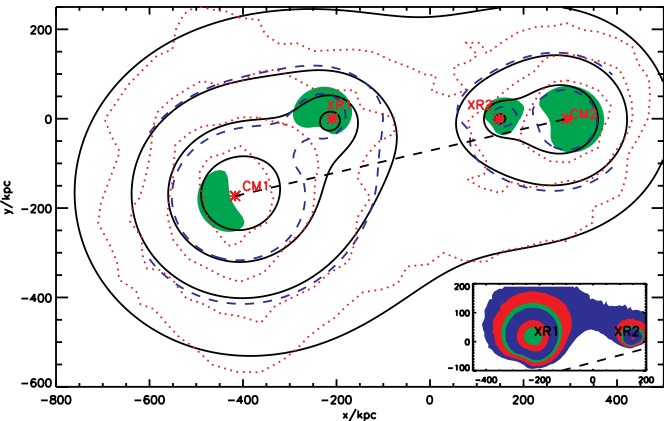

**Figure 29.** Weak lensing convergence map of the Bullet Cluster, shown using contours in a MOND model with 2 eV/$c^2$ sterile neutrinos (solid black lines) and as observed (dotted red lines). A four-centred potential is used, with two centres corresponding to the gas (red stars labelled XR) and two centres corresponding to the neutrinos (red stars labelled CM). The inset shows the gas surface density in this model. Notice the good match to the weak lensing data [665]. Reproduced from Figure 1 of [670] using ApJ author republication rights.

Assuming MOND is the correct theory, we can infer the mass distribution and subtract the baryonic component, thereby identifying where the sterile neutrinos must be hiding. This was attempted by [677], who then addressed whether the inferred density distribution of the sterile neutrinos is consistent with the Tremaine–Gunn limit. Strikingly, the neutrinos just reach this limit at the centre of the cluster in all 30 cases considered, at least if plausible assumptions are made regarding the $M_\star/L$ of the brightest cluster galaxy. This suggests that the sterile neutrinos accreted onto a halo until they reached the Tremaine–Gunn limit, preventing further accretion (at least in the central regions). While some other interpretation could be advanced, we argue that it would be a rather unlikely coincidence if the observations are caused by CDM particles with a vastly higher Tremaine–Gunn limit to their mass density.

The sterile neutrino solution to the MOND cluster problem is also suggested by the MOND galaxy cluster analysis of [678]. Though the overall discrepancy (at large radii) is smaller than in previous studies, their Figure 6 shows that the extra dynamical mass required in MOND is in a centrally concentrated component, as might be expected for a dense sterile neutrino core at the Tremaine–Gunn limit. Moreover, if sterile neutrinos with a mass of 11 eV/$c^2$ were thermalized in the early universe, they would have the same relic abundance as the CDM in the $\Lambda$CDM paradigm, thus explaining the acoustic oscillations in the CMB ([93,674]; see also Section 9.2.1).

There are also some hints for sterile neutrinos in terrestrial experiments (for a review, see [679]). While the latest results disfavour the sterile neutrino interpretation of the anomalies in short-baseline experiments [680], this is far from ruled out at present [681]. One reason for the apparent inconsistencies between different terrestrial experiments could be the approximation of the neutrinos as a plane wave [682]. A recent paper considering a wide range of experiments found $> 5\sigma$ evidence for sterile neutrinos [683]. This is largely due to neutrino flux measurements at two distances from the radioactive chromium-51 source in the Baksan Experiment on Sterile Transitions (BEST [684–686]). These experiments favour sterile neutrino masses significantly above 1 eV/$c^2$, a region of parameter space which is often not analysed due to the difficulties this would pose for $\Lambda$CDM. However, it is important to conduct particle physics experiments without being biased by prior beliefs regarding the law of gravity at low accelerations.

Substructures identified in galaxy cluster lenses could pose a tight constraint on sterile neutrinos due to the Tremaine–Gunn limit. In this regard, the straight arc around Abell 2390 was found to be consistent with 11 eV/$c^2$ sterile neutrinos in a MOND context [687]. Other

cluster-scale lenses could pose further constraints. In principle, the sterile neutrinos could have a higher mass, but their mass should be $\lesssim 300 \, \text{eV}/c^2$ to avoid disrupting MOND fits to the internal dynamics of galaxies [675].

If MOND is correct, some explanation must be advanced for why galaxies obey it but galaxy clusters seemingly do not. We have argued above that this is because galaxy clusters have much deeper potentials and larger sizes, both of which increase the phase space volume available to sterile neutrinos. As a result, sterile neutrinos at the Tremaine–Gunn limit would significantly affect galaxy clusters but not galaxies.

The deeper potentials in galaxy cluster environments may lead to a new regime of gravity that is not apparent in galaxies. This has given rise to the hypothesis known as extended MOND (EMOND [97]), which promotes $a_0$ to a dynamical variable that takes on higher values in regions with a deeper potential. This idea was explored further by [688]. While initial results were somewhat promising, we prefer the sterile neutrino interpretation because it would also explain the high third peak in the CMB power spectrum (Section 9.2.1).

We should of course bear in mind the possibility that some modification to gravity can account for these observations using only the directly observed $M_s$. For example, MOG can account for the weak lensing properties of the Bullet Cluster using only its observed mass [689]. While MOG has now been excluded at high significance by many interlocking lines of evidence (Section 3.6), it remains possible that some other modification to GR could satisfy the relevant constraints (e.g., [690]). This is suggested by the fact that the total matter density in ΛCDM exceeds the baryonic density by a factor very close to $2\pi$ rather than by many orders of magnitude, as might be expected if the mean baryon and CDM densities were set by very different physics [691].

The cluster-scale difficulties with modified gravity approaches can be interpreted as a sign that we should return to the ΛCDM picture, which can statistically account for the CMB anisotropies [80] and seems able to account for the internal dynamics of galaxy clusters thanks to the presence of CDM in large amounts. The problem then becomes how to explain the successes of MOND (e.g., [72,211,307]) and the failures of ΛCDM (e.g., [54,55,379]). A hybrid approach (elaborated in Section 9.2) seems to be the best way to explain available observations across all scales.

*7.2. Probing Structure Formation*

The distribution of galaxy clusters is sensitive to the growth of structure from the small observed density perturbations in the CMB. The observed low-redshift cluster mass function seems broadly consistent with ΛCDM over the mass range $3 \times 10^{12} M_\odot$–$5 \times 10^{14} M_\odot$ [692], though this is subject to the normalization of the relation between cluster mass and X-ray luminosity [693,694]. The general expectation in MOND is that there should be deviations from ΛCDM in terms of galaxy clusters forming with too high a mass too early in cosmic history [74]. This is also apparent from *N*-body simulations [695,696] using a MOND cosmology that we discuss later (Section 9.2.2).

The Bullet Cluster was thought to provide just such a challenge. Initial modelling attempts implied a very high collision velocity [697–699]. This leads to significant tension with ΛCDM expectations [700,701]. It was later shown that the morphology can be reproduced in idealized non-cosmological ΛCDM simulations of the merger with a lower infall velocity [666]. The parameters of their simulation are not thought to be seriously problematic for ΛCDM [702,703]. Indeed, it has been estimated that 0.1 analogues are expected over the full sky out to the Bullet Cluster redshift of 0.30 [704].

A much more problematic galaxy cluster collision is El Gordo [705,706]. This has a higher redshift, mass [707], and ratio of infall velocity to escape velocity [708–710]. Collisions between individually rare massive galaxy clusters should be quite unlikely as it is necessary to form two massive clusters within striking distance. By considering the properties of galaxy cluster pairs that have turned around from the cosmic expansion in the Hubble-volume Jubilee cosmological ΛCDM simulation [711], it was shown that El Gordo

rules out ΛCDM at 6.16σ significance [712]. The falsification is illustrated in their Figure 7, which we reproduce as our Figure 30. This shows the mass–redshift distribution along our past lightcone of galaxy cluster pairs that have turned around from the cosmic expansion with similar dimensionless parameters to El Gordo. The solid red contour passes through the parameters for the pre-merger configuration of El Gordo, shown with a red cross.

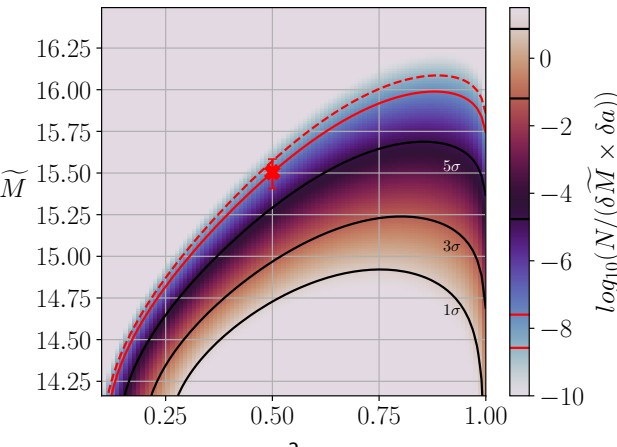

**Figure 30.** The mass–redshift distribution in ΛCDM of El Gordo-like galaxy cluster collisions in a 755 deg² region of sky along our past lightcone. The probability density is shown using the indicated colour scale in the parameter space of the cosmic scale factor *a* when the collision occurred and the total virial mass *M* of the clusters, expressed here as $\widetilde{M} \equiv \log_{10}(M/M_\odot)$. The red cross shows the El Gordo parameters, with the error bar showing a 20% uncertainty on its *M*. The total probability outside each contour is used to find its level of significance, which is indicated on the figure. The contour passing through the El Gordo parameters is shown with a solid red line representing the locus of 6.16σ outliers to ΛCDM expectations, the probability beyond which is $7.51 \times 10^{-10}$. Reproduced from Figure 7 of [712].

The pre-merger configuration was obtained from non-cosmological hydrodynamical simulations of the merger [710], with their best-fitting Model B used in the main analysis. Those authors were aware that their preferred pre-merger parameters would cause tension with ΛCDM, motivating them to consider a wide range of model parameters and check which ones might be compatible with observations of El Gordo. They ran 123 models altogether, with their preferred parameters in agreement with previous studies. However, some other models also gave plausible fits to the observations at some epoch if viewed from some direction. Consequently, it is important to look beyond their best-fitting model and consider a wider range of possible pre-merger configurations that evolve into something resembling the observed morphology of El Gordo. To this end, the parameters of the [710] Model A were also considered by [712], with the lower mass making the pre-merger configuration more likely in Jubilee. However, the tension in this case is still 5.14σ, so the initial configuration of the model could not plausibly arise in ΛCDM cosmology. If the pre-merger parameters are pushed into the range where consistency is gained, then it becomes impossible to explain the observed morphology of El Gordo. For instance, using too low a mass would cause disagreement with the observed X-ray flux, while too low a velocity would be inconsistent with the observed two-tailed morphology. It appears impossible for a Newtonian hydrodynamical simulation to reproduce the observed morphology of El Gordo using a pre-merger configuration that is plausible in ΛCDM.

The high El Gordo mass was recently confirmed in a detailed ΛCDM weak lensing analysis that gives it a total mass of $2.13^{+0.25}_{-0.23} \times 10^{15} M_\odot$ [713]. While slightly lower than the $3.2 \times 10^{15} M_\odot$ assumed in Model B of [710], it is slightly higher than the $1.95 \times 10^{15} M_\odot$ assumed in their Model A, which as mentioned above has initial conditions incompatible with ΛCDM cosmology. It is therefore very surprising that [713] claim their weak lensing measurements reconcile El Gordo with ΛCDM expectations. A closer examination of their

work reveals that this is certainly not the case because the slightly lower weak lensing mass only slightly reduces the tension for ΛCDM. Instead, the main reason for this claim is the use of a very low pre-merger infall velocity, which is not directly measured in a weak lensing analysis. The authors attempted to fit some aspects of the El Gordo morphology using a simplified analysis that does not include dynamical friction, which would be quite considerable for galaxy clusters as massive as the El Gordo progenitors. This might explain why [713] were unable to reproduce the observed locations of the radio relics. It is possible that including dynamical friction would lead to a better match, but then the preferred model parameters might differ greatly from what they obtained. A detailed parameter study of El Gordo using hydrodynamical simulations was indeed conducted previously by [710]. The only problem identified by [713] with this study was that the collision velocities were too high for ΛCDM, which might naturally arise in the MOND scenario due to an isolated point mass having a logarithmic potential in the deep-MOND limit. A Newtonian simulation of the interaction should still be quite accurate because the accelerations are above $a_0$ during the interaction phase and because galaxy clusters are expected to contain a dominant dark matter component even in MOND (Section 9.2). The authors of [710] were aware that their model parameters would be in tension with ΛCDM expectations, so they already tried lowering the mass and infall velocity of the El Gordo progenitors. This is possible to only a limited extent while remaining consistent with the observed morphology of El Gordo. Therefore, we reject the claim of [713] that El Gordo is compatible with ΛCDM—there are currently no Newtonian hydrodynamical simulations of it that match its observed properties and morphology with initial conditions that might plausibly arise in the ΛCDM framework. Rather, the great many hydrodynamical simulations that have been done so far indicate that the observed properties of El Gordo can arise only for a pre-merger configuration which is not feasible in ΛCDM. Some of these simulations use a slightly lower El Gordo mass as favoured by [713]. Moreover, those authors seem to favour the returning scenario for El Gordo whereby the clusters are observed when approaching each other for a second time. This would increase the elapsed time since the clusters first came into contact with each other, so the pre-merger configuration would need to exist at an even earlier stage in cosmic history, worsening the problem for ΛCDM (Figure 30).

In addition to El Gordo, there are several other galaxy clusters which might be problematic for ΛCDM [712]. For instance, scaling the Bullet Cluster results of [704] to the sky area of its discovery survey reveals that it presents $2.78\sigma$ tension [712]. Combined with El Gordo, the tension rises to $6.43\sigma$ (see their Section 3.4). The tension is so serious that it cannot be resolved by the discovery of no further problematic clusters for ΛCDM in the rest of the sky.

An important aspect of [712] was the use of previously conducted *N*-body cosmological MOND simulations [696] to estimate the occurrence rate of El Gordo analogues in MOND. Scaling their results to the effective volume of the survey in which El Gordo was discovered, we expect $\approx 1.2$ El Gordo analogues in MOND cosmology [712]. This is quite consistent with observations (see their Section 4.3). However, the expected number of El Gordo analogues in ΛCDM is only $7.51 \times 10^{-10}$ in the survey region (see their Section 3.3).

This demonstrates that the successes of MOND and the failures of ΛCDM extend well beyond the low-redshift universe and the galaxy RC data which originally motivated the MOND hypothesis [56]. El Gordo is on a large enough scale that it should be well resolved in cosmological simulations. Moreover, the large-scale shock features would be little affected by galaxy-scale baryonic feedback processes. Given the many detailed hydrodynamical simulations of El Gordo done in the past [708–710,714], it seems extremely unlikely that a Newtonian simulation of it will be found to reproduce its detailed morphological properties while also using a pre-merger configuration consistent with ΛCDM cosmology. The very large size of the Hubble-volume Jubilee cosmological simulation and the large number of cluster pairs used in the analysis of [712] argues against uncertainties on the ΛCDM side. Indeed, Figure 8 of [692] shows that very few galaxy clusters with the El Gordo mass are expected at its redshift of 0.87 [706]. Since El Gordo is actually two interacting clusters,

this would be even less likely, and is perhaps the main reason why the Bullet Cluster and especially El Gordo are so problematic for ΛCDM.

## 8. Large-Scale Structure

Since disc galaxy outskirts are already in the low-acceleration regime, tests of MOND should be possible at scales larger than individual galaxies. In addition to virialized systems, the large-scale structure of the universe could provide important constraints. We defer a discussion of the CMB to Section 9 because this does not presently help to distinguish between the ΛCDM and MOND models. For the moment, we focus on later epochs.

It was argued that MOND has difficulty reproducing the internal properties of spectroscopically detected Lyman-α (Ly-α) absorbers (the Ly-α forest [327]). However, those authors noted that consistency could be gained if the EFE is considered. In general, the EFE should have been stronger at higher redshift because the less pronounced cosmic structures at earlier times would be more than compensated by a smaller universe (e.g., [712]). Those authors estimated that $g_e \underset{\sim}{\propto} 1/a$ (see their Section 4.3). Bearing in mind that estimates for $g_e$ today are typically $\approx 0.03\, a_0$ [92] or $0.05\, a_0$ [93], we can take an intermediate value of $0.04\, a_0$ and scale it up by 4 to correspond to the Ly-α results of [327], which were typically at $a = 0.25$ or $z = 3$ (see their Section 2). This yields $g_e = 0.16\, a_0$, which according to them would reconcile the observations with MOND. However, they argued that MOND is strongly inconsistent with the Ly-α observations if the EFE is neglected, demonstrating its crucial role especially in the early universe.

### 8.1. The KBC Void and Hubble Tension

On even larger scales, the accelerations would be even lower, so we might expect MOND to cause a significant enhancement to the density fluctuations on ∼100 Mpc scales [74]. Since observations are easier at smaller distances, we focus on the low-redshift universe to check for significant density fluctuations exceeding what might be expected in ΛCDM. There is quite strong evidence that we are inside an underdensity of radius ≈300 Mpc known as the KBC void [715]. Their work used near-infrared measurements covering 57–75% of the galaxy luminosity function in different redshift slices (see their Figure 9). The apparent luminosity density over the range z = 0.01–0.07 is smaller than the average defined by more distant redshift bins, with an apparent density contrast of $(46 \pm 6)\%$ (see the light blue dot in their Figure 11). Alternatively, one can consider the low-redshift normalization as representative of the cosmic average, but then the higher-redshift results covering out to $z \approx 0.2$ (400–800 Mpc) would correspond to twice the cosmic mean density over a much larger volume, which is far less plausible.

The KBC void (discussed also in [716]) rules out ΛCDM cosmology at 6.04σ [93] based on a comparison to the Millennium XXL (MXXL) simulation [717]. This conclusion is very robust because the length scale corresponds to the linear regime of ΛCDM, where the density fluctuations are expected to be only a few percent. Galaxies with $M_\star \approx 10^{10}\, M_\odot$ should be very reliable tracers of the underlying matter distribution on such a large scale, especially as the comparison in [93] was done based on the abundance matching [718,719] stellar mass in halos with $M_\star > 10^{10}\, h^{-1} M_\odot$, where $h$ is the Hubble constant $H_0$ in units of 100 km/s/Mpc. Moreover, the KBC void has been detected across the entire electromagnetic spectrum, from the radio [720] to X-rays [721,722], as discussed further in Section 1.1 of [93].

Since the universe was fairly homogeneous at early times [723], a large local underdensity can only have arisen due to outflow in excess of a uniform expansion. We therefore expect the locally measured $H_0$ to exceed the value inferred from observations beyond the void, e.g., from CMB anisotropies. In Section 1.1 of [93], it was estimated that this excess in $H_0$ should be ≈ 11%, though the exact value will depend on which alternative model is used to account for the KBC void, which distance range is used to measure the local $H_0$, etc.

Observationally, the Hubble tension (reviewed in [724]) is a statistically significant detection of precisely this excess. We cannot hope to thoroughly review such a vast topic here, but we briefly mention some of the more recent works. The Planck determination of $H_0$ [80] has recently

been independently verified using ground-based surveys covering a smaller portion of the sky at high angular resolution [725]. The low-redshift probes typically rely on some method to calibrate the absolute luminosity of Type Ia supernovae. Cepheid variables can be used for this thanks to a significantly improved calibration of the Leavitt law using Gaia trigonometric parallaxes [726]. Gaia early data release 3 [727] has also allowed for a parallax determination to $\omega$ Centauri, fixing its distance at $5.24 \pm 0.11$ kpc [728]. This sets the absolute magnitude of the tip of the red giant branch, which is better suited for measuring distances to dwarf galaxies that lack recently formed stars and thus Cepheid variables. Both methods of calibrating the supernova distance ladder give consistent results for $H_0$ in the local universe (see also [729]), with other techniques such as megamasers [730] giving a similar result. Figure 1 of [731] nicely illustrates the tension between early and late 'measures' of $H_0$, so we have reproduced this as our Figure 31. Note that the early universe determinations are actually predictions assuming $\Lambda$CDM cosmology. The late universe determinations are less model-dependent, but may still deviate from the true background expansion rate.

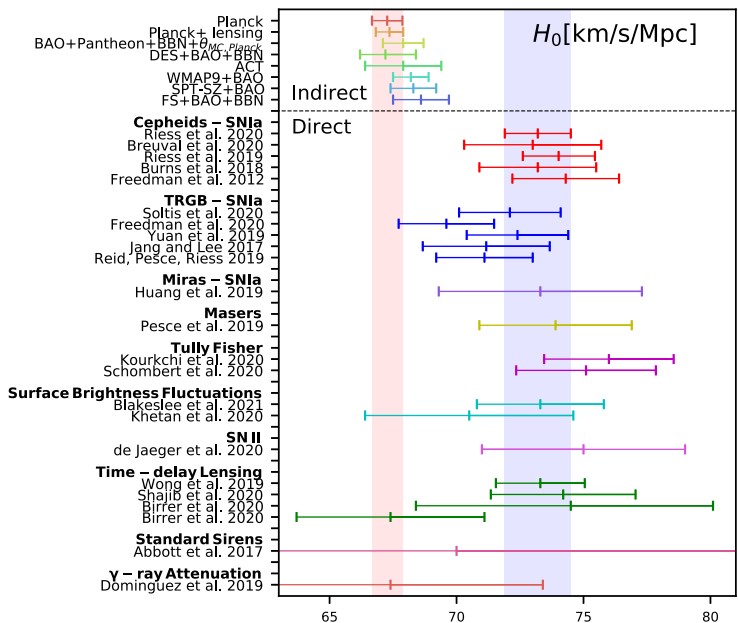

**Figure 31.** Compilation of different $H_0$ measurements. The determinations at the top from early universe probes depend on the assumption of $\Lambda$CDM, making these indirect measurements. The direct measurements (below the dashed line) are from late universe probes, sorted into blocks according to the technique used: Type Ia supernovae calibrated by Cepheid variables [726,732–735], Type Ia supernovae calibrated by the tip of the red giant branch [728,736–739], Type Ia supernovae calibrated by Mira variable stars [740], masers [730], the BTFR [741,742], surface brightness fluctuations [743,744], Type II supernovae [745], strong lensing time delays [746–748], gravitational waves [749], and attenuation of $\gamma$-rays by extragalactic background light [750]. The shaded vertical bands show that the early universe determination (red) is significantly below the late universe determination (blue), with the $\approx 10\%$ discrepancy known as the Hubble tension. Reproduced from Figure 1 of [731].

The KBC void and Hubble tension are two sides of the same coin—a significant local underdensity is caused by outflow in excess of cosmic expansion, which is the Hubble tension. The $\Lambda$CDM relation between an underdensity and the locally inferred $H_0$ is shown in our Figure 32, which reproduces Figure 2 of [93]. The relation passes close to the observed combination of density contrast and Hubble constant excess. This is unlikely if there are significant systematics with both the density contrast measured by [715] and the local measurements of $H_0$ reviewed in [731]. Indeed, the observations could be fit reasonably well in $\Lambda$CDM if we allow ourselves to reside within an $\approx 10\sigma$ underdensity. As this is statistically very unlikely, a better solution would be to explain the observations in a different model that is similar to $\Lambda$CDM with regards to the CMB and expansion rate

history, but has much more structure at late times. The authors of [93] achieved just that, as discussed further in Section 9.2.2.

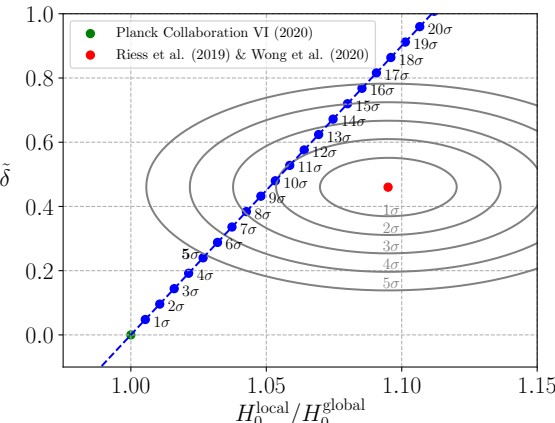

**Figure 32.** For a randomly located observer in a $\Lambda$CDM universe, the dashed blue line shows the relation between the fractional underdensity in a thick spherical shell of radius 40–300 Mpc and the fraction by which the local expansion rate appears to exceed the true cosmic value [80]. The green dot at (1, 0) would arise for an average observer, but some cosmic variance is expected. This is shown with the blue points on a line, with the adjacent text labels indicating the likelihood of such a departure from homogeneity. The observed combination is shown as a red dot, with error ellipses indicating the uncertainty. Notice that a $5\sigma$ density fluctuation in $\Lambda$CDM is not sufficient to get within the $5\sigma$ observationally allowed region. The existence of both a local underdensity and an enhanced local Hubble constant [733,746] argues for a common explanation in a different framework where structure forms more efficiently than in $\Lambda$CDM but the background cosmology is similar. Reproduced from Figure 2 of [93].

If a large local void is responsible for the Hubble tension, we would expect to regain consistency with a Planck background cosmology at $z \gtrsim 0.2$. Locally, this would appear as an unexpectedly large curvature in the Hubble diagram of distance against redshift—the universe would appear to follow a standard background expansion history until $a \approx 0.9$, but then accelerate at late times to a much greater extent than expected in $\Lambda$CDM. This can be parametrized using the acceleration parameter $\bar{q}_0 \equiv a\ddot{a}/\dot{a}^2$, this being the negative of the conventionally defined deceleration parameter $q_0$. In $\Lambda$CDM, we expect that $\bar{q}_0 = 0.53$, but observationally it is $1.08 \pm 0.29$ based on a Taylor expansion to measure $\dot{a}$ and $\ddot{a}$ without strong priors [751,752]. Though consistent with standard cosmology, the preference for larger $\bar{q}_0$ can be interpreted as evidence for a large local void. Moreover, several other workers argued that a low-redshift underdensity is suggested by the supernova Hubble diagram [753–757], and more generally that the inferred $H_0$ declines with redshift (e.g., [758–760]), a trend also apparent in strong lensing time delays [746]. The KBC void has therefore been detected at 0th order (galaxy redshift surveys such as the Two Micron All Sky Survey), 1st order (the Hubble tension), and 2nd order (the anomalously high local $\bar{q}_0$). In principle, it should also be evident at higher order, e.g., in the jerk parameter proportional to $\dddot{a}$. However, this is likely to be quite noisy for the time being given the already significant uncertainty on $\bar{q}_0$.

Since it would be very unlikely for us to lie at the very centre of a large void, another consequence of such a void would be anisotropy in cosmological observables such as the inferred Hubble constant. There is actually strong evidence for precisely that—observations reveal a dipole in $H_0$ with a fractional magnitude comparable to the Hubble tension [761]. The statistical significance of this dipole is $5.9\sigma$ based on ten different scaling relations followed by galaxy clusters that allow us to obtain redshift-independent distances (Table 3 of [762]). Moreover, the expected direction of the dipole is similar to the peculiar velocity of the LG in the CMB frame, which in galactic coordinates is towards $(276°, 30°)$. This

is generically expected if the LG peculiar velocity is largely caused by the void, since the peculiar velocities in this direction would then be different to those at right angles. A more detailed comparison is required to check if the KBC void model proposed in [93] is consistent with observations.

We have seen how extreme galaxy cluster collisions such as El Gordo and the Bullet Cluster suggest that overdensities are more pronounced than expected in ΛCDM (Section 7.2). The $z \lesssim 0.1$ universe complements this by demonstrating that underdensities are larger and deeper than expected in ΛCDM. Considering both the KBC void and the Hubble tension, ΛCDM is ruled out at $7.09\sigma$ confidence [93]. Importantly, their work showed that a MOND cosmological model (which we discuss further in Section 9.2) can fit the locally measured density and velocity field with only $2.53\sigma$ tension. Given the tendency for astronomers to underestimate uncertainties, this represents very good agreement for a theory that was originally designed to address the kpc scales of individual galaxies. It therefore seems very likely that there is more large-scale structure than expected in ΛCDM, i.e., that it underestimates the cosmic variance. This could cause the cluster mass function at $z \lesssim 0.2$ to not be representative, perhaps explaining the lack of lower-redshift analogues to El Gordo.

*8.2. Other Anomalies in Large-Scale Structure*

If structure formation proceeds more rapidly than expected in ΛCDM, we expect it to face additional challenges on large scales. For example, the Copernican principle suggests that the KBC void should not be the only supervoid in the observable universe. This can be addressed with the galaxy two-point correlation function, a measure of how clustered galaxies are on different length scales [763]. This is consistent with ΛCDM, but consistency with a model is of course not definitive evidence of its correctness. The main issue is that galaxy surveys do not directly trace the underlying matter distribution. At large distances, only the brightest galaxies can be detected, so it is necessary to assume a bias factor

$$b \equiv \frac{\delta_g}{\delta}, \tag{70}$$

where $\delta_g$ is the density contrast in galaxies, $\delta$ is the true density contrast in the total matter distribution, and $b$ is the bias, which is usually chosen to match the ΛCDM expectation for density fluctuations on the relevant scale. Consequently, it is quite difficult to obtain an accurate model-independent estimate of $\delta$. This might be possible using deep near-infrared observations that cover the majority of the galaxy luminosity function out to several hundred Mpc (as done by [715]), since galaxies should be unbiased ($b \approx 1$) tracers of the matter distribution on such large scales in ΛCDM [764]—and probably in any cosmological model. However, even the deep near-infrared survey used by [715] only covers 57–75% of the luminosity function (see their Figure 9). This makes it very difficult to perform a similarly detailed analysis at higher redshift.

A further problem is that galaxy distances are generally not known, so their redshift must be used as a proxy. Even then, the redshift is often estimated photometrically rather than measured spectroscopically. Using photometric redshifts increases the uncertainty, blurring structures along the LOS and making it difficult to identify distant supervoids. Nevertheless, supervoids identified in the Dark Energy Survey (DES) do seem to show an enhanced integrated Sachs-Wolfe (ISW) effect [765]. In a stacked analysis of 87 supervoids, those authors found that the effect has an amplitude of $5.2 \pm 1.6$ times the conventional expectation when combined with the earlier results of [766]. These persistent anomalies in the ISW signal have been attributed to the growth of structure on 100 Mpc scales differing from ΛCDM expectations [767].

Possibly related to this is the unexpectedly strong foreground lensing of the CMB [768]. Those authors suggested that the problem could instead be an indication that the universe has a positive curvature, but a closed universe would imply a very low $H_0$ of $54^{+3.3}_{-4.0}$ km/s/Mpc, far below local measurements (see their Figure 7). Including such measurements from either supernovae or baryon acoustic oscillations makes it difficult to reconcile

the preferred cosmological parameters from different datasets [769]. The enhanced lensing amplitude could instead be an imprint of large density fluctuations caused by more rapid growth of structure than expected in ΛCDM. In this scenario, a supervoid would be a more likely explanation for the CMB Cold Spot [770]. Their suggested void profile has a central underdensity of 0.25 and a characteristic size of 280 Mpc (see their equation 1). The Eridanus supervoid has actually been detected with roughly these parameters in DES weak lensing data of the region [771]. The void size and depth are quite similar to the KBC void [93]. Interestingly, the authors of [770] concluded that a supervoid explanation of the CMB Cold Spot is highly unlikely in ΛCDM, supporting the claim of [93] that the similar KBC void is also unlikely in this framework. The Cold Spot is suggestive of enhanced structure formation on large scales, which could be related to the Hubble tension [772]. Note that even if the Eridanus supervoid did form in a ΛCDM universe, the resulting ISW signal would be insufficient to explain the CMB Cold Spot by a factor of ≈5, so the problem is not merely a matter of postulating a rare underdensity [771]. One could postulate that the CMB Cold Spot is mostly a primordial temperature fluctuation, but then a fairly rare supervoid would have to coincidentally align with it. A better explanation might be an alternative gravity theory which enhances structure formation and leads to a different relation between density contrasts and the ISW signal.

On a smaller scale, the formation of galaxy clusters appears to be more efficient than expected in ΛCDM [773–776], a problem which extends down to individual galaxies [777–782]. One example of this is ALESS 073.1, whose gas disc is coherently rotating and has only a small fraction of its kinetic energy in non-circular motion [783]. Their Figure 2 shows that a significant central bulge is required to fit the RC. If bulges form through hierarchical merging, then the merger should have occurred at $z \gtrsim 5.5$ to leave enough time for the post-merger disc to settle down. Another problematic aspect of the observations is that the galaxy retains a dynamically cold gas disc despite undergoing a starburst and likely possessing a central supermassive black hole. These should create significant baryonic feedback effects on the gas if galaxies are to lose a substantial fraction of their original baryons, as required to explain the RAR in the ΛCDM framework (Section 3). Instead, it seems likely that galaxies are only mildly affected by baryonic feedback processes, in which case they would retain most of their original baryon endowment. The large proportion of missing gravity in especially dwarf galaxies would then need to be explained in some other way, perhaps using MOND.

*8.3. Cosmic Shear and the Matter Power Spectrum*

We have already discussed weak lensing by individual galaxies (Section 3.5), but in principle *any* density fluctuation would cause some gravitational distortion to the light from background objects. This is known as cosmic shear (for a review, see [784]). The weak lensing signature is more pronounced at smaller scales where there is more structure [785], yielding a power spectrum that is generally quite consistent with ΛCDM apart from a mild tension in the amplitude $\sigma_8$ of the inferred density fluctuations.

It is not presently clear what the cosmic shear signal would look like in MOND. The slope of the power spectrum is indicative of an inverse square gravity law, which is expected if the EFE dominates (Equation (34)). This should occur at distances beyond a few hundred kpc from a galaxy. For comparison, the weak lensing survey of [785] only considered $z > 0.2$ (see their Section 2), corresponding to an angular diameter distance of 700 Mpc in standard cosmology. Their Figure 3 shows that the results are sensitive to angular scales of $0.1° - 1°$, which corresponds to physical distance scales of $r = 1.2 - 12$ Mpc. We expect the gravitational field from a galaxy to be completely EFE-dominated at this distance— for a galaxy with a typical $v_f = 180$ km/s [786], the corresponding acceleration is only $v_f^2/r = 0.007 \, a_0$ at $r = 1.2$ Mpc, well below plausible estimates of the EFE from large-scale structure (e.g., [92,93,107,319,321]). Therefore, it is quite likely that the MOND gravity from individual galaxies would follow an inverse square law over the spatial scales covered by cosmological weak lensing surveys (Figure 1).

While this may explain the slope of the weak lensing cross-correlation power spectrum, its normalization would in general differ from ΛCDM expectations if MOND is correct. Since the normalization agrees with ΛCDM to within 10%, we can think of this as the probability that a completely different theory achieves a similar level of agreement. In MOND, the expected cosmic shear signal is not presently known. It is likely to vary between observers more so than in ΛCDM due to the enhanced structure formation that is generically expected in MOND (e.g., [74,93,695]). For instance, an observer located deep within a supervoid may measure a different $\sigma_8$ to an observer in an overdense region. Cosmic shear calculations would be rather complicated in MOND due to its non-linearity, with matter outside any particular LOS also contributing to the weak lensing signal along that direction [343]. Given the validity of Equation (43) in MOND, the best solution is probably to apply standard lightcone analysis techniques to the $\rho_{\mathrm{eff}}$ distribution at several snapshots of a cosmological MOND simulation, which would be a valuable undertaking that has never been done.

Therefore, the approximate agreement of the weak lensing results with ΛCDM expectations may be fortuitous (for possible historical parallels, see Section 10.4). This is suggested by recent hints of a tension in $\sigma_8$ [787–789], which is also apparent from galaxy cluster number counts [694]. As discussed further in Section 9.3, a standard late-time matter power spectrum in the linear regime can also be reproduced in the relativistic MOND theory of [790], so a success of ΛCDM does not necessarily imply a failure of MOND. Further work will be required to clarify the expected cosmic shear signal in MOND.

## 9. Cosmological Context

We began this review by briefly discussing the ΛCDM cosmological paradigm (Section 1). The evidence underpinning it places non-trivial cosmological scale constraints on the gravitational physics [791]. However, their work is not directly applicable to MOND for various reasons, and indeed does not mention MOND at all. The major reason is their focus on linear gravitational theories which are alternatives to DM on cosmological scales. MOND is not linear in the matter distribution (Equation (18)), nor is it necessarily an alternative to DM on cosmological scales. In principle, the gravity law and the matter content of the universe are separate issues—it is quite possible for DM particles to follow a Milgromian gravity law. Historically, MOND was proposed as an "alternative to the hidden mass hypothesis" [56], so one can argue that the existence of DM particles would remove the motivation for MOND. However, this is merely a sociological objection, and a weak one at that given the initial focus on galaxies. Scientifically, the question is whether an extra assumption increases the predictive power of a model sufficiently, since otherwise we can apply Occam's Razor and avoid making the extra assumption. MOND is certainly more complicated than Newtonian gravity, while having DM particles would imply extension(s) to the well-tested standard model of particle physics. There is a tendency to focus on observations which favour one or the other of these extensions to established physics, but both could be correct. In MOND, galaxies should contain very little DM to avoid disrupting RC fits, but DM is certainly not ruled out on larger scales such as galaxy clusters (Section 7.1).

The main objection to this hybrid approach is the great deal of theoretical flexibility that arises if one is prepared to add DM *and* modify the gravity law from GR. On the other hand, we should consider the improved agreement with observations across a much broader range of scales than can be achieved if we make only one of these assumptions. In particular, assuming only DM leads to failures of ΛCDM on many scales, while assuming only MOND makes it difficult to explain the Bullet Cluster (Figure 29). In Section 10, we carefully assess whether a hybrid model agrees well with observations *given its theoretical flexibility*, and do a similar analysis for ΛCDM. The basic idea is to impose a penalty not only for poor agreement with observations, but also for having a great deal of theoretical flexibility such that agreement with observations does not lend support to the theory (Equation (72)).

### 9.1. Time Variation of $a_0$

A theoretical uncertainty unique to MOND is whether $a_0$ should always have been equal to its present value. If this is assumed, the LG timing argument works fairly well with a previous MW–M31 flyby (Section 5.6), leaving little room for $a_0$ to have been substantially smaller in the past. If instead $a_0$ was much larger, there would be tension with galaxy RCs at high redshift [85]. Between these extremes, there is certainly scope for $a_0$ to have gradually changed over cosmic time, which would have secular effects on galaxies [792] and might impact the CMB. Ever more precise data at high redshift (e.g., [783]) should put much tighter constraints on any time variation of $a_0$. Such variation would also affect structure formation, perhaps leading to an incorrect frequency of El Gordo analogues given that its calculated frequency is about right with constant $a_0$ [712] according to the cosmological MOND simulations of [696]. If the RAR phenomenology is caused by the complex interplay of baryonic feedback processes in galaxies with CDM halos that obey Newtonian gravity, then the inferred value of $a_0$ should have a particular redshift dependence [793].

### 9.2. The νHDM Model

One of the most promising extensions of MOND to cosmological scales is the neutrino hot dark matter (νHDM) model [674]. Its development is in line with the historical pattern of understanding systems at ever greater distance, starting with the solar system, moving out to galaxies, and then galaxy clusters. As discussed in Section 7.1, clusters require additional undetected mass even in MOND. However, it can explain the equilibrium dynamics of 30 virialized galaxy clusters [677] if sterile neutrinos are assumed to exist and to reach the Tremaine–Gunn phase space density limit [676] at the cluster core.

#### 9.2.1. At High Redshift

Extending this model to even larger scales, a crucial aspect of νHDM is that thermal sterile neutrinos with a mass of 11 eV/$c^2$ would have the same relic density as the CDM in the ΛCDM paradigm, leading to a standard expansion rate history [794–796]. Consequently, νHDM should be able to account for the primordial abundances of light elements (BBN) similarly to ΛCDM, as discussed further in Section 3.1 of [93]. An extra relativistic species would affect the light element abundances, but by a very small amount that is quite difficult to rule out at present (see their Section 3.1.2). We might expect three species of sterile neutrinos similarly to the active neutrinos, but it is typically assumed that two of the three undiscovered sterile neutrino species have a very high mass and would therefore have decayed prior to the BBN era (see Figure 3 of [797]). In the standard gravity context, the third light sterile neutrino species is usually assumed to have a rest energy of order keV rather then the 11 eV considered here [798].

An important constraint on any cosmological model is the CMB. When its constituent photons were last scattered, the gravitational fields were much stronger than $a_0$ (see Figure 1 of [799]), an assumption explained in more detail below. Since MOND was originally designed to do away with CDM, it was initially assumed that the CMB anisotropies in a Milgromian universe would be the same as in GR without the CDM, leading to the prediction that the second peak in the CMB power spectrum is 2.4× lower in amplitude than the first [800]. This predicted amplitude ratio was later confirmed [801]. However, the same model also predicts that the third peak has a much lower amplitude than the second peak [800]. This prediction was soon falsified [802]. It is therefore clear that there must be additional complications in any attempt to explain the CMB anisotropies in MOND.

The authors of [93] explained in detail why the CMB can likely be explained in the νHDM framework (see their Section 3.1.3). There are a few non-trivial aspects to this, which we briefly discuss. To get the relatively strong third peak in the CMB power spectrum, it is necessary to have a sufficient amount of mass in a collisionless component, which the νHDM paradigm does by choice of the sterile neutrino mass. The gravity law is very similar to GR prior to recombination because the universe was much smaller then, more than compensating for the density fluctuations being less pronounced. The typical gravitational field can be estimated

as $g_{CMB} \approx 20\,a_0$, so MOND would not have substantially affected the universe when $z \gtrsim 50$. Due to the importance of this issue, we discuss it in a little more detail below.

The study that originally introduced $\nu$HDM estimated that $g_{CMB} \approx 570\,a_0$ (Section 1 of [674]). However, this is based on the assumption that the angular diameter distance to the CMB is 14 Gpc. In fact, this is the co-moving radial distance to the CMB—the angular diameter distance is smaller by a factor of $a \approx 1/1100$ when the CMB was emitted (Section 3 of [803]). As a result, the $570\,a_0$ estimate in [674] is completely incorrect.

A more careful calculation was presented in Section 3.1.3 of [93]. A complementary way of estimating $g_{CMB}$ is to start with the peculiar velocity of the LG with respect to the CMB. This has been measured at 630 km/s [179], implying that today the typical gravity on the relevant scale is $g_{now} \approx 0.01\,a_0$ based on dividing 630 km/s by the Hubble time of 14 Gyr. The actual value is likely a little larger (see Section 2.2 of [92]), but we continue with this estimate in order to be conservative. We use this to normalize the fractional amplitude of density fluctuations $\delta$ in a $\Lambda$CDM universe, since if $\Lambda$CDM is correct, it must be able to explain the precisely measured LG peculiar velocity (this assumption was verified in, e.g., [100]). We next need to scale $\delta$ to the 150 co-moving Mpc scale of the first acoustic peak in the CMB, and then work backwards in time to the recombination era. The enclosed mass on any scale $\lambda$ is $\propto \lambda^3$, while the fractional density fluctuation is expected to scale as $\delta \propto \lambda^{-1}$ [804,805]. Combining these results with the inverse square gravity law in $\Lambda$CDM, we see that the typical gravitational field from inhomogeneities should depend little on $\lambda$ in the linear regime. Turning now to the time dependence, we get that $g_{CMB} \propto \delta/a^2$, with the factor of $a^{-2}$ coming from the inverse square law and our desire to consider a fixed co-moving scale. $\Lambda$CDM predicts that $\delta \propto a$ during the matter-dominated era. The growth since recombination would be slightly smaller than a factor of 1100 because dark energy slows down the growth of structure at late times, while structure growth is also slower around the time of recombination due to the still significant contribution of radiation to the total mass-energy budget (the Meszaros effect [806]). We therefore assume that $\delta$ in the dominant DM component was typically $600\times$ smaller than today during the recombination era. Combining this with our previous estimate that $g_{now} \approx 0.01\,a_0$ and depends little on scale, we get that

$$g_{CMB} \approx g_{now} \times \frac{1100^2}{600} = 20\,a_0 \,. \tag{71}$$

This matches the estimate in [93], though their estimate used the $\sigma_8$ parameter rather than the CMB-frame peculiar velocity of the LG. The similar result in both cases clarifies that the CMB does not provide a strong test of gravitational physics at low accelerations, so the good fit in $\Lambda$CDM is not an indication that MOND would necessarily fare worse.

In the recombination era, free streaming effects would be small because sterile neutrinos more massive than 10 eV/$c^2$ already have a very small free streaming length [79], so "their effect on the CMB spectra is identical to that of CDM" (see their Section 6.4.3). While some minor differences are to be expected, these could be compensated by slight adjustments to the cosmological parameters [807]. For example, their analysis found that $\nu$HDM prefers a slightly higher $H_0$, which would reduce the Hubble tension but would not solve it (see the CMB fit in their Figure 1, which we reproduce in our Figure 33). Combined with a similar angular diameter distance to the CMB as in $\Lambda$CDM due to a nearly standard expansion rate history, the CMB does not pose obvious problems for MOND in the $\nu$HDM framework.

### 9.2.2. At Low Redshift

Structure formation would be quite non-standard in $\nu$HDM, which generically predicts the formation of supervoids [807] and massive galaxy clusters [695] in the late universe. These predictions were considered problematic, but nowadays the enhanced structure formation in $\nu$HDM seems necessary to form El Gordo [712] and the KBC void [93], which are actually very problematic in $\Lambda$CDM as discussed in Sections 7.2 and 8, respectively. A semi-analytic

calculation in $\nu$HDM showed that a small initial underdensity can evolve over a Hubble time into a large supervoid that matches the observed density profile and velocity field of the KBC void fairly well [93]. The high peculiar velocities in this model provide a natural solution to the Hubble tension. In this context, the 630 km/s CMB-frame peculiar velocity of the LG is actually rather low, but the likelihood of an even lower velocity arising was found to be 1.9% in their Section 2.3.4. The reason is that a local void resolution to the Hubble tension requires an enhancement to the local Hubble constant by $\approx$7 km/s/Mpc (Figure 31). The peculiar velocity is then <630 km/s in the central $\approx$90 Mpc of the void, which extends out to $\approx$300 Mpc [715]. The likelihood of a random observer within this void having a CMB-frame peculiar velocity <630 km/s is then around $(90/300)^3 = 2.7\%$, which is quite close to the 1.9% yielded by a more detailed calculation [93]. Their Figure 10 shows the overall $\chi^2$ budget of the best-fitting void model, which is in only 2.53$\sigma$ tension with the considered observables taken in combination. The most problematic individual observation is the above-mentioned issue of the low LG peculiar velocity, which causes only 2.34$\sigma$ tension.

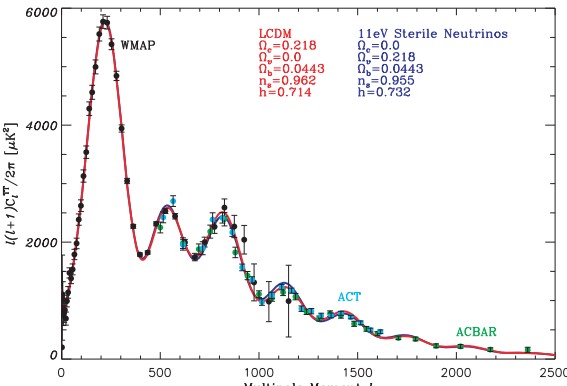

**Figure 33.** Fit to the angular power spectrum of the CMB using $\Lambda$CDM (red) and $\nu$HDM (blue), with the parameters of each model indicated in the figure. Notice that both fit the data well, mainly because the 11 eV/$c^2$ mass sterile neutrinos play much the same role in $\nu$HDM that CDM particles do in $\Lambda$CDM, with both comprising the same fraction of the cosmic critical density. The Milgromian gravity law assumed in $\nu$HDM is expected to depart little from standard gravity during the recombination era due to the high accelerations, as discussed in the text (see also [93]). Reproduced from Figure 1 of [807].

The lack of even more massive analogues to El Gordo at lower redshift than its $z = 0.87$ [706] may well be due to the KBC void, which would cause the $z \lesssim 0.2$ universe to not be representative. This may explain why the $\nu$HDM model produces many more massive galaxy clusters than are observed locally, an issue discussed further in Section 3.1.4 of [93]. It is possible that the cluster mass function inside a $\nu$HDM supervoid looks similar to the low-redshift cluster mass function, especially given the uncertainties on, e.g., the relation between mass and X-ray luminosity (Section 7.2). Perhaps a better test is provided by observations beyond the KBC void (i.e., at $z > 0.2$), since the much faster structure growth in $\nu$HDM cannot be masked in the universe at large. $\nu$HDM is quite consistent with extreme galaxy cluster collisions such as El Gordo and the Bullet Cluster [696], with 1.16 analogues to El Gordo expected in the surveyed region [712]. However, those authors showed that El Gordo rules out $\Lambda$CDM at 6.16$\sigma$, so observations beyond the KBC void seem to favour models such as $\nu$HDM in which structure formation proceeds more rapidly than in $\Lambda$CDM.

In general, any cosmological MOND model should explain why MOND works so well in galaxies (Section 3) but seems to do poorly in galaxy clusters (Section 7). The $\nu$HDM answer is that for a given limit to the phase space density of sterile neutrinos, their allowed number density is higher in galaxy clusters due to their deeper potential wells than field galaxies. As a result, galaxy clusters can have much more substantial amounts of DM than galaxies, even though the central region of a cluster might be comparable in

size to the virial radius of a typical galaxy. The $\nu$HDM model might therefore be able to explain gravitationally bound structures ranging from dwarf galaxies up to large galaxy clusters, with the sterile neutrinos helping to address early universe observables such as BBN and the CMB in the context of a standard expansion history. MOND is not expected to significantly affect the early universe because the typical gravitational field was above $a_0$ until $z \approx 50$, partly due to extra gravity from the sterile neutrinos.

### 9.3. Towards a Relativistic Model

The most natural way to embed the $\nu$HDM framework into a relativistic MOND theory is to use one of the class of such theories in which GWs travel at $c$ [139], as required for consistency with observations [7]. Prior work indicated that MOND fits to the CMB using three fully thermalized ordinary neutrinos with a mass of 2.2 eV/$c^2$ [796] look similar to what might be expected if we neglect MOND effects [674]. This is because the typical gravitational field around the time of recombination was $\approx$20 $a_0$ (Section 9.2.1). Therefore, the main deficiency with the work of [796] was not the gravity law per se, but rather the assumed matter content of the universe. In particular, since $3 \times 2.2 < 11$, the model has less mass in neutrinos than the CDM in standard cosmology, making it difficult to explain why the second and third peaks in the CMB power spectrum have almost the same height [80]. As discussed in Section 9.2.1, this problem can be resolved using sterile neutrinos with a mass of 11 eV/$c^2$, which would also help to explain the cluster-scale evidence for extra collisionless matter in the MOND context (Section 7.1). Therefore, we suggest that the most promising way to construct a relativistic MOND theory is to assume the existence of such sterile neutrinos and incorporate the constraints placed by GWs propagating at $c$ [139].

However, this path is nowadays neglected in favour of another recently proposed relativistic MOND theory which can fit the CMB anisotropies while reproducing a standard matter power spectrum in the linear regime [790]. This is quite promising because it builds on an earlier model in which GWs travel at $c$ [139] and avoids so-called 'ghost' instabilities to quadratic order [790,808]. However, it is not clear how their model can fit data on galaxy clusters such as the Bullet. Moreover, obtaining consistency with $\Lambda$CDM is not the same as agreeing with observations. The model completely ignores the results discussed in Section 8.1 on the KBC void and Hubble tension, so these observations rule out the [790] model in just the same way that they rule out $\Lambda$CDM.

Despite these serious problems, we must bear in mind that further development of their model could lead to different conclusions. In particular, it likely has enough flexibility to fit the CMB using a different $H_0$, thereby alleviating the Hubble tension. In addition, galaxy clusters such as El Gordo are not in the linear regime, so it is possible that structure formation on this scale would be enhanced over $\Lambda$CDM to a sufficient extent that El Gordo is no longer problematic. This would still not explain the KBC void, but in principle we do not expect a correct theory to explain all the data as some of the many published observational results are very likely incorrect. While the detection of the KBC void seems very secure [716], its exact properties are somewhat uncertain. It could be argued that slightly increasing the uncertainties would reduce the $6.04\sigma$ tension with $\Lambda$CDM [93] to $< 5\sigma$, thereby gaining consistency with any theory that has the same predictions as $\Lambda$CDM at scales $\gtrsim$ 100 Mpc. A shallower void would still enhance the local $H_0$ to some extent, which might help to explain the anomalously high local ($z < 0.15$) determination of $\bar{q}_0$ (e.g., [751,752]). It is likely that the model of [790] will need to be more fully developed especially on the galaxy cluster scale before firmer conclusions can be drawn regarding its validity. A solution to the Hubble tension has also not been demonstrated in this framework.

## 10. Comparing $\Lambda$CDM and MOND with Observations

In this section, we summarize the previously discussed evidence from a huge range of astrophysical scales pertaining to the cause(s) of the missing gravity problem. MOND generally works well on smaller scales, while $\Lambda$CDM performs better at larger scales [809]. Both models face challenges when extrapolated beyond the scales for which they were

originally designed, which for MOND means addressing cosmology and for ΛCDM means addressing galaxy scale challenges such as the RAR and the satellite planes.

In the following, we assess how well each model performs against various observational tests, bearing in mind the theoretical uncertainty surrounding what the model predicts for the situation being considered. For this purpose, we define

$$\text{Confidence} \equiv \begin{pmatrix} \text{Level of} \\ \text{agreement} \end{pmatrix} - \begin{pmatrix} \text{Theoretical} \\ \text{flexibility} \end{pmatrix}, \tag{72}$$

where the level of agreement and theoretical flexibility are given an integer score between $-2$ and $+2$. The theoretical flexibility scores have the same meaning as in Section 2.7, with $+2$ indicating 'anything goes'. Better agreement with observations is indicated by a higher score. The scores we assign are somewhat subjective, with further explanation provided in the following sections where the choice was not obvious or runs counter to intuition.

The main idea behind Equation (72) is that if the uncertainty surrounding the theoretical expectation is large, agreement with observations is not particularly impressive. Our measure of the 'confidence' lent to a theory by any particular test captures the generally agreed philosophy of science that agreement is more impressive if the theoretical expectations are clear, and especially if they were made prior to the relevant observations [50]. We will not penalize a model very much if a theory has clearly unavoidable consequences which were, however, not published prior to the relevant astronomical observations, as this is at least partly an accident of history. Progress with hypothesis testing relies mainly on expanding the number of well-observed phenomena with clearly defined theoretical expectations, either by collecting more data or through further calculations. This is because agreement of a theory with observations does not prove the theory correct, but disagreement does prove it wrong. However, the scope for disagreement is non-existent if the theoretical expectations leave room for adjustment to fit any plausible dataset. This is of course not the idea of science as scientific theories should be falsifiable. We nonetheless include a few 'tests' of this sort because the observations are very well known. If nothing else, this serves to show that they are consistent with the theory under consideration, though the epistemic significance of such agreement is low. Some of these tests are occasionally used to argue against a model even though it can accommodate the data in question, so including such tests is also meant to correct the record in this regard.

We can apply Equation (72) to one theoretical framework without considering alternatives. In this case, we need to set a threshold confidence score which a theory should exceed on average in order to be considered realistic. We argue that this threshold is 0. This is because a physically unrealistic theory will still make some predictions, but these will in general disagree with observations. The greater the amount of flexibility, the higher the likelihood that plausible agreement can be obtained. However, strong a priori predictions with little wiggle room will invariably be completely off the mark if the model is not based on the correct physics. Therefore, the level of theoretical flexibility should typically be approximately the same as the level of agreement with observations, leading to a confidence score of 0. A higher value would indicate agreement with observations even when the model does make rather specific predictions. If the average confidence score is clearly positive after consideration of many tests, then it is likely that the model captures some aspects of physical reality.

### 10.1. ΛCDM

The tests we consider for ΛCDM are summarized in Table 3. While most of the assigned scores are fairly clear, a few deserve some clarification. In addition, we need to decide how to fix the free parameters of ΛCDM at both the background and the perturbation level. These can be set in various ways, with the CMB power spectrum being the most common starting point nowadays because it leads to very high precision constraints on the model parameters if we assume the model itself to be valid. Other observations could be used instead, e.g., the expansion rate history and the apparent cosmic acceleration from supernovae, and the ages of globular clusters. Historically, these led to the construction

of $\Lambda$CDM with parameters close to the current concordance values before precise CMB data were available (Section 1). However, the power spectrum of the CMB anisotropies is typically used nowadays, so we take these as fixing the model parameters of $\Lambda$CDM. The CMB is therefore not in favour of $\Lambda$CDM from a model selection perspective. This allows the expansion rate history at $z \gtrsim 0.2$ to be counted as a success for $\Lambda$CDM. If instead the expansion rate history is used to set the model parameters, then this would not be a success for $\Lambda$CDM, while the CMB would become a success story. Which way one looks at this is somewhat subjective, but should have little impact on our overall confidence in $\Lambda$CDM.

Another important test of $\Lambda$CDM in the early universe is provided by BBN, i.e., by the abundances of light elements before the onset of fusion reactions in stars. While this is generally thought to agree well with expectations, the reality is not so straightforward. In addition to the well-known lithium problem [810], the primordial deuterium abundance reported by observers changed significantly from a generation-long consensus value after precise CMB data were published that require a different value in the $\Lambda$CDM context (for a review of pre-CMB constraints, see [811]). This could be due to significant improvements in deuterium observations coinciding with the launch of the Cosmic Background Explorer, but since primordial deuterium abundances are measured very differently, this unlikely coincidence could be a sign of confirmation bias in the community (see Chapter 6 of [50]). Indeed, it is possible that observers studying the early universe are more reliant on $\Lambda$CDM to interpret their results, making them less likely to report results that go against it. Lithium abundances are measured in old galactic stars, so observers studying them are not much reliant on the validity of $\Lambda$CDM. A worrying aspect is that BBN is mainly reliant on lithium and deuterium because the primordial helium abundance is not too sensitive to the exact parameters—and there are essentially no other probes of BBN [47]. We therefore conclude that BBN works well in $\Lambda$CDM with parameters set by the CMB anisotropies, but there is still the possibility of an unpleasant surprise in this area.

The radii of individual Einstein rings were not predicted a priori because the same images that would be necessary to obtain the visible mass distribution also reveal the Einstein ring, unlike, e.g., with RCs, where it is possible to obtain the distribution of $M_s$ before knowing its kinematics. Nonetheless, GR clearly predicts that Equation (43) remains valid even when $g \ll a_0$. This has been confirmed observationally by comparing dynamical mass estimates using lensing and non-relativistic tracers (Section 3.4), which is in favour of $\Lambda$CDM.

The tests which significantly reduce our confidence in $\Lambda$CDM are the RCs and internal velocity dispersions of galaxies (via $\sigma_{\mathrm{LOS}}$ measurements). We have split the disc galaxy RC tests into tests involving HSBs (which tend to be gas-poor like the MW), LSBs, and gas-rich galaxies. This is because the RC data in, e.g., the SPARC sample cover many orders of magnitude in central surface brightness and a wide range of gas fractions [167]. Differences in gas fraction imply differences in the amount of star formation and thus the number of supernovae, which are presumably an important source of feedback in especially dwarf galaxies. Differences in surface brightness at fixed $M_s$ imply the data cover out to different distances within the CDM halo, which should also be a relevant consideration. LSBs are in general strong tests of what causes the acceleration discrepancies in galaxies because the discrepancies are very large in such systems, unlike, e.g., the MW where they are much less pronounced within the optical disc. A significant amount of feedback is essential for $\Lambda$CDM to have any hope of producing realistic galaxies with a much lower baryon fraction than implied by the CMB. Some stochasticity to the feedback is inevitable due to diversity in the star formation history, gas fraction, merger history, etc. [203,204]. This is furthermore required to fit the diversity of dwarf galaxy RC shapes at fixed $v_f$ [205], which in $\Lambda$CDM is a measure of the halo mass. Thus, the lack of spread in the RAR is unusual. We conclude that while $\Lambda$CDM can be argued to plausibly work in HSBs, it is in some tension with LSBs and gas-rich galaxies, which cannot be expected to have blown out the same amount of baryons through supernovae as gas-poor galaxies with a similar baryon distribution.

**Table 3.** Summary of how well ΛCDM fares when confronted with the data and how much flexibility it had in the fit. The open dot shows observations used in theory construction, so this test is not used when giving a numerical score in Section 12.

| | Clear Prior Expectation | Not Predicted, but Follows from Theory | Auxiliary Assumptions Needed, but These Have Little Effect | Auxiliary Assumptions Needed, but These Have a Discernible Effect | Auxiliary Assumptions Allow Theory to Fit Any Plausible Data |
|---|---|---|---|---|---|
| Excellent agreement | 🔴 Gravitational waves travel at $c$ 🔴 Expansion history at $z \gtrsim 0.2$ | 🔴 Einstein ring radii | | ⊙ CMB anisotropies | 🔴 MW escape velocity curve 🔴 MW–M31 timing argument 🔴 Galaxy cluster internal dynamics 🔴 Galaxy two-point correlation function |
| Works well | 🔴 Big Bang nucleosynthesis 🔴 Offset between X-ray and lensing in Bullet Cluster | | | 🔴 Hickson Compact Group abundance | |
| Plausibly works | 🔴 Weak lensing correlation function | | 🔴 Galaxy cluster mass function at low redshift | | 🔴 Weak lensing by galaxies 🔴 HSB disc galaxy RCs |
| Some tension | 🔴 Number of spiral arms in disc galaxies 🔴 External field effect | | | 🔴 Prevalence of thin disc galaxies 🔴 Weakly barred M33 | 🔴 LSB disc galaxy RCs 🔴 Gas-rich galaxy RCs 🔴 Elliptical galaxy RCs 🔴 Spheroidal galaxy $\sigma_{\mathrm{LOS}}$ 🔴 Galaxy group $\sigma_{\mathrm{LOS}}$ |

**Table 3.** *Cont.*

| | Clear Prior Expectation | Not Predicted, but Follows from Theory | Auxiliary Assumptions Needed, but These Have Little Effect | Auxiliary Assumptions Needed, but These Have a Discernible Effect | Auxiliary Assumptions Allow Theory to Fit Any Plausible Data |
|---|---|---|---|---|---|
| Strong disagreement | 🔴 No distinct tidal dwarf mass–radius relation 🔴 Local Group satellite planes 🔴 El Gordo formation 🔴 KBC void 🔴 Local Hubble diagram slope and curvature | 🔴 Galaxy bar pattern speeds 🔴 RV of NGC 3109 association | 🔴 Tidal limit to radii of MW satellites 🔴 Bar fraction in disc galaxies | | |

Weak lensing by galaxies is also consistent with the disc galaxy RAR (Figure 11), but given the larger uncertainties, this poses a less severe fine-tuning problem for ΛCDM. Galaxy groups also have somewhat uncertain $\sigma_i$, but we argue that there is some tension with ΛCDM due to the lack of a significant hot gas halo around Centaurus A, and more generally because of the agreement with a tight RAR despite a large sample size (Section 6.4). In all these cases, the model has a huge amount of theoretical flexibility given that vast amounts of otherwise undetected matter can be postulated to explain the observed kinematics.

We consider ΛCDM to be in strong disagreement with several tests, as discussed previously. One of these failures involves the tidal stability of MW satellites, for which model-independent observables such as the morphology strongly suggest significant tidal disturbance, but this is not expected in ΛCDM [244]. While a significant failure of ΛCDM is of course not good for this paradigm, it is also rewarded by our scoring system for making clear a priori predictions, or in general for having little theoretical flexibility (Equation (72)). This means that the failures of ΛCDM reduce our overall confidence in it to only a small extent, which could be considered too lenient. However, we argue that this is fair because these tests also represent a missed opportunity to significantly increase our confidence in the model, leaving the door open for a rival theory to do just that.

### 10.2. MOND

Similarly to Table 3, the tests we consider for MOND are summarized in Table 4. We again provide a brief explanation for some of the tests where the scores required some more thought than usual. Though MOND has little theoretical flexibility in general, it was still necessary to decide upon acceleration rather than distance as the crucial parameter. RC data were important to this (Section 2). The parameter $a_0$ was found using the RCs of a handful of HSBs [69], with the preferred value remaining the same since then. RCs of HSBs are still the most accurate way to empirically constrain $a_0$, so we consider MOND to take these as an input. LSBs were only discovered after $a_0$ was already fixed. These often have a surface brightness many orders of magnitude below that of HSBs, so LSBs can be considered an independent test of MOND [387]. Unlike with ΛCDM, we do not consider gas-rich galaxies as providing another test because the gas fraction is not supposed to be dynamically relevant in MOND once the distribution of $M_s$ is known.

While MOND works well in galaxies, it is generally considered to work less well in galaxy clusters. However, the equilibrium dynamics of galaxy clusters are not in tension with MOND once we include sterile neutrinos with a mass of 11 eV/$c^2$ [677]. The required sterile neutrino phase space density marginally reaches the Tremaine–Gunn limit at the centres of the 30 galaxy clusters they studied, which is a strong hint for the reality of the sterile neutrinos. We therefore argue that the equilibrium dynamics of galaxy clusters are in good agreement with MOND. Since this success comes at the cost of extra theoretical flexibility arising from the HDM component, we agree with the general intuition that MOND does not work so well in galaxy clusters. Still, it must be borne in mind that the sterile neutrinos cannot be packed more tightly than the Tremaine–Gunn limit, so it is still not possible for MOND to match any arbitrary dataset on galaxy cluster scales. The situation is similar for the offset between the weak lensing and X-ray centroids in the Bullet Cluster (which is consistent with MOND even with a lower sterile neutrino mass of only 2 eV/$c^2$ [670]). Moreover, the sterile neutrinos do not increase the flexibility of MOND with regards to individual galaxies because these are too small and have too low an escape velocity to contain an appreciable amount of HDM [675]. Brightest cluster galaxies could be an exception because the neutrino halo of the cluster could affect the dynamics in the outskirts of such a galaxy, perhaps leading to unexpectedly large light deflection.

**Table 4.** Similar to Table 3, but for MOND.

| | Clear Prior Expectation | Not Predicted, but Follows from Theory | Auxiliary Assumptions Needed, but These Have Little Effect | Auxiliary Assumptions Needed, but These Have a Discernible Effect | Auxiliary Assumptions Allow Theory to Fit Any Plausible Data |
|---|---|---|---|---|---|
| Excellent agreement | ● LSB disc galaxy RCs ● No distinct tidal dwarf mass–radius relation ● External field effect | ● Galaxy bar pattern speeds ☉ HSB disc galaxy RCs ● Elliptical galaxy RCs ● El Gordo formation | | ● Expansion history at $z \gtrsim 0.2$ | ● Gravitational waves travel at $c$ ● Einstein ring radii ● CMB anisotropies |
| Works well | ● Tidal limit to radii of MW satellites ● Freeman limit ● Weak lensing by galaxies ● Binary galaxy $v_{\mathrm{rel}}$ ● Galaxy group $\sigma_{\mathrm{LOS}}$ | ● RV of NGC 3109 association | ● Weakly barred M33 ● Exponential profiles of disc galaxies ● Local Hubble diagram slope and curvature ● Shell galaxies | ● Big Bang nucleosynthesis ● Galaxy cluster internal dynamics ● Offset between X-ray and lensing in Bullet Cluster | |
| Plausibly works | ● Number of spiral arms in disc galaxies ● Spheroidal galaxy $\sigma_{\mathrm{LOS}}$ ● KBC void | ● MW–M31 timing argument | ● Local Group satellite planes | ● MW escape velocity curve | |
| Some tension | | | | | |
| Strong disagreement | | | | | |



Turning to relativistic tests, the validity of Equation (43) is not guaranteed in a theory which reduces to MOND for galaxy RCs. It seems to be correct observationally (Section 3.4)— this was certainly important in the construction of relativistic MOND theories (Section 2.6). Since it would also be possible for the light deflection to receive no extra enhancement in MOND [126], we argue that a wide range of possible observations could have been accommodated. Similarly, the fact that GWs travel at a speed very close to $c$ [7] rules out some versions of MOND that predict a rather slower speed [135]. However, other versions exist which are compatible with this constraint [139,812]. GWs are therefore not a major success for MOND, but neither do they falsify it.

There is rather less flexibility with regards to the LG timing argument, where MOND implies a past close MW–M31 flyby [92,578] due to the almost radial MW–M31 orbit [146,576,577]. It is inevitable that out of the dozens of dwarfs in the LG, some would have been near the spacetime location of the flyby. These dwarfs would have been flung outwards at high speed to distances similar to that of NGC 3109 [500]. Consequently, the existence of some LG dwarfs with an unusually high RV for their position was expected in MOND, and indeed provided an important motivation for their 3D timing argument analysis of the LG in $\Lambda$CDM.

While MOND by itself does not predict that we must be inside a large deep supervoid such as the KBC void [715], such voids were predicted in the $\nu$HDM cosmology at almost the same time as that publication [695,696]. These studies were clearly not in response to the observations—the authors of [695] stated that "there was a catastrophic overproduction of supercluster sized haloes and large voids" in the earlier simulations of [807], which were too early to have been influenced by observational results suggesting a large deep supervoid. We therefore conclude that in MOND, it was predicted a priori that such supervoids should exist, implying that we might reside within one. If so, the local Hubble constant would slightly exceed the global value. The excess depends on the detailed model, but very simple analytic arguments can be used to show that the expected excess is $\approx 10\%$ 10% (Section 1.1 of [93]). A high local $H_0$ thus follows naturally from the KBC void, albeit with some model dependence.

*10.3. Comparing the Models*

We reconsider the tests summarized in Tables 3 and 4 in order to obtain a single 'confidence' score for how each theory performs against each test (Equation (72)), with the results summarized in Table 5. Some tests can be usefully applied to only one theory, so there are slightly more tests for $\Lambda$CDM (red dots) than for MOND (blue dots). As before, we also use an open circle to indicate which observations were crucial to the formulation of each theory or to set its free parameters. These observations do not by themselves lend support to the theory, which will inevitably match them quite well [50]. Since we have many tests, this is only a minor hindrance in our attempt to quantify the confidence we should have in each theory, which we do in Section 12 by adding the confidence lent by each test except that used in theory construction.

**Table 5.** Comparison of ΛCDM (red dots) and MOND (blue dots) with observations based on the tests listed in Tables 3 and 4, respectively. The 2D scores in those tables have been collapsed into a single score for each test using Equation (72). For each theory, the open dot indicates that the data were crucial to theory construction or to fix free parameters, so our final score for each theory (Section 12) uses only the solid dots. The horizontal lines divide tests into those probing smaller or larger scales than the indicated length.

| Astrophysical Scenario | | Confidence ≡ Level of Agreement—Theoretical Flexibility | | | | | | | | |
| --- | --- | --- | --- | --- | --- | --- | --- | --- | --- | --- |
| | −4 | −3 | −2 | −1 | 0 | 1 | 2 | 3 | 4 |
| Big Bang nucleosynthesis | | | | | 🔵 | | | 🔴 | |
| Gravitational waves travel at $c$ | | | | | 🔵 | | | | 🔴 |
| *— pc —* | | | | | | | | | |
| No distinct TDG mass–radius relation | | | | | 🔴 | | | | 🔵 |
| Tidal limit to MW satellite radii | | | 🔴 | | | | | 🔵 | |
| Prevalence of thin disc galaxies | | | 🔴 | | | | | 🔵 | |
| Freeman limit to disc central density | | | | | | | | 🔵 | |
| Number of spiral arms in disc galaxies | | | | | | 🔴 | 🔵 | | |
| Weakly barred M33 | | | 🔴 | | | 🔵 | | | |
| Bar fraction in disc galaxies | | | 🔴 | | | | | | |
| Galaxy bar pattern speeds | | | | 🔴 | | | | 🔵 | |
| *— kpc —* | | | | | | | | | |
| Disc galaxies have exponential profiles | | | | | | 🔵 | | | |
| HSB disc galaxy RCs on RAR | | | 🔴 | | | | | ⊙ | |
| LSB disc galaxy RCs on RAR | | 🔴 | | | | | | | 🔵 |
| Gas-rich galaxy RCs on RAR | | 🔴 | | | | | | 🔵 | |
| Elliptical galaxies on RAR | | 🔴 | | | | | | 🔵 | |
| Spheroidal galaxy $\sigma_i$ | | 🔴 | | | | | 🔵 | | |
| External field effect | | | | | | 🔴 | | | 🔵 |
| MW escape velocity curve | | | | 🔵 | 🔴 | | | | |
| Shell galaxies | | | | | | 🔵 | | | |
| Local Group satellite planes | | | | | 🔴🔵 | | | | |
| Weak lensing by galaxies | | | 🔴 | | | | | 🔵 | |
| Einstein ring radii | | | | | 🔵 | | | 🔴 | |
| *— Mpc —* | | | | | | | | | |
| MW–M31 timing argument | | | | | 🔴 | 🔵 | | | |
| RV of NGC 3109 association | | | | 🔴 | | | 🔵 | | |
| Hickson Compact Group abundance | | | | | 🔴 | | | | |

**Table 5.** *Cont.*

| Astrophysical Scenario | Confidence ≡ Level of Agreement—Theoretical Flexibility | | | | | | | | |
|---|---|---|---|---|---|---|---|---|---|
| | $-4$ | $-3$ | $-2$ | $-1$ | 0 | 1 | 2 | 3 | 4 |
| Binary galaxy $v_{\mathrm{rel}}$ | | | | | | | | 🔵 | |
| Galaxy group $\sigma_i$ | | 🔴 | | | | | | 🔵 | |
| Galaxy cluster internal dynamics | | | | | 🔴 🔵 | | | | |
| Offset between X-ray and lensing in Bullet Cluster | | | | | 🔵 | | | 🔴 | |
| El Gordo formation | | | | | 🔴 | | | 🔵 | |
| Galaxy two-point correlation function | | | | | 🔴 | | | | |
| Galaxy cluster mass function at low redshift | | | | | 🔴 | | | | |
| Weak lensing correlation function (cosmic shear) | | | | | | | 🔴 | | |
| CMB anisotropies | | | | | 🔵 | ⊙ | | | |
| KBC void | | | | | 🔴 | | 🔵 | | |
| Local Hubble diagram slope and curvature | | | | | 🔴 | 🔵 | | | |
| ———————————————— Gpc ———————————————— | | | | | | | | | |
| Expansion history at $z \gtrsim 0.2$ | | | | | | 🔵 | | | 🔴 |

### 10.4. Parallels with the Heliocentric Revolution

Though we do not know how future theoretical and observational results will play into the debate over the appropriate low-acceleration gravity law, some insights may be gained by considering possible historical parallels. There are interesting parallels between the presently unclear situation and that at the dawn of the GR and quantum revolutions in the early 20th century [50,691]. However, in both cases, there were few alternatives to the new theory in the domain for which it was developed (e.g., the Michelson-Morley experiment pre-dated the formulation of GR and had no Newtonian interpretation).

A more accurate parallel might be the debate between the geocentric and heliocentric worldviews in the early 17th century. In both models, orbits were assumed to be circular. As a result, the heliocentric model still required the use of the now-infamous epicycles, albeit to a much smaller extent than the geocentric model. This made it less clear just how much of an advance the heliocentric model really represented, if its main goal was to do away with epicycles altogether. Moreover, the null detections of stellar aberration and parallax were more naturally explained by the geocentric model, which in any case is certainly correct with regards to the Moon. We conclude three major lessons from this:

1. All currently proposed models are surely wrong at some level, but it is still worthwhile to find a model which is more nearly correct as this would form a more reliable stepping stone to a more complete theory.
2. At an early stage of development, the more realistic model will not be able to explain everything it seeks to explain.
3. Even if both models are fully developed, the less realistic model will provide a better explanation of some observables, similarly to how a broken clock tells the correct time twice each day.

This suggests that it is better to use one universal force law in all galaxies than to use multiple tunable feedback parameters to achieve at best post-hoc explanations for their RCs, a procedure which bears strong similarities to the epicycles required in the geocentric model [50]. The fact that the heliocentric model also initially required epicycles due to the assumption of circular orbits might be analogous to how DM of some form still seems necessary in MOND, with the possible exception of the [790] model if it can be shown to work in galaxy clusters (Section 9). Moreover, the seemingly correct (until the 1830s) strong prediction of zero stellar parallax in the geocentric model might be analogous to the cosmic shear results so far agreeing with ΛCDM (Section 8.3).

The analogy with a clock may be particularly apt here—one can identify circumstances where a working clock gives the correct time, but this is also possible with a broken clock. Therefore, correct predictions can be expected in the correct model, but also in the wrong model. One difference is that only a very limited number of examples can be provided of a broken clock giving the correct time, while a working clock would do so much more often. However, the most important difference is that the broken clock will give an extremely incorrect time after only a limited amount of observation, allowing it to be ruled out as a viable time-keeping device. This raises the critical issue that in science, making a correct prediction does not prove a hypothesis correct, because there might be other explanations. However, an incorrect prediction would prove the hypothesis wrong, if the prediction is theoretically secure and the empirical data are observationally secure.

The failure of a hypothesis would not prove its leading alternative correct. For example, the working clock may also ultimately fail due to, e.g., leap seconds. However, it would be more worthwhile to make some adjustments to this clock than to provide multiple often contradictory post-hoc explanations for why the broken clock gave the wrong time of day, and to point out the thousands of successes that it built up over several decades as evidence of its remarkable predictive power. We leave the reader to decide if the broken clock in this analogy represents ΛCDM or MOND. Conclusive results should be provided by future experiments and observations, some of which we describe next.

## 11. Future Tests of MOND

Despite the wide array of currently available astronomical evidence and its high precision in some cases, the true cause of the missing gravity problem is still not definitively known. Further theoretical work would help in some instances, but we expect that additional observational results are necessary to reach general agreement. To avoid theoretical uncertainties dominating, these results should pertain to areas which are theoretically clear in both ΛCDM and MOND (Section 2.7). Since the application of MOND to extragalactic scales carries some uncertainty, we focus on smaller scales. This also has the advantage of limiting the role of DM particles should they exist. The future tests described in this section are ordered so the most near-term tests are described earlier, based on our understanding of the relative difficulty and the technological advances that may be required.

### 11.1. Galaxy Cluster Collision Velocities

The ΛCDM prediction for the growth of structure could be falsified by a sufficiently energetic galaxy cluster collision between sufficiently massive galaxy clusters early enough in cosmic history. El Gordo is one such example (Section 7.2), with others discussed in [712]. The collision velocities need to be inferred from hydrodynamical simulations, which creates some uncertainty due to issues such as projection effects [813]. To obtain the collision velocity more directly, we need to find some way to obtain the transverse velocity within the sky plane. This is normally done with proper motions, but these are too small at cosmological distances. Another possibility is to use precise redshifts of a background galaxy multiply imaged by a foreground moving lens, whose time-dependent potential causes the images to have slightly different redshifts [814]. This moving cluster effect (MCE) could be a way to directly obtain the transverse velocity of a galaxy cluster [815]. It may actually be more direct than a proper motion measurement because the signal would arise from the dominant matter component. The MCE has been suggested as a way to measure the present kinematics of the Bullet Cluster [816], where the expected signal is equivalent to a velocity of order 1 km/s.

If we only consider the redshifts of the multiple images, the MCE is degenerate with other effects such as differential magnification across the source galaxy [817]. However, those authors argued that the degeneracy can be broken with detailed spectral line profiles from the multiple images. For this, the two spectra should be compared with each other, so absolute redshifts are not required at the 1 km/s level. The techniques described could be applied to even more problematic examples such as El Gordo. Since its properties already rule out ΛCDM at high significance [712], other ways should be found to quantify its properties that do not rely on the assumption of ΛCDM. More generally, the properties of extreme objects can help to constrain the cosmological model.

### 11.2. Dynamically Old TDGs

The self-gravity of a dynamically old yet securely identified TDG is a very promising way to find the true cause of the missing gravity in galaxies (Table 2). It was prematurely claimed that this test has already been done using three TDGs around NGC 5291, with the results decisively favouring MOND [818–820]. However, these TDGs have disturbed velocity fields unsuitable for a traditional RC analysis, or more generally "are not enough virialized to robustly challenge cosmological scenarios" [821]. It is therefore important to check both the identification of a dwarf galaxy as a TDG and the reliability with which its self-gravity can be estimated observationally.

A more promising example is NGC 5557, where the TDGs are thought to be 4 Gyr old [526], making them much more likely to be virialized. Their TDG nature is also fairly secure because they lie within a faint tidal tail and are anomalously metal-rich for their mass. As a result, follow-up spectroscopic observations to determine their internal kinematics would be very valuable.

More generally, it is important to take deeper observations to try and identify tidal features and TDGs, and to then take follow-up spectroscopic observations. This approach

would require an advance on existing observational and data analysis techniques rather than completely new ones, making it less speculative than the other possible tests of MOND discussed next.

### 11.3. Wide Binaries

The currently most promising test of MOND is probably that involving wide binary stars in the solar neighbourhood. The basic idea is that because MOND posits an acceleration-dependent departure from Newtonian gravity, the departure sets in beyond a rather small distance in a system with a small mass (Equation (18)). In particular, the MOND radius of the Sun is only $r_M = 7$ kAU = 0.034 pc, much smaller than the galaxy. Since the Newtonian gravity at the edge of a uniform density sphere scales with its size and the galactic CDM halo is thought to cause an extra acceleration of $\approx a_0$ at a distance of $\approx 10$ kpc, the acceleration at the edge of a 0.1 pc radius CDM sphere with the same mean density would be $\approx 10^{-5} a_0$ [822]. However, MOND effects of order $a_0$ are expected in the solar neighbourhood because the galactic EFE is only slightly larger than $a_0$ [1] and the transition from Newtonian to Milgromian gravity is rather gradual (Figure 3).

To visualize the predicted effects, we use Figure 34 to show the expected MOND boost to the radial gravity of a point mass as a function of position in units of $r_M$. The results are based on Figure 1 of [823], whose Figure 2 shows the angle between the gravity and the radially inward direction (the maximum is $\approx 8°$). The results are shown in a way that is independent of the central mass, but do depend on the assumed external field and the interpolating function. The galactic external field on the solar neighbourhood (towards $+x$ in the figure) is set by kinematic constraints on the galactic RC [148], and indeed has been measured directly based on the acceleration of the solar system with respect to distant quasars [1]. The interpolating function is also well constrained by the RAR (Figure 8) and other considerations (see Section 7.1 of [111]). The simple form (Equation (15)) adopted in Figure 34 has a fairly shallow transition between the Newtonian and MOND regimes, as required to fit the RC data. Therefore, we can be confident that in the solar neighbourhood, MOND does indeed significantly enhance the gravity of a point mass beyond its MOND radius.

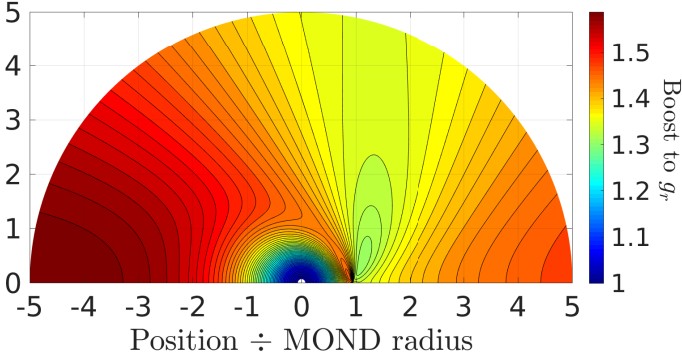

**Figure 34.** The colour shows the factor by which MOND enhances the radially inward Newtonian gravity from the Sun (white dot at the origin). The galactic gravity is included as a constant background field towards the $+x$ direction, about which the Milgromian solar potential is axisymmetric. Distances are shown in units of the MOND radius (Equation (18)), so the results here do not depend on the mass of the central object. They do depend on the assumed EFE strength and on the interpolating function, the simple form of which is used here (Equation (15)). Reproduced from Figure 1 of [823].

To test this prediction, we need a tracer. Fortunately, stars often have binary companions at kAU separations. Even the nearest star to the Sun (Proxima Centauri) is actually in a wide binary around $\alpha$ Centauri A and B [824], which for this problem can be treated as a single mass as their orbital semi-major axis is only 23 AU [825]. Thus, the orbital motion of Proxima Centauri should be subject to significant MOND effects [826,827].

It is worth considering whether the detection of such effects could be accommodated within the Newtonian framework. This is generally very difficult because wide binaries are so small that they should hardly be affected by CDM particles, even if these comprise most of the galaxy's mass. The main issue is the low density of the hypothetical galactic CDM halo, which would need to be much denser around the stars in a wide binary to appreciably affect its dynamics. It is unclear how such stellar CDM halos can form around newly born stars in the galactic disc, since the velocity dispersion of CDM particles in the galactic halo would be $\approx$200 km/s (e.g., [828]), well above the Sun's escape velocity at kAU distances. If such a halo nonetheless formed, it would need to be quite dense in order to mimic the MOND signal. As a rough guide, this requires an extra acceleration of $g_h \approx 0.3\, a_0$ at a typical separation between wide binary companions of $r_c = 10$ kAU. Applying the Copernican principle, such a stellar CDM halo should also exist around the Sun. This would cause a planet at orbital radius $r_p$ to experience an extra Sunwards acceleration of magnitude $\alpha a_0$, where

$$\alpha \;=\; 0.3 \left( \frac{r_p}{r_c} \right). \tag{73}$$

It has been demonstrated in equations 8 and 11 of [369] that the resulting displacement to the planet's position after half its orbital period is

$$d \;=\; \frac{2\alpha a_0 r_p{}^3 \sqrt{1 + \pi^2}}{GM_\odot}, \tag{74}$$

which for Saturn is just over 5 km as $r_p = 9.58$ AU. Accurate radio tracking of the Cassini orbiter around Saturn over approximately half its orbital period has revealed no such anomalous acceleration, with the ephemeris accurate to 31.6 m (Table 11 of [829]). Therefore, observations strongly exclude an extra 5 km displacement due to a solar CDM halo. Moreover, stellar CDM halos are not expected. It is therefore clear that Newtonian gravity cannot be reconciled with a significant detection of the MOND-predicted extra gravity in solar neighbourhood wide binaries.

11.3.1. Using the Velocity Distribution

The long orbital periods of wide binaries make it difficult to use their orbital acceleration as a test of the low-acceleration gravity law. For instance, the Keplerian orbital period of Proxima Centauri around $\alpha$ Centauri A and B is $\approx$550 kyr [824], so observations would be limited to a parabolic arc during which the acceleration hardly changes. However, this is only one wide binary. Statistical analysis of the relative velocity distribution in a large sample of wide binaries should reveal a larger velocity dispersion than Newtonian expectations if Milgromian gravity applies [328]. Those authors prematurely concluded in favour of MOND, but it was later shown that many of the claimed wide binaries are chance alignments, so strong conclusions cannot yet be drawn [830]. More recently, Gaia data release 2 [831] has provided a large sample of wide binaries (e.g., [832]). This was again used to argue in favour of MOND [833], but several problems were pointed out with their analysis [834,835], including especially the reliance on systems separated by $\gtrsim$100 kAU which would be quite prone to tidal disturbance [836]. It is also very important to look at the entire distribution of relative velocities, not just the velocity dispersion [837]. Those authors recommended a focus on the parameter

$$\widetilde{v} \;\equiv\; v_{\mathrm{rel}} \div \overbrace{\sqrt{\frac{GM}{r}}}^{\text{Newtonian } v_c}, \tag{75}$$

where $M$ is the total mass of a wide binary with relative velocity $v_{\mathrm{rel}}$ and separation $r$. In Newtonian mechanics, $\widetilde{v} < \sqrt{2}$ for a bound orbit, but higher values are possible in MOND

or other modified gravity theories [837]. Observationally, it would be easier to work with $\widetilde{v}_{\mathrm{sky}}$, which is calculated similarly to Equation (75) but using only the sky-projected separation and relative velocity [111]. In what follows, we will use $\widetilde{v}$ to mean $\widetilde{v}_{\mathrm{sky}}$ and will not consider the full 3D relative velocities of wide binaries, partly because RVs are subject to uncertain zero-point offsets from gravitational redshift [838]. Even with this restriction to $\widetilde{v}_{\mathrm{sky}}$, the systemic RV is still important due to perspective effects [834]—but this need not be known very precisely [835]. It can also be assigned based on the location within the galactic disc, with an uncertainty commensurate with the local stellar velocity dispersion tensor. However, it is anticipated that the systemic RVs of many wide binaries will become available with future Gaia data releases at $\approx 1\,\mathrm{km/s}$ precision.

An important aspect of the wide binary test of gravity is the galactic EFE [111]. Using analytic and numerical methods, those authors showed that the circular orbital velocity of wide binaries in the solar neighbourhood should exceed Newtonian expectations by 20%. Uncertainties in galactic parameters hardly alter this, and expectations differ little between AQUAL and QUMOND (see their Table 3). The unknown eccentricity distribution of the wide binaries has a much smaller impact on their $\widetilde{v}$ distribution than the gravity law (see their Figure 3, reproduced here as our Figure 35). The galactic EFE certainly makes this test more challenging, but we argue that it is still extremely promising, especially in light of Gaia early data release 3 [727]. Indeed, data on wide binaries have already provided important constraints and could probably have decisively tested MOND by now without the galactic EFE (Section 3.3).

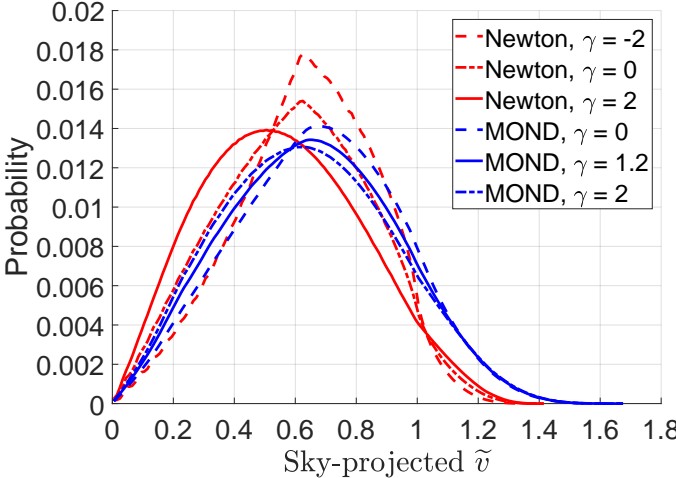

**Figure 35.** The predicted distribution of $\widetilde{v}$ (Equation (75)) for solar neighbourhood wide binaries in Newtonian gravity (red) and MOND (blue), with only sky-projected quantities used here. The different linestyles show different distributions of the orbital eccentricity $e$, whose prior is parametrized as $1 + \gamma(e - 1/2)$. If one of the results shown here is correct, it cannot be matched in the other gravity theory simply by varying $\gamma$, whose value is indicated in the legend. Reproduced from Figure 3 of [111].

As with other tests of gravity, the wide binary test is subject to contamination. The main contamination was expected to come from undetected close binary companions to one or both of the stars in a wide binary, which would cause an extended tail going well beyond $\widetilde{v} = 3$, the limit for plausible modifications to gravity [111]. The observed $\widetilde{v}$ distribution does in fact have such a tail [329]. This is very unlikely to be caused by LOS contamination (see their Section 3.1). Moreover, their Table 1 shows that the $r_{\mathrm{sky}}$ dependence of the contamination is not $\propto r_{\mathrm{sky}}$ as would be expected for LOS contamination, arguing very strongly against this hypothesis. Rather, the number of wide binaries with unphysically high $\widetilde{v}$ depends on $r_{\mathrm{sky}}$ similarly to the genuine wide binary population, whose contribution can be estimated from the peak in the $\widetilde{v}$ distribution at $\widetilde{v} \approx 1$. This is very suggestive of the expected close binary contamination [839]. In addition, stars in binaries with high $\widetilde{v}$

have a poorer Gaia astrometric fit (parallax + proper motion), suggestive of photocentre acceleration induced by an undetected close binary companion [840].

The issue of close binaries probably underlies a recent very problematic claim to have confirmed MOND and ruled out Newtonian gravity with wide binaries [841]. The study claimed to confirm a version of MOND without the EFE, which as argued in Section 3.3 is already strongly excluded observationally [329] based on the shape of the $\widetilde{v}$ distribution—which is not shown in [841]. They instead use its root mean square value (i.e., the velocity dispersion). This statistic is not meaningful because it would be significantly inflated by the inevitable low amplitude tail going out to $\widetilde{v} \gg 1$ (Figure 10). This tail is probably due to close binary contamination and will be very difficult to remove altogether even with detailed follow-up campaigns (see Section 8.2 of [111]). A better strategy would be to focus closely on the main peak evident at $\widetilde{v} \approx 1$, which should be broadened in MOND and lead to an excess of systems in the crucially important range $\widetilde{v}$ = 1–1.3.

It will therefore be important to jointly fit the wide binary and close binary populations, though it is probably safe to treat the two as independent and the close binaries as Newtonian. A careful statistical treatment is needed to determine whether wide binaries can be fit using Newtonian gravity or require a helping hand from Milgrom (for a detailed plan, see [842]). Properties of the close binary population can be deduced from the tail at high $\widetilde{v}$ and from systems with low $r_{\text{sky}}$, where MOND effects should be insignificant. The idea would be to fit the observed distribution of $\left( r_{\text{sky}}, \widetilde{v} \right)$ using a forward model including both close and wide binaries, with the latter having a parametrized gravity law in between Newtonian and Milgromian that the analysis could vary so as to best match the data. The analysis could also be restricted to different subsamples which should be differently affected by MOND. A possible distinct signature of MOND would be if a subsample with wide binaries separated by more than their MOND radius prefers, e.g., a significantly different close binary fraction than wide binaries with smaller separations.

### 11.3.2. Using the Acceleration of Proxima Centauri

Accurate observations of our nearest wide binary should allow for a decisive test of the low-acceleration gravity law [843]. By considering Proxima Centauri as a test particle subject to the combined gravity of the much more massive $\alpha$ Centauri A and B, those authors showed that its orbital acceleration should be $0.60\,a_0$ in Newtonian gravity but $0.87\,a_0$ in MOND, with a very similar expected direction in both cases.

This subtle effect might be detectable with the proposed Theia mission [844] or something similar [845]. Due to the low temperature of Proxima Centauri, the prospects are somewhat better if using near-infrared wavelengths [846]. After a decade of observations, there would be a difference in RV of 0.52 cm/s between the models, while that in sky position would be 7.18 μas. An astrometric measurement is probably better because the signal to noise ratio would rise with time $T$ as $T^{5/2}$, but the improvement is only $T^{3/2}$ for the RV. This is because a constant acceleration changes the position $\propto T^2$, whereas the velocity only changes $\propto T$. In either case, the measurement would constitute a direct test of MOND because the acceleration would be measured directly. It is interesting to note that if we had evolved around Proxima Centauri and developed similar technology, the measurement of our parent star's acceleration relative to distant quasars to a precision of $0.13\,a_0$ [1] would already place significant tension on Newtonian gravity if MOND were correct, and vice versa.

### 11.4. Solar System Ephemerides

Due to the high accuracy of solar system ephemerides (e.g., [829]), these place important constraints on allowed modifications to GR. One might naively expect that the corrections arising from MOND can be exponentially suppressed using an interpolating function such as Equation (48). It is certainly true that this transition function would exponentially suppress the local value of $\nu - 1$ in regions where $g_N \gg a_0$. However, the gravitational field at any location also depends on the behaviour of $\nu$ elsewhere (Equation (2)). The

solar system might thus be affected by the PDM distribution at large heliocentric distances of order the solar $r_M = 7$ kAU. The solar PDM distribution departs significantly from spherical symmetry at these distances due to the galactic EFE (e.g., Figure 34). This creates a divergence-free tidal stress or anomalous quadrupole on the solar system [847]. The magnitude of this dipole is usually denoted by $Q_2$, which is defined so that up to a constant the MOND correction to the potential within the solar system is

$$\Delta\Phi_M = -\frac{Q_2}{2} r^i r^j \left( \widehat{g}_e^i \widehat{g}_e^j - \frac{1}{3}\delta^{ij} \right), \tag{76}$$

where $\widehat{g}_e$ is the direction of the galactic external field, $i$ and $j$ are spatial indices which in 3D can each take on three possible values, $\delta^{ij}$ is the Kronecker $\delta$ function with value 1 if $i = j$ and 0 otherwise, and the summation convention is used over $i$ and $j$. This form of $\Delta\Phi_M$ creates no extra divergence to the potential.

The MOND prediction for $Q_2$ has to be determined numerically and depends on the choice of interpolating function [848]. Their work shows that many previously considered MOND interpolating functions are strongly excluded by solar system ephemerides. Subsequent work using Cassini radio tracking data [849] tightly constrains $Q_2$ to the range $(3 \pm 3) \times 10^{-27}$ s$^{-2}$ [119]. The interpolating function in Equation (48) is called $\hat{\nu}_1$ in Table 2 of [120], which shows that the resulting $Q_2 = 31 \times 10^{-27}$ s$^{-2}$. While this seems to be at odds with Cassini radio tracking data, we should bear in mind that Figure 6 of [119] shows that the typical uncertainties from Cassini data in different time intervals are slightly above $10^{-26}$ s$^{-2}$, with those authors considering three time intervals of approximately 3 years each. It is also possible that larger values of $Q_2$ are permissible in a full fit to solar system ephemerides where the masses of planets and asteroids are allowed to vary so as to accommodate the anomalous quadrupole predicted by MOND. In particular, an undiscovered planet [850] or significant amount of mass in asteroids in the outer solar system could create a tidal stress that masks the MOND effect. The hypothesis of a distant ninth planet was originally motivated by an apparent clustering in the orbital elements of Kuiper Belt Objects, but the data are subject to significant selection biases [851]. The latest analyses including such biases cast very serious doubt on the original claims of significant clustering, undermining the motivation for an extra planet [852,853]. This of course does not prove that we have identified all the mass present in the outer solar system. Finally, there is always the possibility that the interpolating function can be adjusted to further reduce the predicted $Q_2$. In particular, Table 2 of [120] demonstrates that even sharper transition functions could easily satisfy the constraint from [119], whose equation 7 shows that the predicted $Q_2$ should lie in the range $(2.1 - 41) \times 10^{-27}$ s$^{-2}$. The constraint on $Q_2$ derived in [119] can thus easily be accommodated in MOND even if we consider only a few families of interpolating functions (see their equation 5).

We therefore argue that solar system ephemerides do not currently pose a significant challenge to MOND. However, this conclusion could change if the precision improves another order of magnitude and MOND effects are still not detected despite being correctly included in the model.

### 11.5. Spacecraft Tests

In the longer term, spacecraft can be sent to low-acceleration regions in order to test MOND. Tracking of the spacecraft and/or onboard measurements should give definitive results. One difficulty is that the nearest low-acceleration MOND bubbles of appreciable size are still rather distant, though accurate data from a spacecraft sent out to 150 AU could still be very valuable and would probe lower accelerations than ever before [854]. Larger distances should eventually be attainable, with the MOND radius (7 kAU) perhaps providing an important intermediate goal on the road to missions that reach other stars [855]. Such truly interstellar missions would need to reach at least 270 kAU, which is the aim of the Breakthrough Starshot initiative [856]. Importantly, some potentially usable MOND

bubbles are closer than distances that have already been traversed by spacecraft in the last millennium, as discussed next.

### 11.5.1. Within the Solar System

Since the MOND radius of the Sun is 7 kAU, we might naively expect that MOND effects would be very small in the solar system. While this is generally correct [119,120], more significant effects might be apparent in regions where there is a cancellation between the solar gravitational field and the gravity from a planet, i.e., at a saddle point in the gravitational potential [857]. In principle, it is not necessary to reach the point where the gravity completely vanishes. This is because the simple interpolating function (Equation (15)) preferred by observations [111,141,211,295] implies significant MOND effects in the saddle region, the volume within which $g < a_0$. Therefore, spacecraft tests of MOND could aim for the saddle region between a planet and the Sun (for a review, see [858]).

One such proposal was to send the Laser Interferometer Space Antenna (LISA) Pathfinder mission [859] through the Earth–Sun saddle region [860,861]. However, it was later shown that a null detection of MOND effects by LISA Pathfinder would be consistent with a wide range of MOND interpolating functions [120,862]. The main reason is that the combined Earth–Sun tidal stress would be quite significant, so even if it was possible to precisely identify the Earth–Sun saddle point, the MOND bubble around it would be very small. For any planet of mass $M_p \ll M_\odot$ on a circular orbit of size $r_p$ maintained by the solar gravitational field $g_\odot = GM_\odot/r_p{}^2$, the planetary gravity $g_p$ must be very close to $g_\odot$ at the saddle point in order to achieve a cancellation. The distance of the saddle point from the planet is then $d_s \approx r_p\sqrt{M_p/M_\odot} \ll r_p$, so the tidal stress orthogonal to the Sun–planet line is $g' \approx g_p/d_s$. This is because the solar tide of $\approx g_\odot/r_p$ is negligible in comparison to the planetary tide. The saddle region thus has a width orthogonal to the Sun–planet line of $w_s = a_0/g'$, which combining the earlier results gives

$$ w_s \;=\; \frac{a_0 r_p{}^3 \sqrt{M_p}}{GM_\odot{}^{3/2}} \,. \tag{77} $$

The saddle region would extend half as much along the Sun–planet line because the tidal stress is twice as strong this way for an inverse square gravity law.

Quantitatively, the Earth–Sun saddle region has a size of only 4.4 m along the Earth–Sun line [858]. This would be crossed very rapidly given the significant orbital velocity of the LISA pathfinder—or indeed any spacecraft at heliocentric distances of $r \approx 1$ AU. This severely limits the prospects for a decisive test of gravity in the low-acceleration regime. Similar issues are likely to arise with the Earth–Moon saddle region [863]. Those authors suggested measuring the time taken for light to cross this region using retroreflectors installed on the Moon. The Sun would generally move the saddle region off the Earth–Moon line, but alignment might be restored during a lunar or a solar eclipse (see their Figure 1). Even so, other planets need to be considered as well, so it would be extremely fortunate if any anomalous signal were detected. This would likely make it very difficult to repeat the experiment. The small size of the saddle region also means that the extra Shapiro delay due to MOND would be very small.

These difficulties can largely be overcome by considering the saddle point between the Sun and a gas giant planet [858]. Not only is the Keplerian velocity smaller, the saddle region is also larger because its linear dimensions are $\propto r^3 \sqrt{M_p}$, where $M_p$ is the planet mass (see its equation 37 and our Equation (77)). To estimate the crossing time, consider a spacecraft at the saddle point in both position and velocity, i.e., at rest in a reference frame rotating around the Sun at the Keplerian angular velocity of the planet. The spacecraft does not accelerate in the heliocentric frame, but since the planet accelerates towards the Sun at $g_p$, the saddle point almost does so as well. In the rotating frame, this appears as the spacecraft accelerating away from the Sun at $g_p$ while the saddle point remains fixed. Due to the very low velocity in the rotating frame compared to the planet's Keplerian velocity, Coriolis forces would be negligible, so the spacecraft would simply accelerate

directly away from the Sun at $g_p$ in the rotating frame. If the spacecraft starts at rest in this frame a distance $w_s$ Sunwards of the saddle point, then it will reach the same distance on the anti-solar side after a duration

$$T_s = \frac{2a_0^{1/2} r_p^{5/2} M_p^{1/4}}{GM_{\odot}^{5/4}} \, . \tag{78}$$

The Sun–Jupiter saddle region has an extent of 10.2 km along the Sun–Jupiter line, so a spacecraft on a freely falling trajectory should have 5 hours in the Jovian saddle region [858]. The prospects are even better for the Sun–Neptune saddle region, which extends 475 km along the Sun–Neptune line. Perturbations to the saddle region's location from Triton can be calculated fairly precisely because its mass is well known thanks to the Voyager 2 close spacecraft flyby in August 1989. A freely falling spacecraft could spend 209 hours in the Sun–Neptune saddle region, which should yield quite definitive results.

Once in the saddle region, the most obvious test is to conduct a Cavendish-style active gravitational experiment where two masses freely fall towards each other and both fall towards the spacecraft. To limit uncertainties from solar radiation pressure, one or both of the masses could be hollow so they have the same ratio of mass to surface area, or identical masses could be used. The gravity $g_{\text{rel}}$ between the masses could be determined by observing their separation decline with time. As an example, a 1 kg mass acting on a test particle 1 m away induces a Newtonian acceleration of $0.56 \, a_0$. If the masses are initially at rest relative to each other, their separation would decrease by 1.1 cm after five hours. MOND would change this by tens of percent, depending on how close the spacecraft gets to the saddle point. The predicted MOND effects should be readily detectable using, e.g., laser metrology. Moreover, the masses would generally experience a lateral acceleration in MOND (see, e.g., Equation (34)). If detected, the resulting change in the orientation of the masses would be very difficult to explain in Newtonian gravity and would give strong constraints on what should replace it [65]. It may also be possible to repeat the experiment during the same saddle region crossing, perhaps to constrain the interpolating function by exploring how the deviation from Newtonian gravity correlates with how deep inside the saddle region the spacecraft is.

To further explore the proposal of [858], we set up a simulation of a spacecraft crossing the Sun–Neptune saddle region, with all motions assumed to lie within the orbital plane of Neptune. We use semi-analytic force calculations based on numerically determined interpolations between the various asymptotic limits in which analytic solutions are available. The initial spacecraft position is assumed to be $w_s$ Sunwards of the saddle point and also $w_s$ off the Sun–Neptune line, to mimic the effect of a targeting error that causes the spacecraft to slightly miss the saddle point. The spacecraft is assumed to start exactly at rest with respect to the saddle point. Coriolis and centrifugal forces are included on the spacecraft, which is assumed to have a mass of 100 kg and an otherwise conventional trajectory. The two test masses are assumed to be 1 kg Tungsten spheres, one of which we place 20 m from the spacecraft orthogonal to the Sun–Neptune line. The second test mass is 3 m from the first in the anti-Sunwards direction, so the barycentre of the test masses is initially just over 20 m from the spacecraft. The Newtonian accelerations in the experiment are thus $\approx 0.1 \, a_0$. In addition to gravity, we also consider radiation pressure effects by assuming the Tungsten spheres absorb all sunlight incident on them and reradiate completely isotropically, with some of this reradiation falling on the other test mass. We neglect radiation from the spacecraft and from Neptune. The separation between the test masses and the distance from their barycentre to the spacecraft evolve with time under the effect of gravity and radiation pressure, as shown in Figure 36. The evolution differs significantly depending on whether we assume Newtonian or Milgromian gravity, with the separation between the test masses providing the larger signal. They almost collide after 100 hours, so we terminate the experiment then. At that point, the test masses could perhaps be restacked with the robotic arm to start another experiment, better using the remaining $\approx 100$ h.

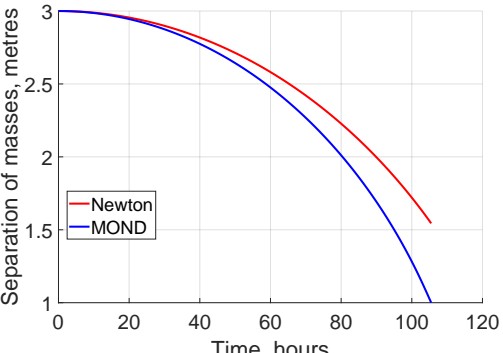 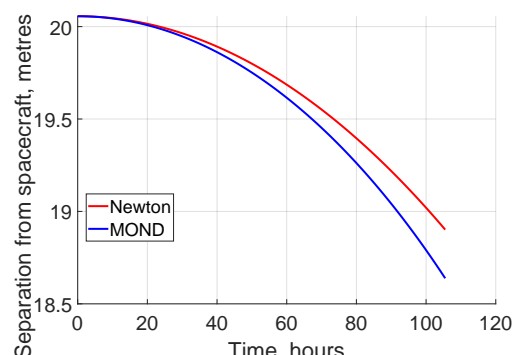

**Figure 36.** The expected behaviour of a Cavendish-style active gravitational experiment onboard a spacecraft sent to the Sun–Neptune saddle region in Newtonian gravity (red) and MOND (blue) with the MLS form of the interpolating function (Equation (48)). The panels show the distance between the two 1 kg Tungsten test masses (**left**) and between their barycentre and the spacecraft's 100 kg main body (**right**) as a function of time since the start, when the spacecraft is assumed to be at rest relative to the saddle point (see the text). The experiment is stopped while the spacecraft is still in the saddle region because the test masses almost collide. In principle, the experiment could then be rerun in the remaining $\approx$100 hours within the saddle region (not shown).

We also consider using instead the Sun–Jupiter saddle region, but its much smaller size and the resulting shorter crossing time of just under 5 hours means that the Newtonian and Milgromian displacements differ by only $\approx$1 mm, which would be much more difficult to detect. The $50\times$ smaller $w_s$ also means that the targeting accuracy would need to be much better. Another complication is that radiation pressure would be much more significant than around Neptune, though in both cases the resulting uncertainty could be reduced by repeating the experiment outside the saddle region crossing. Regardless of which planet is used, the difference in the expected test mass acceleration between the two theories is expected to be of order $0.1\,a_0$, so the resulting difference in position scales quadratically with the duration of the saddle region crossing. Consequently, Neptune offers far better prospects for obtaining decisive results (Equation (78)).

The initial configuration for this test involves reaching the Sun–Neptune saddle region with the same heliocentric angular velocity as Neptune, thereby moving at the same velocity as the saddle point. The required velocity is slightly smaller than that of Neptune by a fraction close to $\sqrt{M_p/M_\odot}$, while the Keplerian velocity around the Sun at the saddle point is fractionally higher than that of Neptune by approximately half as much. The spacecraft would therefore have to reach the aphelion of an almost circular orbit around the Sun with a semi-major axis slightly below that of Neptune by a fraction close to $4\sqrt{M_p/M_\odot}$. Due to Neptune's long orbital period, ensuring an acceptable mission duration would likely entail using ion engines to accelerate away from the Sun and then decelerate to the required velocity. A gravity assist at another gas giant could be used to reduce the fuel required. The overall profile would be similar to a Neptune orbiter, which indeed the spacecraft could become after conducting an active gravitational experiment in the Sun–Neptune MOND bubble. A Neptune orbiter could lead to much tighter constraints than Cassini around Saturn on any anomalous tidal stress created by modified gravity theories, perhaps leading to another highly sensitive test of MOND (Section 11.4). It is therefore quite possible to experimentally test MOND using existing technologies deployed much closer than the distances to several currently operational spacecraft.

### 11.5.2. Beyond the Solar System

Looking to the more distant future, further tests will become possible as spacecraft reach greater distances. In particular, an interstellar precursor mission travelling at $0.01\,c$ would take 11.1 years to reach the Sun's MOND radius 7 kAU away. Outside the Newtonian bubble created by the solar gravity, strong MOND effects should become apparent, though

still limited by the galactic EFE. Simply tracking the spacecraft would provide important constraints because the Sunwards gravity should be stronger in MOND, reducing the two-way light travel time by $\approx 0.1$ s after 20 years [823]. In addition, the EFE would break isotropy, causing the spacecraft to undergo a characteristic lateral drift of order 0.1 mas on the terrestrial sky. This is illustrated in our Figure 37, which reproduces their Figure 5. Since the solar potential would still retain axisymmetry about the external field, the Sun, spacecraft, and galactic centre would all remain in the same plane. The lateral drift is caused by the fact that at the same heliocentric distance, the MOND potential is typically deeper along the external field direction (see, e.g., Figure 34). The drift would thus be in a particular direction that can be calculated in advance based on the spacecraft trajectory well within the MOND radius but beyond perturbative effects from the gas giants, potentially allowing for a highly distinctive test by launching multiple spacecraft in different directions. The angular deflection could be detected by comparing the arrival times of signals at receiving stations around the world, and possibly also on the Moon [864].

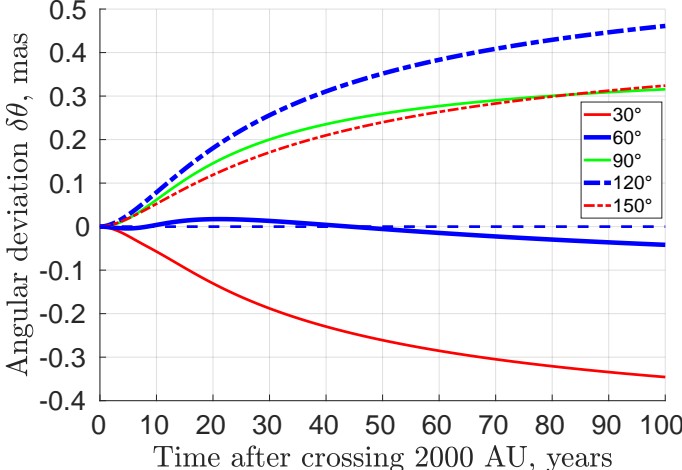

**Figure 37.** Difference between the Newtonian and Milgromian heliocentric sky position of an interstellar precursor mission travelling at $0.01\,c$ as a function of time after crossing 2 kAU from the Sun. Different lines correspond to different angles between the launch direction and that towards the galactic centre, as indicated in the legend. If this angle rises during the mission, then $\delta\theta > 0$ as the Newtonian prediction is for the spacecraft to travel on a straight line, shown here as the flat dashed blue line at $\delta\theta = 0$. Non-zero values arise in MOND because it generally predicts a deeper solar potential on the external field axis than at right angles (Equation (34); see also [65]). Reproduced from Figure 5 of [823].

Without signals emitted by the spacecraft, it would still have some capacity to transit across background stars, perhaps allowing its sky position to be determined. However, even if we assume the spacecraft has a sail of radius 100 m, this represents an angular radius of only $\theta_s = 0.014\,\mu$as at a distance of 10 kAU, slightly beyond the MOND radius. A Sun-like star in the galactic bulge 8.2 kpc away [147] would have a much larger angular radius of $\theta_\star = 0.567\,\mu$as, making the fractional transit depth only 0.06%. Due to the Earth's motion, the spacecraft would appear to move in parallactic ellipses. Assuming a launch towards an ecliptic pole, these would become circles of radius $p = 20.6''$, though the radius would shrink as the spacecraft recedes. Due to this parallax, the transit would last only 0.283 s. Moreover, a transit would require the centre of a star to come within an angle $(\theta_s + \theta_\star)$ of the spacecraft, so we can imagine it defining a narrow track on the sky of twice this width and whose length over a year is $2\pi p$. The area of this track is then $1.51 \times 10^{-4}$ square arcseconds. Assuming optimistically that the spacecraft is launched towards a very crowded direction with $10^9$ stars in a $2° \times 2°$ field of view, there would be 19.3 stars per square arcsecond. The likelihood of a transit in any given year is then 0.29%,

which would decrease quickly as the spacecraft becomes smaller on the sky and has less annual parallax.

Accurate tracking of the spacecraft will therefore be possible only with a functioning transmitter. Fortunately, it need not send signals very regularly if the main goal is to test the tangential component of the solar gravitational field predicted in MOND [823]. Additional instruments onboard the spacecraft could allow even more definitive tests. The above-mentioned lateral drift might be easier to detect using a camera on the spacecraft pointed towards the Sun, which in MOND would move relative to background stars. It might be easier for the spacecraft to detect the Sun against background stars and report its findings than for terrestrial observers to precisely locate the spacecraft.

A low-acceleration Cavendish-style active gravitational experiment beyond the Sun's MOND radius should behave quite differently to Newtonian expectations. A test particle 1 m away from a 1 kg test mass experiences a Newtonian acceleration below $a_0$. Such an experiment might be feasible to set up by, e.g., deploying two masses using a robotic arm. Their mutual gravity would not only be enhanced in MOND, it would generally not point along the line separating them. By varying the orientation of the experiment and its distance from the spacecraft, it would be possible to build up a detailed picture of the potential generated by the masses in the experiment as a function of the external field across it. Since an acceleration of $a_0$ would cause a particle at rest to move by 8 cm after ten hours, it should be possible to obtain results in a reasonable timeframe once the experiment is underway. In principle, it would be possible to repeat the experiment indefinitely because the solar gravity would become even weaker as the spacecraft continues receding from the Sun. This would be a major advantage over spacecraft tests in saddle regions of solar system planets, which need to be conducted over a limited duration (approximately given by Equation (78)). With technological advances, it is clear that conclusive results will ultimately be obtained using space-based laboratory experiments, though by then the debate over the cause of the missing gravity problem might already have been settled.

## 12. Conclusions

We are now in a position to weigh up the evidence for MOND and whether it might do better than the currently popular ΛCDM framework. Adding together the confidence scores in Table 5 gives an overall measure of confidence in each theory, which we summarize as its final score (Table 6). For ΛCDM, we do not use the CMB in the calculation of its final score as the CMB is crucial to setting the parameters of the ΛCDM model. For MOND, the test involving HSB disc galaxy RCs has been excluded as these were used in theory construction, even though only a small proportion of the presently available RC data were available in the early 1980s. Slightly more tests have been applied for ΛCDM due to it being better developed, so the most useful quantity for both theories is the average confidence listed in the final column.

**Table 6.** The total confidence in ΛCDM and MOND based on how well each theory performs against each test, bearing in mind its theoretical flexibility (Table 5). The test used to construct each theory is not counted here. Slightly more tests are possible for ΛCDM because it is theoretically better developed. We therefore use the final column to show the average confidence score for each theory across all the tests considered in this review. It is clear that overall, MOND significantly outperforms ΛCDM.

| Theory | Total Score | Number of Tests | Average Score |
|---|---|---|---|
| ΛCDM | −8 | 32 | −0.25 |
| MOND | +49 | 29 | +1.69 |

ΛCDM yields an average confidence of −0.25 across 32 tests. This indicates that clear prior expectations are generally strongly excluded by the latest data, or that areas with good agreement involve a significant amount of theoretical flexibility regarding the

calculations, many of which were obviously done in full view of the observational facts that needed to be explained. It is not very common that clear prior predictions in the ΛCDM paradigm are subsequently confirmed, though a small number of such cases exist and have been included.

In contrast, the 29 tests applied to MOND return an average confidence of +1.69, which corresponds to plausible agreement between observations and a clear prior prediction. A confidence of +2 can also mean excellent agreement where additional assumptions beyond MOND are required but these have only a small effect on the results. Therefore, the latest data support MOND given its low theoretical flexibility. The main reason is its capacity to predict observations that have not yet been made, which is widely considered a hallmark of a good scientific theory [50].

Another way to consider the situation is that the success of a model in explaining some observables does not imply the model is correct, especially if there is a significant amount of flexibility. However, a clear failure to explain some observables does imply that the model is wrong. Therefore, the correctness of a model should not be judged primarily by its successes, least of all those it was designed to achieve (the CMB in ΛCDM and the HSB galaxy RAR in MOND). Much more important is whether the model provides a plausible explanation for its weakest aspects, or whether these represent genuine falsifications of the paradigm. For instance, it is unlikely that the LG satellite planes will ever be accommodated in ΛCDM (Section 5.6), whereas it is quite plausible that MOND will address cosmological observables such as the CMB and the Bullet Cluster that are sometimes considered challenging for it (Section 9.2). Because of this, many tests give ΛCDM a negative confidence, but this is rare in MOND.

We conclude that observations distinct from those used to set up ΛCDM and MOND strongly disfavour the ΛCDM hypothesis because there are now several independent highly significant falsifications of this paradigm, a conclusion reached independently by other authors (e.g., [53–55,731]). These falsifications point rather specifically to failure of the GR and CDM assumptions, even if various other assumptions such as dark energy may be more secure. Looking instead at the successes of each paradigm paints a similar picture, since those for MOND were generally clear a priori predictions with negligible theoretical flexibility. Meanwhile, some of the claimed successes of ΛCDM involved a great deal of adjustments to various free parameters in full view of the data, such that a wide range of observations could plausibly have been accommodated. This lends little confidence to the theory, unlike a clear prior prediction that is subsequently confirmed [50]. Moreover, observations used in theory construction or to set free parameters cannot be argued to increase our confidence in the theory. We have accounted for this by not considering the CMB when assessing ΛCDM and the HSB disc galaxy RAR when assessing MOND. Bearing this in mind, the vast array of evidence presented in this review on balance strongly prefers a breakdown in GR at low accelerations, falsifications of which range from the kpc scales of galaxy bars to the Gpc scale of the KBC void and Hubble tension. It therefore seems inevitable that a MOND-based cosmological framework will soon supersede ΛCDM, thereby providing a much better stepping stone on the quest to understand the fundamental quantum gravitational laws governing our universe.

**Author Contributions:** I.B. led the review and wrote most of it. H.Z. wrote the part on the neutrino-based non-relativistic Lagrangian and the covariant version of this. He also helped to edit the theoretical section of this review. All authors have read and agreed to the published version of the manuscript.

**Funding:** This research was funded by Science and Technology Facilities Council grant number ST/V000861/1.

**Acknowledgments:** I.B. is supported by Science and Technology Facilities Council grant ST/V000861/1, which also partially supports H.Z. I.B. acknowledges support from a "Pathways to Research" fellowship from the University of Bonn. He is grateful to Elena Asencio for providing two of the figures used here and for helping to decide the scores assigned to ΛCDM and MOND for each test. The

authors are grateful for permission to reproduce several figures from previous publications. They also thank Benoit Famaey, Moti Milgrom, Nils Wittenburg, Pavel Kroupa, Raphaël Errani, Stacy McGaugh, and Alfie Russell for helpful discussions. I.B. would like to thank Luis Acedo for inviting this review. The authors are grateful for helpful comments from the anonymous referees.

**Conflicts of Interest:** The authors declare no conflict of interest with this review.

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
