# Peer review of "From Galactic Bars to the Hubble Tension: Weighing Up the Astrophysical Evidence for Milgromian Gravity"

_symmetry, doi:10.3390/sym14071331_

Round 1
Reviewer 1 Report
This manuscript presents an exhaustive review of cosmology and extragalactic astronomy in the context of the modified Newtonian dynamics hypothesized by MOND. It is extremely thorough and clearly written, and should be published after some minor but substantive fixes.
Small omission: in the last paragraph of page 31 "The density wave theory should also apply in LSBs." Yes. Yes it should. This was explicitly used by McGaugh & de Blok (1998) to predict that conventional analyses would find uncomfortably high stellar M*/L, as later found by Fuchs and also Saburova & Zasov (2013). This was a successful a priori prediction of MOND, so should be mentioned given this manuscript's emphasis on the scientific method.
Section 3.3: I find this discussion very strange. It is nominally about "observational signatures of the EFE" but starts by talking about one galaxy, DF2 - an object that they've already discussed at great length earlier in the manuscript. This is like talking about a branch in a forest without mentioning all the trees. DF2 is one ordinary if very overhyped dwarf galaxy with dodgy data that have changed continually since it van Dokkum's group decided to make the proverbial mountain out of a molehill. The opening sentence is untrue, that DF2 can only be reconciled with MOND with the EFE. That depends on its distance from NGC 1052, and what the inclination of its rotating globular cluster system is. One can easily extract from their data an equivalent velocity dispersion of 13 km/s (the EFE prediction) or 20 km/s (the isolated prediction) depending on the inclination. Which of these apply depends on the distance, which is hotly debated. There is, at present, no credible test with this galaxy. I can understand the desire to address this case, if not to do it again, and it is a very strange place to start a discussion of evidence for the EFE. A more obvious starting point would be Milgrom's earlier (1995) "seven dwarfs" paper, and McGaugh & Milgrom (2013b). The latter paper is cited in several places but not in this section, despite having a discussion of photometrically matched dwarfs that are or are not subject to the EFE... an effect that was essential to the successful a priori prediction of the M31 dwarfs with lower velocity dispersions - the very same ones that are puzzling in LCDM, and are dismissed by conventionalists as conveniently out of equilibrium. If that is fair for LCDM, then it is also fair for MOND, which brings us to the discussion of Crater 2, which is both bizarre and wrong. Advocates of dark matter often dismiss the successful predictions of MOND as some kind of magic, as if they'd rather believe in magic than science if it means giving credit to MOND. Here the authors - in a lengthy review advocating MOND - engage in this same magical thinking. There is a long discussion of how it shouldn't be possible to predict the velocity dispersion of this object because of tidal effects, thus dismissing the successful a priori prediction of the velocity dispersion of this object. This discussion grossly misrepresents the work of McGaugh (2016), and illustrates a rather naive static thinking on the part of the authors. Science is a series of increasingly accurate approximations. They are correct that Crate 2 is subject to tidal effects. Indeed, now that we know it had a close pericentric passage, it is pretty clear that it has achieved its current ultradiffuse form as a result of that, evolving along the [non-equilibrium, non-adiabatic] lines anticipated by the modeling of Brada & Milgrom (2000). McGaugh (2016) discussed this (albeit using the criterion of Brada & Milgrom rather than the one they choose) and said OK, can't know this will work, but the timescales are so long it is worth trying the EFE approximation. That this indeed works out would, in the normal course of scientific discussion, be interpreted as evidence that this approach was indeed an adequate approximation, bearing in mind that no approximation is ever perfect. Unexamined here are the predictions (or lack thereof) of LCDM, which by any of several methods anticipated a much higher velocity dispersion for this object (nominally 17 km/s; see section 4.2 of McGaugh 2016). So we wind up in the strange situation where the authors are advocating MOND as a theory, have a case with a clear predictive difference between LCDM (17) and MOND (2), the observation is 2.7, and what they say is that the MOND prediction can't possible be right despite being right while not even mentioning how for off the conventional prediction is. All in a section about evidence for the EFE, of which there are many better examples. By the fourth paragraph they have returned to their senses, realizing that there is a very narrow window in which the static EFE prediction might hold, and the rest of the section is find. The first three paragraphs need to be completely rewritten, preferably in a more general way that doesn't dwell on a couple of unicorn cases and preferably in a way that doesn't undermine their own case.
A minor but significant misstatement appears in section 10.2: "the travel speed of GWs came as a surprise to the MOND community" No, it did not - certainly not to this reviewer. Perhaps it came as a surprise to people who mistakenly conflated TeVeS with MOND, but that was a mistake I most commonly heard made by people in the LCDM community. I don't know how widespread this misconception was in the MOND community (I don't even know who that is), but I never heard it asserted as it is phrased here. This is just a false statement.
Such overstatements are sprinkled throughout the text, but it is too exhausting to track them all down. These few at least have to be fixed.
That said, a larger concern is section 10 and the associated tables. This is a good faith effort to objectively assess the predictive successes and accommodative flexibility of both LCDM and MOND. It is a laudable attempt, and I like how the table utilizes the language of the philosophy of science. It is also an impossible task. This is immediately illustrated by the absence of both Lambda and CDM as auxiliary hypotheses. The authors are starting from the reasonable position that LCDM is the current standard cosmology. But dark matter is itself an auxiliary hypotheses invented to save FRW cosmology. Lambda is an auxiliary hypotheses invented by Einstein to hold the universe static, then rehabilitated to save the flat geometry of Inflation. How bad is this? Do each of the lines in the tables deserve equal weight in eqn 72? Is the offset between X-ray and lensing centroids in the bullet cluster as important as needing to invent an invisible form of mass that requires novel physics outside the Standard Model of particle physics?
These are unanswerable questions, so I don't expect the authors to answer them. I am only making the point that they are accepting uncritically a lot of history... if they had attempted to make this same table 25 years ago, their starting assumptions would themselves be items to be listed and debated. I won't try to tell them what should and shouldn't be here, but I do expect that there not be errors in the listed entities, of which there are several.
Table 3: CMB anisotropies are listed as being used in theory construction. This is historically inaccurate. The only input that CMB anisotropies had in the construction of CDM was there absence at the level expected for a baryonic, GR-only universe. COBE only detected the first few multipoles in 1992; this was one of many observations that led to the change from SCDM to LCDM. This transition was accomplished by a variety of traditional cosmological observations culminating in confirmation by the type Ia SN experiments. Contrary to some accounts, the accelerated expansion rate of LCDM was entirely expected by 1998 - so much so that I would count that among the successful predictions of LCDM. It wasn't until 2000 that CMB observations provided any real input, by which time LCDM was already well established. In no way is it correct to say that these were used in theory construction. Worse, the earliest CMB data both confirmed and refuted LCDM. The location of the first peak was a strong corroboration of the flat geometry that was assumed/predicted by LCDM. The amplitude of the second peak was smaller than predicted. This leads us to the first entry in Table 3: BBN, which is listed as having a clear prior expectation and being in excellent agreement. That's wrong. The highest pre-CMB estimate was Tytler (1999) Omega_b*h^2 = 0.019 +/- 0.001. The unexpectedly small second peak required revision of BBN as an auxiliary hypothesis that allowed the theory to fit any plausible data. The CMB-fit value of 0.0225 is double the Know value from BBN and at least 3.5 different. The excellent agreement with D/H only emerged after CMB fits had dictated the "right" answer: it looks like a classic case of confirmation bias. So right off there are two entries in the top line of Table 3 that are simply wrong, and must be revised.
Table 4: In Table 4, disc galaxy rotation curves are listed as informing the theory. As with CMB fluctuations, that is an oversimplification. The only aspect of rotation curves that informed theory construction, and indeed motivated it, was that rotation curves tended towards asymptotic flatness. The shapes of rotation curves were not then well known, and a successful, completely a priori prediction of Milgrom (1983b) was for the shape of LSB galaxy rotation curves. This belongs in the top left of the table (a clear prior expectation that is in excellent agreement with data) but it is absent. Similarly, there is no mention of the Tully-Fisher relation. That the relation existed was already known, and played some role in theory construction (mostly that its slope was not 2). That it is really a relation between total baryonic mass and Vflat with zero dependence on scale length or surface brightness or gas fraction was a strong a priori prediction that also goes unmentioned. Do the authors really think that the lack of a distinct tidal dwarf mass-radius relation is more important than these? There are no entires at all in the "some tension" and "strong disagreement" rows of Table 4. Really? MOND has no problems at all? Should not (at least) the residual cluster mass discrepancy be listed here? Or are they so confident that sterile neutrinos will explain everything that this doesn't even warrant mention?
Table 5: same. On the first line, the ranking of MOND and LCDM for BBN is reversed. BBN was revised to save LCDM. It did not need to be revised - and is still consistent with lithium etc. - in order to successfully predict the amplitude of the second peak with practically zero flexibility (see McGaugh 2004 and references therein). The current depiction is flat out wrong and must be corrected. Further down, it is asserted that disc galaxy rotation curves on RAR were used in theory construction for MOND. Again, this is not true. "Flat", yes. On the RAR, no - that was a prediction. Yes, the flat part is on the low acceleration part of the RAR, but the shapes of rotation curves did not have to be consistent with a single RAR, nor did LSB galaxies have to fall exclusively on the low end of the RAR (as predicted by MOND but not LCDM). Similarly, CMB anisotropies were not used in constructing LCDM, though they are certainly used these days to specify its parameters to absurd precision. There seems to be some conflation of Planck 2018 cosmological parameters with LCDM itself, which is obviously much older.
On the whole, this is a good paper. In a review of such length, some overreach is inevitable. Please fix the relatively few items I have called out above.
Author Response
Referee 1:
This manuscript presents an exhaustive review of cosmology and extragalactic astronomy in the context of the modified Newtonian dynamics hypothesized by MOND. It is extremely thorough and clearly written, and should be published after some minor but substantive fixes.
Small omission: in the last paragraph of page 31 "The density wave theory should also apply in LSBs." Yes. Yes it should. This was explicitly used by McGaugh & de Blok (1998) to predict that conventional analyses would find uncomfortably high stellar M*/L, as later found by Fuchs and also Saburova & Zasov (2013). This was a successful a priori prediction of MOND, so should be mentioned given this manuscript's emphasis on the scientific method.
We have added a reference to the Saburova & Zasov (2013) work. The number of spiral arms in LSBs is already listed as one of the tests with a clear prior expectation in MOND because the Toomre stability condition follows fairly directly from the gravitational physics. Similarly, the work of McGaugh & de Blok (1998) is already cited, but we added some clarifications regarding this.
Regarding the prediction in section 3.3 of McGaugh & de Blok (1998), this pertains to the amount of mass required in the disk to maintain a thin disk. But since the amount of baryons which should be in the disk in LCDM can be worked out from a cosmological LCDM simulation, another way to look at this is that thin disk galaxies should be more common in MOND because the enhanced vertical gravity would reduce the aspect ratio. We have discussed elsewhere that the aspect ratio distribution of galaxies is not compatible with LCDM, which could be due to disks in this framework having too little vertical gravity and thus being thicker than observed disks.
Section 3.3: I find this discussion very strange. It is nominally about "observational signatures of the EFE" but starts by talking about one galaxy, DF2 - an object that they've already discussed at great length earlier in the manuscript. This is like talking about a branch in a forest without mentioning all the trees. DF2 is one ordinary if very overhyped dwarf galaxy with dodgy data that have changed continually since it van Dokkum's group decided to make the proverbial mountain out of a molehill. The opening sentence is untrue, that DF2 can only be reconciled with MOND with the EFE. That depends on its distance from NGC 1052, and what the inclination of its rotating globular cluster system is. One can easily extract from their data an equivalent velocity dispersion of 13 km/s (the EFE prediction) or 20 km/s (the isolated prediction) depending on the inclination. Which of these apply depends on the distance, which is hotly debated. There is, at present, no credible test with this galaxy. I can understand the desire to address this case, if not to do it again, and it is a very strange place to start a discussion of evidence for the EFE. A more obvious starting point would be Milgrom's earlier (1995) "seven dwarfs" paper, and McGaugh & Milgrom (2013b). The latter paper is cited in several places but not in this section, despite having a discussion of photometrically matched dwarfs that are or are not subject to the EFE... an effect that was essential to the successful a priori prediction of the M31 dwarfs with lower velocity dispersions - the very same ones that are puzzling in LCDM, and are dismissed by conventionalists as conveniently out of equilibrium. If that is fair for LCDM, then it is also fair for MOND, which brings us to the discussion of Crater 2, which is both bizarre and wrong. Advocates of dark matter often dismiss the successful predictions of MOND as some kind of magic, as if they'd rather believe in magic than science if it means giving credit to MOND. Here the authors - in a lengthy review advocating MOND - engage in this same magical thinking. There is a long discussion of how it shouldn't be possible to predict the velocity dispersion of this object because of tidal effects, thus dismissing the successful a priori prediction of the velocity dispersion of this object. This discussion grossly misrepresents the work of McGaugh (2016), and illustrates a rather naive static thinking on the part of the authors. Science is a series of increasingly accurate approximations. They are correct that Crate 2 is subject to tidal effects. Indeed, now that we know it had a close pericentric passage, it is pretty clear that it has achieved its current ultradiffuse form as a result of that, evolving along the [non-equilibrium, non-adiabatic] lines anticipated by the modeling of Brada & Milgrom (2000). McGaugh (2016) discussed this (albeit using the criterion of Brada & Milgrom rather than the one they choose) and said OK, can't know this will work, but the timescales are so long it is worth trying the EFE approximation. That this indeed works out would, in the normal course of scientific discussion, be interpreted as evidence that this approach was indeed an adequate approximation, bearing in mind that no approximation is ever perfect. Unexamined here are the predictions (or lack thereof) of LCDM, which by any of several methods anticipated a much higher velocity dispersion for this object (nominally 17 km/s; see section 4.2 of McGaugh 2016). So we wind up in the strange situation where the authors are advocating MOND as a theory, have a case with a clear predictive difference between LCDM (17) and MOND (2), the observation is 2.7, and what they say is that the MOND prediction can't possible be right despite being right while not even mentioning how for off the conventional prediction is. All in a section about evidence for the EFE, of which there are many better examples. By the fourth paragraph they have returned to their senses, realizing that there is a very narrow window in which the static EFE prediction might hold, and the rest of the section is find. The first three paragraphs need to be completely rewritten, preferably in a more general way that doesn't dwell on a couple of unicorn cases and preferably in a way that doesn't undermine their own case.
The discussion of observational evidence for the EFE was indeed not handled appropriately, so it has been significantly revised. DF2 is mentioned much less in this section – the uncertainties are large enough that it is unwise to conclude about fundamental physics using this example. Crater 2 is discussed in a different way – the observed low velocity dispersion would be even lower without tidal heating, so there is significant tension with the MOND prediction assuming no Galactic EFE. The prediction with the EFE works fine. What we were trying to say previously is that this is not a strong test of MOND because even if the self-gravity were negligible, the observations could still be explained. But the observations are accurate enough to distinguish between MOND with and without the EFE. To help our discussion, we have estimated that the line of sight velocity dispersion should be the quadrature sum of the virial value and that due to tides alone. The purpose of this section is not to compare with LCDM, but we have briefly mentioned that the very low velocity dispersion of Crater 2 is hard to understand in LCDM (Borukhovetskaya+ 2022, Errani+ 2022).
There are some earlier works which the referee mentions which we have also included in this section, though they had been cited elsewhere. The McGaugh & Milgrom (2013a,b) works are helpful here, while the older Milgrom (1995) work showed that it could be important to include the EFE when predicting the velocity dispersions of Galactic satellites. Their work does not clarify whether neglecting the EFE would have given realistic results, but for historical reasons we do still mention it here.
A minor but significant misstatement appears in section 10.2: "the travel speed of GWs came as a surprise to the MOND community" No, it did not - certainly not to this reviewer. Perhaps it came as a surprise to people who mistakenly conflated TeVeS with MOND, but that was a mistake I most commonly heard made by people in the LCDM community. I don't know how widespread this misconception was in the MOND community (I don't even know who that is), but I never heard it asserted as it is phrased here. This is just a false statement.
This has been deleted and the surrounding text rephrased. It is still the case that not all versions of MOND predict GWs to travel at c, so the fact that they do in reality and can be made to do so in MOND is not a major win for MOND. It is consistent with this observation, but clearly there exists significant theoretical flexibility.
Such overstatements are sprinkled throughout the text, but it is too exhausting to track them all down. These few at least have to be fixed.
That said, a larger concern is section 10 and the associated tables. This is a good faith effort to objectively assess the predictive successes and accommodative flexibility of both LCDM and MOND. It is a laudable attempt, and I like how the table utilizes the language of the philosophy of science. It is also an impossible task. This is immediately illustrated by the absence of both Lambda and CDM as auxiliary hypotheses. The authors are starting from the reasonable position that LCDM is the current standard cosmology. But dark matter is itself an auxiliary hypotheses invented to save FRW cosmology. Lambda is an auxiliary hypotheses invented by Einstein to hold the universe static, then rehabilitated to save the flat geometry of Inflation. How bad is this? Do each of the lines in the tables deserve equal weight in eqn 72? Is the offset between X-ray and lensing centroids in the bullet cluster as important as needing to invent an invisible form of mass that requires novel physics outside the Standard Model of particle physics?
The LCDM model indeed has various free parameters and has been constructed over the years with various fudge factors motivated by observations, as the referee points out. How bad this is should in principle be considered through the theoretical flexibility scores. If some observations were matched only by post-hoc adjustments to some parameters or through introducing new ones, then these observations are not counted as increasing confidence in the theory. This could be achieved if the same parameters now fixed by those observations were used to make clear a priori predictions in a different setting, and if these predictions were later confirmed. Issues like the flatness of the universe are not counted as good for LCDM, but it is difficult to argue this is necessarily bad for the model either, mainly because this is about physics at such an early time that both LCDM and MOND might be meaningless, putting the epoch beyond the scope of this review.
We have decided to assign equal weight to different lines of evidence because it could create biases otherwise. If it is justified to assign a lower weight to some line of evidence, this is presumably because it is uncertain either theoretically or observationally, or both. In this case, it should end up with similar theoretical flexibility and observational agreement scores, and thus contribute little overall.
Regarding the particular issue of the Bullet Cluster, it could be argued that this alone is not sufficient to invent dark matter, but it is not the only example. If it was, then the additional flexibility would reduce the confidence from many different tests while leading to improved agreement on only one test. We hope that we have provided a framework by which people can judge whether a simpler model that agrees slightly less well with the data could still be preferred. In general, we take the view that the offsets between X-ray and weak lensing centroids should be explained somehow – if a theory fails to do this, then it cannot be considered a good theory merely because it is more elegant and has less flexibility. However, our framework does not enshrine this assumption – one would have to work out how much the confidence scores are improved from the reduced flexibility, and how much confidence is lost due to disagreement with observations in this regard.
These are unanswerable questions, so I don't expect the authors to answer them. I am only making the point that they are accepting uncritically a lot of history... if they had attempted to make this same table 25 years ago, their starting assumptions would themselves be items to be listed and debated. I won't try to tell them what should and shouldn't be here, but I do expect that there not be errors in the listed entities, of which there are several.
The tables are an attempt to assess the currently prevailing standard cosmological paradigm and MOND supplemented by light sterile neutrinos. Inevitably the models considered would have been different 25 years ago, but hopefully the referee agrees that they are accurate today after the modifications we have made.
Table 3: CMB anisotropies are listed as being used in theory construction. This is historically inaccurate. The only input that CMB anisotropies had in the construction of CDM was there absence at the level expected for a baryonic, GR-only universe. COBE only detected the first few multipoles in 1992; this was one of many observations that led to the change from SCDM to LCDM. This transition was accomplished by a variety of traditional cosmological observations culminating in confirmation by the type Ia SN experiments. Contrary to some accounts, the accelerated expansion rate of LCDM was entirely expected by 1998 - so much so that I would count that among the successful predictions of LCDM. It wasn't until 2000 that CMB observations provided any real input, by which time LCDM was already well established. In no way is it correct to say that these were used in theory construction. Worse, the earliest CMB data both confirmed and refuted LCDM. The location of the first peak was a strong corroboration of the flat geometry that was assumed/predicted by LCDM. The amplitude of the second peak was smaller than predicted. This leads us to the first entry in Table 3: BBN, which is listed as having a clear prior expectation and being in excellent agreement. That's wrong. The highest pre-CMB estimate was Tytler (1999) Omega_b*h^2 = 0.019 +/- 0.001. The unexpectedly small second peak required revision of BBN as an auxiliary hypothesis that allowed the theory to fit any plausible data. The CMB-fit value of 0.0225 is double the Know value from BBN and at least 3.5 different. The excellent agreement with D/H only emerged after CMB fits had dictated the "right" answer: it looks like a classic case of confirmation bias. So right off there are two entries in the top line of Table 3 that are simply wrong, and must be revised.
Our logic is that by LCDM we mean LCDM with the parameters fixed, since without that a lot of its predictions become very uncertain (as indeed in MOND with a_0 left arbitrary). We have taken the view that the parameters should be fixed using CMB data. If instead they are fixed using other data, then these data would not count in favour of LCDM while the CMB would. We feel that this would not make too much difference overall. Usually LCDM theorists fix their model parameters using the CMB and then try to account for other observations like the local H_0. It may be that a different approach was typical in the past and other lines of evidence led to reasonably accurate constraints on some of the model parameters prior to precise CMB data, but this seems to be the approach today. We have clarified this in the text.
The expansion rate history of the Universe is already counted as a success for LCDM at redshifts above 0.2. Much of the actual data underpinning this are from supernovae as the referee points out, but we have not gone into this in detail because there are also other probes which give consistent results. However, the expansion rate history is just one function, so there are limits to how compelling it is for a theory if it gets this right to within the uncertainties. For LCDM, there is also a significant tension at redshifts below 0.2 that is called the Hubble tension.
The primordial deuterium abundance does seem to have jumped after the CMB data were released, so we have reduced how well this can be said to agree with LCDM expectations. This is also discussed in the Philosophy of MOND book and could be a case of confirmation bias, which we briefly mention.
Table 4: In Table 4, disc galaxy rotation curves are listed as informing the theory. As with CMB fluctuations, that is an oversimplification. The only aspect of rotation curves that informed theory construction, and indeed motivated it, was that rotation curves tended towards asymptotic flatness. The shapes of rotation curves were not then well known, and a successful, completely a priori prediction of Milgrom (1983b) was for the shape of LSB galaxy rotation curves. This belongs in the top left of the table (a clear prior expectation that is in excellent agreement with data) but it is absent. Similarly, there is no mention of the Tully-Fisher relation. That the relation existed was already known, and played some role in theory construction (mostly that its slope was not 2). That it is really a relation between total baryonic mass and Vflat with zero dependence on scale length or surface brightness or gas fraction was a strong a priori prediction that also goes unmentioned. Do the authors really think that the lack of a distinct tidal dwarf mass-radius relation is more important than these?
The rotation curves of many galaxies have been used to test MOND, covering many orders of magnitude in the surface brightness, which in MOND is important as it is closely related to the acceleration. Because of this, it is warranted to split the rotation curve tests into HSB and LSB categories. The Begeman, Broeils & Sanders (1991) paper used HSB galaxies to fix the a_0 parameter, so it is still the case that not only the asymptotic flatness of rotation curves but other data besides were used in theory construction, by which we also mean fixing theoretical parameters. However, LSB galaxies were observed later. Since the amount of star formation should be related to the amount of feedback in LCDM and this should affect the position on the RAR, rotation curve tests for gas-rich and gas-poor galaxies are also considered distinct. The agreement of MOND in both cases thus lends further support to it and cannot be considered likely in LCDM. We disagree with the referee that the scale length is another relevant parameter because this is mathematically related to the surface brightness, and also that the Tully-Fisher relation is an independent test compared to the RAR for similar reasons. The historical role of the former is hopefully clear from the text.
While these are important issues, they should not detract from other observations like the mass-size relation of tidal dwarf galaxies being similar to that of field dwarfs, which would mostly be primordial in LCDM. It is difficult to compare which is more important – these are both aspects of the Universe which should be understood in the correct theory. The issue with rotation curve tests is that in principle one can construct a dark matter halo that matches the rotation curve. But tidal dwarf galaxies should lack dark matter. In LCDM, this means they need to be much smaller at fixed baryonic mass to remain bound. This does not seem to be the case, even though many tidal dwarfs are known by now. The point is that this is an almost direct window on how much self-gravity tidal dwarfs have, which is a rather compelling test of MOND and LCDM. In the future, direct measurements of the internal kinematics of tidal dwarf galaxies could well settle the issue. Indeed, this was part of the motivation for the MOND community meeting in Cleveland having dwarf galaxies in the conference name. We feel it is important to note that tidal dwarfs appear to have about as much self-gravity as primordial dwarfs. This is contrary to LCDM because the so-called dual dwarf galaxy theorem is fairly clear on the cosmological simulation side (Haslbauer_2019). While the referee is correct to point out some of the deficiencies in the previous version of our review with regards to rotation curve tests, the importance of tidal dwarf galaxies to the debate should not be underestimated.
There are no entires at all in the "some tension" and "strong disagreement" rows of Table 4. Really? MOND has no problems at all? Should not (at least) the residual cluster mass discrepancy be listed here? Or are they so confident that sterile neutrinos will explain everything that this doesn't even warrant mention?
We have mentioned why the cluster discrepancies for MOND are not listed as being in tension with MOND. The reason is that with sterile neutrinos, there is no tension (Angus, Famaey & Diaferio 2010). The required neutrino density marginally reaches the Tremaine-Gunn limit at the centre, which is actually a strong hint in favour of their existence. This is why we argue that galaxy clusters work well in MOND. Having said that, our scores are consistent with the general intuition expressed by the referee that galaxy clusters are not good for MOND. This is because the sterile neutrinos increase the flexibility that MOND has on galaxy cluster scales. However, the flexibility is limited because of the Tremaine-Gunn limit to the phase space density. Therefore, we have assigned the second-highest theoretical flexibility score. This means the overall confidence score from this test is zero, which on average is basically what LCDM achieves – the equilibrium dynamics of galaxy clusters are not counted as being in favour of MOND. But they are not in tension either.
Table 5: same. On the first line, the ranking of MOND and LCDM for BBN is reversed. BBN was revised to save LCDM. It did not need to be revised - and is still consistent with lithium etc. - in order to successfully predict the amplitude of the second peak with practically zero flexibility (see McGaugh 2004 and references therein). The current depiction is flat out wrong and must be corrected. Further down, it is asserted that disc galaxy rotation curves on RAR were used in theory construction for MOND. Again, this is not true. "Flat", yes. On the RAR, no - that was a prediction. Yes, the flat part is on the low acceleration part of the RAR, but the shapes of rotation curves did not have to be consistent with a single RAR, nor did LSB galaxies have to fall exclusively on the low end of the RAR (as predicted by MOND but not LCDM). Similarly, CMB anisotropies were not used in constructing LCDM, though they are certainly used these days to specify its parameters to absurd precision. There seems to be some conflation of Planck 2018 cosmological parameters with LCDM itself, which is obviously much older.
Table 5 follows from the previous tables, corrections to which have been fed through to corrections here. While LCDM is older than precise CMB data, specifying the parameters to good precision – and thus allowing precise tests of LCDM on other scales – does rely on the CMB data. Moreover, as explained above, even if it was possible to use other lines of evidence to set the LCDM parameters, these tests would then not count in favour of LCDM while the CMB would, which would have little effect on the overall score.
The referee also mentioned the McGaugh (2004) work. This has now been cited in Section 9.2.1 where we first consider the CMB. However, we also mention that the model which was claimed to work in that paper also predicted other features which were subsequently refuted, i.e. the height of the third peak in the CMB power spectrum. This motivates the rest of the discussion on adding HDM. Note that we have also discussed a way to do the CMB in MOND without dark matter, but we do not consider this very promising due to producing the same large-scale structure as LCDM and not providing a good explanation for the dynamics of galaxy clusters. This is still under investigation though, so we should keep an open mind.
On the whole, this is a good paper. In a review of such length, some overreach is inevitable. Please fix the relatively few items I have called out above.
We thank the referee for his/her comments and have addressed them as explained above.

Reviewer 2 Report
This is a well written paper and I felt excited reading it, but I felt that this is a contribution in the field of Astronomy, and not in mathematics. The authors review the MOND theory, which makes predictions using data from astronomical observations (galaxies, clusters, etc.). I did not see any mathematical methodology, throughout the text, which lacks completely the rigor of spacetime geometry, topology, geometric analysis, even statistical mechanics (that are widely used in GR and theories beyond it).
Author Response
Referee 2:
This is a well written paper and I felt excited reading it, but I felt that this is a contribution in the field of Astronomy, and not in mathematics. The authors review the MOND theory, which makes predictions using data from astronomical observations (galaxies, clusters, etc.). I did not see any mathematical methodology, throughout the text, which lacks completely the rigor of spacetime geometry, topology, geometric analysis, even statistical mechanics (that are widely used in GR and theories beyond it).
This review is for a special issue on modified gravity theories and applications to astrophysics and cosmology, so the referee is correct that it is not a contribution to mathematics. There are many equations in the text and a detailed description of the mathematical methodology. Though the focus is on non-relativistic systems, there are covariant equations in the review, and more importantly references to other works which describe relativistic MOND theories. The velocity dispersions are an aspect of statistical mechanics (the spherical Jeans equation; see our Section 3.2.1). MOND is not as developed as GR in some ways, but nonetheless the review tries to be as mathematically rigorous as possible while leaving some aspects open to reflect the current lack of knowledge in certain areas.

Reviewer 3 Report
In this article, the authors review and outline the existing astrophysical evidence for MOND and compare it to the LCDM model. Overall, the article is very well written and thought out and surely deserves publication as soon as a couple of points have been clarified.
- One minor comment about Eq.26: I assume that the crossed out terms cancel or are dropped, but this is not commented on in the text, and it would help the reader if the reason was explicitly stated.
- My main concern is the 'Confidence' score defined in Eq.72, which is used by the authors to determine how well MOND fares in a specific test, as compared to LCDM. There is no derivation or citation accompanying Eq.72, and I do not see how it could be derived from e.g. Bayesian evidence, since it seems to be a discrete classifier. I ask that the authors motivate and clarify the use of the Confidence measure. I may have simply missed it when reading the paper, as it is very long.
Author Response
Referee 3:
In this article, the authors review and outline the existing astrophysical evidence for MOND and compare it to the LCDM model. Overall, the article is very well written and thought out and surely deserves publication as soon as a couple of points have been clarified.
One minor comment about Eq.26: I assume that the crossed out terms cancel or are dropped, but this is not commented on in the text, and it would help the reader if the reason was explicitly stated.
The crossed-out terms are neglected for reasons that we now explain. The chemical potential mu is assumed to be negligible as the neutrinos decoupled while relativistic, at which time they were tightly coupled to photons with zero chemical potential. The mass is neglected compared to the momentum for a similar reason: the neutrinos at the early times considered here are ultra-relativistic. This has been clarified in the text.
My main concern is the 'Confidence' score defined in Eq.72, which is used by the authors to determine how well MOND fares in a specific test, as compared to LCDM. There is no derivation or citation accompanying Eq.72, and I do not see how it could be derived from e.g. Bayesian evidence, since it seems to be a discrete classifier. I ask that the authors motivate and clarify the use of the Confidence measure. I may have simply missed it when reading the paper, as it is very long.
The theoretical flexibility and observational agreement scores are in any case subjective, though we have tried to justify the choices made where these were not obvious or counterintuitive. Subtracting one score from the other is also not mathematically rigorous, as the referee points out. However, we have further clarified why we have done it this way.
Note Eq. 72 applies only to one theory at a time, so in principle it can be used to test LCDM without considering MOND. Then the way to read it is that a physically meaningless theory can still agree with observations fairly well if there are many free parameters and much flexibility in the model, but if it has very little flexibility, then there will typically be a very significant disagreement with observations. Therefore, a theory with some physically realistic aspects should get a score above 0 on average. This is regardless of whether other theories exist.

Reviewer 4 Report
About fourty years ago, in order to explain the nature of the phenomenon of hidden mass in galaxies, the concept of modified gravity (MOND theory) was proposed. For decades, this concept has been viewed by the scientific community as somewhat marginal. Noticeably more preference was given to the idea of ​​the existence of some hypothetical dark matter. However, in the past few years, due to the fact that (i) dark matter particles have not been detected, and (ii) due to the emergence of significant disagreements in the determination of the Hubble constant using different observational techniques, the concept of modified gravity has found a new lease of life.
Both MONDian gravity and dark matter approaches are not a panacea for all the problems associated with interpreting observations. Both concepts are relatively successful on some scales but become unsatisfactory on others.
Just now, in the current year 2021, the idea of ​​a hybrid concept has become popular, allowing the simultaneous coexistence of both MONDian gravity and dark matter particles (see, for example, Tobias Mistele, Cherenkov radiation from stars constrains hybrid MOND-dark-matter models). The authors of the peer-reviewed paper advocate one such hybrid concept.
I should note that the hybrid model loses the elegance of both models, since it increases the number of fit parameters (the number of degrees of freedom) of the model, but it allows to explain a larger amount of observational data.
I am confident that the Astrophysics Special Issue of Symmetry could provide a suitable forum for discussion between the modified-gravity and dark-matter communities, so I recommend this article for publication.
Somewhat alarming is the fact that the text of the article was typeset not in the template dictated by the rules for formatting articles in the Symmetry journal, but in the template of another journal. I hope that the authors will once again confirm that the article is not (or is not already) under consideration in the Monthly Notices of the Royal Astronomical Society.
Author Response
Referee 4:
About fourty years ago, in order to explain the nature of the phenomenon of hidden mass in galaxies, the concept of modified gravity (MOND theory) was proposed. For decades, this concept has been viewed by the scientific community as somewhat marginal. Noticeably more preference was given to the idea of the existence of some hypothetical dark matter. However, in the past few years, due to the fact that (i) dark matter particles have not been detected, and (ii) due to the emergence of significant disagreements in the determination of the Hubble constant using different observational techniques, the concept of modified gravity has found a new lease of life.
Both MONDian gravity and dark matter approaches are not a panacea for all the problems associated with interpreting observations. Both concepts are relatively successful on some scales but become unsatisfactory on others.
Just now, in the current year 2021, the idea of a hybrid concept has become popular, allowing the simultaneous coexistence of both MONDian gravity and dark matter particles (see, for example, Tobias Mistele, Cherenkov radiation from stars constrains hybrid MOND-dark-matter models). The authors of the peer-reviewed paper advocate one such hybrid concept.
We do advocate a hybrid approach, but not a unified solution for the MOND phenomenology on galaxy scales and the dark matter phenomenology on larger scales, e.g. in the Bullet Cluster. A unified solution is likely to run into severe problems, as discussed in the reference mentioned above (which we cited) and explained further by its author in this guest blog post:
https://darkmattercrisis.wordpress.com/2021/12/01
I should note that the hybrid model loses the elegance of both models, since it increases the number of fit parameters (the number of degrees of freedom) of the model, but it allows to explain a larger amount of observational data.
That is indeed the trade-off, though it is still consistent with Occam’s Razor because this is not about removing model parameters from a working model to come up with a simpler model which then fails to fit the observations. We have tried to account for the extra flexibility in such a hybrid model through the confidence score in Section 10. One important thing to understand is that although allowing sterile neutrinos increases the flexibility of MOND, these would not affect galaxies, so the extra degrees of freedom are effectively frozen out on this scale, where MOND remains very predictive. In galaxy clusters, the sterile neutrinos cannot be used to explain any plausible dataset because their phase space density is limited by the Tremaine-Gunn limit. Therefore, while the referee is correct, the situation is not as bad as it could be if e.g. the sterile neutrinos also allowed almost anything to be possible in MOND on galaxy scales, which is basically the situation in LCDM because it assumes much higher mass dark matter particles with an essentially infinite Tremaine-Gunn limit.

Round 2
Reviewer 3 Report
My comments have been adequately addressed, and the submission has been improved.